# WHU-SGCC: A novel approach for blending daily satellite (CHIRP) and precipitation observations over the Jinsha River Basin

Gaoyun Shen[1], Nengcheng Chen[1,2], Wei Wang[1,2], Zeqiang Chen [1,2]

[1] State Key Laboratory of Information Engineering in Surveying, Mapping and Remote Sensing, Wuhan University, 129 Luoyu Road, Wuhan 430079, China

[2] Collaborative Innovation Center of Geospatial Technology, Wuhan 430079, China

*Correspondence to*: ZeqiangChen@whu.edu.cn; Tel.: +86 13871025965; Fax: +86 27 68778229.

**Abstract.** Accurate and consistent satellite based precipitation estimates blended with rain gauge data are important for regional precipitation monitoring and hydrological applications, especially in regions with limited rain gauges. However, the existing fusion precipitation estimates often have large uncertainties over mountainous areas with complex topography and sparse rain gauges, and most of the existing data blending algorithms are not good at removing the day by day errors. Therefore, the development of effective methods for high accuracy precipitation estimates over complex terrain and at a daily scale is of vital importance for mountainous hydrological applications. This study aims to offer a novel approach for blending daily precipitation gauge data and the Climate Hazards Group Infrared Precipitation (CHIRP; daily, 0.05°) satellite derived precipitation developed by UC Santa Barbara over the Jinsha River Basin from 1994 to 2014. This method is called the Wuhan University Satellite and Gauge precipitation Collaborated Correction (WHU-SGCC). The results show that the WHU-SGCC method is effective for liquid precipitation bias adjustments from points to surfaces as evaluated by multiple error statistics and from different perspectives. Compared with CHIRP and CHIRP with station data (CHIRPS), the precipitation adjusted by the WHU-SGCC method has greater accuracy, with overall average improvements of the Pearson's correlation coefficient (PCC) by 0.0082-0.2232 and 0.0612-0.3243, respectively, and decreases in the root mean square error (RMSE) by 0.0922-0.65 mm and 0.2249-2.9525 mm, respectively. In addition, the Nash-Sutcliffe efficiency coefficient (NSE) of the WHU-SGCC provides substantial improvements than CHIRP and CHIRPS, which reached 0.2836, 0.2944 and 0.1853 in the spring, fall and winter. Daily accuracy evaluations indicate that the WHU-SGCC method has the best ability to reduce precipitation bias, with average reductions of 21.68% and 31.44% compared to CHIRP and CHIRPS, respectively. Moreover, the accuracy of the spatial distribution of the precipitation estimates derived from the WHU-SGCC method is related to the complexity of the topography. The validation also verifies that the proposed approach is effective at detecting major precipitation events within the Jinsha River Basin. In spite of the correction, the uncertainties in the seasonal precipitation forecasts in the summer and winter are still large, which might be due to the homogenization attenuating the extreme rain events estimates. However, the WHU-SGCC approach may serve as a promising tool to monitor daily precipitation over the Jinsha River Basin, which contains complicated mountainous terrain with sparse rain gauge data, based on the spatial correlation and the historical precipitation characteristics. The daily precipitation estimations at the 0.05° resolution over the Jinsha River Basin during all four seasons from 1990 to 2014, derived from WHU-SGCC are available at the PANGAEA Data Publisher for Earth & Environmental Science portal (https://doi.pangaea.de/10.1594/PANGAEA.905376).

## 1 Introduction

Accurate and consistent estimates of precipitation are vital for hydrological modelling, flood forecasting and climatological studies in support of better planning and decision making (Agutu et al., 2017; Cattani et al., 2018; Roy et al., 2017). In general, ground-based gauge networks include a substantial number of liquid precipitation observations measured with high accuracy, high temporal resolution, and long historical records. However, the sparse distribution and point measurements limit the accurate estimation of spatially gridded rainfall (Martens et al., 2013).

Due to the sparseness and uneven spatial distribution of rain gauges and the high proportion of missing data, satellite-derived precipitation data are an attractive supplement offering the advantage of plentiful information with high spatio-temporal resolution over widespread regions, particularly over oceans, high elevation mountainous regions, and other remote regions where gauge networks are difficult to deploy. However, satellite estimates are susceptible to systematic biases that can influence hydrological modelling and the retrieval algorithms are relatively insensitive to light rainfall events, especially in complex terrain, resulting in underestimations of the magnitudes of precipitation events (Behrangi et al., 2014; Thiemig et al., 2013; Yang et al., 2017). Without adjustments, inaccurate satellite-based precipitation estimates will lead to unreliable assessments of risk and reliability (AghaKouchak et al., 2011).

Accordingly, many kinds of precipitation estimates combining multiple sources and datasets are available. Table 1 shows the temporal and spatial resolution of current major satellite-based precipitation datasets. Since 1997, the Tropical Rainfall Measurement Mission (TRMM) has improved satellite-based rainfall retrievals over tropical regions (Kummerow et al., 1998; Simpson et al., 1988). High spatial and temporal resolution multi-satellite precipitation products have been developed continuously during the TRMM era (Maggioni et al., 2016), including: (1) the TRMM Multisatellite Precipitation Analysis (TMPA) products, which are derived from gauge-satellite fusing (Huffman et al., 2010; Vila et al., 2009); (2) the Climate Prediction Center (CPC) morphing technique (Joyce et al., 2004; Joyce and Xie, 2011; Xie et al., 2017), which integrates geosynchronous infrared (GEO IR) and polar-orbiting microwave (PMW) sensor data and is available three hourly on a grid with a spatial resolution of 0.25°; (3) the Precipitation Estimation from Remotely Sensed Information using Artificial Neural Networks - Climate Data Record (PERSIANN-CDR) produced by the PERSIANN algorithm, which has daily temporal and 0.25° × 0.25° spatial resolutions (Ashouri et al., 2015); and (4) the Global Satellite Mapping of Precipitation (GSMaP) project, which produces global rainfall estimates in near-real time and applies the motion vector Kalman filter based on physical models (GSMaP-NRT and GSMaP-MVK, respectively) (Aonashi et al., 2009; Ushio et al., 2009; Ushio and Kachi, 2010). In 2014, the Global Precipitation Measurement (GPM) satellite was launched after the success of the TRMM satellite by a cooperation between the National Aeronautics and Space Administration (NASA) and Japan Aerospace Exploration Agency (JAXA) (Mahmoud et al., 2018; Ning et al., 2016). The main core observatory satellite (GPM) integrates advanced radar and radiometer systems to obtain the precipitation physics and takes advantages of TMPA, the Climate Prediction Center morphing technique (CMORPH), and PERSIANN algorithms to offer high spatiotemporal resolution products (0.1° × 0.1°, half hourly) of global real time precipitation estimates (Huffman et al., 2018; Skofronick-Jackson et al., 2017; Hou et al., 2014). Nevertheless, the major aforementioned products have only been available since 1998, which limits long term climatological studies. Only the PERSIANN-CDR data set has temporal coverage since 1983. However, the spatial resolution of PERSIANN-CDR is relatively coarse, and the data resolution must be degraded to achieve high accuracy in precipitation monitoring. To fill the gap in high resolution and long term global multi-satellite precipitation monitoring, the Multi-Source Weighted-Ensemble Precipitation (MSWEP) product (Beck et al., 2017; Beck et al., 2019), and the Climate Hazards Group Infrared Precipitation with Station data (CHIRPS) product from UC Santa Barbara (Funk et al., 2015 a) were developed. MSWEP is a precipitation data set with global coverage available at 0.1° spatial resolution and at three hourly, daily, and monthly temporal resolutions. MSWEP is multi-source data that takes advantage of the complementary strengths of gauge-, satellite-, and reanalysis-based data. However, to provide precipitation estimates at a higher spatial resolution, the CHIRPS data set is used in this study.

CHIRPS is a longer length precipitation data series with a higher spatial resolution (0.05°) that, merges three types of information: global climatology, satellite estimates and in situ observations. The CHIRPS precipitation dataset with several temporal and spatial scales has been evaluated in Brazil (Nogueira et al., 2018; Paredes-Trejo et al., 2017), Chile (Yang et al., 2016; Zambrano-Bigiarini et al., 2017), China (Bai et al., 2018), Cyprus (Katsanos et al., 2016a; Katsanos et al., 2016b), India (Ali and Mishra, 2017; Prakash, 2019) and Italy (Duan et al., 2016). However, the temporal resolutions of these applications

were mainly at seasonal and monthly scales, lacking the evaluation and correction of daily precipitation. Additionally, despite the great potential of gauge-satellite fusing products for large scale environmental monitoring, there are still large discrepancies with ground observations at the sub-regional level where these data have been applied. Furthermore, the CHIRPS product's reliability has not been analysed in detail over the Jinsha River Basin in China, particularly at a daily scale. The Jinsha River Basin is a typical study area with complex and varied terrain, an uneven spatial distribution of precipitation, and a sparse spatial distribution of rain gauges, which limit high accuracy precipitation monitoring. The existing research indicates that estimations over mountainous areas with complex topography often have large uncertainties and bias due to the topography, seasonality, climate impact and sparseness of rain gauges (Derin et al., 2016; Maggioni and Massari, 2018; Zambrano-Bigiarini et al., 2017). Moreover, Bai et al. (2018) evaluated CHIRPS over mainland China and indicated that the performance of CHIRPS is poor over the Sichuan Basin and the Northern China Plain, which have complex terrain with substantial variations in elevation. Additionally, Trejo et al. (2016) shows that CHIRPS overestimates low monthly rainfall and underestimates high monthly rainfall using several numerical metrics and that the rainfall event frequency is overestimated outside the rainy season.

**Table 1** Coverage and spatiotemporal resolutions of major satellite precipitation datasets.

| Product | Temporal resolution | Spatial resolution | Period | Coverage |
|---|---|---|---|---|
| TRMM 3B42-RT | 3 Hourly | 0.25° | 1998-present | 50°S-50°N |
| CMORPH | 0.5 Hourly/3 Hourly/Daily | 8 km/0.25° | 1998- | 60°S-60°N |
| PERSIANN-CDR | daily | 0.25° | 1983-(delayed) present | 60°S-60°N |
| GsMaP-NRT | Hourly | 0.01° | 2007 | 60°S-60°N |
| GsMaP-MVK | Hourly | 0.01° | 2000 | 60°S-60°N |
| GPM | 0.5 Hourly/Hourly/ 3 Hourly/Daily/3 Day/ 7 Day/Monthly | 0.1°/0.25°/0.05°/5° | 2014-present | 60°S-60°N 70°N-70°S 90°N-90°S |
| MSWEP | 3 Hourly/Daily/Monthly | 0.1° | 1979-2017 | 90°N-90°S |
| CHIRPS | Daily/Pentad/Dekad/ Monthly/Annual | 0.05°/0.25° | 1981- present | 50°S-50°N |

To overcome these limitations, many studies have focused on proposing effective methodologies for blending rain gauge observations, satellite-based precipitation estimates, and sometimes radar data to take advantage of each dataset. Many numerical models have been established with these datasets for high accuracy precipitation estimations, such as bias adjustment by a quantile mapping (QM) approach (Yang et al., 2016), Bayesian kriging (BK) (Verdin et al., 2015) and a conditional merging technique (Berndt et al., 2014). The QM approach is a distribution-based approach, which works with historical data for bias adjustment and is effective at reducing the systematic bias of regional climate model precipitation estimates at monthly or seasonal scales (Chen et al., 2013). However, the QM approach offers very limited improvement in removing day by day errors. The BK approach provides very good model fit with precipitation observations, but the Gaussian assumption of the BK model is invalid for daily scales. Overall, there is a lack of effective methods for high-accuracy precipitation estimates over complex terrain at a daily scale.

As such, due to the poor performance at the sub-regional scale, the gauge-satellite fusing algorithms can be assumed to limit high accuracy estimations in the process of CHIRPS data production. Therefore, the aim of this article is to present a novel approach for reblending daily liquid precipitation gauge data and the Climate Hazards Group Infrared Precipitation (CHIRP) satellite-derived precipitation estimates developed by UC Santa Barbara, over the Jinsha River Basin. We use precipitation to denote liquid precipitation throughout the text. The CHIRP data are the raw data of CHIRPS before blending with the rain gauge data. The objective is to build corresponding precipitation models that consider terrain factors and precipitation characteristics to produce high quality precipitation estimates. This novel method is called the Wuhan University Satellite and Gauge precipitation Collaborated Correction (WHU-SGCC) method. We present this method by applying it to daily precipitation over the Jinsha River Basin in the different seasons from 1990 to 2014. The results support the validity of the proposed approach for producing refined satellite-gauge precipitation estimates over mountainous areas.

The remainder of this paper is organized as follows: Section 2 describes the study region, rain gauges and CHIRPS dataset used in this study. Section 3 presents the principle of the WHU-SGCC approach for high accuracy daily precipitation estimates.

The results and discussion are analysed in Section 4, the data available are described in Section 5, and the conclusions and
future work are presented in Section 6.
**2 Study Region and Data**
**2.1 Study Region**
The Yangtze River is one of the largest and most important rivers in Southeast Asia, originating on the Tibetan Plateau and
extending approximately 6300 km eastward to the East China Sea. The river's catchment covers an area of approximately ~180
$\times$ 10$^4$ km$^2$ and the average annual precipitation is approximately 1100 mm (Zhang et al., 2019).The Yangtze River is divided
into nine sub-basins, the upper drainage basin is the Jinsha River Basin, which flows through the provinces of Qinghai, Sichuan,
and Yunnan in western China. Within the Jinsha River Basin, the total river length is 3486 km, accounting for 77% of the
length of the upper Yangtze River, and covering a watershed area of 460 $\times$ 10$^3$ km$^2$. The location of the Jinsha River Basin is
shown in Fig. 1, and it covers the eastern part of the Tibetan Plateau and part of the Hengduan Mountains. The southern portion
of the river basin is the Northern Yunnan Plateau and the eastern portion includes a wide area of the southwestern margin of
the Sichuan Basin. Crossing complex and varied terrains, the elevation of the Jinsha River ranges from 263 to 6575 m above
sea level, which results in significant temporal and spatial climate and weather variations inside the basin. The average annual
precipitation of the Jinsha River Basin is approximately 710 mm, the average annual precipitation of the lower reaches is
approximately 900-1300 mm, and the average annual precipitation of the middle and upper reaches is approximately 600-800
mm (Yuan et al., 2018). The Jinsha River Basin has four seasons: spring (March-April-May), summer (June-July-August), fall
(September-October-November) and winter (December-January-February). Therefore, the blending of satellite estimations
with gauge observations during the different seasons is the main focus of this research.

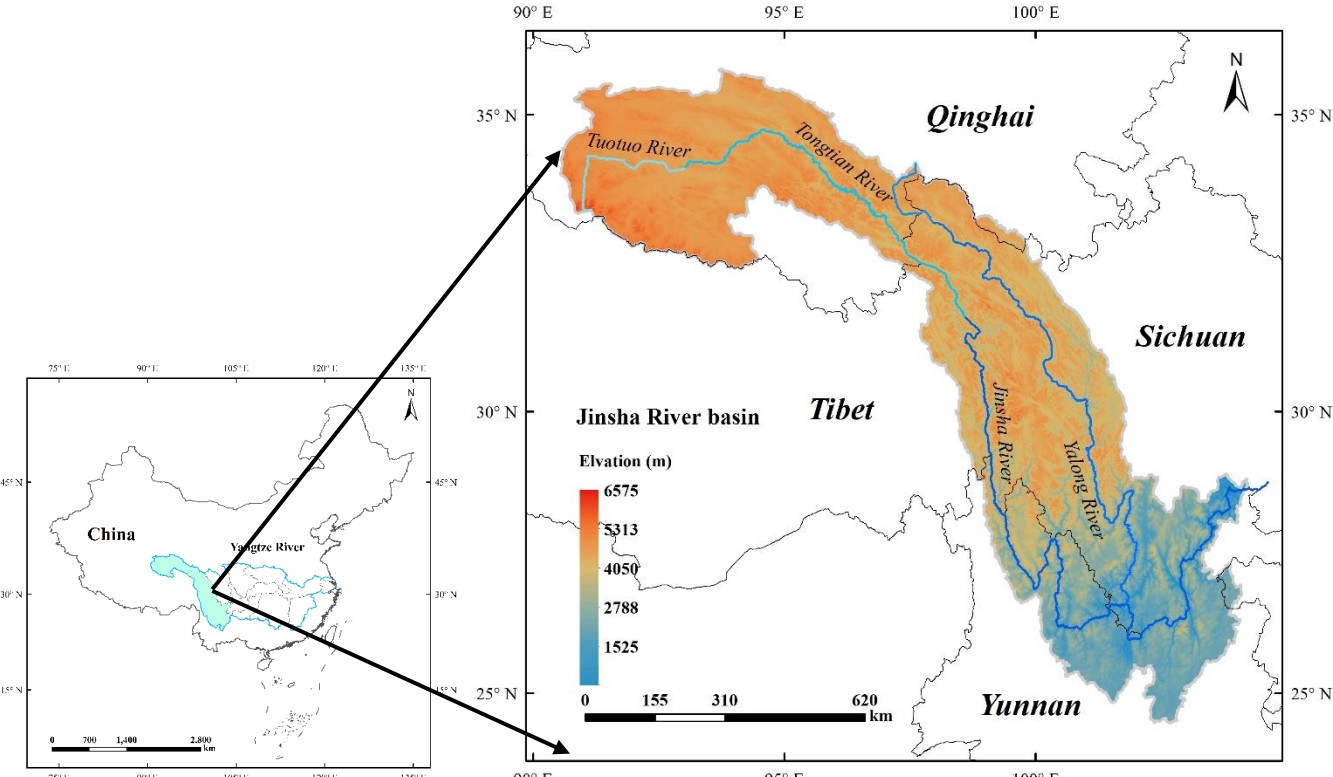

**Figure 1** Location of the study area with key topographic features.
**2.2 Study Data**
**2.2.1 Precipitation gauge observations**

Daily rain gauge observations at 30 national standard rain stations within the Jinsha River Basin from 1 March 1990 to February 2015 were provided by the National Climate Centre (NCC) of the China Meteorological Administration (CMA) (http://data.cma.cn/data/cdcdetail/dataCode/SURF_CLI_CHN_MUL_DAY_V3.0.html, last access: 10 December, 2018), which imposes strict quality control at the station, provincial and state levels. The process of quality control conducted by the CMA is as follows: (1) Climate threshold or allowable value check; (2) Extreme values at gauge stations check; (3) Internal consistency check between fixed value, daily average value and daily extreme value; (4) Time consistency check; and (5) Manual verification and correction. The station identification numbers and relevant geographical characteristics are shown in Appendix A, and their uneven spatial distribution is shown in Fig. 2. The selected rain gauges are located in Qinghai, Tibet, Sichuan and Yunnan Provinces but are mainly scattered in Sichuan Province, and the northern river basin contains fewer rain gauges than the southern river basin. In this study, the daily rain gauge observations were used as the reference data for the bias correction of satellite precipitation estimations.

The multi-annual (1990-2014) average seasonal precipitation over the Jinsha River Basin increases from north to south (Fig. 2). The dynamic and uneven distribution of precipitation is influenced distinctly by the seasonal climate. Most of the precipitation falls in the summer, with the average seasonal precipitation ranging from less than 250 mm to more than 600 mm, while the average seasonal precipitation during the winter is no more than 50 mm. The average seasonal precipitation and spatial distribution in the spring are similar with those in the fall, with values concentrated in the range of 50 mm to 200 mm.

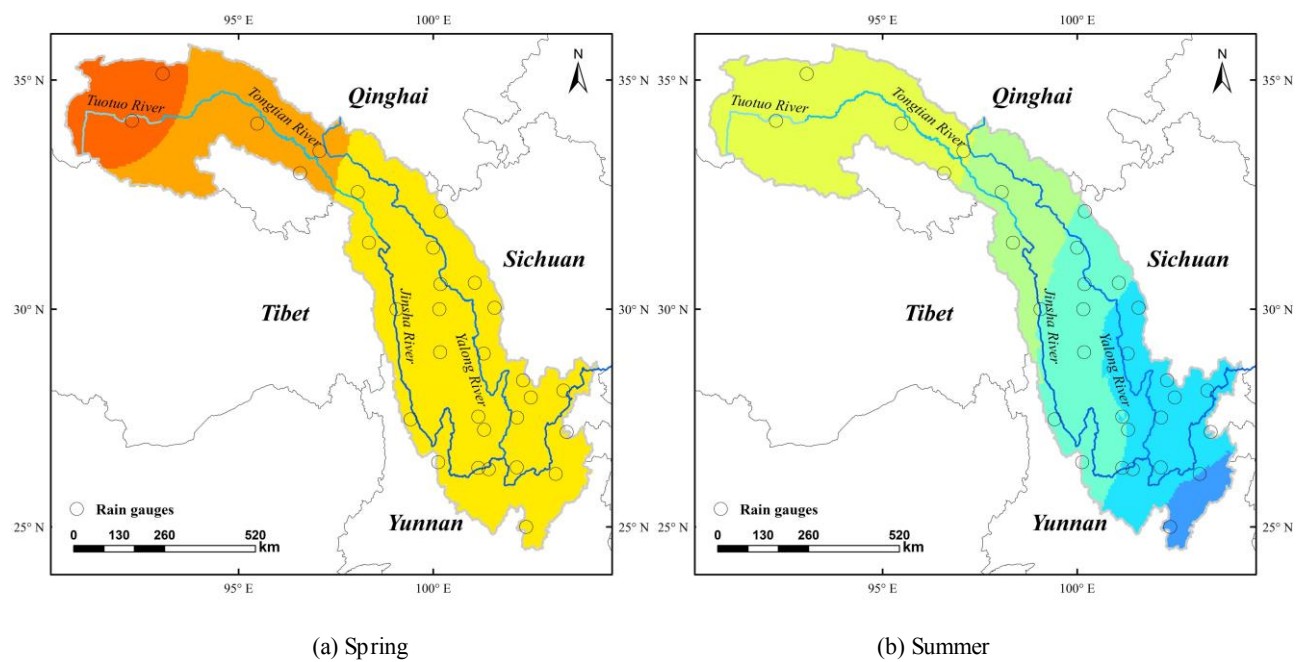

(a) Spring                                    (b) Summer

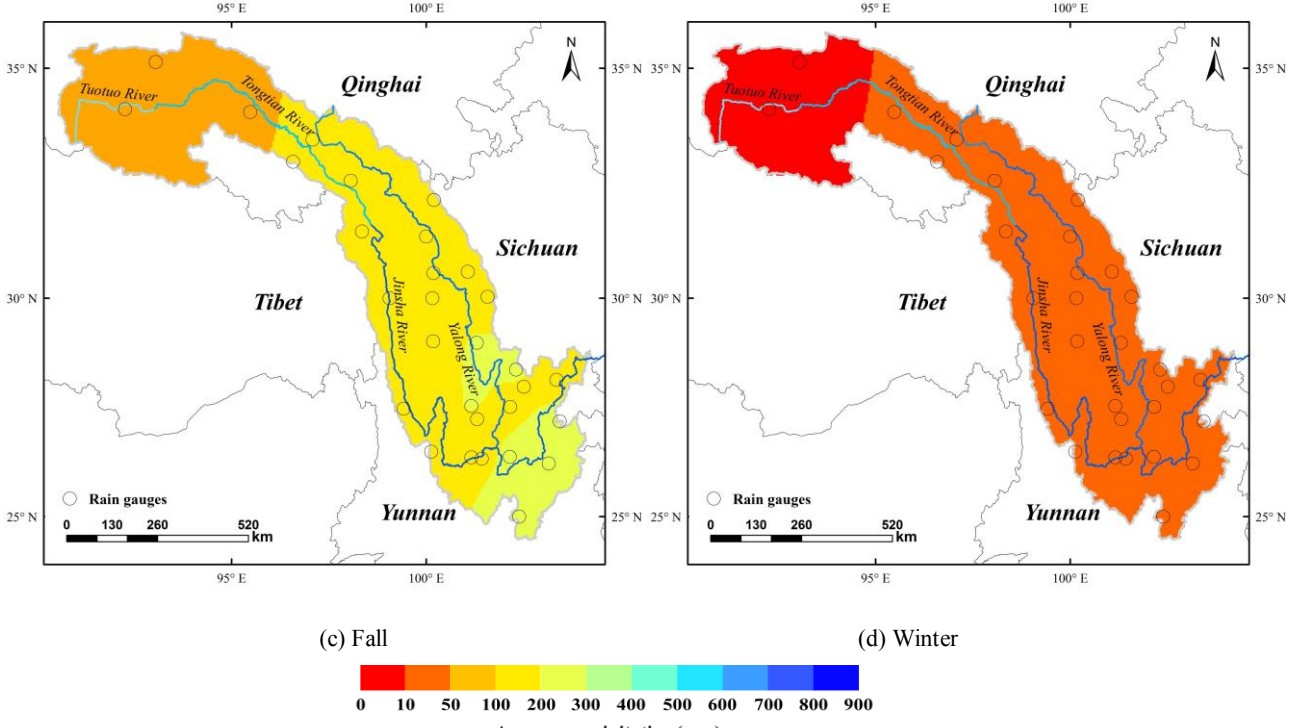

(c) Fall                                    (d) Winter


**Figure 2** The multi-annual (1990-2014) average seasonal precipitation over the Jinsha River Basin interpolated from 30 rain gauges
downloaded from the China Meteorological Administration stations.
**2.2.2 CHIRPS satellite-gauge fusion precipitation estimates**
The CHIRPS v.2 dataset, a satellite-based daily rainfall product, is available online at
ftp://ftp.chg.ucsb.edu/pub/org/chg/products/CHIRPS-2.0/global_daily/tifs/p05/ (last access: 10 December, 2018). It covers a
quasi-global area (land only, 50° S-50° N) a several temporal scales (daily, pentad, dekad, monthly and annual temporal
resolutions) and a high spatial resolution (0.05°) (Rivera et al., 2018). This dataset contains a wide variety of satellite-based
rainfall products derived from a multiple data sources and incorporates five data types: (1) the monthly precipitation from
CHPClim v.1.0 (Climate Hazards Group's Precipitation Climatology version 1) derived from a combination of satellite fields,
gridded physiographic indicators, and in situ climate normal with the geospatial modelling approach based on moving window
regressions and inverse distance weighting interpolation (Funk et al., 2015 b); (2) quasi-global geostationary thermal infrared
(IR) satellite observations; (3) the TRMM 3B42 product (Huffman et al., 2007); (4) the CFS (Climate Forecast System, version
2) atmospheric model rainfall fields from NOAA; and (5) surface-based precipitation observations from various sources
including national and regional meteorological services. The differences from other frequently used precipitation products are
the higher resolution of 0.05° , wider coverage and longer length data series from 1981 to near-real time (Funk et al., 2015 a).
CHIRPS is the blended product of a two-part process. First, IR precipitation (IRP) pentad rainfall estimates are fused with
corresponding CHPClim pentad data to produce an unbiased gridded estimate, called CHIRP, which is available online at
ftp://ftp.chg.ucsb.edu/pub/org/chg/products/CHIRP/daily/ (last access: 10 December, 2018). In the second part of the process,
the CHIRP data are blended with ground-based precipitation observations obtained from a variety of sources, including
national and regional meteorological services by means of a modified inverse-distance weighting algorithm to create the final
blended product, CHIRPS (Funk et al., 2014). The daily CHIRP satellite-based data over the Jinsha River Basin from 1990.02
to 2015.02 were selected as the input for WHU-SGCC blending with rain observations, and the corresponding daily CHIRPS
blended data was used for comparisons of the precipitation accuracy.
The blended in situ daily precipitation observations of the CHIRPS data come from a variety of sources, such as the daily
GHCN archive (Durre et al., 2010), the Global Summary of the Day dataset (GSOD) provided by NOAA's National Climatic
Data Center, the World Meteorological Organization's Global Telecommunication System (GTS) daily archive provided by
NOAA CPC, and more than a dozen national and regional meteorological services. However, the stations for daily CHIRPS
data have a different spatial distribution than those downloaded from the CMA, and the precipitation values used for CHIRPS
production are the monthly values available online (ftp://ftp.chg.ucsb.edu/pub/org/chg/products/CHIRPS-
2.0/diagnostics/monthly_station_data/). For the daily precipitation adjustments over the Jinsha River Basin, the daily gauge
observations from the CMA are blended with the daily CHIRP data due to the unknown spatial distribution and precipitation
values of gauge stations used in the process of daily CHIRPS merging.

**3 Methods**

**3.1 The WHU-SGCC approach**

In this study, the WHU-SGCC approach estimates the precipitation at every pixel by blending satellite estimates and rain gauge
observations considering the terrain factors and precipitation characteristics. Due to the significant seasonal difference of
precipitation, the WHU-SGCC method was applied in the different seasons. Four steps were used to establish the numerical
relationship between the gauge stations and the corresponding satellite pixels and for the interpolation of the remaining pixels.
The WHU-SGCC method identifies the geographical locations and topographical features of each pixel and applies the four
classification and blending rules. A flowchart of the WHU-SGCC method is shown in Fig. 3. The proposed approach was
evaluated over the Jinsha River Basin based on 30 gauge stations and CHIRP satellite-based precipitation estimations in the
different seasons from 1990 to 2014. The leave-one-out cross validation step was applied to compute the out-of-sample
adjusted bias with the gauge stations. The WHU-SGCC algorithm was repeated 30 times, each time leaving one station as the
validation station.
The basic description of the WHU-SGCC method is given below, and the details are illustrated separately in later sections:
(1) Classify all regional pixels into four types: C1 (pixels including one gauge station in their area), C2 (pixels statistically
similar to C1), C3 (pixels statistically similar to C2) and C4 (remaining pixels).
(2) Analyze the relationships between the precipitation observations and the C1, C2, and C3 pixel types, and interpolate for
the C4 pixels. These relationships are described by four rules, which are described below as Rules 1 through 4.
(3) Establish statistical models and screen the target pixels based on the four rules.
(4) Correct all of the precipitation pixels in the daily regional precipitation images.

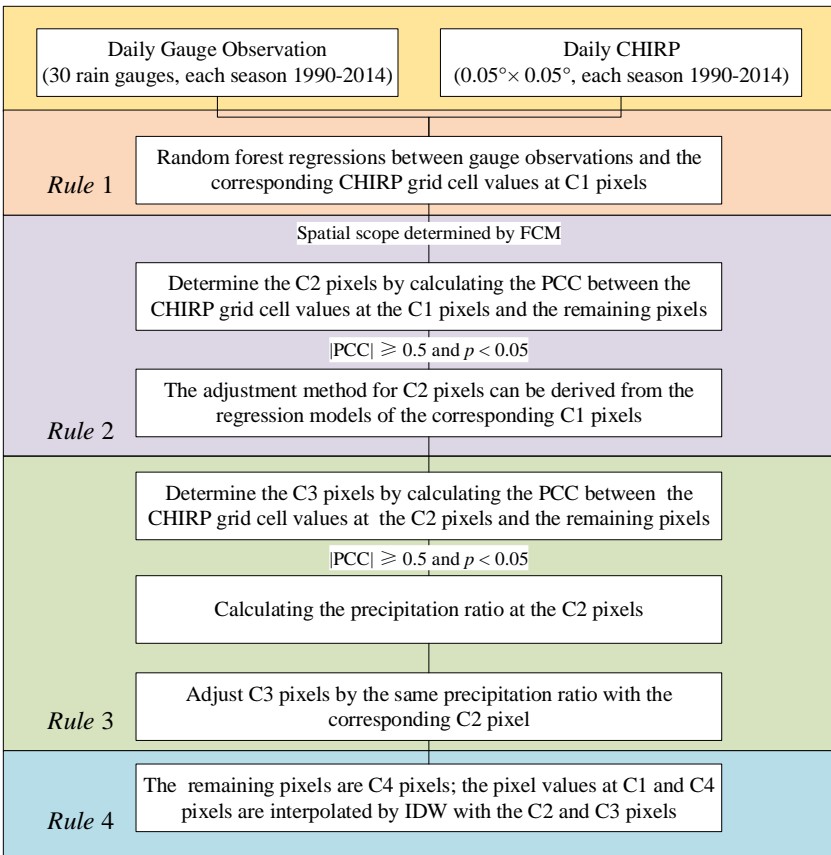


**Figure 3** Flowchart of the WHU-SGCC approach with the four rules applied in this study.
**3.1.1 Assumptions**
(1) Gauge observations are the most accurate, or "true", values for reference purposes. However, the sparseness of the
gauges, their uneven spatial distribution, and the high proportion of missing data may limit high accuracy estimation in rainfall
monitoring.
(2) No major terrain changes occurred during the twenty years (Appendix B).
(3) There are no abnormal values at one pixel in the CHIRP dataset during the long time series, so Pearson's Correlation
Coefficient (PCC) can represent the statistical similarity of the rainfall characteristics among the pixels in a certain spatial area
at a seasonal scale.
**3.1.2 Rule 1 of the WHU-SGCC method**
In general, the satellite precipitation estimations deviated from the ground-based measurements, which were assumed to be
the true values. Rule 1 aims to establish a regression model between the historical observations at each gauge and the
corresponding CHIRP grid cell values. The regression relationship was derived by random forest regression (RFR) at each
gauge station. RFR is a machine-learning algorithm for a predictive model with a large set of regression trees in which each
tree in the ensemble is grown from a bootstrap sample (Johnson, 1998) drawn with a replacement from the training set. In the
process of establishing regression trees, a subset of variables for each node is selected to avoid overfitting. The final prediction
is obtained by combining the results of the prediction methods applied to each bootstrap sample (Genuer et al., 2017). The
predicted value is calculated by the average of the values from all of the decision trees. Each tree can be expressed as
$$Tree_k = f_{RFR}(Y_o, \Theta_{Y_{s_k}}), \quad k=1...n \qquad (1)$$
where $Y_o$ denotes the historical observations at each gauge at the C1 pixels, $\Theta_{Y_{s_k}}$ is a randomly selected vector from $Y_s$,
$Y_s$ denotes the corresponding CHIRP grid cell values at the C1 pixels, $n$ is the number of trees, and $f_{RFR}$ is constructed

from the time series $Y_o$ (dependent variable) and $Y_s$ (independent variable) by means of RFR. The bootstrap sample will be the training set used for growing the tree. The error rate (out-of-bag, OOB) left out of one-third of the training data is also monitored to determine the number of decision trees. In this study, the minimum OOB error rate was reached when the number of decision trees $n$ was less than 500 (Appendix C).

Rule 1 builds the statistical relationships between the gauge observations and the corresponding CHIRP grid cell values, which is the key idea in correcting the satellite-based precipitation estimations in the entire study area. In the process of Rule 1, the regression relationships at the C1 pixels were established at 30 gauge stations separately. The values of the C1 pixels are not corrected in Rule 1 but are interpolated in Rule 4.

### 3.1.3 Rule 2 of the WHU-SGCC method

It is reasonable to assume that some pixels are statistically similar to the historical precipitation characteristics of the C1 pixels within a certain area. Therefore, it is feasible to adjust the satellite estimation bias of the C2 pixels by referring to the appropriate regression relationships at the corresponding C1 pixels based on Rule 1.

First, the spatial area in which pixels may have highly similar characteristics is established. Several studies indicate that the geographical location, elevation and other terrain information influence the spatial distribution of rainfall, especially in mountainous areas with complex topography (Anders et al., 2006; Long and Singh, 2013). The size of the spatial range is an important parameter to distinguish the spatial similarity and heterogeneity. In the WHU-SGCC method, the fuzzy c-means (FCM) clustering approach was used to determine the spatial range considered for each pixel's terrain factors, including longitude, latitude, elevation, slope, aspect and curvature. The FCM method was developed by J.C. Dunn in 1973 (Dunn, 1973), and improved in 1983 (Wang, 1983). It is an unsupervised fuzzy clustering method and its steps are as follows (Pessoa et al., 2018):

1) Choose the number of clusters $c$. The optimum number of clusters is determined by $L(c)$, which is derived from the inter-distance and inner distance of the samples in Eq. (2). It is ensured that the distance between similar samples is smaller, while the distance between different samples is larger.

$$L(c) = \frac{\sum_{i=1}^{c}\sum_{j=1}^{n} w_{ij}^m \parallel c_i - \bar{x} \parallel^2 /(c-1)}{\sum_{i=1}^{c}\sum_{j=1}^{n} w_{ij}^m \parallel x_j - c_i \parallel^2 /(n-c)} \tag{2}$$

In Eq. (2), the denominator is the inner-distance, and the numerator is the inter-distance. The initial value of $c$ is 1 and the maximum value of $c$ is the number of gauge stations in the study area. The optimum number of clusters was optimized to maximize $L(c)$. For this reason, the value of $c$ is varied from 1 to the number of gauge stations with an increment of 1 in this study.

2) Assign coefficients randomly to each data point $x_i$ for the degree to which it belongs in the $i$-th cluster $w_{ij}(x_i)$:

$$c_i^{(t)} = \frac{\sum_{j=1}^{n} w_{ij}^m x_i}{\sum_{j=1}^{n} w_{ij}^m} \tag{3}, \qquad w_{ij} = \frac{1}{\sum_{k=1}^{c}(\frac{\parallel x_i - c_i \parallel}{\parallel x_i - c_k \parallel})^{\frac{2}{m-1}}} \tag{4}, \qquad \bar{x} = \frac{\sum_{i=1}^{c}\sum_{j=1}^{n} w_{ij}^m x_j}{n} \tag{5}$$

where $x$ is a finite collection of $n$ elements that will be partitioned into a collection of $c$ fuzzy clusters, $c_i$ is the centre of each cluster, $m$ is the hyper-parameter that controls the level of cluster fuzziness, $w_{ij}$ is the degree to which element $x_i$ belongs to $c_i$, and $\bar{x}$ is the centre vector of the collection. In Eq. (3), $c_j^{(t)}$ represents the cluster centre in iteration $t$. If the

minimum improvement in the objective function between two consecutive iterations satisfies the following equation, the
algorithm terminates in iteration $t$ (Eq. (6)):

$$\| c_i^{(t)} - c_i^{(t+1)} \| < \varepsilon \tag{6}$$

3) Minimize the objective function $F_c$ to achieve data partitioning.

$$F_c = \sum_{j=1}^{n} \sum_{i=1}^{c} w_{ij}^{m} \| x_j - c_i \|^2 \tag{7}$$

The results of the FCM are the degree of membership of each pixel to the cluster centre as represented by numerical values.
The pixels in each cluster have similar terrain features and precipitation characteristics.
Second, as mentioned above, the aim of Rule 2 is to derive an adjustment method for the C2 pixels based on learning from
Rule 1. With the establishment of a regression relationship between the gauge observations and the corresponding CHIRP grid
cell values of the C1 pixels by the RFR method, the determination of the C2 pixels follows a complicated procedure. With the
exception of the C1 pixels, the remaining pixels in each cluster represent potential C2 pixels, which are called R pixels. The
Pearson's correlation coefficient (PCC) and $p$-values between the satellite estimations (multi-annual daily CHIRP grid cell
values) at the R pixels and the C1 pixels are the criteria for the final determination of the C2 pixels. The PCC is defined as
follows:

$$PCC_{x,y} = \frac{\sum_{i=1}^{n}(x_i - \overline{x})(y_i - \overline{y})}{\sqrt{\sum_{i=1}^{n}(x_i - \overline{x})^2} \sqrt{\sum_{i=1}^{n}(y_i - \overline{y})^2}} \tag{8}$$

where $n$ is the number of samples, $x_i$ and $y_i$ are individual samples (CHIRP grid cell values at the C1 and C2 pixels,
respectively), $\overline{x}$ is the arithmetic mean of $x$ calculated by $\overline{x} = \frac{1}{n}\sum_{i=1}^{n} x_i$, and $\overline{y}$ is the arithmetic mean of $y$ calculated by
$\overline{y} = \frac{1}{n}\sum_{i=1}^{n} y_i$.
The PCC ranges between -1 and +1. If there are no repeated data values, a perfect PCC of +1 or −1 occurs when each of the
variables is a perfect monotonic function of the other. However, if the value is close to zero, there is zero correlation. In
addition, the correlation is not only determined by the value of the correlation coefficient but also by the correlation test's $p$-
value. The critical values for the PCC and $p$-value are 0.5 and 0.05, respectively; thus, a PCC value higher than 0.5 and a $p$-
value lower than 0.05 indicate that the data are significantly correlated (Zhang and Chen, 2016). Therefore, the final
determination of the C2 pixels must meet the following criteria:

$$|PCC| \geq 0.5 \quad and \quad p < 0.05 \tag{9}$$

Each R pixel has $m$ PCC and $p$-values (the number of C1 pixels in the cluster), and the subset of C2 pixels is identified by
excluding the data that failed the correlation test and retaining both the data with a maximum PCC of at least 0.5 and a $p$-value
lower than 0.05, and the corresponding index of C1 pixels. The selected C2 pixels can then be considered statistically similar
to the precipitation characteristics of the corresponding C1 pixels in their defined spatial area.
After identifying the C2 pixels and their corresponding C1 pixels, the adjustment method for the C2 pixels is derived from
the regression model for the C1 pixels:

$$C2_{as} = \frac{1}{n}\sum_{k=1}^{n} Tree_{k_{C1}}(Y_{s_{C2}}) \tag{10}$$

where $Tree_{k_{C1}}$ is the decision tree derived from the RFR algorithm at the corresponding C1 pixel, $Y_{s_{C2}}$ is the CHIRP grid cell

value at the C2 pixels, and $C2_{as}$ is the adjusted satellite precipitation estimate calculated by the average of the values from the RFR decision trees.

### 3.1.4 Rule 3 of the WHU-SGCC method

Recognizing that precipitation has a spatial distribution, the assumption that the C3 pixels are statistically similar to the precipitation characteristics of the C2 pixels is adopted to establish the adjustment method for the C3 pixels.

First, the determination of the C3 pixels in each spatial cluster is based on the selection of C2 pixels. The satellite-based estimation values at the pixels other than the C1 and C2 pixels are used to calculate the PCC and $p$-value with the satellite-based estimation values at the C2 pixels in the same cluster. The results of each pixel's $k$ PCC and $p$-value (the number of C2 pixels in the cluster) are evaluated based on the correlation test (Eq. (9)) that the pixels have a maximum PCC of at least 0.5 and a $p$-value is of no more than 0.05, and the corresponding index of C2 pixels is retained. The selected pixels are called C3 pixels, which are statistically similar to the precipitation characteristics of the corresponding C2 pixels in the defined spatial area.

After identifying the C3 pixels, a method for merging the CHIRP grid cell values at the C3 pixels ($Y_s$) and the target reference values of $C2_{as}$ at the corresponding C2 pixels is applied to estimate the adjusted precipitation values at the C3 pixels. This method combines the $Y_s$ and $C2_{as}$ values into one variable, as shown in Eq. (11):

$$w_i = \frac{C2_{as_i} + \lambda}{Y_{s_i} + \lambda} \quad i=1,\ldots, \ n \tag{11}$$

where $\lambda$ is a positive constant set to 10 mm (Sokol, 2003), $C2_{as}$ is the adjusted precipitation values at the C2 pixels, $Y_{s_i}$ is extracted from the CHIRP grid cell values at the corresponding location of the C2 pixels, and $n$ is the number of C2 pixels in each spatial cluster.

Each $w$ of the C3 pixels is assigned the same value as the corresponding C2 pixel. Therefore, the values of the C3 pixels are derived from Eq. (12):

$$C3_{as} = \max(w \times (Y_s + \lambda) - \lambda, 0) \tag{12}$$

where $C3_{as}$ is the adjusted target precipitation value at one C3 pixel, and $Y_s$ is the corresponding CHIRP grid cell value. To avoid precipitation estimates below 0, Eq. (12) sets negative values to 0.

### 3.1.5 Rule 4 of the WHU-SGCC method

The pixels other than the C1, C2 and C3 pixels are called C4 pixels and they are adjusted by inverse distance weighting (IDW). IDW is based on the concept of the first law of geography from 1970, which was defined as *everything is related to everything else, but near things are more related than distant things*. Therefore, the attribute value of an unsampled point is the weighted average of the known values within the neighbourhood, and the distance weighting can be determined by means of IDW (Lu and Wong, 2008). In Rule 4, IDW is used to interpolate the unknown spatial precipitation data from the adjusted precipitation values at the C2 and C3 pixels. The IDW formulas are given as Eq. (13) and Eq. (14).

$$R_{as} = \sum_{i=1}^{n} w_i R_i \tag{13}$$

$$w_i = \frac{d_i^{-\alpha}}{\sum_{i=1}^{n} d_i^{-\alpha}} \quad \text{with} \quad \sum_{i=1}^{n} w_i = 1 \tag{14}$$

where $R_{as}$ is the unknown spatial precipitation data, $R_i$ is the adjusted precipitation values at the C2 and C3 pixels, $n$ is the number of C2 and C3 pixels, $d_i$ is the distance from each C2 or C3 pixel to the unknown grid cell, and $\alpha$ is the power which is generally specified as a geometric form for the weight. Several studies (Simanton and Osborn 1980; Tung 1983) have experimented with variations in the power; a the small $\alpha$ tends to estimate values with the averages of sampled grids in the neighbourhood, while a large $\alpha$ tends to give larger weights to the nearest points and increasingly down-weights points farther away (Chen and Liu, 2012; Lu and Wong, 2008). The value of $\alpha$ has an influence on the spatial distribution of the information from precipitation observations. For this reason, $\alpha$ is varied in the range of 0.1 to three (0.1, 0.3, 0.5, 1.0, 1.5, 2.0, 2.5 and 3.0) in this study.

Note that the unknown spatial precipitation data include C1 and C4 pixels because the C1 pixels values were not adjusted in Rule 1.

After applying these four rules, we obtained complete daily adjusted regional precipitation maps for the four seasons over the Jinsha River basin.

**3.2 Accuracy assessment**

The performance of the WHU-SGCC adjusted precipitation estimates was evaluated by eight mathematic metrics: the Pearson's correlation coefficient (PCC), root mean square error (RMSE), mean absolute error (MAE), relative bias (BIAS), Nash-Sutcliffe efficiency coefficient (NSE), probability of detection (POD), false alarm ratio (FAR) and critical success index (CSI). The results of accuracy assessment are the average values validated by the leave-one-out cross method. Each validated pixel will probably be a C2, C3 or C4 pixel in the process of the WHU-SGCC algorithm. The PCC, RMSE, MAE and BIAS were used to evaluate how well the WHU-SGCC method adjusted the satellite estimation bias, while POD, FAR and CSI were used to evaluate the performance of precipitation forecasting (Su et al., 2011). The PCC measures the strength of the correlation relationship between the satellite estimations and observations. The RMSE is an absolute measurement used to compare the difference between the satellite estimations and observations, and the MAE represents the average magnitude of error estimations considering both systematic and random errors. The NSE (Nash and Sutcliffe, 1970) determines the relative magnitude of the variance of the residuals compared to the variance of the observations, bounded by minus infinity and 1; a negative value indicates a poor precipitation estimate, and a value of 1 indicates an optimal estimate. The BIAS measures the mean tendency of the estimated precipitation to be larger (positive values) or smaller (negative values) than the observed precipitation and has an optimal value of 0. The POD, also known as the hit rate, represents the probability of rainfall detection, and the FAR is defined as the ratio of the false alarm of rainfall to the total number of rainfall events. All of the accuracy assessment metrics are shown in Table 2.

**Table 2** Accuracy assessment metrics.

| Accuracy assessment Index | Unit | Formula | Range | Optimal value |
|---|---|---|---|---|
| Pearson's Correlation Coefficient (PCC) | NA | $PCC = \dfrac{\sum_{i=1}^{n}(Y_{oi} - \bar{Y}_o)(C_i - \bar{C})}{\sqrt{\sum_{i=1}^{n}(Y_{oi} - \bar{Y}_o)^2} \cdot \sqrt{\sum_{i=1}^{n}(C_i - \bar{C})^2}}$ | [-1,1] | 1 |
| Root Mean Square Error (RMSE) | mm | $RMSE = \sqrt{\dfrac{1}{n}\sum_{i=1}^{n}(C_i - Y_{oi})^2}$ | [0,+∞) | 0 |
| Mean Absolute Error (MAE) | mm | $MAE = \dfrac{1}{n}\sum_{i=1}^{n}|C_i - Y_{oi}|$ | [0, +∞) | 0 |
| Relative Bias (BIAS) | NA | $BIAS = \dfrac{\sum_{i=1}^{n}(C_i - Y_{oi})}{\sum_{i=1}^{n}Y_{oi}}$ | (-∞, +∞) | 0 |
| Nash-Sutcliffe Efficiency Coefficient (NSE) | NA | $NSE = 1 - \dfrac{\sum_{i=1}^{N}(C_i - Y_{oi})^2}{\sum_{i=1}^{N}(C_i - \bar{Y}_o)^2}$ | (-∞,1] | 1 |
| Probability of Detection (POD) | NA | $POD = H/(H+M)$ | [0,1] | 1 |

| | NA | FAR=F/(H+F) | [0,1] | 0 |
| False Alarm Ratio (FAR) | | | | |
| Critical Success Index (CSI) | NA | CSI=H/(H+M+F) | [0,1] | 1 |

Note: $Y_{oi}$ is the observation data; $C_i$ is the adjusted value using the WHU-SGCC method for the test sample pixel; $\bar{Y}_o$ is the arithmetic mean of $Y_o$ and is given by $\bar{Y}_o = \frac{1}{n}\sum_{i=1}^{n} Y_{oi}$; $\bar{C}$ is the arithmetic mean of $C$ and is given by $\bar{C} = \frac{1}{n}\sum_{i=1}^{n} C_i$; H represents the number of both observed and estimated precipitation events (successfully forecasted); F is the number of false alarms when the observed precipitation was below the threshold and estimated precipitation was above threshold (false alarms; and. M is the number of events in which the estimated precipitation was below the threshold and observed precipitation was above the threshold (missed forecasts). The POD and FAR values are dimensionless numbers ranging from 0 to 1. The precipitation threshold (event/no event) was set to 0.1 mm/day.

## 4 Results and Discussion

A total of 18,482 daily pixels were adjusted by blending the satellite estimations (CHIRP) and observations (rain gauge stations) using the WHU-SGCC approach over the Jinsha River Basin from 1990 to 2014. The percentage of pixels adjusted by each rule in the WHU-SGCC method is shown in Table 3. The number of C1 pixels was the number of training gauge stations, which accounted for 0.16% of the total pixels (18,482) within the basin. Due to the leave-one-out cross validation step, the different training samples will have different numbers of C2, C3 and C4 pixels within the Jinsha River Basin. The percentage of C2 and C3 pixels are highest in fall, followed by summer, spring and winter. In the spring, the average percentage of C2 pixels was approximately 21.27%, the average percentage of C3 pixels was approximately 17.12%, and the percentage of C4 pixels was approximately 61.46%. In the summer, the percentage of C2 pixels was approximately 17.86%, the percentage of C3 pixels was approximately 23.43%, and the percentage of C4 pixels was approximately 58.55%. In the full, the average percentage of C2 pixels was approximately 31.40%, the average percentage of C3 pixels was approximately 21.77%, and the average percentage of C4 pixels was approximately 46.68%. In the winter, the average percentage of C2 pixels was approximately 15.60%, the average percentage of C3 pixels was approximately 19.23%, and the average percentage of C4 pixels was approximately 65.01%. Besides, the pixel type of the validation gauge station is shown in Table D1 and the spatial distribution of C1-C3 pixels in Figure D1 with the most uniform in the fall and, while the sparsest in the winter. Each validation gauge station could be identified as either C2, C3 or C4 pixels to evaluate the performances of all the rules in the WHU-SGCC method.

**Table 3** The percentage of each class pixels adjusted by each rule using the WHU-SGCC method within the Jinsha River Basin.

| Validation gauge station | C2 Pixels (%) | | | | C3 Pixels (%) | | | | C4 Pixels (%) | | | |
|---|---|---|---|---|---|---|---|---|---|---|---|---|
| | Spring | Summer | Fall | Winter | Spring | Summer | Fall | Winter | Spring | Summer | Fall | Winter |
| 52908 | 20.80% | 16.59% | 29.15% | 15.52% | 17.76% | 22.85% | 20.82% | 18.16% | 61.29% | 60.40% | 49.87% | 66.16% |
| 56004 | 20.89% | 15.59% | 29.40% | 15.65% | 16.29% | 22.24% | 20.64% | 18.83% | 62.66% | 62.01% | 49.81% | 65.36% |
| 56021 | 21.38% | 17.91% | 32.46% | 15.65% | 17.55% | 24.40% | 21.85% | 19.91% | 60.91% | 57.53% | 45.53% | 64.28% |
| 56029 | 21.77% | 18.06% | 32.60% | 16.03% | 17.31% | 24.06% | 21.61% | 19.64% | 60.76% | 57.72% | 45.63% | 64.18% |
| 56034 | 21.09% | 17.86% | 31.22% | 14.86% | 17.78% | 23.95% | 23.07% | 20.19% | 60.97% | 58.03% | 45.55% | 64.79% |
| 56038 | 20.48% | 17.36% | 30.72% | 15.56% | 16.12% | 21.72% | 23.74% | 17.63% | 63.23% | 60.76% | 45.39% | 66.65% |
| 56144 | 21.42% | 18.11% | 31.97% | 16.00% | 16.46% | 24.03% | 21.78% | 19.38% | 61.96% | 57.70% | 46.09% | 64.46% |
| 56146 | 21.33% | 17.22% | 31.77% | 15.70% | 17.12% | 24.24% | 21.42% | 18.34% | 61.39% | 58.38% | 46.65% | 65.81% |
| 56152 | 21.32% | 17.17% | 31.27% | 15.57% | 17.56% | 22.59% | 22.32% | 18.94% | 60.96% | 60.08% | 46.26% | 65.34% |
| 56167 | 21.46% | 18.19% | 32.36% | 15.84% | 16.90% | 23.51% | 21.72% | 19.03% | 61.48% | 58.14% | 45.76% | 64.98% |
| 56247 | 21.66% | 18.32% | 31.44% | 16.10% | 17.16% | 23.89% | 22.19% | 19.55% | 61.03% | 57.63% | 46.21% | 64.20% |
| 56251 | 21.09% | 17.86% | 31.28% | 15.73% | 17.39% | 23.53% | 22.88% | 18.50% | 61.36% | 58.46% | 45.68% | 65.62% |

| 56257 | 21.17% | 17.93% | 30.99% | 15.95% | 16.15% | 21.88% | 23.55% | 19.13% | 62.53% | 60.04% | 45.30% | 64.77% |
| 56357 | 21.62% | 18.14% | 31.59% | 15.64% | 17.12% | 23.75% | 22.54% | 19.52% | 61.10% | 57.95% | 45.71% | 64.68% |
| 56374 | 21.52% | 18.08% | 31.92% | 14.32% | 17.38% | 23.23% | 21.90% | 19.20% | 60.95% | 58.53% | 46.02% | 66.32% |
| 56459 | 21.30% | 18.10% | 32.14% | 15.64% | 16.92% | 23.45% | 21.16% | 19.17% | 61.62% | 58.29% | 46.54% | 65.03% |
| 56462 | 21.67% | 18.29% | 32.68% | 15.92% | 17.28% | 23.68% | 21.55% | 19.14% | 60.90% | 57.87% | 45.61% | 64.78% |
| 56475 | 21.49% | 18.10% | 32.49% | 15.98% | 16.36% | 23.50% | 22.08% | 19.53% | 62.00% | 58.24% | 45.28% | 64.33% |
| 56479 | 20.42% | 17.88% | 31.34% | 15.69% | 16.35% | 22.79% | 19.36% | 18.74% | 63.07% | 59.17% | 49.14% | 65.41% |
| 56485 | 21.44% | 18.36% | 32.78% | 15.64% | 17.43% | 23.91% | 21.82% | 19.85% | 60.97% | 57.57% | 45.24% | 64.35% |
| 56543 | 21.52% | 18.25% | 32.51% | 15.87% | 16.97% | 23.72% | 21.78% | 18.90% | 61.35% | 57.87% | 45.56% | 65.06% |
| 56565 | 21.21% | 17.54% | 30.93% | 15.52% | 17.81% | 24.08% | 23.55% | 19.96% | 60.83% | 58.23% | 45.36% | 64.37% |
| 56571 | 21.62% | 17.89% | 31.31% | 14.94% | 17.03% | 23.07% | 20.83% | 18.94% | 61.19% | 58.89% | 47.70% | 65.97% |
| 56586 | 21.73% | 18.33% | 21.73% | 15.49% | 17.35% | 23.99% | 17.35% | 19.59% | 60.76% | 57.53% | 60.76% | 64.77% |
| 56651 | 20.90% | 18.07% | 32.46% | 15.38% | 17.78% | 23.98% | 22.13% | 19.95% | 61.16% | 57.79% | 45.25% | 64.51% |
| 56664 | 20.94% | 18.22% | 32.43% | 15.50% | 16.64% | 23.06% | 21.00% | 18.70% | 62.26% | 58.56% | 46.42% | 65.64% |
| 56666 | 21.06% | 17.98% | 31.59% | 15.39% | 18.03% | 23.97% | 22.41% | 19.64% | 60.76% | 57.89% | 45.84% | 64.82% |
| 56671 | 20.71% | 18.16% | 32.55% | 15.67% | 16.53% | 23.63% | 21.89% | 20.03% | 62.61% | 58.06% | 45.41% | 64.14% |
| 56684 | 21.36% | 18.04% | 32.65% | 15.46% | 17.72% | 23.15% | 21.95% | 19.28% | 60.76% | 58.65% | 45.24% | 65.10% |
| 56778 | 21.63% | 18.11% | 32.25% | 15.91% | 17.31% | 23.14% | 22.11% | 19.52% | 60.90% | 58.59% | 45.48% | 64.41% |

**4.1 Model performance based on overall accuracy evaluations**

The multi-annual (1990-2014) average seasonal precipitation over the Jinsha River Basin interpolated from WHU-SGCC, CHIRP and CHIRPS is shown in Fig. 4. There exist some differences in the spatial pattern of precipitation estimates. Overall, the WHU-SGCC method exhibits the similar spatial distribution of precipitation to the CHIRP and CHIRPS, while the WHU-SGCC method attenuated the intense rain in the central area. The statistical accuracy evaluations are needed to further analyze the performance of the WHU-SGCC method.

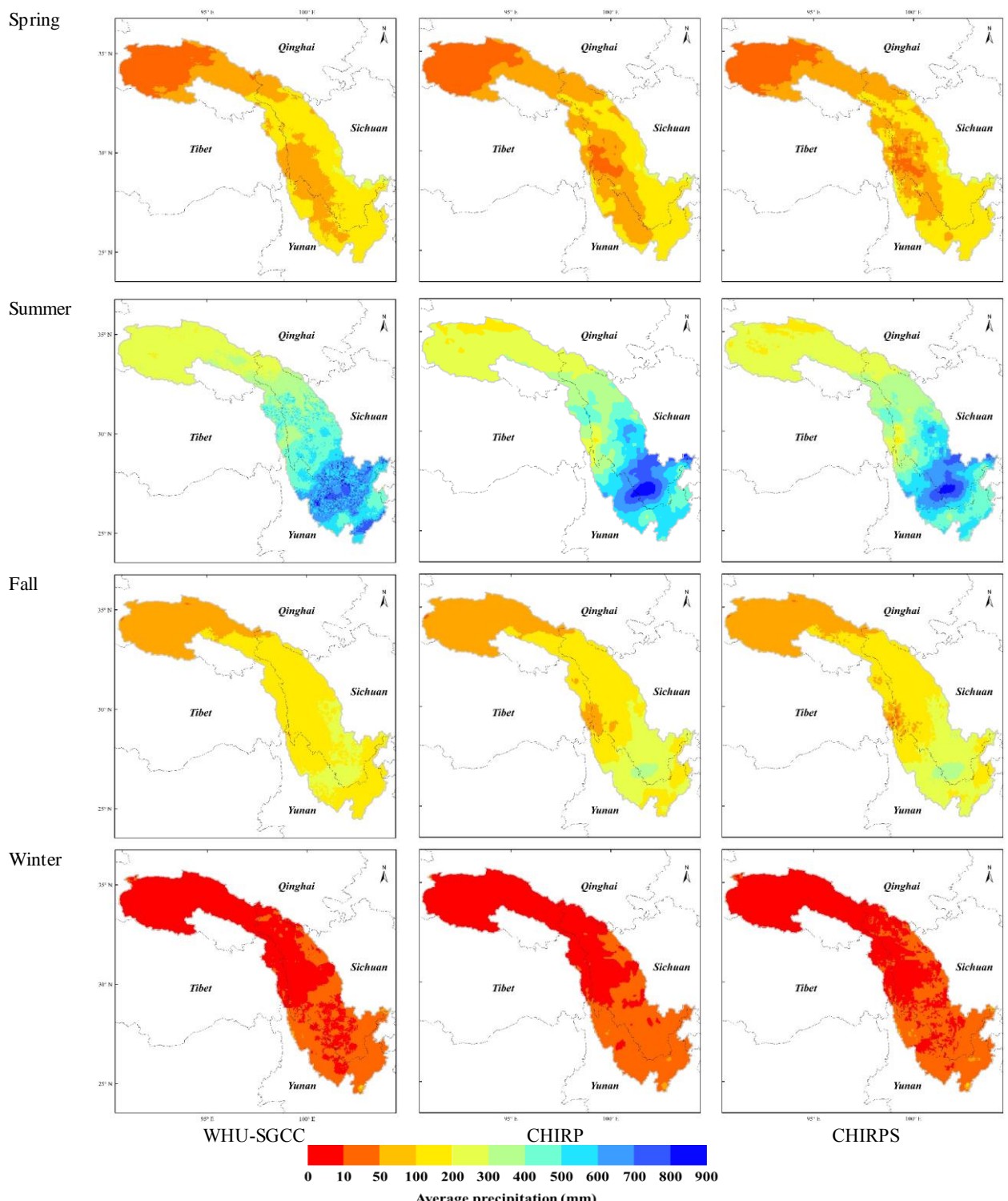

**Figure 4** The multi-annual (1990-2014) average seasonal precipitation over the Jinsha River Basin interpolated from WHU-SGCC, CHIRP and CHIRPS.

To test the performance of the WHU-SGCC method for precipitation estimates, the PCC, RMSE, BAE, BIAS, NSE, POD,

FAR, and CSI were calculated and are presented in Table 4 (the results were derived from the 22 clusters for the FCM in Rule 2, as shown in Appendix E, and $\alpha$ =0.1 for the IDW in Rule 4 after the comparison of the RMSE). After the correction, the PCC in the WHU-SGCC method shows an improvement relative to the CHIRP and CHIRPS estimates. The spring and fall have better correlations than the summer and winter. In addition, the NSE of the WHU-SGCC provides substantial improvements over CHIRP and CHIRPS, especially in the spring and fall which were better than the summer and winter. The RMSE and MAE are the largest in the summer, followed by the fall, spring, and winter; however, the performances of the BIAS in the summer and fall are better than those in the spring and winter, which might be influenced by the greater precipitation in the summer and fall than in the spring and winter. The assessments of the POD and CSI are lowest, and the FAR is largest in the winter due to the overestimation of no rain events estimated by the satellite-based data set.

Compared with the estimates of CHIRP and CHIRPS, the PCCs of the WHU-SGCC method are improved to more than 0.5 in the spring and fall and to approximately to 0.5 in the winter, with overall average improvements of the Pearson's correlation coefficient (PCC) by 0.0082-0.2232 and 0.0612-0.3243, respectively. In addition, the RMSE and MAE of the WHU-SGCC were all lower than those of CHIRP and CHIRPS, with overall average decreases in the root mean square error (RMSE) by 0.0922-0.65 mm and 0.2249-2.9525 mm, respectively. The absolute values of the BIAS of the WHU-SGCC are substantial improved in the spring, followed by the summer, winter and fall. Although the absolute value of the BIAS of the WHU-SGCC in fall are not significantly better than those of CHIRP and CHIRPS, all of the values are approximately 0. The NSEs of the WHU-SGCC reached 0.2836, 0.2944 and 0.1853 in the spring, fall and winter, respectively, which are substantially better than the negative or zero values of CHIRP and CHIRPS. In the summer, the NSE of the WHU-SGCC is still negative, but it is improved to be nearly zero, which indicates that the adjusted results are similar to the average level of the rain gauge observations. It is worth noting that in the spring, summer and fall, the POD values of the WHU-SGCC are in the range of 0.95 to 1, better than CHIRP and CHIRPS, and the FAR values of the WHU-SGCC are no more than 0.3, lower than CHIRP and CHIRPS; these results represent the better ability of the WHU-SGCC method to predict precipitation events. The rainfall detection ability is the worst in the winter compared to the other seasons. This can be explained by the seasonal distribution of precipitation in the Jinsha River Basin, in which the most rainfall occurs in the summer, followed by the fall, spring and winter. In addition, the spatial distribution of C2 and C3 pixels might slightly impact the overall accuracy in different seasons that the sparsest in the winter, while more uniform in the summer. However, the performances of PCC, RMSE, MAE and NSE in the winter are better than those in the summer. The worst errors of forecasting performance in the summer may be attributed to the highest precipitation. The limited precipitation events detection in the winter could also be explained by the lowest precipitation (Xu et al., 2019).

**Table 4** Overall accuracy assessments for the four seasons from 1990 to 2014.

| Statistic | Spring | | | Summer | | | Fall | | | Winter | | |
| --- | --- | --- | --- | --- | --- | --- | --- | --- | --- | --- | --- | --- |
| | WHU-SGCC | CHIRP | CHIRPS | WHU-SGCC | CHIRP | CHIRPS | WHU-SGCC | CHIRP | CHIRPS | WHU-SGCC | CHIRP | CHIRPS |
| PCC | 0.5376 | 0.3644 | 0.2132 | 0.2536 | 0.2454 | 0.1924 | 0.5508 | 0.3889 | 0.2661 | 0.4722 | 0.2490 | 0.1716 |
| RMSE | 2.9526 | 3.4332 | 5.1926 | 8.7608 | 9.4108 | 11.3354 | 4.7981 | 4.9038 | 7.7506 | 0.8120 | 0.9042 | 1.0569 |
| MAE | 1.3380 | 1.5426 | 1.9948 | 5.4564 | 5.8415 | 7.0088 | 2.0973 | 2.2943 | 2.9925 | 0.2093 | 0.7398 | 0.6905 |
| BIAS | -0.1148 | -0.2490 | -0.1783 | -0.0167 | -0.0443 | -0.0134 | -0.0566 | -0.0563 | -0.0231 | -0.1775 | -0.2083 | -0.3093 |
| NSE | 0.2836 | 0.0745 | -1.0817 | -0.0139 | -0.2083 | -0.8293 | 0.2944 | 0.0168 | -1.4692 | 0.1853 | 0.0161 | -0.3098 |
| POD | 0.9605 | 0.8572 | 0.2918 | 0.9932 | 0.9578 | 0.4351 | 0.9612 | 0.9047 | 0.2326 | 0.6988 | 0.5786 | 0.2076 |
| FAR | 0.2416 | 0.4515 | 0.3888 | 0.1146 | 0.2323 | 0.1601 | 0.2386 | 0.4301 | 0.2638 | 0.5242 | 0.7082 | 0.6381 |
| CSI | 0.6928 | 0.5001 | 0.2335 | 0.8799 | 0.7405 | 0.401 | 0.7089 | 0.5303 | 0.2144 | 0.3668 | 0.2210 | 0.1352 |

The spatial distributions of the statistical comparisons between the observations and the WHU-SGCC precipitation estimations are shown in Fig. 5 and Fig. 6. Overall, the variation in the PCC shows low correlations in areas with lower elevation, particularly in the southeast Jinsha River Basin, where there is higher precipitation and a greater density of rain

gauges. The PCC is highest in the fall, followed by the spring and winter, and finally by summer. The higher correlations are located in the north-central area along the Tongtian River, Jinsha River and upstream part of the Yalong River, which has complex terrain and few rain gauges. The RMSE is lowest in the winter than in the spring, fall and summer, which can be attributed to the lower precipitation in the winter and the greatest in the summer. The spatial distribution of the RMSE shows that, the smaller errors are scattered in the northwest area of the river basin, with values lower than 5 mm, while the highest errors are located along the border between the lower reaches of the Jinsha Jiang River and the river basin. This is related to the climate regimes of the Jinsha River Basin, which includes more rainfall in the south and southeast areas than in the north, and northwest.

The results show that the WHU-SGCC method improves the correlation relative to CHIRP and CHIRPS, especially in central and southeast river basin during the spring, fall and winter, with most of the PCC values falling between 0.4 and 0.8 (Fig. 5). As shown by the RMSE (Fig. 6), the WHU-SGCC can also correct the precipitation bias in the central and southeast river basin, especially along the downstream part of the Yalong River. In addition, the WHU-SGCC slightly improved the RMSE around the convergence of the rivers, where it is less than 5 mm in the spring and fall, and most of RMSE values are less than 1 mm in the winter. In spite of the correction, the RMSE values in the summer are still substantial.

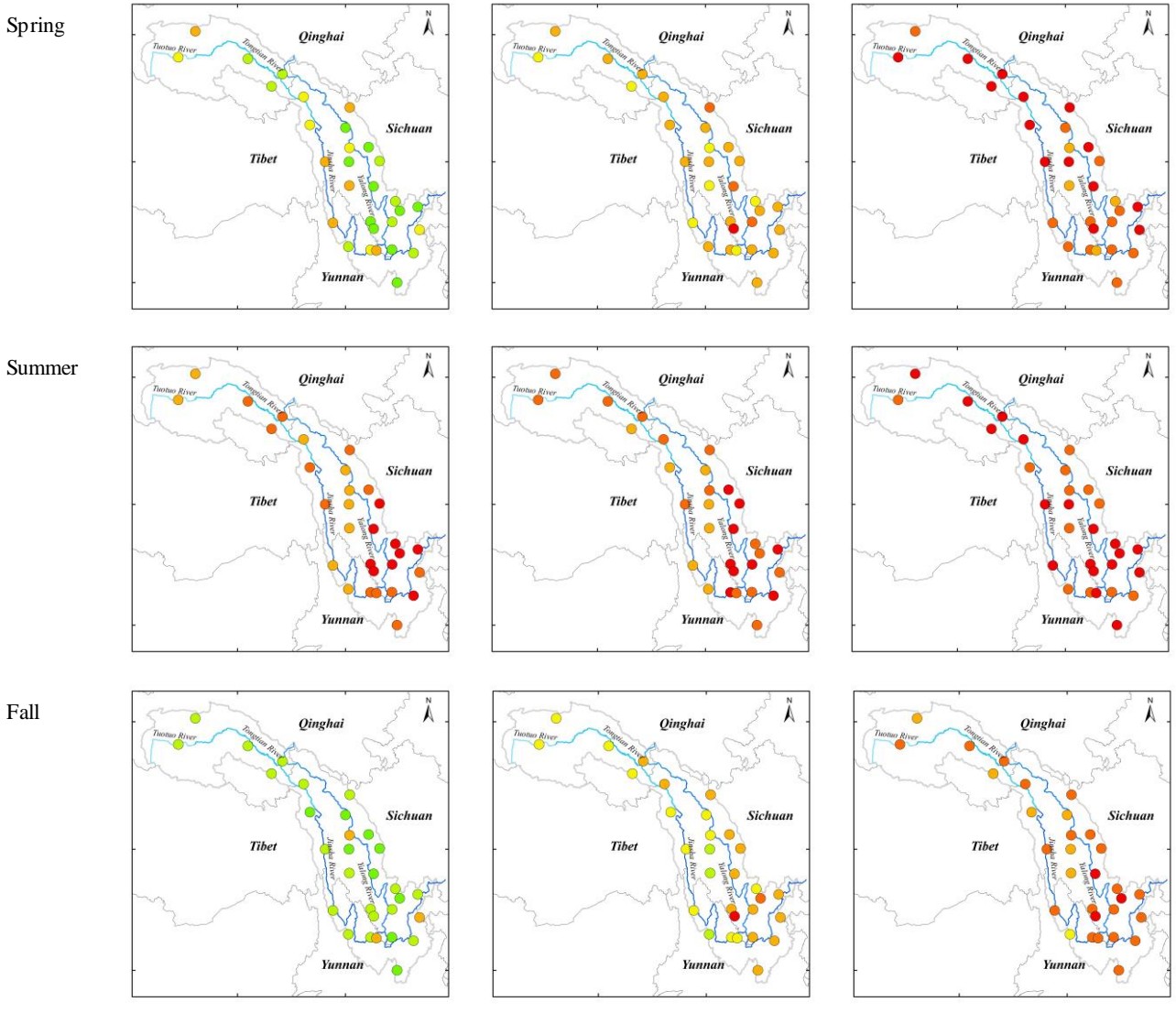

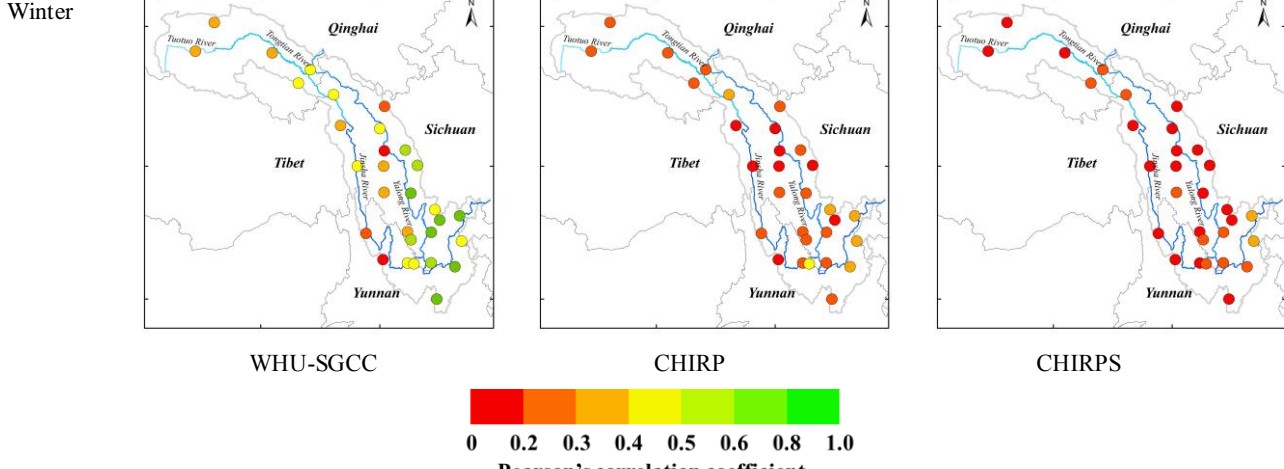

Winter

WHU-SGCC   CHIRP   CHIRPS

Pearson's correlation coefficient

**Figure 5** Spatial distribution of the Pearson's correlation coefficient of the overall agreement between observations and the WHU-SGCC,
CHIRP and CHIRPS estimations in the four seasons from 1990 to 2014.
All of the spatial distribution statistics indicate that the statistical relationships established during the process of the WHU-
SGCC method are susceptible to the mode values of the rain gauge stations data, especially in the summer. Although the
average summer precipitation in the southern Jinsha River Basin was more than 600 mm (Fig. 2), days of light rain still
represent a large percentage, which causes large biases and limits the performance over the south, while there are sufficient
data with similar precipitation features for the WHU-SGCC in the north. Nevertheless, the WHU-SGCC approach is still
effective at adjusting the satellite biases by blending the data with the observations, particularly in the complicated
mountainous regions, where higher PCCs correspond to lower RMSEs.

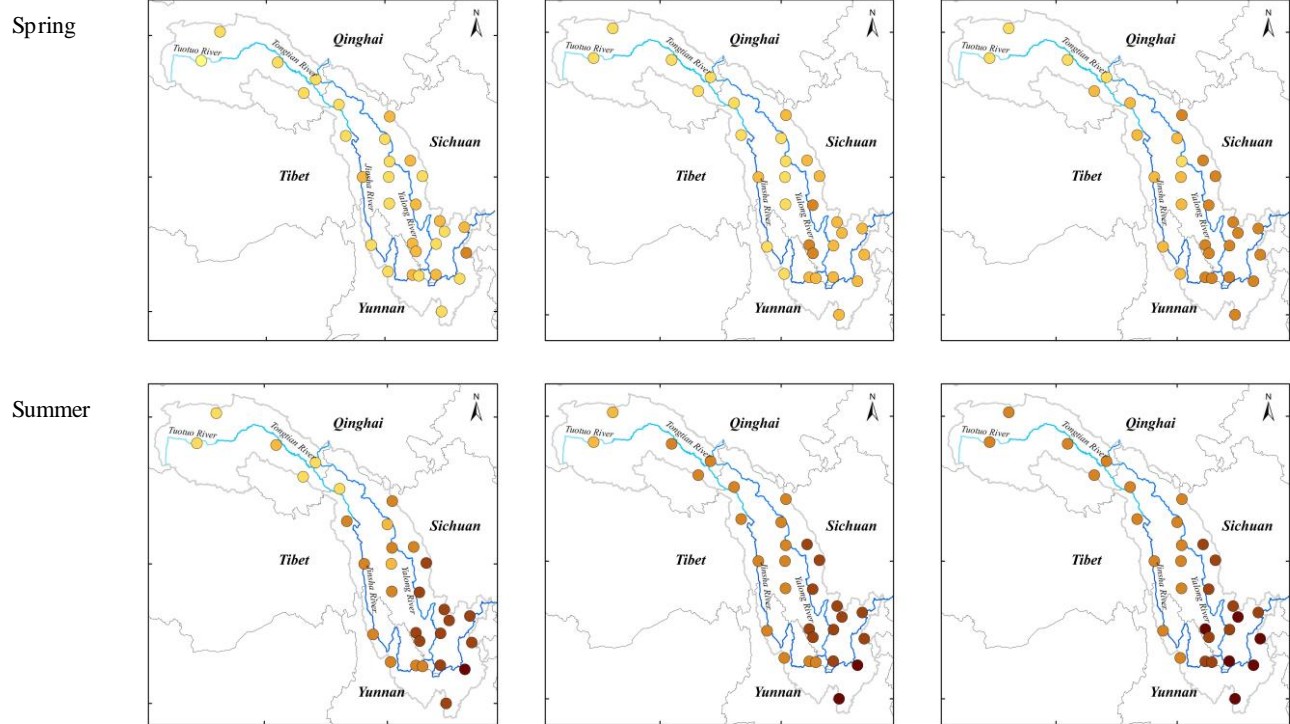

Spring

Summer

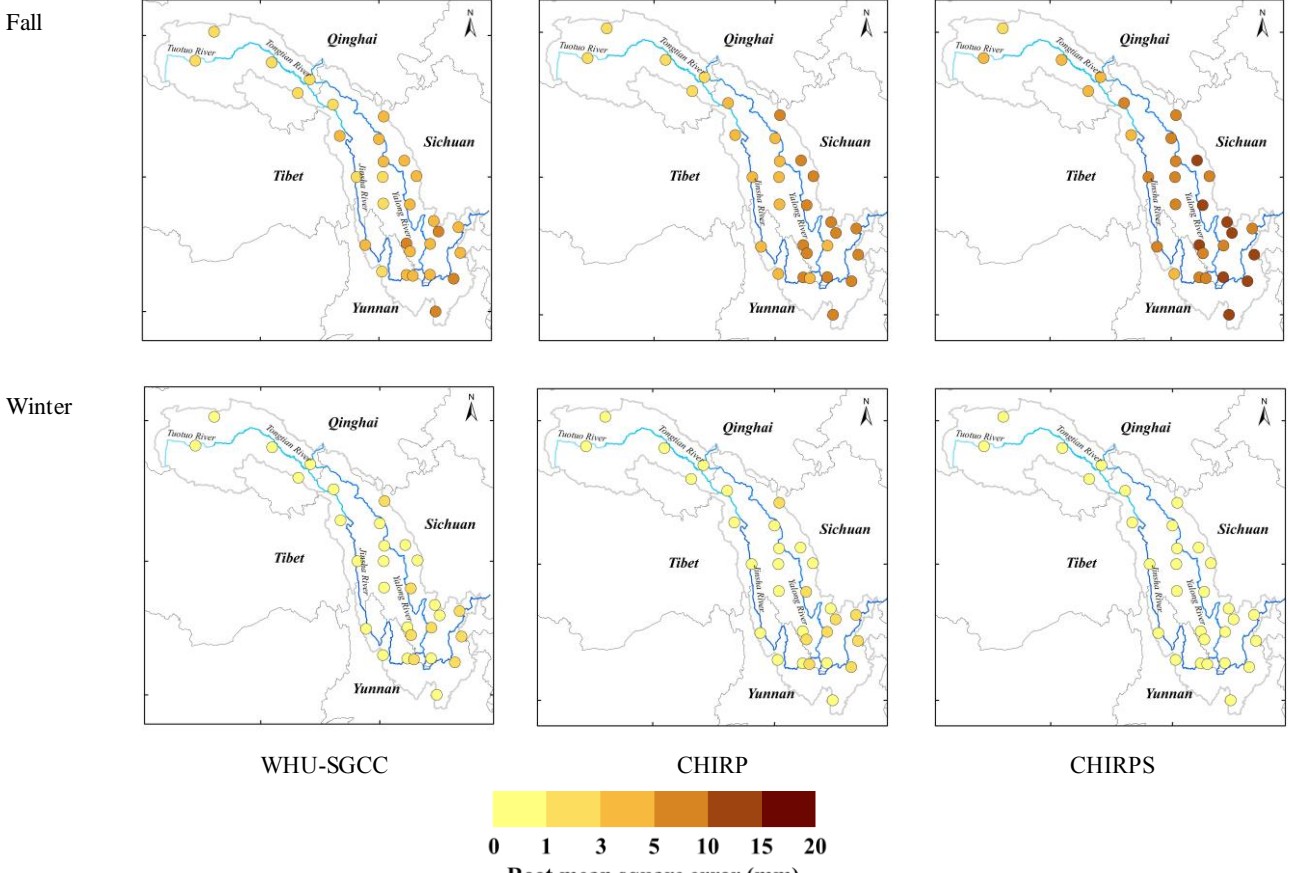

Fall

Winter

WHU-SGCC      CHIRP      CHIRPS

0  1  3  5  10  15  20
**Root mean square error (mm)**


**Figure 6** Spatial distribution of the root mean square errors of the overall agreement between observations and the WHU-SGCC, CHIRP
and CHIRPS estimations in the four seasons from 1990 to 2014.
**4.2 Model performance based on daily accuracy evaluations**
After the overall accuracy evaluations were conducted, further evaluations of the daily accuracy in the four seasons were
conducted, and the results are shown in Fig. 7. The evaluation of the daily accuracy indicates that the PCCs of the WHU-
SGCC were slightly better than those of CHIRP and CHIRPS in the spring, fall and winter but were not as good in the summer
and winter. The WHU-SGCC had lower RMSEs and MAEs than CHIRP and CHIRPS, especially compared to CHIRPS. The
daily RMSE and MAE in the summer are the highest, although the WHU-SGCC still corrects the bias. Figure 7 indicates that
there is a slight increase in the PCC, with average improvements of 0.0249-0.0405 and 0.0456-0.1355, respectively; however,
the PCC is a relative metric of the magnitude of the association between paired variables, and a relative consistency may not
indicate absolute proximity. Thus, the absolute measure indicated by the RMSE may be more reasonable. In this study, the
RMSE and MAE derived from the WHU-SGCC are reduced by approximately 14.47% and 33.87% on average compared to
CHIRP and CHIRPS, respectively. As for BIAS, WHU-SGCC method can correct the CHIRP precipitation bias in the spring,
fall and winter, but the results are not as good compared with CHIRPS. The larger BIAS values and higher PCCs in the spring
and fall may be attributed to the seasonal variations, when the CHIRP is highly consistent with the observations but subject to
large biases. After the correction, a substantial decrease in BIAS occurs in the winter, and there is no significant reduction in
the summer; all of the median and average adjusted values are approximately 0. The WHU-SGCC method provides an obvious
improvement in the NSE, with average improvements of 0.1742-13.8322 and 2.0131-14.7052 relative to CHIRP and CHIRPS,
though the median and average values are still less than 0, which may be due to the inherent uncertainty in the CHIRP.
Moreover, in terms of the POD, FAR and CSI, except for the results in winter, the WHU-SGCC method appears to be better
at detecting precipitation than CHIRP and CHIRPS; the results of POD and CSI are closest to 1, although FAR is worse than
CHIRPS on some days. However, the overall result of FAR is the best in the WHU-SGCC. The POD and FAR results are the

worst in the winter, and the CSI is slightly higher, which may be attributed to the overestimation of no-rain events and the inherent uncertainty in the CHIRP.

    Overall, the WHU-SGCC approach can be regarded as an effective tool for daily precipitation adjustments.

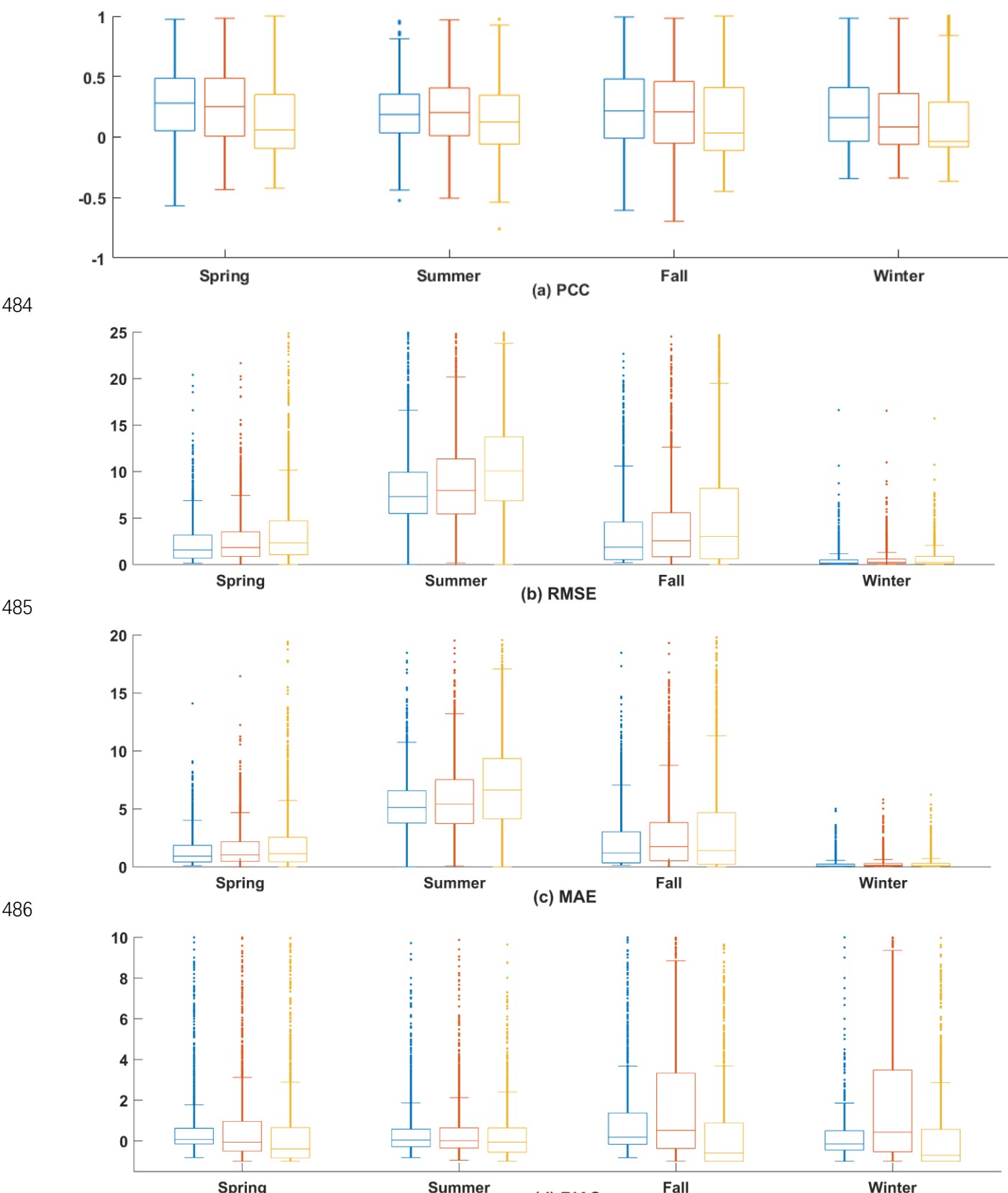

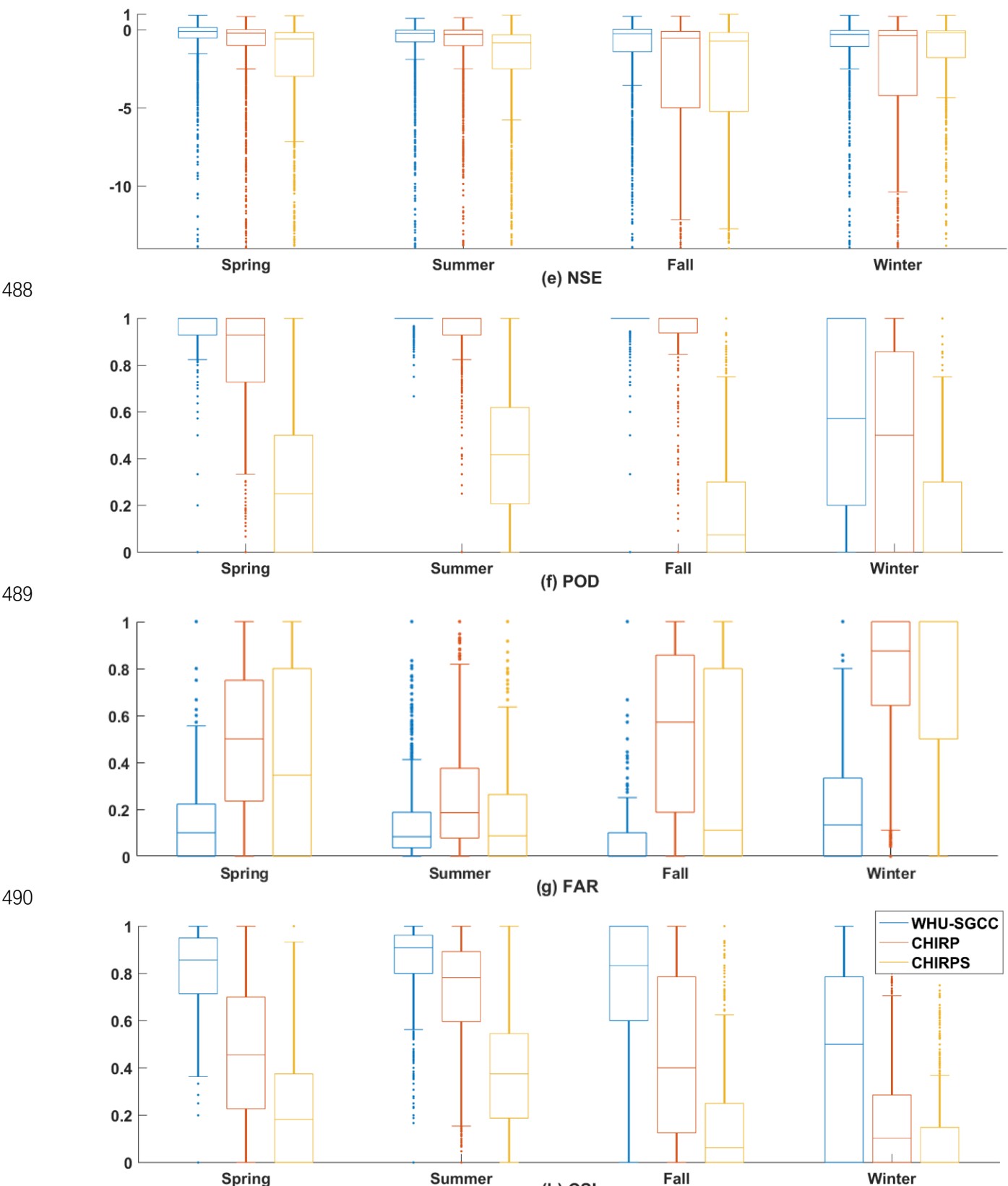

**Figure 7** Statistical analysis of the agreement between the daily observations and the WHU-SGCC, CHIRP and CHIRPS estimates with the leave-one-out cross validation: a) Pearson's correlation coefficient, b) root mean square error, c) mean absolute error, d) relative bias, e) Nash-Sutcliffe efficiency coefficient, f) probability of detection, g) false alarm ratio, and h) critical success index.

## 4.3 Model performance on rain events predictions

To measure the WHU-SGCC performance on predicting rain events, daily precipitation thresholds of 0.1, 10, 25, and 50 mm were considered, and the results are shown in Table 5 and Table 6. The average percentage of each class of rain events at the

validation gauge station during the four seasons from 1990 to 2014 are shown in Table 5. The major rain events within the Jinsha River Basin were no rain (<0.1 mm) and light rain (0.1-10 mm), which accounted for more than 80% of the total days (the average percentage of rain event days of the total days at each gauge station), while the number of days with daily precipitation greater than the 50 mm was the smallest (no more than 1% of the total days) and fewer than 5% of the days had daily precipitation in the range of 25 mm to 50 mm. In the spring, fall and summer, significantly more no-rain days occurred than rainy days, and approximately 5% of the days had daily precipitation of 10-50 mm. The seasonal distribution of rainfall was concentrated in the summer, and 54.76%, 14.01% and 3.62% of the days had daily precipitation of 0.1-10, 10-25, and 25-50 mm, respectively. The results indicated that the average daily precipitation was less than 10 mm throughout the years of the study.

**Table 5** The average percentage of each class of rain events at the validation gauge station during the four seasons from 1990 to 2014 within the Jinsha River Basin.

| Rain event (mm) / Season | <0.1 | [0.1,10) | [10,25) | [25,50) | >=50 |
|---|---|---|---|---|---|
| Spring | 57.87% | 38.43% | 3.29% | 0.39% | 0.02% |
| Summer | 26.89% | 54.76% | 14.01% | 3.62% | 0.72% |
| Fall | 57.32% | 36.62% | 4.99% | 0.93% | 0.14% |
| Winter | 85.78% | 13.99% | 0.21% | 0.01% | 0.00% |

The WHU-SGCC approach had lower errors than CHIRP and CHIRPS, as indicated by the RMSE, MAE and BIAS, but the performance of WHU-SGCC is not promising for events with total rainfall greater than 25 mm in the summer (Fig. 8). This negative performance for total rainfall higher than 25 mm in the summer might be attributed to the overestimation of rainfall by CHIRP and CHIRPS. For the seasonal distribution of precipitation (Table 5), the average daily precipitation within the basin less than 10 mm over the study period, which results in numerous rain gauge station data with values lower than 10 mm, which had a significant impact on the establishment of statistical relationships for the WHU-SGCC. Besides the WHU-SGCC dataset has almost always a negative bias, while CHIRP and CHIRPS has a positive bias in the different rain events. After bias correction of the WHU-SGCC, some precipitation estimates are lower than observations. The estimates of extreme rain events might also be attenuated during the process of WHU-SGCC adjustment.

Besides, the POD and CSI results of CHIRPS are the worst, while the results of the WHU-SGCC are the highest, which indicate its superiority for the detection of precipitation events. As for the results of the WHU-SGCC, the assessments of POD and CSI are the best in the summer, followed by the fall, spring, and winter, which are related to the seasonal rainfall pattern of more rain in the summer and less in the winter.

Therefore, the WHU-SGCC approach is applicable for the detection of rainfall events in the Jinsha River Basin, while in the summer it is better with rainfall less than or approximately equal to the average daily precipitation. Due to the homogenization of the WHU-SGCC method, its performance for short intense and extreme rain events was poorer than those of CHIRP and CHIRPS, which should be improved in a future study.

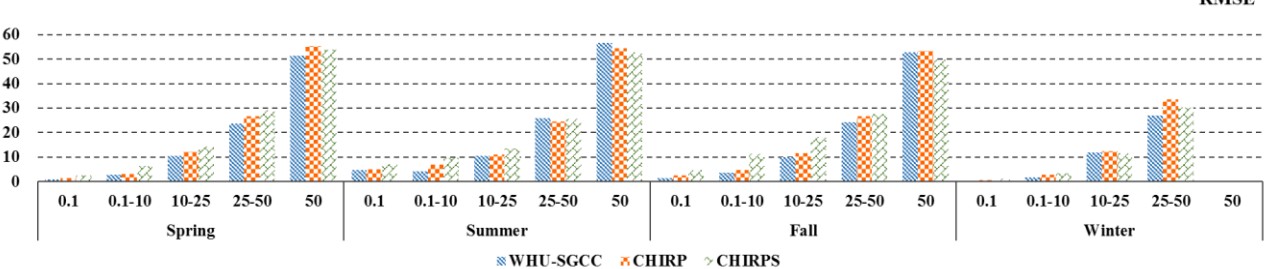

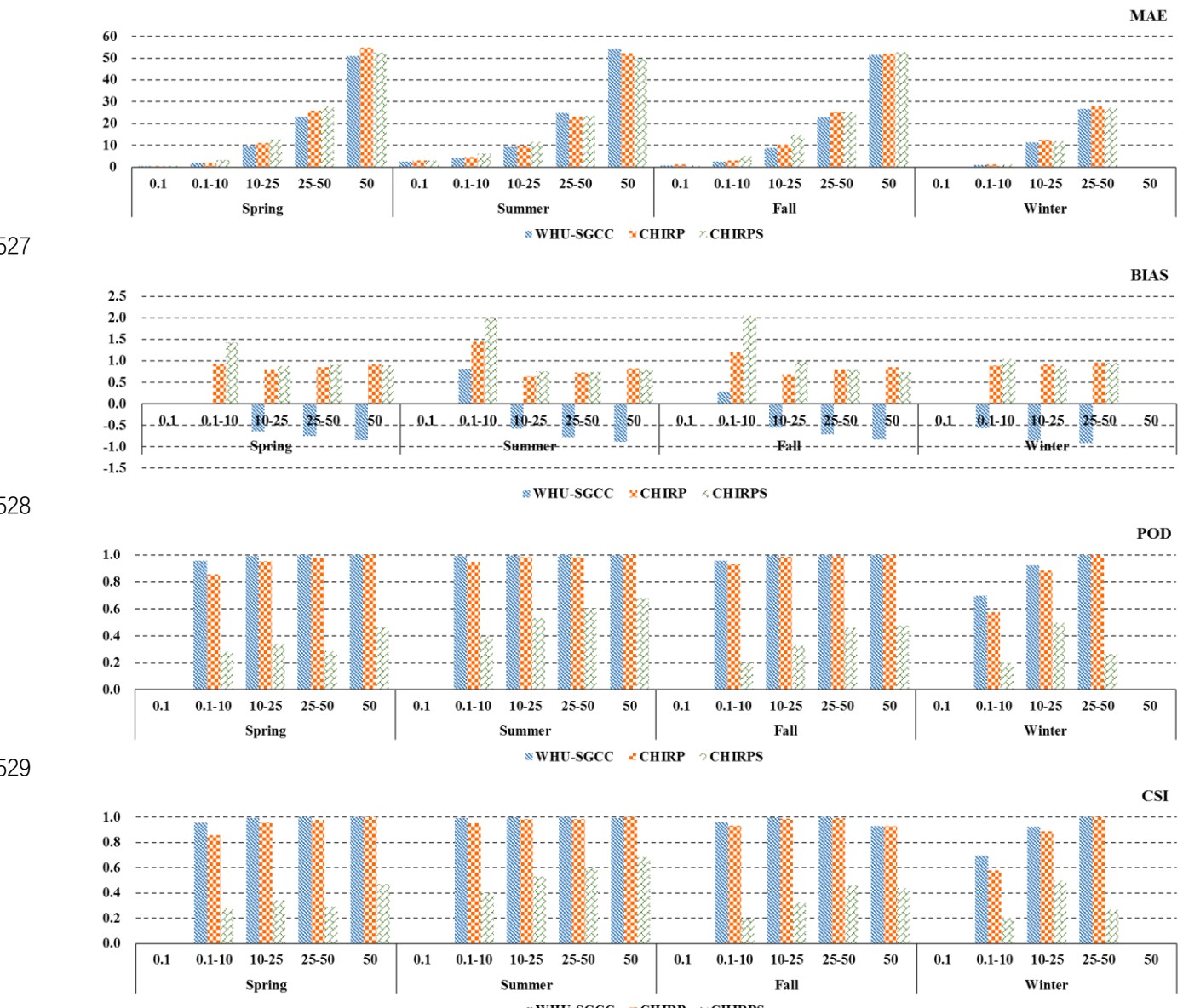




**Figure 8** Accuracy assessment of liquid precipitation events from 1990 to 2014.
**5 Data availability**
All the resulting dataset derived from the WHU-SGCC approach is available on PANGAEA, with the following DOI:
https://doi.pangaea.de/10.1594/PANGAEA.905376 (Shen et al., 2019). The high-resolution (0.05°) daily precipitation
estimation data over the Jinsha River Basin from 1990 to 2014 can be downloaded in TIFF format.
**6 Conclusions**
This study provides a novel approach, the WHU-SGCC method, for merging daily satellite-based precipitation estimates with
observations. A case study of the Jinsha River Basin was conducted to verify the effectiveness of the WHU-SGCC approach
during all four seasons from 1990 to 2014, and the adjusted precipitation estimates were compared to CHIRP and CHIRPS.
The WHU-SGCC method aims to reduce the bias and uncertainties in CHIRP data over regions with complicated mountainous
terrain and sparse rain gauges. To the best of the authors' knowledge, this study is the first to use daily CHIRP and CHIRPS
data in this area.
According to our findings, the following conclusions can be drawn: (1) The WHU-SGCC method is effective for the

adjustment of precipitation biases from points to surfaces. The precipitation adjusted by the WHU-SGCC method can achieve greater accuracy compared with CHIRP and CHIRPS, with average improvements of Pearson's correlation coefficient (PCC) of 0.0082-0.2232 and 0.0612-0.3243, respectively. The PCCs were improved to more than 0.5 in the spring and fall and to approximately 0.5 in the winter, and they were the worst in the summer, which may be attributed to the greater precipitation in the summer and lower precipitation in the winter. In addition, the NSE of the WHU-SGCC provides substantial improvements over CHIRP and CHIRPS, which reached 0.2836, 0.2944 and 0.1853 in the spring, fall and winter, respectively. In the summer, the NSE of the WHU-SGCC is still negative, but it is improved to be nearly zero, which indicates that the adjusted results are similar to the average level of the rain gauge observations. All of the measured errors were reduced except for the BIAS, which showed no significant improvement in the summer but was approximately 0. Overall, the WHU-SGCC approach achieves good performance in error correction of CHIRP and CHIRPS. (2) The spatial distribution of the precipitation estimate accuracy derived from the WHU-SGCC method is related to the topographic complexity. These errors over the lower elevation regions and the large size of light precipitation events with short durations resulted in a limited improvement in accuracy, with PCC values less than 0.3. However, higher PCCs and lower errors were observed over the north-central part of the river basin, which is a drier region with complex terrain and sparse rain gauges. The spatial distribution statistics indicate that the WHU-SGCC method is promising for the adjustment of satellite biases by blending with the observations over regions of complex terrain. (3) The leave-one-out cross validation of WHU-SGCC on daily rain events confirmed that the model is effective in the detection of precipitation events that are less than or approximately equal to the average annual precipitation in the Jinsha River Basin. The WHU-SGCC approach achieves reductions of the RMSE, MAE and BIAS metrics, while on rain events less than 25 mm in the summer. Specifically, the WHU-SGCC has the best ability to reduce precipitation bias for daily accuracy evaluations, with average reductions of 21.68% and 31.44% for compared to CHIRP and CHIRPS, respectively. As for the results of the WHU-SGCC, the assessments of POD and CSI are the best in the summer, followed by the fall, spring, and winter, which are related to the seasonal rainfall pattern of more rain in the summer and less in the winter. In spite of the corrections, the performance of the WHU-SGCC for short intense and extreme rain events was poorer than those of CHIRP and CHIRPS, and the bias in the precipitation forecasts in the summer are still large, which may due to the homogenization attenuating the extreme rain events estimates.

In conclusion, the WHU-SGCC approach can help adjust the biases of daily satellite-based precipitation estimates over the Jinsha River Basin, which contains complicated mountainous terrain with sparse rain gauges. This approach is a promising tool to monitor daily precipitation over the Jinsha River Basin, considering the spatial correlation and historical precipitation characteristics between raster pixels in regions with similar topographic features. Future development of the WHU-SGCC approach will focus on the following three aspects: (1) the improvement of the adjusted precipitation quality to better monitor extreme rainfall events by blending multiple data sources for different rain events; (2) the introduction of more climatic factors and multi-model ensembles to achieve more accurate spatial distributions of precipitation; and (3) investigations of the performance over other areas and for particular hydrological cases to validate the applicability of WHU-SGCC approach.

**Appendix A: Geographical characteristics of rain stations**
The station identification numbers and relevant geographical characteristics are shown in Table A1.

**Table A1** Geographical characteristics of the rain stations.

| Station number | Province | Lat (°N) | Lon (°E) | Elevation (m) |
|---|---|---|---|---|
| 52908 | Qinghai | 35.13 | 93.05 | 4823 |
| 56004 | Qinghai | 34.13 | 92.26 | 4744 |
| 56021 | Qinghai | 34.07 | 95.48 | 5049 |
| 56029 | Qinghai | 33.00 | 96.58 | 4510 |
| 56034 | Qinghai | 33.48 | 97.08 | 4503 |
| 56144 | Tibet | 31.48 | 98.35 | 4743 |
| 56038 | Sichuan | 32.59 | 98.06 | 4285 |
| 56146 | Sichuan | 31.37 | 100.00 | 4703 |
| 56152 | Sichuan | 32.17 | 100.20 | 4401 |
| 56167 | Sichuan | 30.59 | 101.07 | 3374 |
| 56247 | Sichuan | 30.00 | 99.06 | 2948 |
| 56251 | Sichuan | 30.56 | 100.19 | 4284 |
| 56257 | Sichuan | 30.00 | 100.16 | 3971 |
| 56357 | Sichuan | 29.03 | 100.18 | 4280 |
| 56374 | Sichuan | 30.03 | 101.58 | 3902 |
| 56459 | Sichuan | 27.56 | 101.16 | 3002 |
| 56462 | Sichuan | 29.00 | 101.30 | 4019 |
| 56475 | Sichuan | 28.39 | 102.31 | 1850 |
| 56479 | Sichuan | 28.00 | 102.51 | 2470 |
| 56485 | Sichuan | 28.16 | 103.35 | 2060 |
| 56565 | Sichuan | 27.26 | 101.31 | 2578 |
| 56571 | Sichuan | 27.54 | 102.16 | 1503 |
| 56666 | Sichuan | 26.35 | 101.43 | 1567 |
| 56671 | Sichuan | 26.39 | 102.15 | 1125 |
| 56543 | Yunnan | 27.50 | 99.42 | 3216 |
| 56586 | Yunnan | 27.21 | 103.43 | 2349 |
| 56651 | Yunnan | 26.51 | 100.13 | 2449 |
| 56664 | Yunnan | 26.38 | 101.16 | 1540 |
| 56684 | Yunnan | 26.24 | 103.15 | 2184 |
| 56778 | Yunnan | 25.00 | 102.39 | 1975 |

**Appendix B: Multi-annual land cover type**
The multi-annual land cover types in the Jinsha River Basin from 2001 to 2013 are shown in Fig. B1. All of the land cover
type maps were derived from the MODIS/Terra+Aqua Land Cover Type Yearly L3 Global 500 m SIN Grid V051 data set,
which is available online at https://search.earthdata.nasa.gov/search/granules?p=C200106111-
LPDAAC_ECS&q=MCD12&ok=MCD12 (last access: 23 July 2019). Fig. B1 shows that the land use had no obvious changes
over the study period. In addition, the upstream area of the Jinsha River is an untraversed region that has not been affected
significantly by human activities. Thus, the land use in the study area has hardly changed.

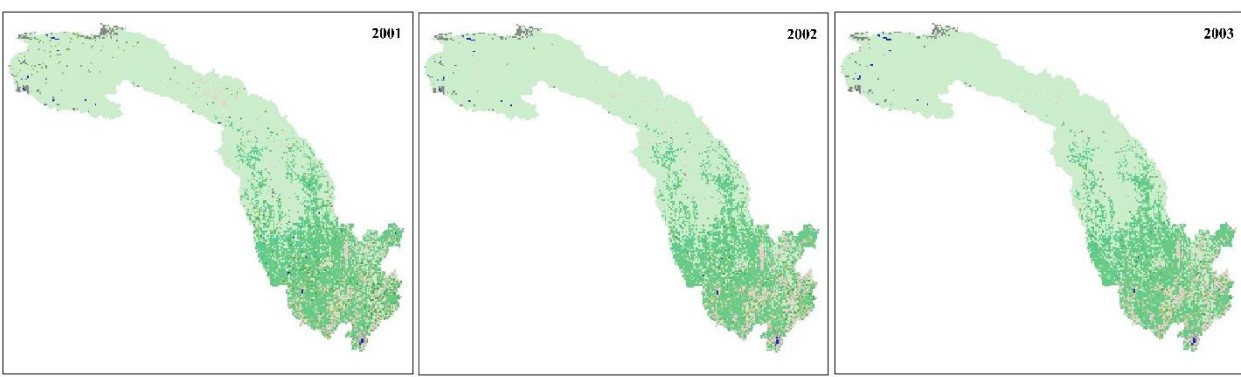


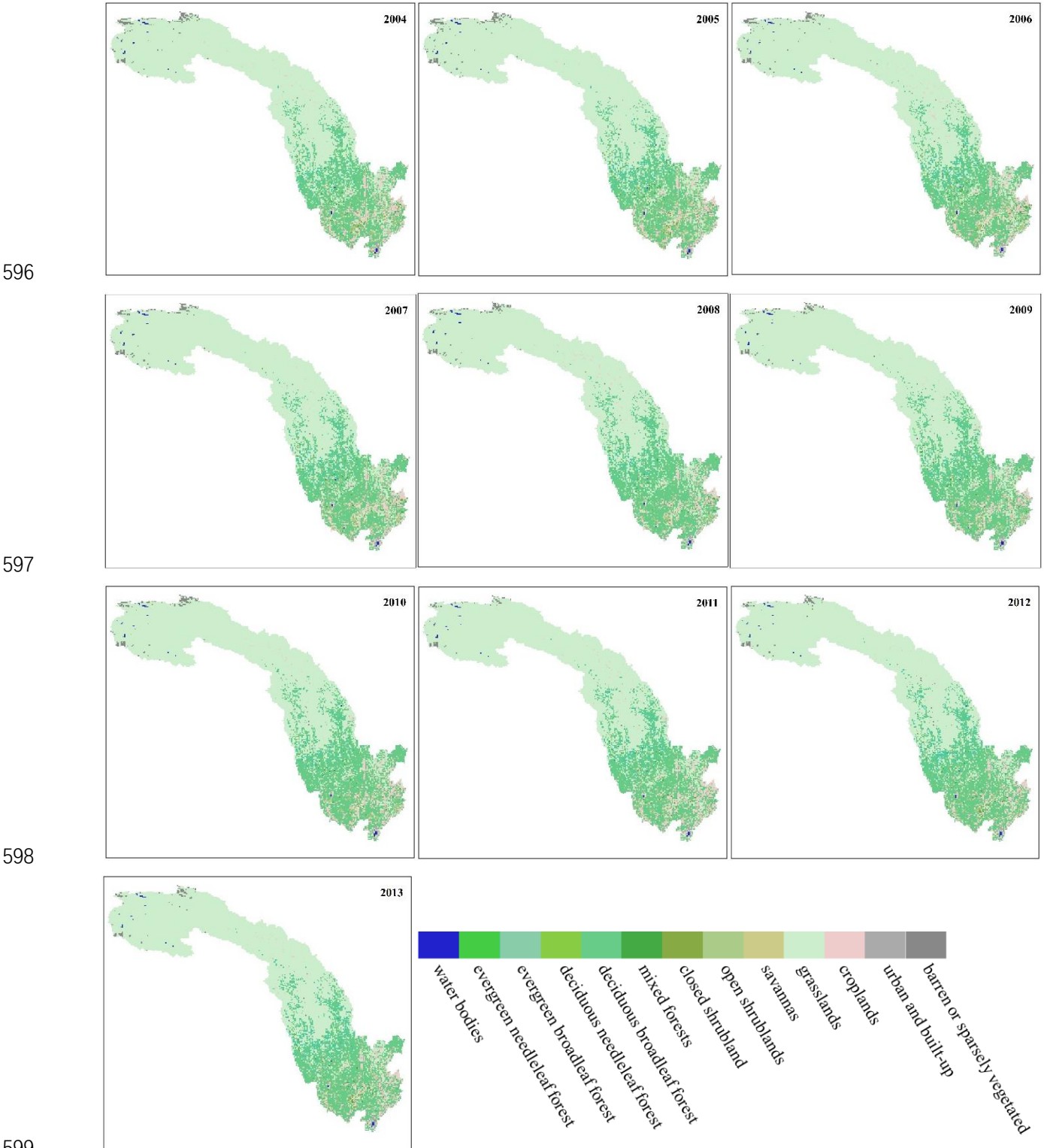

**Figure B1** Land cover types over the Jinsha River Basin from 2001 to 2013.












**Appendix C: Selection of decision trees for random forest regression**

52908

56004

56021

56029

56374

56459

56462

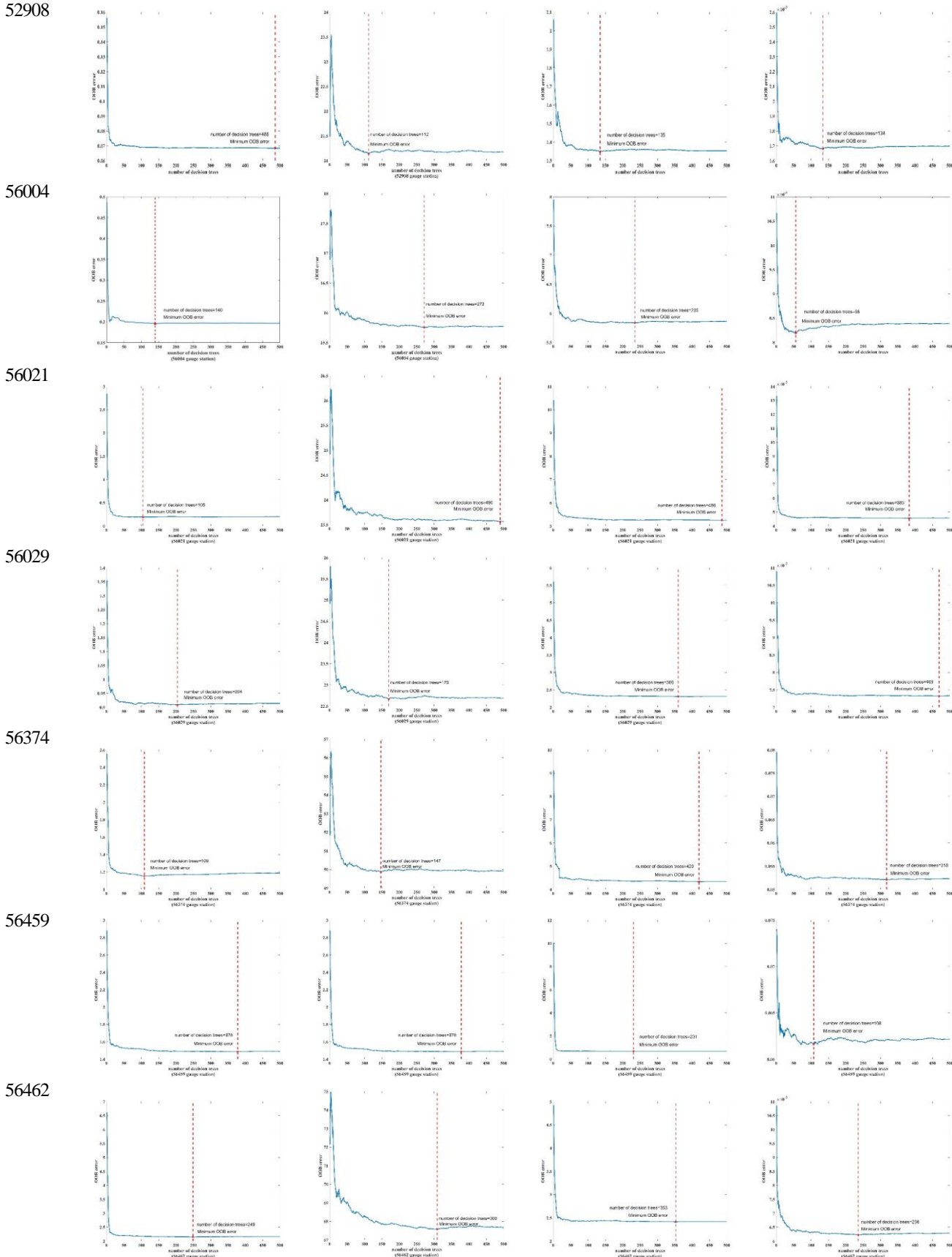

56475

56479

56485

56543

56565

56571

56684

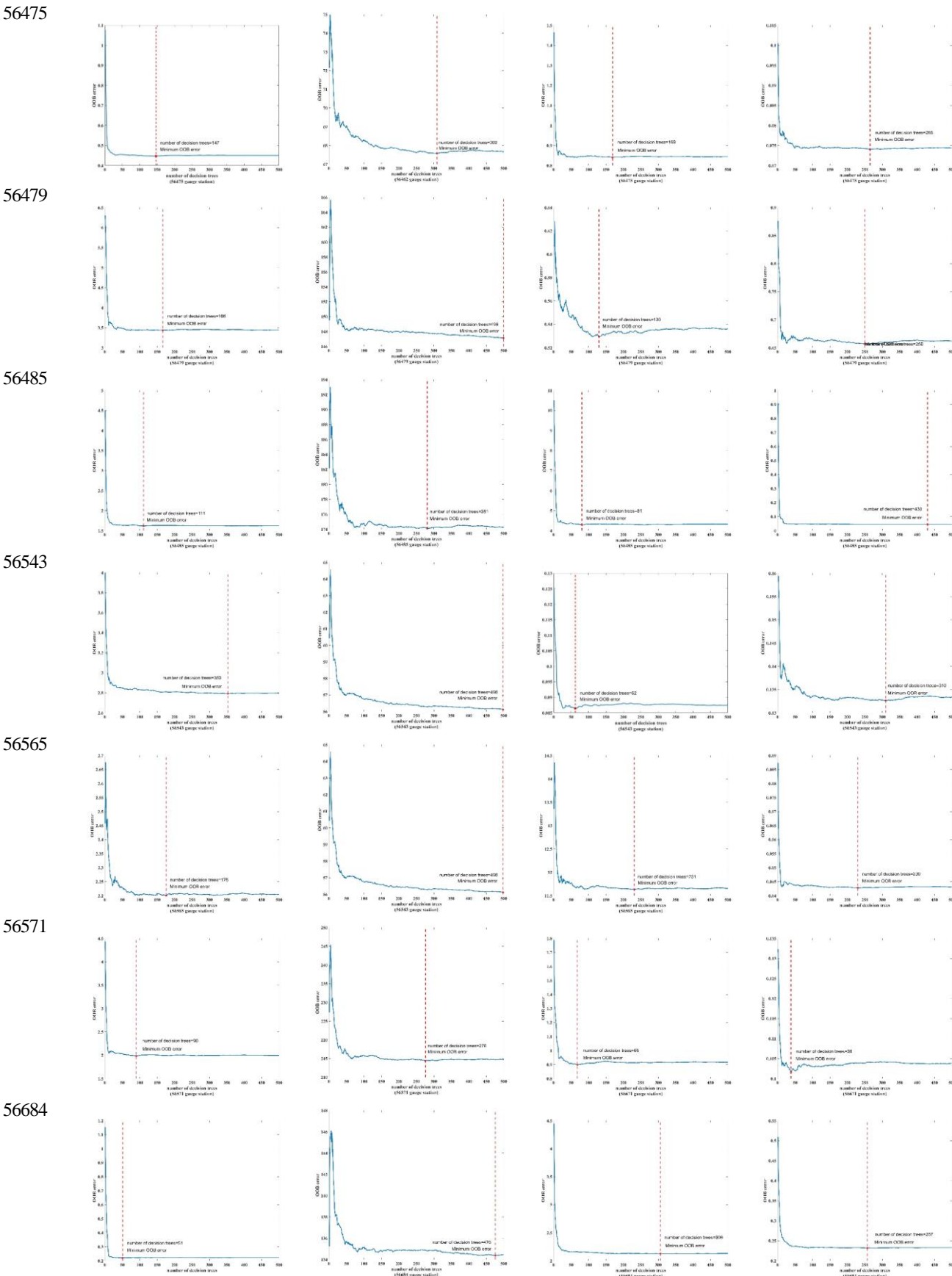

56778

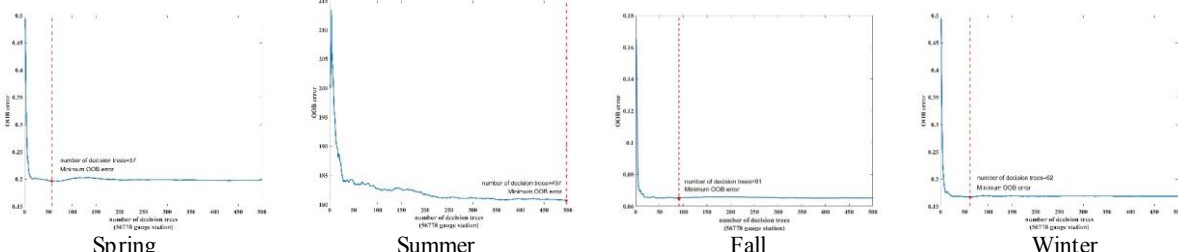

| Spring | Summer | Fall | Winter |

**Figure C1** Changes in the out-of-bag (OOB) error with increasing number of decision trees by means of random forest regression at each
gauge station.
**Appendix D: Spatial distribution of C1, C2 and C3 pixels**

Table D1 Pixel type of the validation gauge station.

| Validation gauge station / Pixel type | Spring | Summer | Fall | Winter |
|---|---|---|---|---|
| 52908 | C4 | C4 | C4 | C4 |
| 56004 | C4 | C4 | C4 | C4 |
| 56021 | C2 | C2 | C2 | C3 |
| 56029 | C2 | C3 | C2 | C3 |
| 56034 | C2 | C3 | C2 | C3 |
| 56038 | C4 | C4 | C4 | C4 |
| 56144 | C4 | C4 | C4 | C4 |
| 56146 | C4 | C4 | C4 | C4 |
| 56152 | C2 | C3 | C3 | C4 |
| 56167 | C4 | C2 | C2 | C4 |
| 56247 | C4 | C4 | C4 | C4 |
| 56251 | C2 | C2 | C3 | C3 |
| 56257 | C4 | C4 | C4 | C4 |
| 56357 | C4 | C4 | C4 | C4 |
| 56374 | C4 | C4 | C3 | C4 |
| 56459 | C4 | C4 | C4 | C4 |
| 56462 | C4 | C4 | C4 | C4 |
| 56475 | C4 | C3 | C3 | C3 |
| 56479 | C4 | C4 | C4 | C4 |
| 56485 | C3 | C2 | C2 | C3 |
| 56543 | C3 | C3 | C4 | C4 |
| 56565 | C2 | C2 | C3 | C3 |
| 56571 | C2 | C4 | C4 | C4 |
| 56586 | C2 | C3 | C2 | C3 |
| 56651 | C3 | C2 | C2 | C3 |
| 56664 | C4 | C4 | C4 | C4 |
| 56666 | C3 | C3 | C3 | C3 |
| 56671 | C3 | C2 | C2 | C3 |
| 56684 | C2 | C2 | C2 | C4 |
| 56778 | C4 | C3 | C3 | C4 |


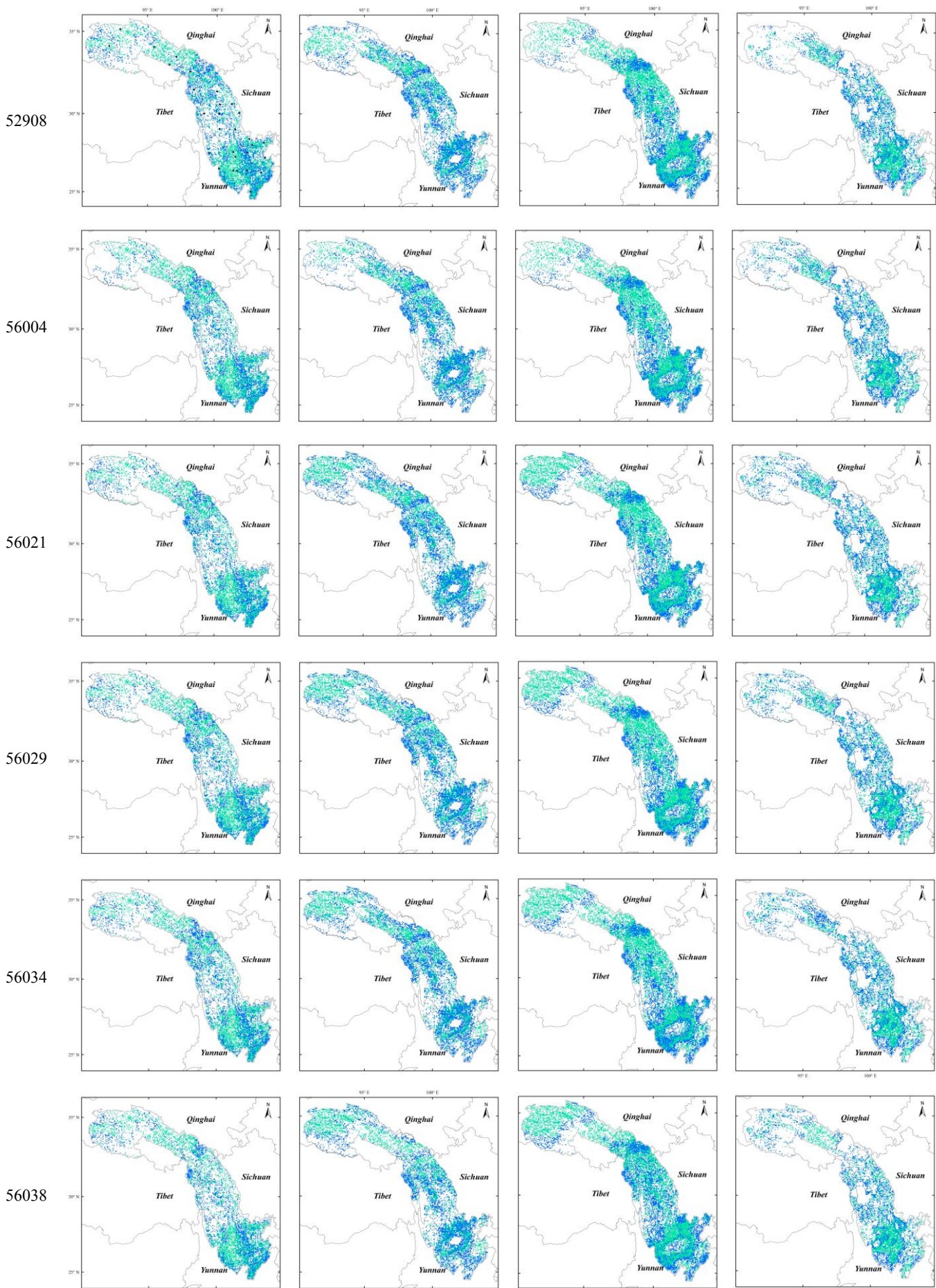

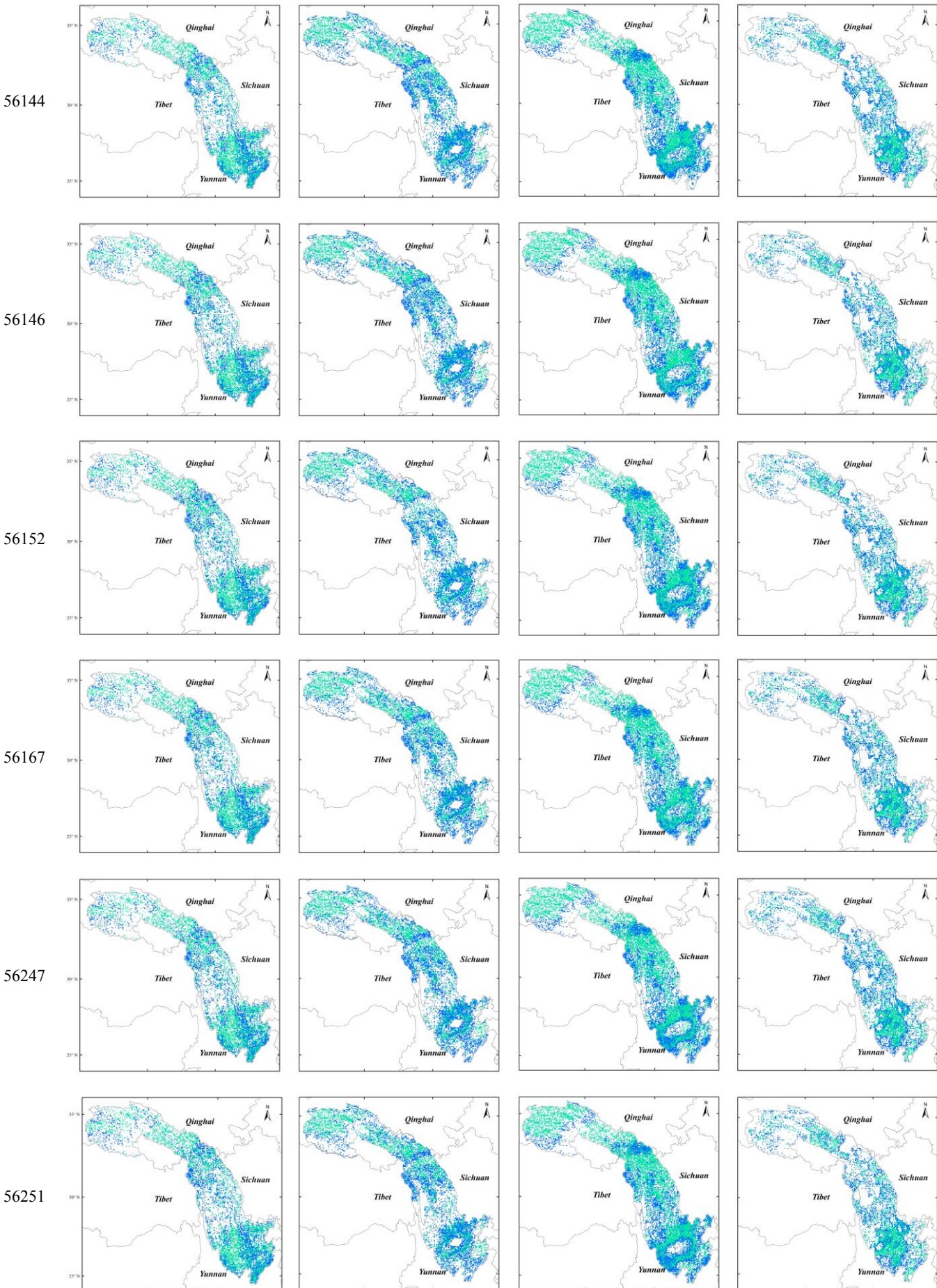

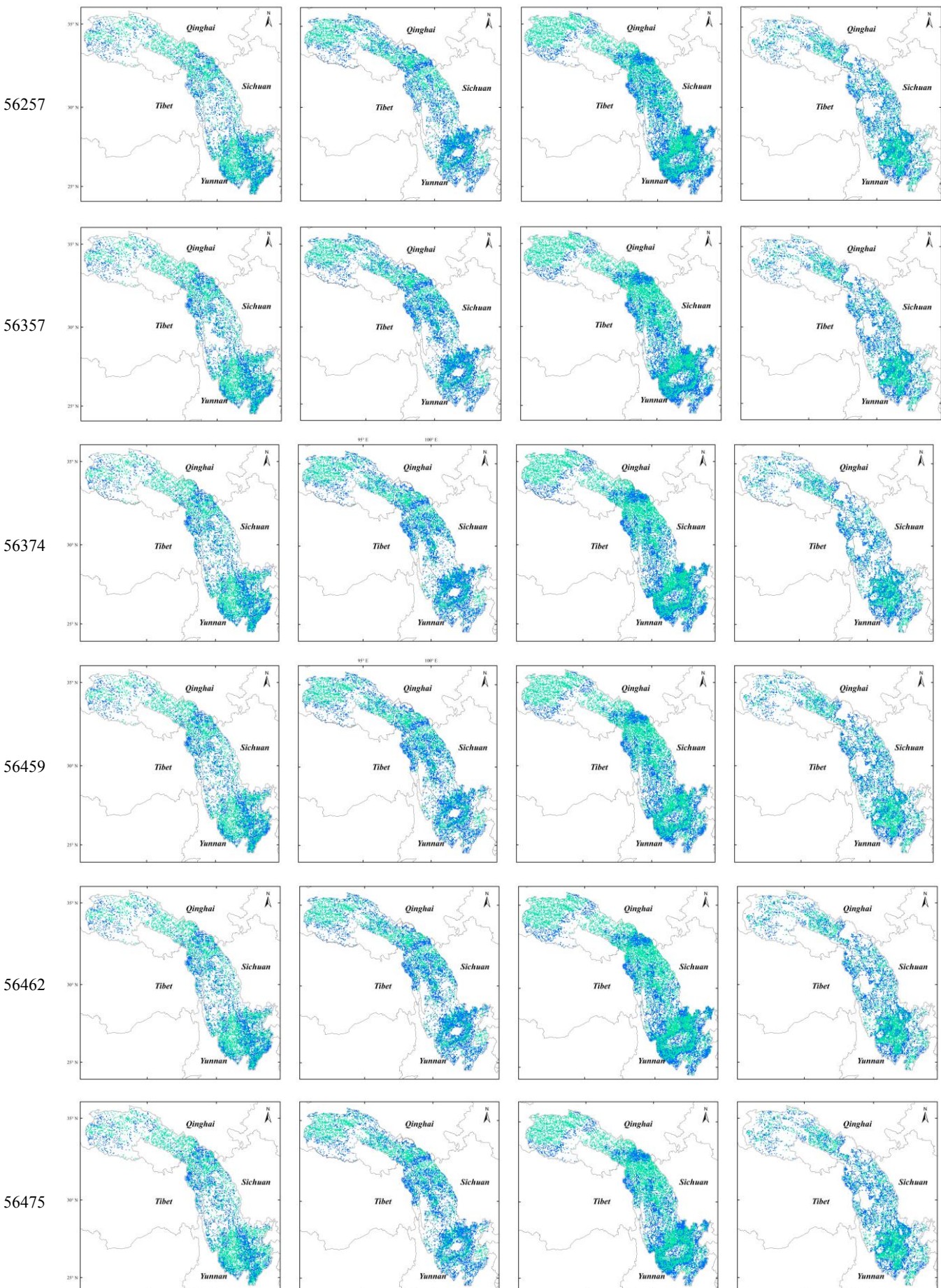

56257

56357

56374

56459

56462

56475

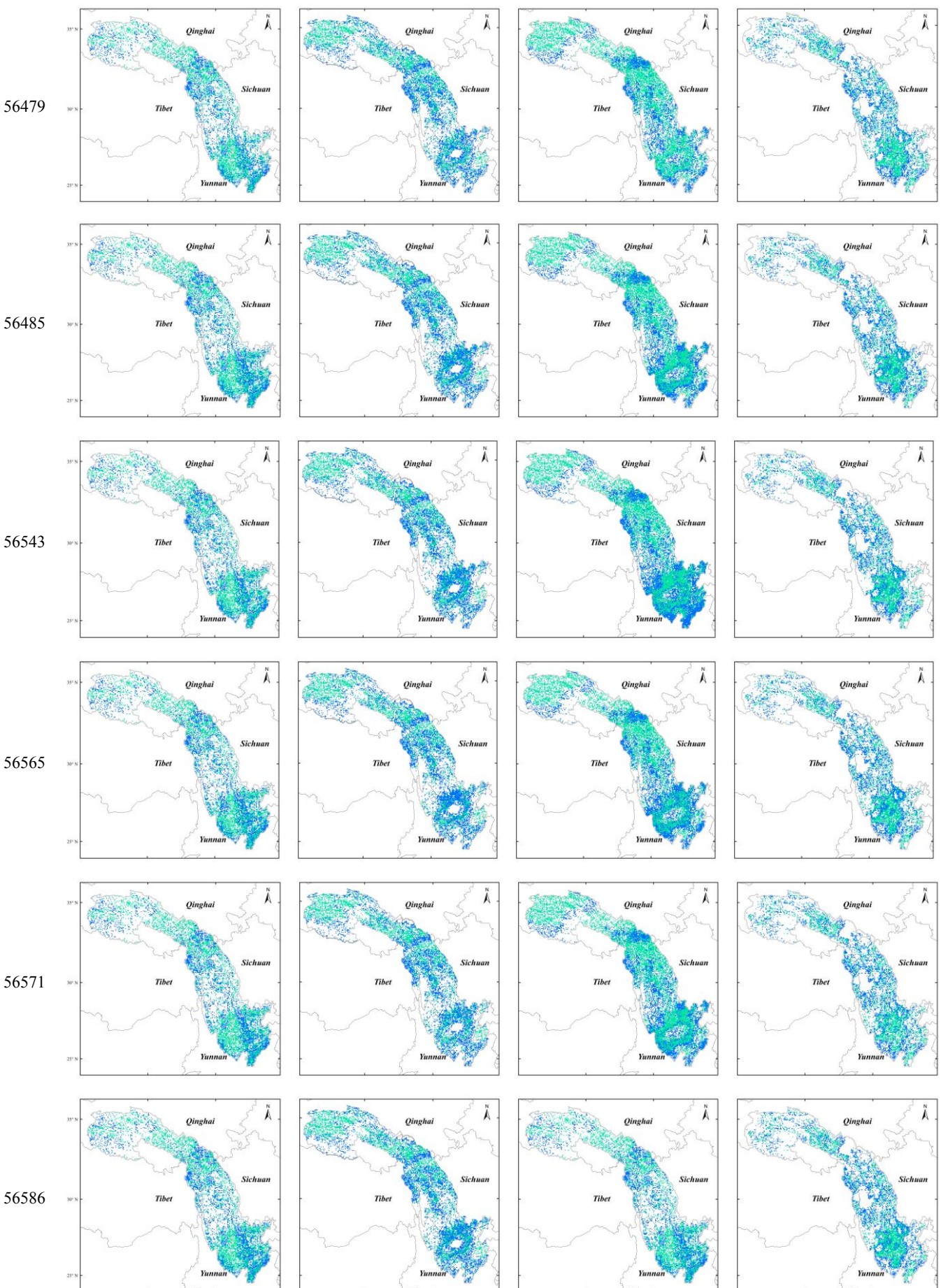

56479

56485

56543

56565

56571

56586

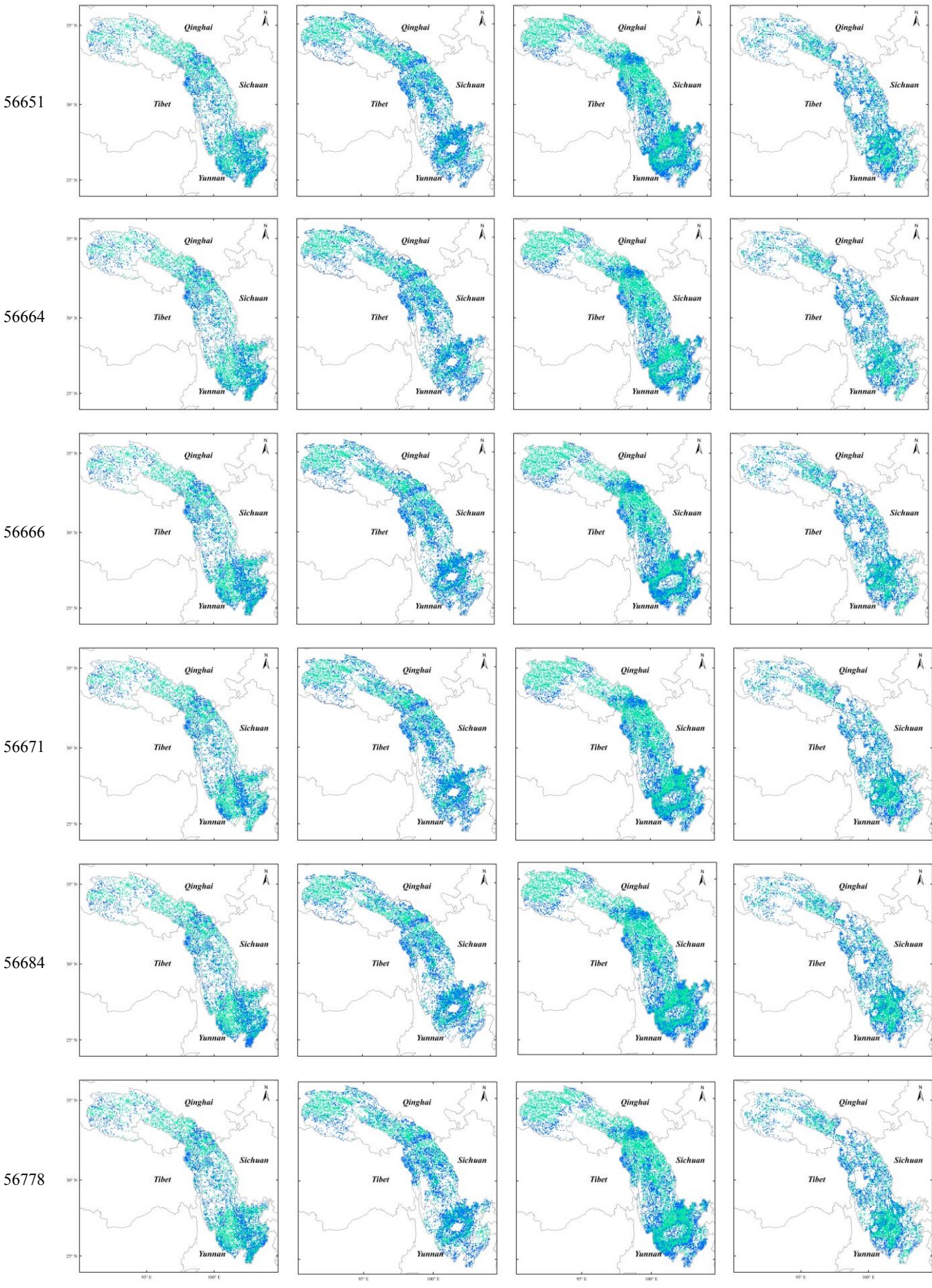

56651

56664

56666

56671

56684

56778

Spring            Summer            Fall            Winter


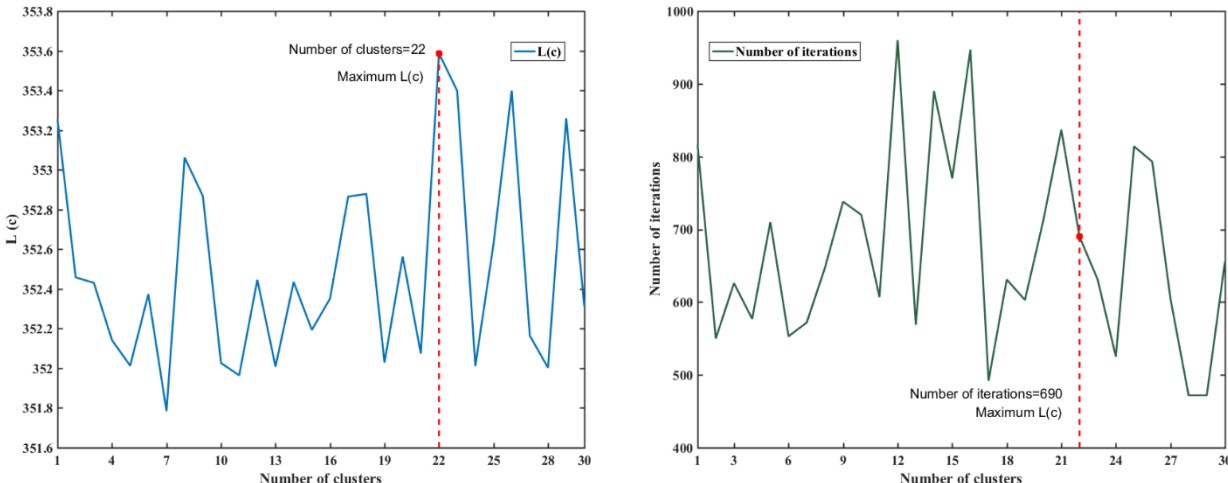

**Figure D1** Spatial distribution of each class of pixels adjusted by each rule using the WHU-SGCC method in the Jinsha River Basin.
**Appendix E: Spatial Clustering from the FCM method**

**Figure E1** Optimum number of clusters determined by the maximum $L(c)$ with the iterative process.

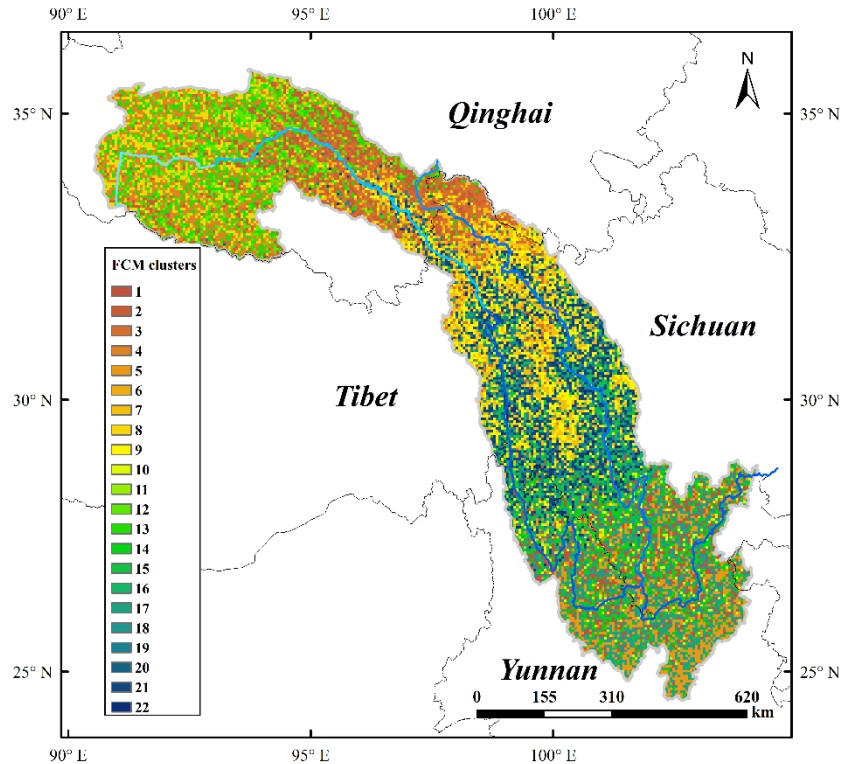


**Figure E2** Spatial clustering as defined by the FCM method in the Jinsha River Basin.
This appendix shows how to set the number of clusters in the FCM method.
To adjust the pixels other than those of the gauge stations, the pixels that are statistically similar to the C1 pixels were
selected. According to Rule 2, the C2 pixels were identified in a spatial area defined by the FCM method. In the following
experiments of Rule 2, we set the parameters $m = 2, \varepsilon = 0.00001$, and the maximum number of iterations was set 1000 ( a
sufficiently large value considering the algorithm efficiency). To determine the optimal numbers of clusters, the value of $c$ was
varied from 1 to 30 with an increment of 1. The values of $L(c)$ during the running of the FCM are shown in Fig E1. The optimum
number of clusters was 22, and the number of iterations was 690 less than the maximum number of iterations.
Therefore, the number of clusters was set to 22, and the number of iterations was set to 1000 for full operation by means of
the FCM. The spatial clustering results considering the terrain factors are shown in Fig. E2. In general, the spatial results of
the FCM have many of the same characteristics as the areas defined by the terrain variations, especially with respect to the
slope and runoff directions, which may influence the regional rainfall.

**7 Acknowledgments**

This work was supported by the National Natural Science Foundation of China program (no. 41771422), the Nature Science
Foundation of Hubei Province (no. 2017CFB616), the fundamental research funds for the central universities (no.
2042017kf0211), and the LIESMARS Special Research Funding.
The authors would like to thank data support: the Climate Hazards Group at the University of California, Santa Barbara, for
providing CHIRP and CHIRPS datasets (http://chg.ucsb.edu/data/), and the National Climate Center (NCC) of the China
Meteorological Administration (CMA) for providing the daily rain gauged observations and gridded precipitation observations
(http://data.cma.cn/). The authors also thank the PANGAEA Data Publisher for Earth & Environmental Science platform for
providing the storage to disseminate the data generated in this experiment.
The authors are grateful for the editor and anonymous reviewers for their useful suggestions that clearly improved this paper.

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
