# Peer review of "WHU-SGCC: A novel approach for blending daily satellite (CHIRP) and precipitation observations over the Jinsha River Basin"

_Earth System Science Data, 2018_

## Referee Comment (RC1) · Anonymous Referee #1 · 20 Feb 2019

Minor comments - Line 14: By mentioning the CHIRP database the University of Santa Barbara needs to be cited as developer. Thus, the sentence has to be changed in: . . . the Climate Hazards Group InfraRed Precipitation (CHIRP, daily 0.05°) satellite-derived precipitation developed by the UC Santa Barbara - Line 52: When the CHIRPS dataset has been mentioned the developer (UC Santa Barbara et al.) has to cited as well. - Line 109: Section 2.2 can to be compacted in only 2 subsections for a better reading: 1) precipitation gauged observations and 2) gridded precipitation+CHIRPS - Line 159: what's "SICR approach"? - Line 162 "C3 (pixel physically similar to C1C2)". What does it mean "physically"? - Line 180 ". . .satellite precipitation estimations deviated from observed data . . .". Really satellite precipitation even though retrieved are always mea-

sured data. Thus, it is better replacing the above sentence with: "satellite precipitation estimations deviated from ground-based measurements" - Section 4 – This section is too much subdivided getting quite difficult the reading. Please, let you group the discussion. - Table 6: What's "wet precipitation?" You mean, probably liquid precipitation, right?

Major comments to Authors The proposed manuscript tries to improve the performance of the CHIRP/S datasets by statistically adjusting the original data over complex terrain. The general statistics described in table 5 reveals very light improvements even though WHU-SGCC performs better and CHIRPS dataset seems to be worse also respect to raw data (CHIRP). Skipping to the performance evaluation for rain categories, how do you justify the inversion of BIAS tendency from the category (5,10) to > 40 (see table 6)? The accuracy of WHU-SGCC method seems to be limited to low precipitation (<10, not >20) where the model tends to overestimate. For precipitation greater than 10 the WHU-SGCC starts to underestimate. Please, let u clarify this! Really, the validation of the WHU-SGCC method is only limited to the Jinsha River Basin in summertime 2016 thus new and more accurate validation campaigns have to be done. On that, the challenging efforts to apply and validate a new method over orographically complex terrain have to be supported by new application on similar morphology where the rain-gauges are typically sparse. Furthermore, since during the monsoon season precipitation is typically higher than 20 mm, how the WHU-SGCC will perform? Of course, this question needs to be exhaustively answered by a new validation using the same methodology described in the manuscript.
* * *

---

## Referee Comment (RC2) · Anonymous Referee #2 · 21 Feb 2019

General Comments The manuscript describes a very interesting method to correct random and systematic errors from satellite infrared precipitation estimates based on gauge data and grid points from interpolated precipitation fields. The manuscript is well structured and present links to access all the data that is used in the work. However, I found that the analysis period (summer of 2016, JJA) is too short to make significant conclusions about this precipitation dataset. Also, the manuscript could have explored another ways to analyze and evaluate the dataset other than the very conventional use of statistical metrics, that are definitely needed, but the authors could have explored beyond that. Maybe using a case study or exploring the usefulness of the data for a particular hydrological application would be helpful. The dataset published with the

manuscript is of good quality and is stored in a standard and easy to read format. I suggest that the authors make another English revision, especially regarding the use of articles and prepositions.

Specific Comments Line 18: It was evaluated not only by categorical indices. Line 28: The number of gauge stations is actually very limited, especially in regions with complex terrain and in the case of gauges that measure solid precipitation. Accuracy of this gauges is also not very good in the case of solid precipitation. It would be a good idea to state that the paper focus is on liquid precipitation (rainfall), and use this term throughout the text. Line 35: I found this line confusing in the way it is phrased. I think this sentence could be phrased this way: "Satellite estimates are susceptible to systematic biases that can influence hydrological modelling." In the introduction, I think that a better description of what is available to estimate precipitation from satellites is missing. For example, GOES-R and GPM are missing in the description. Line 62: That are other sources of uncertainty in the monitoring of rainfall in complex terrain (e.g., orographic enhancement). Line 94: I was not able to understand the average precipitation over the Yangtze River Basin. You could be more specific about what statistic you are presenting here. Usually what is presented is the spatially averaged annual accumulation of precipitation as an indication of precipitation climatology for the region. Line 98: I was not able to comprehend the units for the area of the basin. It should be presented as km2. Line 102: Topography would not exert a temporal variation in climate, since this is not a very dynamic feature of the Earth's surface. It could exert a temporal variation in weather though. Lines 102 and 103: Please try be consistent with the statistics you are using here. Line 134: Maybe explain better what is the CHPClim product. Line 146: Which CHIRPS data? The data from its stations? Or the blended product? Please make this clear here. Line 162: What do you mean about physically similar? Is this means that these pixels are related to others based on its physical attributes (lat, long, elevation, slope, aspect, and curvature)? Or this means that is similar is terms of rainfall distribution? If is in terms of rainfall, I think a better world would be statistically similar rather than physically similar, since

this is based on a cluster analysis. Line 170: Please explain why you chose 30% of the stations/grid points for validation. Since the stations and grid points are of limited number, it would not be better to do a Bootstrap validation instead? Then you are able to use all the stations/grid points in your correction algorithm. Lines 224 and 225: Since your method relies heavily on the cluster algorithm, it would not be better to use some sort of statistical metric to define the number of clusters? Line 234: Pixels should also be similar regarding precipitation characteristics, is that right? Line 244: I think you should use the word relationship or correlation instead of confidence in this line. Line 259: Why the number of decision trees was set to 500? Line 264: Do C1 and C2 pixels included in the R pixel category? Lines 267 and 280: Are you also considering only SCC values with p-value lower than 0.05 here? Line 287: Why did you choose the 10 mm value? Could you please explain the meaning of this constant better? Line 298: Please use the actual percentage here. Line 326: You should be clearer on what the numbers of pixels are here. Is this the number of pixels inside the basin multiplied by the number of days? It would be a good idea to describe the exact number of pixels for each class along with its actual percentage in the text. Line 341: It would be better to use the same x axis scale in both plots. It seems that the gridded data observation has similar biases as CHIRP and thus their CDFs are more similar, providing less improvement in the adjusted dataset. Line 355: It would be a good idea to explore and discuss more the statistics presented in Figure 7. Line 373: NSE values have increased, but still not very good (i.e., still negative). Line 376: Is not intuitive that the evaluation metrics are better for CHIRP than CHIRPS, since CHIRPS adds stations data to their dataset. Could you please clarify this? I am seeing here that magnitude evaluation metrics have not changed considerably, probably because the improvement is seem in the low magnitude events. SCC has a considerable increase, but still cannot explain much of the variability of rainfall in the region. POD values are good. Line 383: Could this means that because the daily precipitation in the lowland region of the basin is higher, the RMSE values are also higher? Line 388: The fact that your method performs well in complex terrain is a very positive point in your manuscript,

but you will need a longer study period to confirm this finding. Line 402: The boxplots do not show that the higher reduction is seen in the Bias metric. Line 417: I am still confused about which rainfall value is presented is this figure. It does not seem realistic that the average daily rainfall would be $\sim$ 200 mm. Please be more specific about which rainfall statistic you are using in this figure. Line 430: The evaluation metrics for the threshold values are quite similar among the three precipitation products. Just because the values for WHU-SGCC become slightly worse for precipitation higher than 20 mm/day does not mean they are significantly different from the other products. Since the values are very similar, I would suggest to test their differences statistically before making the assumption that WHU-SGCC works better for low magnitude events. This might also be caused by the limitation of the short study period. There is a tendency to use the word scale for a temporal dimension, better to use period instead. I think it would be interesting to add a map with the accumulated precipitation for the study period. The analysis period of 92 days is too short to make conclusive assumptions about the dataset usefulness in the region, which are made several times throughout the text. I understand why the authors want to focus on the summer months to avoid the higher biases introduced by solid precipitation, but I did not understand why they only performed the evaluation for one summer season. Is there a particular reason for that? I still think that is hard to make the conclusions you made based on 92 days, if you add more seasons to the analysis this can become a very interesting manuscript and dataset. The dataset seems to be of good quality. A few comments about it are the following. For a spatial extent of this magnitude I think it would be better to use a geographic coordinate system, rather than a Mercator projection. There is also some artifacts (0 precipitation values) that appear at the same location at multiple days. I was wondering if this is a limitation from the negative values of Rule 4. Is there any way to correct this? This dataset could be very useful if its period is extended to multiple years.

Technical Corrections Line 2: Change "over Jinsha" for "over the Jinsha". Line 31: Change "distributed" for "spatial distribution". Line 37: "without adjustment" is mentioned twice. Line 68: In table 1, it should be written "PERSIANN-CDR" instead of "PRESSIANN-CDR". Line 84: Change "in summer 2016" for "in the summer of 2016". Line 93: Change "proximately covers an area" for "covers an area of approximately". Line 95: Change "sub-regions" for "sub-basins". Line 136: Change "precipitation observations" for "surface based precipitation observations". Line 157: Change "other pixels" for "the remaining pixels". Line 159: The acronym "SIRC" meaning was not mentioned before. Line 169: The first sentence could be placed before the item 1. Line 171: Is the same phrase that is shown in line 163. Line 172: The flowchart: CHIRP resolution should be 0.05 x 0.05. In the first box of rule 3, change "and gauged" for "with gauged". In the last box of rule 3, change "can derive" for "can be derived". In rule 4, change "ration" for "ratio" (this happens twice here). Line 189: Change "as" for "in". Line 246: Change "p" for "p-value". Line 283: The word "method" is repeated twice. Line 300: Change "for summer (JJA) 2016" for "for the summer (JJA) of 2016". Line 315: Change "as" for "in". Line 326: Change "to be adjusted" for "adjusted". Line 340: Change "study" for "studies". Line 400: Change "with especially greatly decreases compared to CHIRPS" for "with greater decreases when compared to CHIRPS". Line 451: Change "in summer 2016" for "in the summer of 2016". Line 456: Change "over region has" for "over a region that has". Line 465: Change "of the precipitation region" for "of precipitation events in the region". Line 466: Change "short" for "short duration". Line 468: Change "complicated mountainous" for "complex terrain". Line 480: Change "topographic and long time series climatic factors" for "topographic factors and longer time series".

---

## Referee Comment (RC3) · Anonymous Referee #3 · 21 Feb 2019

The manuscript presents a new method for combining high-resolution daily satellite precipitation estimates with rain gauge observations. The method is applied and evaluated over the Jinsha River Basin for the summer period in 2016 (June, July August). The performance of the method is compared to already existing satellite datasets CHIRP, which is also the base for the new dataset, and CHIRPS. The evaluation reveals an improvement in accuracy of precipitation estimates with rain rates of less than 20 mm per day compared to CHIRP and CHIRPS, however, the chosen time period of just 3 months seems to be rather short for this somewhat general conclusion. For heavy precipitation, however, no improvement could be found. The dataset and the blending method are described and the data is available for free.

The manuscript fits in the scope of ESSD, but some issues need to be addressed. I recommend taking the following suggestions and comments into account:

1. It is not quite clear to me what exactly is the reference dataset in this study. On page 6, line 170 the authors state that 70% of the total gauged stations and gridded points were used as the training dataset and the remaining 30% serve as reference dataset. How was decided which station / grid point was used for training and which station / grid point was used for evaluation? As I understand it is a mixture between actual station measurements and gridded, i.e. interpolated, station data. Is the ratio for both data types also 70% training and 30% reference data points? Is there a difference in performance metrics when only one of the two datasets is used for evaluation? Direct measurements from stations might be even more accurate than the interpolated data. A more detailed description of the reference dataset and decision making process is desirable, e.g. a map with the mean or the sum of precipitation during the observation period at the reference grid points and stations.

As far as I understand, using this evaluation dataset implies that only C1 and C2 grid points are evaluated, because they contain either a rain gauge station or a grid point of the interpolated station data. Is that correct? Can the authors give an assessment on the quality of the method at C3 and C4 pixels?

Does the selection of the stations and grid points for training have an influence on the model performance? Depending on the location of the points for adjustment the quality of the blended dataset may vary. An ensemble study using different compositions of the pool of training stations / grid points would give statistically more robust results.

2. CHIRP data is used as basis for the WHU-SGCC dataset and it is shown that the blending approach leads to better (light and moderate rainfall) or similar (heavy precipitation) results compared to measurements. CHIRPS, however, seems to perform much worse than the original CHIRP dataset although it is also adjusted to rain gauges. Can the authors give an explanation for that? It would also be desirable to expand the

investigated period to get more robust results, e.g. add more summer seasons from other years.

Specific comments:

- P.1, L.37: There is twice "without adjustment" in the sentence

- P.2, L.63 and 65: Remove the brackets at Bai et al. and Trejo et al.

- P.3, L.89: Section 5 is about data availability. Section 6 presents conclusions

- P.3, L.102-103: I'm a bit confused here. Does "average annual precipitation", "annual precipitation" and "total annual precipitation" mean the same thing? Or is the total (for me this refers to the sum) of the precipitation north of Shigu almost four times smaller than the mean annual precipitation in the whole Jinsha River Basin?

- P.6, L169: I would remove the numbering here, as it doesn't seem to be another part of the method, but refers to the overview of steps 1-4.

- P.11, L.309: Nash and Sutcliffe(1970) is missing in the references

- P.14, Table 4: How is the accuracy assessment of C3 pixels done? What is the reference here? Why is SCC < 0.5?

- P.17, Fig.10: It might be helpful to present the percentage deviation from the observations for clarification of the model performance. It seems that at some days, all three datasets deviate more than 70% from the observations.

---

## Author Comment (AC1) · 2 Apr 2019

Dear Reviewer,

Thank you for your insightful comments and suggestions. We have modified the manuscript accordingly. We trust that all of your comments have been addressed accordingly in the revised manuscript. If you have further suggestions for changes, please let us know. The detailed corrections are listed below point by point:

All changes in the manuscript are marked with red color in the attachment: 1) The reply to reviewer: essd-2018-150 Reply to Reviewer 1.pdf 2) The revised maniscript:

essd-2018-150-manuscript-WHU-SGCC for blending CHIRP and observations-Earth System Science Data-Revised.pdf 3) The clean manuscript: essd-2018-150-manuscript-WHU-SGCC for blending CHIRP and observations-Earth System Science Data-Clean.pdf 4) The appendix of the manuscript: essd-2018-150-Appendix for manuscript.pdf

Please also note the supplement to this comment:
https://www.earth-syst-sci-data-discuss.net/essd-2018-150/essd-2018-150-AC1-supplement.zip

---

## Editor Comment (EC1) · Giulio G.R. Iovine (Editor) · 31 Jul 2019

Please, consider even the following comments (received via email by one of the Referees) during the revision of your manuscript:

<[authors should] explain more clearly how they did the leave-one-out validation as this important information is not emphasized in the paper. By leaving-one-out if I understand correctly, they will have to repeat the algorithm 30 more times with each time leaving one station as validation station. The pixel that has that station therefore could be identified as either C2, C3 or even C4 pixels. If this is the case, this validation will allow them to validate corrections for all types of pixels. I would then highly rec-

ommend that they add (C2, C3, C4 pixel type) near the station number in Table 4. If however, all the station shows as C2 pixels (can't be C1), then the validation is indeed not complete.>

—————————————————————

---

## Author Response (AR1)

**Reply to Reviewer 1**

**Manuscript ID:** essd-2018-150
**Title:** WHU-SGCC: A novel approach for blending daily satellite (CHIRP) and precipitation observations over Jinsha River Basin
**Journal:** Earth System Science Data
**Type:** Article

Dear Reviewer,

Thank you for your insight comments and suggestions. We have modified the manuscript accordingly. We trust that all of your comments have been addressed accordingly in the revised manuscript. If you have further suggestions for changes, please let us know. The detailed corrections are listed below point by point:

All changes in the manuscript are marked with red color.

**Minor comments**

(1)- Line 14: By mentioning the CHIRP database the University of Santa Barbara needs to be cited as developer. Thus, the sentence has to be changed in: the Climate Hazards Group InfraRed Precipitation (CHIRP, daily 0.05) satellite-derived precipitation developed by the UC Santa Barbara

**Answer:** Thanks. Done. The University of Santa has been cited when mentioning the CHIRP database.

**Change:** changed "the Climate Hazards Group Infrared Precipitation (CHIRP, daily, 0.05°) satellite-derived precipitation estimates" to "the Climate Hazards Group Infrared Precipitation (CHIRP, daily, 0.05°) satellite-derived precipitation developed by the UC Santa Barbara over the Jinsha River Basin for the period of June-July-August in 2016"

(2)- Line 52: When the CHIRPS dataset has been mentioned the developer (UC Santa Barbara et al.) has to cited as well.

**Answer:** Thanks. Done. The University of Santa has been cited when mentioning the CHIRP database.

**Change:** changed "the Climate Hazards Group Infrared Precipitation with Station data (CHIRPS)" to "the Climate Hazards Group Infrared Precipitation with Station data (CHIRPS) developed by the UC Santa Barbara"

(3)- Line 109: Section 2.2 can to be compacted in only 2 subsections for a better reading: 1) precipitation gauged observations and 2) gridded precipitation+CHIRPS

**Answer:** Thanks. Done. Because the gridded precipitation used here was from China

Meteorological Data Service, interpolated from 2472 rain gauge stations. The interpolated data with some errors was less accurate than the direct measurements from stations, for example, daily precipitation was more than 1000 mm at one interpolated grid point. So only the rain gauge observations were used to the new experiments.

**Change:** We have only used rain gauge stations to conduct the WHU-SGCC method over the Jinsha River Basin during the summer seasons from 1990 to 2014. So we delete the relative sections (2.2.2 Gridded precipitation observations; 3.1.3 Rule 2 of the WHU-SGCC method). And we changed the classifications of the target pixels from "1) Classify all regional pixels into five types: C1 (pixel including one gauged station in its area), C2 (pixel including one gridded point), C3 (pixel physically similar to C1C2), C4 (pixel physically similar to C3) and C5 (remaining pixels)." to "Classify all regional pixels into four types: C1 (pixel including one gauge station in its area), C2 (pixel statistically similar to C1), C3 (pixel statistically similar to C2) and C4 (remaining pixels). "

(4)- Line 159: what's "SICR approach"?

**Answer:** Sorry. Thanks. The "SICR" approach must be clerical error.

**Change:** This sentence has been changed "On this basis, the WHU-SGCC method identifies the geographical locations and topographical features of each pixel and applies the classification principles of the SICR approach, including five classification and blending rules." to "On this basis, the WHU-SGCC method identifies the geographical locations and topographical features of each pixel and applies the four classification and blending rules."

(5)- Line 162 "C3 (pixel physically similar to C1C2)". What does it mean "physically"?

**Answer:** Thanks. Some studies indicate that pixels have similar precipitation features in certain spatial scope. And the size of spatial range can be determined by similar geographical location, elevation and other terrain information with the method of fuzzy c-means (FCM) clustering in this study. Because we deleted the gridded precipitation observations and changed the pixels classifications to "Classify all regional pixels into four types: C1 (pixel including one gauge station in its area), C2 (pixel statistically similar to C1), C3 (pixel statistically similar to C2) and C4 (remaining pixels)." So in the new experiments, we can assume that **the C2 pixels have similar precipitation features (e.g. rainfall distribution) with C1 pixels in the same cluster**, which may be better called **statistically simila**r rather than physically similar.

**Change:** We changed "C3 (pixel physically similar to C1C2), C4 (pixel physically similar to C3)" to "C2 (pixel statistically similar to C1), C3 (pixel statistically similar to C2)"

(6)- Line 180 "∶ ∶ ∶satellite precipitation estimations deviated from observed data ∶ ∶ ∶". Really satellite precipitation even though retrieved are always measured data. Thus, it is better replacing the above sentence with: "satellite precipitation estimations deviated from ground-based measurements"

**Answer:** Thanks. Done.

**Change:** changed "satellite precipitation estimations deviated from observed data" to "satellite precipitation estimations deviated from ground-based measurements".

(7)- Section 4 – This section is too much subdivided getting quite difficult the reading. Please, let you group the discussion.

**Answer:** Thanks. Done.

Change: Because of the modification of the WHU-SGCC approach, the section 4 was adjusted accordingly.

Now the section 4 was divided into 3 parts: 4.1 Model performance based on overall accuracy evaluations, 4.2 Model performance based on daily accuracy evaluations and 4.3 Model performance on rain events predictions, which may be simpler for reading.

(8)- Table 6: What's "wet precipitation?" You mean, probably liquid precipitation, right?

**Answer:** Yes. Thanks. It is a good idea to state that the paper focus is on liquid precipitation (rainfall) and this term would be used throughout.

**Change:** changed "wet precipitation" to "liquid precipitation".

And we stated in the introduction "Here, we will use precipitation to name liquid precipitation throughout the text."

**Major comments to Authors**

The proposed manuscript tries to improve the performance of the CHIRP/S datasets by statistically adjusting the original data over complex terrain. The general statistics described in table 5 reveals very light improvements even though WHU-SGCC performs better and CHIRPS dataset seems to be worse also respect to raw data (CHIRP). Skipping to the performance evaluation for rain categories, how do you justify the inversion of BIAS tendency from the category (5,10) to > 40 (see table 6)? The accuracy of WHU-SGCC method seems to be limited to low precipitation (<10, not >20) where the model tends to overestimate. For precipitation greater than 10 the WHU-SGCC starts to underestimate. Please, let u clarify this! Really, the validation of the WHU-SGCC method is only limited to the Jinsha River Basin in summertime 2016 thus new and more accurate validation campaigns have to be done. On that, the challenging efforts to apply and validate a new method over orographically complex terrain have to be supported by new application on similar morphology where the rain-gauges are typically sparse. Furthermore, since during the monsoon season precipitation is typically higher than 20 mm, how the WHU-SGCC will perform? Of course, this question needs to be exhaustively answered by a new validation using the same methodology described in the manuscript.

**Answer:** The CHIRPS was derived from blending in-suit precipitation observations and the CHIRP data, with a modified inverse-distance weighting algorithm at a quasi-global area (land only, 50° S-50° N). The blended data (CHIRPS) has an effective performance on a large scale region according to existing studies, such as at the national scale, but there are still large discrepancies with ground observations at the sub-regional level, especially at the river basin scale. The performance and applicability of CHIRPS at the sub-regional level still need to be validated. What's more, the interpolation performance from the limited and sparse rain gauge stations will be affected by more errors which was evaluated with rain gauge stations shown in Table 5.

As such, due to the poor performance of CHIRPS data at the sub-regional scale and the shortcomings of the modified inverse-distance weighting algorithm, the aim of this article is to offer a novel blending approach to improve the precipitation estimated accuracy at the river basin scale.

The Jinsha River Basin is a typically study area, with the complex and varied terrains that the range of elevation is from 263 to 6575 m above sea level, which results in significant temporal and spatial weather variation within the basin. What's more, the multi-year (1990-2014) average annual precipitation increases from north to south and the spatial distribution of precipitation is uneven, with an average annual precipitation ranging from less than 250 mm to more than 600 mm during the summer seasons over the Jinsha River Basin. However, the number of rain gauges stations is limited inside the river basin which cause precipitation estimations bias a lot.

In the previous experiment, the rain gauge stations and gridded points were used as the reference precipitation data. From that data, the training samples represented 70% of total gauged stations and gridded points, and the remaining data were used to verify the model performance. And the WHU-SGCC approach was evaluated for the Jinsha River Basin for JJA 2016.

However, the gridded precipitation used here was from China Meteorological Data Service, interpolated from 2472 rain gauge stations, which was less accurate than the rain gauge stations observations, for example, daily precipitation was more than 1000 mm at one interpolated grid point. So only the 30 rain gauge stations were used in the new WHU-SGCC experiments. In the new experiment, selecting 30% of the stations for validation was not an appropriate validation method, while the leave-one-out cross validation was a better instead for using all the stations in WHU-SGCC correction algorithm. What's more, in order to evaluate the model performance more reasonably, the study period was changed from summer of 2016, JJA to a longer study period during June-July-August from 1990 to 2014.

In the results, the days of each class of rain events at the validation gauge station during the JJA from 1990 to 2014 were shown in Table 6 in the paper and the following figure. The major rain events inside the Jinsha River Basin were light rain (0.1-10 mm), accounting for 54.76% of the total days (the average percentage of rain event days in its total days at each gauge station), while the days with daily precipitation over the 50 mm was least, only accounting for 0.72%. And the percentage of the daily precipitation of <0.1, 10-25, and 25-50 mm were 26.89%, 14.01% and 3.62% respectively. The result indicated that the average daily precipitation was less than 10 mm, though in the summer seasons during the multi-year. As well as, the spatial distribution of precipitation was also uneven, with an increase from north to south. In terms of performance with respect to different daily rain events, the WHU-SGCC approach had the lowest error, as indicated by RMSE, MAE and BIAS for events with total rainfall less than 25 mm which can represents the mainly precipitation conditions over the

Jinsha River basin.

[Figure]

The WHU-SGCC approach blended daily precipitation gauge data and the CHIRP satellite-derived precipitation, considering the spatial correlation and the historical precipitation characteristics. Therefore, the applicability of the WHU-SGCC method over the complicated mountainous terrain with sparse rain gauge data could be confirmed by the multi-year validation.

It is quite a worthy advice that applying and validating the WHU-SGCC method over the other similar terrain area where the sparse rain-gauges layout. But during the he revision period, our major work was on the study period extension and the method modification, the validation on the other area would be carried on further research.

.

**Reply to Reviewer 2**

**Manuscript ID:** essd-2018-150
**Title:** WHU-SGCC: A novel approach for blending daily satellite (CHIRP) and precipitation observations over Jinsha River Basin
**Journal:** Earth System Science Data
**Type:** Article

Dear Reviewer,

Thank you for your insight comments and suggestions. We have modified the manuscript accordingly. We trust that all of your comments have been addressed accordingly in the revised manuscript. If you have further suggestions for changes, please let us know. The detailed corrections are listed below point by point:

All changes in the manuscript are marked with red color.

**General Comments**
The manuscript describes a very interesting method to correct random and systematic errors from satellite infrared precipitation estimates based on gauge data and grid points from interpolated precipitation fields. The manuscript is well structured and present links to access all the data that is used in the work. However, I found that the analysis period (summer of 2016, JJA) is too short to make significant conclusions about this precipitation dataset. Also, the manuscript could have explored another ways to analyze and evaluate the dataset other than the very conventional use of statistical metrics, that are definitely needed, but the authors could have explored beyond that. Maybe using a case study or exploring the usefulness of the data for a particular hydrological application would be helpful. The dataset published with the manuscript is of good quality and is stored in a standard and easy to read format. I suggest that the authors make another English revision, especially regarding the use of articles and prepositions.
**Answer:**
   Thank you for your advices on the analysis period, method, validation and the syntax modification.
   The WHU-SGCC approach is a promising tool to monitor the summer precipitation over the Jinsha River Basin, considering the spatial correlation and historical precipitation characteristics.
   However, in the previous experiment, the analysis period (summer of 2016, JJA) was too short to make significant conclusions. Therefore, in order to evaluate the model performance more reasonably, the study period was changed from summer of 2016, JJA to a longer study period during June-July-August from 1990 to 2014.
   What's more, the rain gauge stations and gridded points were used as the reference precipitation data in the previous experiment. Now, due to the gridded precipitation interpolated from 2472 rain gauge stations, which was less accurate than the rain gauge stations observations, for example, daily precipitation was more than 1000 mm at one interpolated grid point (This data can be obtained from the China Meteorological Data Service). In hence, only the 30 rain gauge stations were used in the new WHU-SGCC experiments and the validation method was changed from "selecting 30% of the stations for validation" to leave-one-out cross validation for using all the rain gauge stations. Based on this, the WHU-SGCC was also modified for better precipitation correction.

It is quite a worthy advice that using a case study or exploring the usefulness of the data for a particular hydrological application. But the monitoring data for water level and velocity at the gauge stations are not available online, which limits the input data for hydrological model. Nevertheless, we are applying to hydro-graphic office. The validation on a particular hydrological case would be carried on further research.

In the new experiment, the applicability of the WHU-SGCC method over the complicated mountainous terrain with sparse rain gauge data could be confirmed by the multi-year statistical validation over the Jinsha River Basin.

**Specific Comments**

(1)- Line 18: It was evaluated not only by categorical indices.

**Answer:** Thanks. The performance of WHU-SGCC approach was evaluated by multiple error statistics and from different perspectives, such as overall accuracy, daily accuracy and performance on different rain events.

**Change:** So according to your advice, we changed "which is evaluated by categorical indices" to "which is evaluated by multiple error statistics and from different perspectives".

(2)- Line 28: The number of gauge stations is actually very limited, especially in regions with complex terrain and in the case of gauges that measure solid precipitation. Accuracy of this gauges is also not very good in the case of solid precipitation. It would be a good idea to state that the paper focus is on liquid precipitation (rainfall), and use this term throughout the text.

**Answer:** Thanks. It is a good idea to state that the paper focus is on liquid precipitation (rainfall) and this term would be used throughout.

**Change:** We changed "In general, ground-based gauge networks include a substantial number of precipitation observations measured with high accuracy" to "In general, ground-based gauge networks include a substantial number of liquid precipitation observations measured with high accuracy"

**Changed** "As such, the aim of this article is to offer a novel approach for blending daily precipitation gauge data, gridded precipitation data and the Climate Hazards Group Infrared Precipitation (CHIRP) satellite-derived precipitation estimates developed by the UC Santa Barbara, over the Jinsha River Basin." to "As such, the aim of this article is to offer a novel approach for blending daily liquid precipitation gauge data and the Climate Hazards Group Infrared Precipitation (CHIRP) satellite-derived precipitation estimates developed by the UC Santa Barbara, over the Jinsha River Basin."

**Added** "Here, we will use precipitation to name liquid precipitation throughout the text."

(3)- Line 35: I found this line confusing in the way it is phrased. I think this sentence could be phrased this way: "Satellite estimates are susceptible to systematic biases that can influence hydrological modelling."

**Answer:** Thanks. Done.

**Change:** We changed "However, the retrieval algorithms for satellite-based precipitation estimates are susceptible to systematic biases in hydrologic modelling and are relatively insensitive to light rainfall events" to "However, satellite estimates are susceptible to systematic biases that can influence hydrological modelling and the retrieval algorithms are relatively insensitive to light rainfall events".

(4)- In the introduction, I think that a better description of what is available to estimate precipitation from satellites is missing. For example, GOES-R and GPM are missing in the description.

**Answer:** Thanks. Done. The description of GEOS-R and GPM has been added into introduction.

**Change:**

**Added the description into introduction:**

1) The Global Precipitation Measurement (GPM) satellite was launched after the success of the TRMM satellite by the cooperation of National Aeronautics and Space Administration (NASA) and Japan Aerospace Exploration Agency (JAXA) on February 27, 2014 (Mahmoud et al., 2018;Ning et al., 2016). The main core observatory satellite (GPM) cooperates with the ten other satellites (partners) to offer the high spatiotemporal resolution products (0.1° × 0.1°- half- hourly) of the global real-time precipitation estimates (Mahmoud et al., 2019).

2) 2) The Geostationary Operational Environmental Satellite (GOES)-R Series is the geostationary weather satellites, which significantly improves the detection and observation of environmental phenomena. The Advanced Baseline Imager (ABI) onboard the GOES-R platform will provide images in 16 spectral bands, spatial resolution of 0.5 to 2 km (2 km in the infrared and 1–0.5 km in the visible), and full-disk scanning every 5 minutes over the continental United States. The GOES-R Series will offer the enhanced capabilities for satellite-based rainfall estimation and nowcasting (Behrangi et al., 2009;Schmit et al., 2005).

**Added the relevant references:**

Behrangi, A., Hsu, K. L., Imam, B., Sorooshian, S., Huffman, G. J., and Kuligowski, R. J.: PERSIANN-MSA: A Precipitation Estimation Method from Satellite-Based Multispectral Analysis, J. Hydrometeorol., 10, 1414-1429, 10.1175/2009jhm1139.1, 2009.

Mahmoud, M. T., Al-Zahrani, M. A., and Sharif, H. O.: Assessment of global precipitation measurement satellite products over Saudi Arabia, Journal of

Hydrology, 559, 1-12, 10.1016/j.jhydrol.2018.02.015, 2018.

Mahmoud, M. T., Hamouda, M. A., and Mohamed, M. M.: Spatiotemporal evaluation of the GPM satellite precipitation products over the United Arab Emirates, Atmospheric Research, 219, 200-212, 10.1016/j.atmosres.2018.12.029, 2019.

Ning, S., Wang, J., Jin, J., and Ishidaira, H.: Assessment of the Latest GPM-Era High-Resolution Satellite Precipitation Products by Comparison with Observation Gauge Data over the Chinese Mainland, Water, 8, 481-497, doi:10.3390/w8110481,2016.

Schmit, T. J., Gunshor, M. M., Menzel, W. P., Gurka, J. J., Li, J., and Bachmeier, A. S.: Introducing the next-generation Advanced Baseline Imager on goes-R, Bulletin of the American Meteorological Society, 86, 1079-+, 10.1175/bams-86-8-1079, 2005.

(5)- Line 62: That are other sources of uncertainty in the monitoring of rainfall in complex terrain (e.g., orographic enhancement).

**Answer:** Thanks. Done. In existing studies, they found that topography, seasonality, and climate impacted on the satellite-based precipitation estimations performance.

**Change:** We changed "estimations over mountainous areas with complex topography often have large uncertainties and systematic errors due to the sparseness of rain gauges (Zambrano-Bigiarini et al., 2017)" to "estimations over mountainous areas with complex topography often have large uncertainties and systematic errors due to the topography, seasonality, climate impact and sparseness of rain gauges (Derin et al., 2016;Maggioni and Massari, 2018;Zambrano-Bigiarini et al., 2017)" and we added the relevant references.

(6)- Line 94: I was not able to understand the average precipitation over the Yangtze River Basin. You could be more specific about what statistic you are presenting here. Usually what is presented is the spatially averaged annual accumulation of precipitation as an indication of precipitation climatology for the region.

**Answer:** Thanks. Done. We have used the average annual precipitation as an indication of precipitation climatology for the study region.

**Change:**

1) We changed "The river's catchment proximately covers an area of ~180 × $10^4$ km$^2$.In 2016, the average precipitation in the Yangtze River Basin was 12053 mm and the total precipitation was 21478.7195 billion m3, which is 10.9% higher than the annual average total precipitation" to "The river's catchment proximately covers an area of approximately ~180 × $10^4$ km$^2$ and the average annual precipitation is approximately 1100 mm (Zhang et al., 2019)."

2)We changed "Average annual precipitation in the Jinsha River Basin is approximately 3433.45 mm, the total annual precipitation north of Shigu is 937.25 mm, while south of Shigu annual precipitation is 2496.20 mm." to "The average annual precipitation of the Jinsha River Basin is approximately 710 mm, the average annual precipitation of the lower reaches is approximately 900-1300 mm, while the average annual precipitation of the middle and upper reaches is approximately 600-800 mm (Yuan et al., 2018)."

(7)- Line 98: I was not able to comprehend the units for the area of the basin. It should be presented as km2.
**Answer:** Thanks.  The units for watershed area (the area of the basin) are $km^2$.
**Change:** We changed "covering a watershed area of $460 \times 103$ $km^{2}$" to "covering a watershed area of $460 \times 10^3$ $km^{2}$".

(8)- Line 102: Topography would not exert a temporal variation in climate, since this is not a very dynamic feature of the Earth's surface. It could exert a temporal variation in weather though.
**Answer:** Thanks. It is indeed that complex and varied terrains would not exert a temporal variation in climate due to relatively stable feature of Earth's surface. However, a temporal variation in weather would be susceptible to topography.
**Change:** We changed "which results in significant temporal and spatial climate variation within the basin" to "which results in significant temporal and spatial weather variation within the basin"

(9)- Lines 102 and 103: Please try be consistent with the statistics you are using here.
**Answer:** Thanks. Done. We have been consistent with the statistics and used the average annual precipitation as an indication of precipitation climatology for the study region.
**Change:**
We changed "Average annual precipitation in the Jinsha River Basin is approximately 3433.45 mm, the total annual precipitation north of Shigu is 937.25 mm, while south of Shigu annual precipitation is 2496.20 mm." to "The average annual precipitation of the Jinsha River Basin is approximately 710 mm, the average annual precipitation of the lower reaches is approximately 900-1300 mm, while the average annual precipitation of the middle and upper reaches is approximately 600-800 mm (Yuan et al., 2018)."

(10)- Line 134: Maybe explain better what is the CHPClim product.
**Answer:** Thanks. Done. The reference (Funk, C., Verdin, A., Michaelsen, J., Peterson, P., Pedreros, D., and Husak, G.: A global satellite-assisted precipitation climatology, Earth Syst. Sci. Data, 7, 275-287, 10.5194/essd-7-275-2015, 2015) can better explain the CHpClim product. We have added the reference to the sentence (monthly precipitation from CHPClim) and make some changes.
**Change:** We have changed "monthly precipitation from CHPClim" to "monthly precipitation from CHPClim v.1.0 (Climate Hazards Group's Precipitation Climatology version 1) derived from the combination of the satellite fields, gridded physiographic indicators, and in situ climate normal with the geospatial modelling approach based on moving window regressions and inverse distance weighting interpolation (Funk et al., 2015 b)"

(11)- Line 146: Which CHIRPS data? The data from its stations? Or the blended product? Please make this clear here.

**Answer:** Thanks. It was stated from section 2.2.2 that CHIRPS data is a blended product interpolating from CHIRP data and in situ precipitation observations obtained from a variety of sources including national and regional meteorological services (Funk et al., 2014).

**Change:** We changed a more accurate explain for CHIRPS data used for comparisons of precipitation accuracy. Changed "and the corresponding daily CHIRPS data was used for comparisons of precipitation accuracy" to "and the corresponding daily CHIRPS blended data was used for comparisons of precipitation accuracy"

(12)- Line 162: What do you mean about physically similar? Is this means that these pixels are related to others based on its physical attributes (lat, long, elevation, slope, aspect, and curvature)? Or this means that is similar is terms of rainfall distribution? If is in terms of rainfall, I think a better world would be statistically similar rather than physically similar, since this is based on a cluster analysis.

**Answer:** Thanks. We assumed that the C2 pixels have similar precipitation features (e.g. rainfall distribution) with C1 pixels in the same cluster, which may be better called statistically similar rather than physically similar.

**Change:** We changed "C3 (pixel physically similar to C1C2), C4 (pixel physically similar to C3)" to "C2 (pixel statistically similar to C1), C3 (pixel statistically similar to C2)"

(13)- Line 170: Please explain why you chose 30% of the stations/grid points for validation. Since the stations and grid points are of limited number, it would not be better to do a Bootstrap validation instead? Then you are able to use all the stations/grid points in your correction algorithm.

**Answer:** Thanks. The number of rain gauge stations over the Jinsha River Basin is limited. And because the gridded precipitation used here was from China Meteorological Data Service, interpolated from 2472 rain gauge stations, which was less accurate than the rain gauge stations observations, for example, daily precipitation was more than 1000 mm at one interpolated grid point. So only the 30 rain gauge stations were used to the new experiments. In the new experiment, selecting 30% of the stations for validation was not an appropriate validation method, while the leave-one-out cross validation was a better instead for using all the stations in WHU-SGCC correction algorithm. And the analysis period was changed from (summer of 2016, JJA) to "during the JJA from 1990 to 2014".

**Change:** We changed the validation method from "The proposed approach was evaluated for the Jinsha River Basin for JJA 2016. From that data, the training samples represented 70% of total gauged stations and gridded points, and the remaining data were used to verify the model performance." to "The proposed approach was evaluated over the Jinsha River Basin based on 30 gauge stations and CHIRP satellite-based precipitation estimations during the JJA from 1990 to 2014. The leave-one-out cross validation step was applied to computing the out-of-sample adjusted error with gauge stations."

(14)- Lines 224 and 225: Since your method relies heavily on the cluster algorithm, it would not be better to use some sort of statistical metric to define the number of clusters?
**Answer:** Thanks. Done. The optimum number of clusters was determined by L(c) which was derived from the inter-distance and inner-distance of samples in the following equation. It is ensured that the distance between the same samples is smaller, while the distance between the different samples is larger.

$$L(c) = \frac{\sum\limits_{i=1}^{c}\sum\limits_{j=1}^{n} w_{ij}^{m} \| c_i - \bar{x} \|^2 /(c-1)}{\sum\limits_{i=1}^{c}\sum\limits_{j=1}^{n} w_{ij}^{m} \| x_j - c_i \|^2 /(n-c)}$$

In that equation, the denominator is inner-distance and the molecular is inter-distance. The initial value of c is 1 and the maximum value of c is the number of gauge stations in this study area. The optimum number of clusters was optimized to maximize the L(c). For this reason, c value is conducted in the range from 1 to the number of gauge stations with an incremental interval value of 1 in this study. This result was shown in Appendix B.
**Change**: We added the L(c) metric to determine the optimum number of clusters and the Fig. B1 (The optimum number of clusters determined by the maximum L(c) with the iterative process) was given. Based on this, the optimum number of clusters was set 22 in this study.

[Figure]

**Figure** The optimum number of clusters determined by the maximum $L(c)$ with the iterative process.

(15)- Line 234: Pixels should also be similar regarding precipitation characteristics, is that right?
**Answer:** Thanks. Some studies indicate that pixels have similar precipitation futures in certain spatial scope. And the size of spatial range can be determined by similar geographical location, elevation and other terrain information with the method of fuzzy c-means (FCM) clustering in this study. Therefore, in each cluster, pixels both have the similar terrain features and precipitation characteristics.

**Change:** We changed "Pixels in each cluster have similar terrain features" to "Pixels in each cluster have similar terrain features and precipitation characteristics".

(16)- Line 244: I think you should use the word relationship or correlation instead of confidence in this line.

**Answer:** Thanks. Done.

**Change:** We changed "confidence is not only determined by the value of the correlation coefficient but also from the correlation test's $p$ value" to "correlation is not only determined by the value of the correlation coefficient but also from the correlation test's $p$-value".

(17) Line 259: Why the number of decision trees was set to 500?

**Answer:** Thanks. Done. The number of decision trees was set to 500, which was determined by out-of-bag (OOB) error (Appendix A). The OOB error reached the minimum value when the number of decision trees was less than 500.

**Change**: We calculated the OOB-error of Random forest regression with the increase of the number of decision trees from 1 to 500 at each rain gauge station and the Fig. A1 shown the change of out-of-bag (OOB) error with the number of decision trees increase in appendix.

(18)- Line 264: Do C1 and C2 pixels included in the R pixel category?

**Answer:** No. Thanks. In the previous experiment, C1 and C2 pixels were not included in the R pixels. Now, in the new experiment, we changed the classification of C2, C3 and C4 pixels. Because the gridded precipitation used here was from China Meteorological Data Service, interpolated from 2472 rain gauge stations. The interpolated data with some errors was less accurate than the direct measurements from stations, for example, daily precipitation was more than 1000 mm at one interpolated grid. So only the rain gauge observations were used to the new experiments. And we changed the classifications of the target pixels from "1) Classify all regional pixels into five types: C1 (pixel including one gauged station in its area), C2 (pixel including one gridded point), C3 (pixel physically similar to C1C2), C4 (pixel physically similar to C3) and C5 (remaining pixels)." to "Classify all regional pixels into four types: C1 (pixel including one gauge station in its area), C2 (pixel statistically similar to C1), C3 (pixel statistically similar to C2) and C4 (remaining pixels). "

**Change:** So with the new experiment, we changed "pixels in each cluster represent potential C3 pixels, with exception of the C1 and C2 pixels and are called R pixels" to "With exception of the C1 pixels, the remaining pixels in each cluster represent potential C2 pixels called R pixels"

(19)- Lines 267 and 280: Are you also considering only SCC values with p-value lower than 0.05 here?

**Answer:** Yes. Thanks. The correlation coefficient value higher than 0.5 and the p-value lower than 0.05 were considered for C2 pixels selection.

In the new experiment, we changed the statistical metric of SCC to Pearson's correlation coefficient (PCC) because PCC measures the linear correlation between two series better than SCC to evaluate the adjusted precipitation accuracy.

**Change**: Changed "both the data with a maximum SCC of at least 0.5 and the corresponding index of C1 and C2 pixels" to "both the data with a maximum PCC of at least 0.5 and a p-value lower than 0.05 (Zhang and Chen, 2016)"

(20)- Line 287: Why did you choose the 10 mm value? Could you please explain the meaning of this constant better?

**Answer:** Thanks. In Eq.(11), the relationship between C2 pixels and the corresponding CGURP grid cells is expressed by the ratio:

$$w_i = \frac{C2_{as_i} + \lambda}{Y_{s_i} + \lambda} \quad i=1,...,n \tag{11}$$

where $\lambda$ is a positive constant to avoid the denominator value being 0 when CHIRP grid cell value was 0. (Sokol, 2003) tested various $\lambda$ values indicated that the selected value was not too closely related to the calibration set.

In Rule 3, the values of C3 pixels are derived from Eq. (12):

$$C3_{as} = \max(w \times (Y_s + \lambda) - \lambda, 0) \tag{12}$$

In this equation, the $\lambda$ set to 10 mm made the calculating simpler, and other values are also available.

(21)- Line 298: Please use the actual percentage here.

**Answer:** Thanks. In the previous experiment, each C5 pixel value is set to be the same as the CHIRP grid cell value at the corresponding position, because of the few number of the C5 pixels. However, in the new experiment, we abandoned the gridded precipitation observations and only the 30 rain gauge stations with four rules were used to conduct the WHU-SGCC approach. The C5 pixels were changed to C4 pixels for Rule 4 and the percentage of C4 pixels is around 60% of the total number of pixels over the study area. (Due to the leave-one-out cross validation step, the different training samples will have the different number of C2, C3 and C4 pixels respectively inside the Jinsha River Basin). So that, the Inverse Distance Weighted (IDW) method was used to obtain the C4 pixels values.

**Change:** We added the Table 4 (The number of each class pixels adjusted by each rule using the WHU-SGCC method inside the Jinsha River Basin.) which lists the clearer number and percentage of each class pixels. And the Fig.4 in the previous paper was deleted.

**Table 4** The number of each class pixels adjusted by each rule using the WHU-SGCC method inside the Jinsha River Basin.

| Validation gauge station | C1 Pixels (%) | C2 Pixels (%) | C3 Pixels (%) | C4 Pixels (%) |
|---|---|---|---|---|
| 52908 | 29 (0.16%) | 3066 (16.59%) | 4224 (22.85%) | 11163 (60.40%) |
| 56004 | 29 (0.16%) | 2882 (15.59%) | 4111 (22.24%) | 11460 (62.01%) |
| 56021 | 29 (0.16%) | 3311 (17.91%) | 4510 (24.40%) | 10632 (57.53%) |
| 56029 | 29 (0.16%) | 3338 (18.06%) | 4447 (24.06%) | 10668 (57.72%) |
| 56034 | 29 (0.16%) | 3300 (17.86%) | 4427 (23.95%) | 10726 (58.03%) |
| 56038 | 29 (0.16%) | 3209 (17.36%) | 4014 (21.72%) | 11230 (60.76%) |
| 56144 | 29 (0.16%) | 3347 (18.11%) | 4442 (24.03%) | 10664 (57.70%) |
| 56146 | 29 (0.16%) | 3183 (17.22%) | 4480 (24.24%) | 10790 (58.38%) |
| 56152 | 29 (0.16%) | 3173 (17.17%) | 4176 (22.59%) | 11104 (60.08%) |
| 56167 | 29 (0.16%) | 3362 (18.19%) | 4346 (23.51%) | 10745 (58.14%) |
| 56247 | 29 (0.16%) | 3385 (18.32%) | 4416 (23.89%) | 10652 (57.63%) |
| 56251 | 29 (0.16%) | 3301 (17.86%) | 4348 (23.53%) | 10804 (58.46%) |
| 56257 | 29 (0.16%) | 3313 (17.93%) | 4043 (21.88%) | 11097 (60.04%) |
| 56357 | 29 (0.16%) | 3352 (18.14%) | 4390 (23.75%) | 10711 (57.95%) |
| 56374 | 29 (0.16%) | 3341 (18.08%) | 4294 (23.23%) | 10818 (58.53%) |
| 56459 | 29 (0.16%) | 3345 (18.10%) | 4334 (23.45%) | 10774 (58.29%) |
| 56462 | 29 (0.16%) | 3380 (18.29%) | 4377 (23.68%) | 10696 (57.87%) |
| 56475 | 29 (0.16%) | 3345 (18.10%) | 4344 (23.50%) | 10764 (58.24%) |
| 56479 | 29 (0.16%) | 3305 (17.88%) | 4212 (22.79%) | 10936 (59.17%) |
| 56485 | 29 (0.16%) | 3393 (18.36%) | 4419 (23.91%) | 10641 (57.57%) |
| 56543 | 29 (0.16%) | 3373 (18.25%) | 4384 (23.72%) | 10696 (57.87%) |
| 56565 | 29 (0.16%) | 3241 (17.54%) | 4450 (24.08%) | 10762 (58.23%) |
| 56571 | 29 (0.16%) | 3306 (17.89%) | 4263 (23.07%) | 10884 (58.89%) |
| 56586 | 29 (0.16%) | 3387 (18.33%) | 4434 (23.99%) | 10632 (57.53%) |
| 56651 | 29 (0.16%) | 3340 (18.07%) | 4432 (23.98%) | 10681 (57.79%) |
| 56664 | 29 (0.16%) | 3368 (18.22%) | 4262 (23.06%) | 10823 (58.56%) |
| 56666 | 29 (0.16%) | 3323 (17.98%) | 4431 (23.97%) | 10699 (57.89%) |
| 56671 | 29 (0.16%) | 3356 (18.16%) | 4367 (23.63%) | 10730 (58.06%) |

(22)Line 326: You should be clearer on what the numbers of pixels are here. Is this the number of pixels inside the basin multiplied by the number of days? It would be a good idea to describe the exact number of pixels for each class along with its actual percentage in the text.

**Answer:** Thanks. Done.

**Change:** We added the Table 4 (The number of each class pixels adjusted by each rule using the WHU-SGCC method inside the Jinsha River Basin.) with the exact number of pixels for each class along with its actual percentage. And the Fig.4 in the previous paper was deleted.

The sentence was added into the paper to describe the number and the percentage of each class pixels inside the basin "The number of C1 pixels was the number of training gauge stations accounting 0.16% of the total pixels (18482) inside the basin. Due to the leave-one-out cross validation step, the different training samples will have the different number of C2, C3 and C4 pixels respectively inside the Jinsha River Basin. The number of C4 pixels was approximately 10822 with the percentage around 60%, the number of C3 pixels was approximately 4331 with the percentage ranging from 21.72% to 24.40%, and the number of C2 pixels was approximately 3300 with the percentage ranging from 15.59% to 18.36%."

(23)- Line 341: It would be better to use the same x axis scale in both plots. It seems that the gridded data observation has similar biases as CHIRP and thus their CDFs are more similar, providing less improvement in the adjusted dataset.
**Answer:** Thanks. According to the new experiment, we changed the Rule 1for the C1 pixels without Adj-QM, so the CDFs were not needed.
**Change:** We changed the Rule 1 from Adj-QM to establishing the regression relationships between each gauge historical observations and the corresponding CHIRP grid cell value by means of Random Forest Regression. And we deleted the relative sections 4.1 and 4.2 in the previous paper. So the section 4: Results and Discussion only include 4.1 Model performance based on overall accuracy evaluations, 4.2 Model performance based on daily accuracy evaluations, and 4.3 Model performance on rain events predictions.

(24)- Line 355: It would be a good idea to explore and discuss more the statistics presented in Figure 7.
**Answer:** Because of the multi-year period studied in the new experiment, we modified the WHU-SGCC method. In the new experiment, due to the leave-one-out cross validation step using all the stations, the performance of WHU-SGCC method would be evaluated on the overall accuracy, not on a certain class of pixels. So we didn't evaluate the C3 pixels separately.
**Change:** The evaluation of C3 pixels and Figure 7 were deleted.

(25)- Line 373: NSE values have increased, but still not very good (i.e., still negative).
**Answer:** Thanks. The NSE (Nash and Sutcliffe, 1970) determines the relative magnitude of the variance of the residuals compared to the variance of the observations, bounded by minus infinity to 1.
**Change:** In the new experiment, the NSE of WHU-SGCC method was -0.0137 with an increase of 93.33% and 98.32% to CHIRP and CHIRPS, respectively.
Although the NSE of WHU-SGCC was far less than 1, it was improved to be 0 that indicates the adjusted results were close to the average level of the rain gauge observations, while the NSEs of CHIRP and CHIRPS were much worse.

(26)- Line 376: Is not intuitive that the evaluation metrics are better for CHIRP than CHIRPS, since CHIRPS adds stations data to their dataset. Could you please clarify this? I am seeing here that magnitude evaluation metrics have not changed considerably, probably because the improvement is seem in the low magnitude events. SCC has a considerable increase, but still cannot explain much of the variability of rainfall in the region. POD values are good.

**Comments 1):** Is not intuitive that the evaluation metrics are better for CHIRP than CHIRPS, since CHIRPS adds stations data to their dataset. Could you please clarify this?

**Answer:** The CHIRPS was derived from blending in-suit precipitation observations and the CHIRP data, with a modified inverse-distance weighting algorithm at a quasi-global area (land only, 50° S-50° N). The blended data (CHIRPS) has an effective performance on a large scale region, such as at the national scale, but there are still large discrepancies with ground observations at the sub-regional level, especially at the river basin scale. The performance and applicability of CHIRPS at the sub-regional level still need to be validated. What's more, the interpolation performance from the limited and sparse rain gauge stations will be affected by more errors which was evaluated with rain gauge stations shown in Table 5.

As such, due to the poor performance of CHIRPS data at the sub-regional scale and the shortcomings of the modified inverse-distance weighting algorithm, the aim of this article is to offer a novel blending approach to improve the precipitation estimated accuracy at the river basin scale.

**Change:** We changed the sentence from "As such, the aim of this article is to offer a novel approach for blending daily precipitation gauge data, gridded precipitation data and the Climate Hazards Group Infrared Precipitation (CHIRP) satellite-derived precipitation estimates over Jinsha River Basin." to "As such, due to the poor performance of CHIRPS data at the sub-regional scale and the shortcomings of the existing blending algorithms, the aim of this article is to offer a novel approach for blending daily liquid precipitation gauge data and the Climate Hazards Group Infrared Precipitation (CHIRP) satellite-derived precipitation estimates developed by the UC Santa Barbara, over the Jinsha River Basin." for better explanation.

**Comments 2):** I am seeing here that magnitude evaluation metrics have not changed considerably, probably because the improvement is seem in the low magnitude events. SCC has a considerable increase, but still cannot explain much of the variability of rainfall in the region. POD values are good.

**Answer:** In the previous experiment, the training samples represented 70% of total gauged stations and gridded points, and the remaining data were used to test model performance. This validation was not able to use all the rain gauge stations and the same validation set may not fully explain the performance on the Jinsha River Basin. As such, in the presented experiment, the leave-one-out cross validation step was a better instead for using all the stations in WHU-SGCC correction algorithm.

What's more, the analysis period (summer of 2016, JJA) is too short to make significant conclusions about this precipitation dataset, so we changed the period during June-July-August from 1990 to 2014, 92 days per year for 25 years totally.

**Change:** We use the leave-one-out cross validation to instead the fixed training and testing sets. And we also changed a longer study period during June-July-August from 1990 to 2014, to evaluate the model performance on different rainfall events.

In the results, the days of each class of rain events at the validation gauge station during the JJA from 1990 to 2014 were shown in Table 6 in the paper and the following figure. The major rain events inside the Jinsha River Basin were light rain (0.1-10 mm), accounting for 54.76% of the total days (the average percentage of rain event days in its total days at each gauge station), while the days with daily precipitation over the 50 mm was least, only accounting for 0.72%. And the percentage of the daily precipitation of <0.1, 10-25, and 25-50 mm were 26.89%, 14.01% and 3.62% respectively. The result indicated that the average daily precipitation was less than 10 mm, though in the summer seasons during the multi-year. As well as, the spatial distribution of precipitation was also uneven, with an increase from north to south. In terms of performance with respect to different daily rain events, the WHU-SGCC approach had the lowest error, as indicated by RMSE, MAE and BIAS for events with total rainfall less than 25 mm which can represents the precipitation conditions over the Jinsha River basin.

[Figure]

(27)- Line 383: Could this means that because the daily precipitation in the lowland region of the basin is higher, the RMSE values are also higher?

**Answer:**

In terms of performance with respect to different daily rain events, the WHU-SGCC approach had the lowest error, as indicated by RMSE, MAE and BIAS for events with total rainfall lower than and 25 mm, but WHU-SGCC performance for total rainfall higher than 25mm did not improve compared to CHIRP and CHIRPS (Table 7), though it was better than that of CHIRPS. This negative performance on the total rainfall higher than 25 mm was probably caused by the precipitation conditions inside the Jinsha River Basin (Table 6). The average daily precipitation was less than 10 mm inside the basin, during the multi-year summer seasons, which provided a large amount of rain gauge stations data with the values lower than 10 mm, that caused a significantly impact on the statistical relationships establishment for WHU-SGCC. In hence, the approach of WHU-SGCC is applicable for the detection of rainfall events over the Jinsha River Basin, with the average daily precipitation less than 10 mm, or even than 25mm. Due to the 4.34% of summer days with the daily precipitation over the 25 mm, the performance of WHU-SGCC on these rain events was poorer than the results of CHIRP and CHRPS.

(28)- Line 388: The fact that your method performs well in complex terrain is a very positive point in your manuscript, but you will need a longer study period to confirm this finding.

**Answer:** Thanks. Done.

**Change:** we changed the study period from summer of 2016, JJA to a longer study period during June-July-August from 1990 to 2014, to evaluate the model performance more reasonably.

(29)- Line 402: The boxplots do not show that the higher reduction is seen in the Bias metric.

**Answer:** We redraw the boxplots of the statistical analysis of the agreement between daily observations and WHU-SGCC, CHIRP and CHIRPS estimates on leave-one-out cross validation during the JJA from 1990 to 2014.

And now the section 4 was divided into 3 parts: 4.1 Model performance based on overall accuracy evaluations, 4.2 Model performance based on daily accuracy evaluations and 4.3 Model performance on rain events predictions.

**Change:** Redraw the boxplots.

The slight reduction was reflected in the BIAS, with an 8% to 45% reduction compared to CHIRP and CHIRPS, while all the values were concentrated between -0.5 and 0.5. Therefore, all the precipitation estimations derived from WHU-SGCC, CHIRP, and CHIRPS represented well agreement with the observations in relative bias.

(30)- Line 417: I am still confused about which rainfall value is presented is this figure. It does not seem realistic that the average daily rainfall would be ~ 200 mm. Please be more specific about which rainfall statistic you are using in this figure.

**Answer:** Thanks. This may the error in gridded precipitation observations.

Because the gridded precipitation used in pervious experiment was from China Meteorological Data Service, interpolated from 2472 rain gauge stations. The interpolated data with some errors was less accurate than the direct measurements from stations, for example, daily precipitation was more than 1000 mm at one interpolated grid.

**Change:** We have only used 30 rain gauge stations to conduct the WHU-SGCC method over the Jinsha River Basin during the summer seasons from 1990 to 2014.

(31)- Line 430: The evaluation metrics for the threshold values are quite similar among the three precipitation products. Just because the values for WHU-SGCC become slightly worse for precipitation higher than 20 mm/day does not mean they are significantly different from the other products. Since the values are very similar, I would suggest to test their differences statistically before making the assumption that WHU-SGCC works better for low magnitude events. This might also be caused by the limitation of the short study period. There is a tendency to use the word scale for a temporal dimension, better to use period instead. I think it would be interesting to add a map with the accumulated precipitation for the study period. The analysis period of days is too short to make conclusive assumptions about the dataset usefulness in the region, which are made several times throughout the text. I understand why the authors want to focus on the summer months to avoid the higher biases introduced by solid precipitation, but I did not understand why they only performed the evaluation for one summer season. Is there a particular reason for that? I still think that is hard to make the conclusions you made based on 92 days, if you add more seasons to the analysis this can become a very interesting manuscript and dataset.

**Answer:** Thanks. In the previous experiment, the analysis period (summer of 2016, JJA) is too short to comment about this precipitation dataset.

**Changed:**

1) We added more summer seasons from 1990 to 2014, 92 days per year for 25 years totally to make a more reasonable conclusive.

2) And we added a map with the multi-year (1990-2014) average annual precipitation (Fig. 2). The multi-year average annual precipitation increases from north to south and the spatial distribution of precipitation is uneven, with an average annual precipitation ranging from less than 250 mm to more than 600 mm during the summer seasons over the Jinsha River Basin.

[Figure]

**Figure 2** The multi-year (1990-2014) average annual precipitation during JJA over the Jinsha River Basin. 30 rain stations were provided by the China Meteorological Administration stations, the other 18 CHIRPS fusion stations were provided by the Climate Hazards Group UC Santa Barbara online at ftp://ftp.chg.ucsb.edu/pub/org/chg/products/CHIRPS-2.0/diagnostics/global_monthly_station_density/tifs/p05/ (last access: 10 December, 2018).

3) The result indicated that the average daily precipitation was less than 10 mm, though in the summer seasons during the multi-year. As well as, the spatial distribution of precipitation was also uneven, with an increase from north to south. In terms of performance with respect to different daily rain events, the WHU-SGCC approach had the lowest error, as indicated by RMSE, MAE and BIAS for events with total rainfall less than 25 mm which can represents the precipitation conditions over the Jinsha River basin.

According to the comparison, the WHU-SGCC approach achieves error reductions for the RMSE, MAE and BIAS statistics for rain events less than 25 mm. Specifically, compared with CHIRP, the RMSE value was reduced by approximately by 5.92%-39.44%, the MAE value by 4.28%-12.41%, and the absolute   BIAS value by 9.15%-44.43%; compared with CHIRPS, the RMSE and MAE values were reduced by 11.04%-56.61%, and the absolute BIAS value by 23.77% -59.58%.

**Table 7** Accuracy assessment on liquid precipitation events during the JJA from 1990 to 2014

| Rain Event | RMSE | | | MAE | | | BIAS | | |
|---|---|---|---|---|---|---|---|---|---|
| | WHU-SGCC | CHIRP | CHIRPS | WHU-SGCC | CHIRP | CHIRPS | WHU-SGCC | CHIRP | CHIRPS |
| <0.1 | 4.7253 | 5.0802 | 7.1643 | 2.5927 | 2.9562 | 2.9145 | / | / | / |
| [0.1,10) | 4.1661 | 6.8684 | 9.6022 | 3.9885 | 4.5534 | 6.2462 | 0.8021 | 1.4435 | 1.9842 |
| [10,25) | 10.4281 | 11.0848 | 13.4427 | 9.2722 | 9.6866 | 11.5909 | -0.5762 | 0.6342 | 0.7559 |
| [25,50) | 25.7494 | 24.5600 | 25.4975 | 24.8386 | 23.0967 | 23.4927 | -0.7784 | 0.7250 | 0.7388 |
| ≥50 | 56.6072 | 54.5037 | 52.7875 | 54.4168 | 52.1557 | 49.4318 | -0.8861 | 0.8297 | 0.7852 |

(32)- The dataset seems to be of good quality. A few comments about it are the following. For a spatial extent of this magnitude I think it would be better to use a geographic coordinate system, rather than a Mercator projection. There is also some artifacts (0 precipitation values) that appear at the same location at multiple days. I was wondering if this is a limitation from the negative values of Rule 4. Is there any way to correct this? This dataset could be very useful if its period is extended to multiple years.
**Answer:** Thanks. The average annual precipitation of the Jinsha River Basin is less and the spatial distribution of precipitation is uneven, with an average annual precipitation ranging from less than 250 mm to more than 600 mm during the summer seasons. So there are also possible no rain in some locations at multiple days over the north of Jinasha River Basin. The negative values derived from Rule 3 (in the new experiment, the Rule 4 was changed to the Rule 3) were not too closely related to the zero precipitation values appearing at the same location at multiple days.
**Change:** The results images of the WHU-SGCC method were changed from "Mercator projection" to "geographic coordinate system: WGS_84"

**Technical Corrections**
(1) Line 2: Change "over Jinsha" for "over the Jinsha".
**Answer:** Thanks. Done.
**Change:** We changed "over Jinsha" to "over the Jinsha".

(2) Line 31: Change "distributed" for "spatial distribution".
**Answer:** Thanks. Done.
**Change:** We changed "their uneven distributed" to "their uneven spatial distribution".

(3) Line 37: "without adjustment" is mentioned twice.

**Answer:** Thanks. Deleted one "without adjustment".

**Change:** We changed "Without adjustments, inaccurate satellite-based precipitation estimates without adjustment will lead to unreliable assessments of risk and reliability" to "Without adjustments, inaccurate satellite-based precipitation estimates will lead to unreliable assessments of risk and reliability"

(4) Line 68: In table 1, it should be written "PERSIANN-CDR" instead of "PRESSIANN-CDR".

**Answer:** Thanks. Done.

**Change:** We changed "PRESSIANN-CDR" to "PERSIANN-CDR"

(5) Line 84: Change "in summer 2016" for "in the summer of 2016".

**Answer:** Thanks. Done.

**Change:** We added the seasons in analysis period, so we changed "in summer 2016" to "over the Jinsha River Basin during the summer seasons from 1990 to 2014"

(6) Line 93: Change "proximately covers an area" for "covers an area of approximately".

**Answer:** Thanks. Done.

**Change:** We changed "The river's catchment proximately covers an area of ~180 × $10^4$ km$^2$" to "The river's catchment covers an area of approximately ~180 × $10^4$ km$^2$"

(7) Line 95: Change "sub-regions" for "sub-basins".

**Answer:** Thanks. Done.

**Change:** We changed "sub-regions" to "sub-basins"

(8) Line 136: Change "precipitation observations" for "surface based precipitation observations".

**Answer:** Thanks. Done.

**Change:** We changed "precipitation observations" to "surface based precipitation observations".

(9) Line 157: Change "other pixels" for "the remaining pixels".

**Answer:** Thanks. Done.

**Change:** We changed "other pixels" to "the remaining pixels".

(10) Line 159: The acronym "SIRC" meaning was not mentioned before.

**Answer:** Sorry. Thanks. The "SICR" approach must be clerical error.

**Change:** This sentence has been changed "On this basis, the WHU-SGCC method identifies the geographical locations and topographical features of each pixel and applies the classification principles of the SICR approach, including five classification and blending rules." to "On this basis, the WHU-SGCC method identifies the geographical locations and topographical features of each pixel and applies the five classification and blending rules."

(11) Line 169: The first sentence could be placed before the item 1.

**Answer:** Thanks. Done.

**Change:** We changed this sentence into the first phase in section 3.1, as the reference to the overview of items 1-4. And we changed the validation method from "The proposed approach was evaluated for the Jinsha River Basin for JJA 2016. From that data, the training samples represented 70% of total gauged stations and gridded points, and the remaining data were used to verify the model performance." to "The proposed approach was evaluated for over the Jinsha River Basin based on 30 gauge stations and CHIRP satellite-based precipitation estimations during the JJA from 1990 to 2014. The leave-one-out cross validation step was applied to computing the out-of-sample adjusted error with gauge stations."

(12) Line 171: Is the same phrase that is shown in line 163.

**Answer:** Thanks. Line 171 and line 163 are repeated.

**Change:** We deleted the repeated phrase in line 163.

(13) Line 172: The flowchart: CHIRP resolution should be 0.05 x 0.05. In the first box of rule 3, change "and gauged" for "with gauged". In the last box of rule 3, change "can derive" for "can be derived". In rule 4, change "ration" for "ratio" (this happens twice here).

**Answer:** Thanks. Done. Because we changed the rules of WHU-SGCC, the flowchart was redrawn.

**Change:** In the flowchart, we changed the CHIRP resolution from "$0.5° \times 0.5°$" into "$0.05° \times 0.05°$"

changed "can derive" to "can be derived"

changed "ration" to "ratio" (two modifications)

**The modified flowchart is as follows:**

[Figure]

(14) Line 189: Change "as" for "in".
**Answer:** Thanks. Done.
**Change:** We changed the "as Eq. (2)" to "in Eq. (2)".

(15) Line 246: Change "p" for "p-value".
**Answer:** Thanks. Done.
**Change:** We changed the "*p* value" to "p-value".

(16) Line 283: The word "method" is repeated twice.
**Answer:** Thanks. Done.
**Change:** We deleted the repeated word "method". Changed "a method for merging method the CHIRP grid cell values" to "a method for merging the CHIRP grid cell values"

(17) Line 300: Change "for summer (JJA) 2016" for "for the summer (JJA) of 2016".
**Answer:** Thanks. Done.
**Change:** We changed the "for summer (JJA) 2016" to "for the summer (JJA) of 2016".

(18) Line 315: Change "as" for "in".
**Answer:** Thanks. Done.
**Change:** We changed the "All of the accuracy assessment indices are shown as Table 3" to "All of the accuracy assessment indices are shown in Table 3".

(19) Line 326: Change "to be adjusted" for "adjusted".
**Answer:** Thanks. Done.
**Change:** We changed the "There were 18482 daily pixels to be adjusted" to "There were 18482 daily pixels adjusted".

(20) Line 340: Change "study" for "studies".
**Answer:** Thanks. Done.
**Change:** We changed the "supports further study" to "supports further studies".

(21) Line 400: Change "with especially greatly decreases compared to CHIRPS" for "with greater decreases when compared to CHIRPS".
**Answer:** Thanks. Done.
**Change:** We changed the "with especially greatly decreases compared to CHIRPS" to "especially the greater decreases when compared to CHIRPS".

(22) Line 451: Change "in summer 2016" for "in the summer of 2016".
**Answer:** Thanks. Done.
**Change:** We changed the "in summer 2016" to "in the summer from 1990 to 2014".

(23) Line 456: Change "over region has" for "over a region that has".
**Answer:** Thanks. Done.
**Change:** We changed the "over region has" to "over a region that has".

(24) Line 465: Change "of the precipitation region" for "of precipitation events in the region".
**Answer:** Thanks. Done.
**Change:** We changed the "of the precipitation region" to "the large size of light precipitation events with short duration rainstorms in the region resulted in a limited improvement in accuracy".

(25) Line 466: Change "short" for "short duration".
**Answer:** Thanks. Done.
**Change:** We changed the "during short rainstorms" to "with short duration rainstorms".

(26) Line 468: Change "complicated mountainous" for "complex terrain".
**Answer:** Thanks. Done.
**Change:** We changed the "complicated mountainous region" to "complex terrain".

(27) Line 480: Change "topographic and long time series climatic factors" for "topographic factors and longer time series".
**Answer:** Thanks. Done. Due to the longer time series data has been taken into the new experiment, the future development was changed.
**Change:** We changed the "topographic and long time series climatic factors" to "more climatic factors and mulit-model ensemble".

**Reply to Reviewer 3**

**Manuscript ID:** essd-2018-150
**Title:** WHU-SGCC: A novel approach for blending daily satellite (CHIRP) and precipitation observations over Jinsha River Basin
**Journal:** Earth System Science Data
**Type:** Article

Dear Reviewer,

Thank you for your insight comments and suggestions. We have modified the manuscript accordingly. We trust that all of your comments have been addressed accordingly in the revised manuscript. If you have further suggestions for changes, please let us know. The detailed corrections are listed below point by point:

All changes in the manuscript are marked with red color.

The manuscript presents a new method for combining high-resolution daily satellite precipitation estimates with rain gauge observations. The method is applied and evaluated over the Jinsha River Basin for the summer period in 2016 (June, July August). The performance of the method is compared to already existing satellite datasets CHIRP, which is also the base for the new dataset, and CHIRPS. The evaluation reveals an improvement in accuracy of precipitation estimates with rain rates of less than 20 mm per day compared to CHIRP and CHIRPS, however, the chosen time period of just 3 months seems to be rather short for this somewhat general conclusion. For heavy precipitation, however, no improvement could be found. The dataset and the blending method are described and the data is available for free.
The manuscript fits in the scope of ESSD, but some issues need to be addressed. I recommend taking the following suggestions and comments into account:

1.

(1)- It is not quite clear to me what exactly is the reference dataset in this study. On page 6, line 170 the authors state that 70% of the total gauged stations and gridded points were used as the training dataset and the remaining 30% serve as reference dataset. How was decided which station / grid point was used for training and which station / grid point was used for evaluation? As I understand it is a mixture between actual station measurements and gridded, i.e. interpolated, station data. Is the ratio for both data types also 70% training and 30% reference data points? Is there a difference in performance metrics when only one of the two datasets is used for evaluation? Direct measurements from stations might be even more accurate than the interpolated data.
**Answer:** In the previous experiment, the 30 rain gauge stations and 170 gridded points were used as the "true" precipitation values. However, the gridded precipitation data was from China Meteorological Data Service, interpolated from 2472 rain gauge stations, which was less accurate than the direct measurements from stations, for example, daily precipitation was more than 1000 mm at one interpolated grid point. So only the rain gauge observations were used to the new experiments. What's more, selecting 30% of the stations for validation was not an appropriate validation method, while the leave-one-out cross validation step was a better instead for using all the stations in WHU-SGCC correction algorithm

**Change:** We have only used 30 rain gauge stations as the reference precipitation values to conduct the WHU-SGCC method. We changed from "The proposed approach was evaluated for the Jinsha River Basin for JJA 2016. From that data, the training samples represented 70% of total gauged stations and gridded points, and the remaining data were used to verify the model performance." to "The proposed approach was evaluated over the Jinsha River Basin based on 30 gauge stations and CHIRP satellite-based precipitation estimations during the JJA from 1990 to 2014. The leave-one-out cross validation step was applied to computing the out-of-sample adjusted error with gauge stations."

(2)- A more detailed description of the reference dataset and decision making process is desirable, e.g. a map with the mean or the sum of precipitation during the observation period at the reference grid points and stations.

**Answer:** Thanks. Done.

**Changed:** We added a map with the multi-year (1990-2014) average annual precipitation (Fig. 2). The multi-year average annual precipitation increases from north to south and the spatial distribution of precipitation is uneven, with an average annual precipitation ranging from less than 250 mm to more than 600 mm during the summer seasons over the Jinsha River Basin.

[Figure]

**Figure 2** The multi-year (1990-2014) average annual precipitation during JJA over the Jinsha River Basin. 30 rain stations were provided by the China Meteorological Administration stations, the other 18 CHIRPS fusion stations were provided by the Climate Hazards Group UC Santa Barbara online at ftp://ftp.chg.ucsb.edu/pub/org/chg/products/CHIRPS-2.0/diagnostics/global_monthly_station_density/tifs/p05/ (last access: 10 December, 2018).

(3)- As far as I understand, using this evaluation dataset implies that only C1 and C2 grid points are evaluated, because they contain either a rain gauge station or a grid point of the interpolated station data. Is that correct? Can the authors give an assessment on the quality of the method at C3 and C4 pixels?

Does the selection of the stations and grid points for training have an influence on the model performance? Depending on the location of the points for adjustment the quality of the blended dataset may vary. An ensemble study using different compositions of the pool of training stations / grid points would give statistically more robust results.

**Answer:** Thanks. In the new experiment, the leave-one-out cross validation step using all the stations was used to evaluate the performance of the WHU-SGCC algorithm. The training set was used to establish statically relationships when conducting the WHU-SGCC method, and the remaining one gauge station was used to evaluate. The adjusted process shown that the adjustment method for C2 pixels was derived from C1 pixels, the adjustment method for C3 pixels was derived from C2 pixels, and the adjusted values for C1 and C4 pixels were interpolated by IDW with C2 and C3 pixels. There were statistically relationship among C1, C2, C3 and C4 pixels. Thus, the performance of WHU-SGCC method would be evaluated on the overall accuracy, not on a certain class of pixels.

2.

(1)- CHIRP data is used as basis for the WHU-SGCC dataset and it is shown that the blending approach leads to better (light and moderate rainfall) or similar (heavy precipitation) results compared to measurements. CHIRPS, however, seems to perform much worse than the original CHIRP dataset although it is also adjusted to rain gauges. Can the authors give an explanation for that?

**Answer:** The CHIRPS was derived from blending in-suit precipitation observations and the CHIRP data, with a modified inverse-distance weighting algorithm at a quasi-global area (land only, 50° S-50° N). The blended data (CHIRPS) has an effective performance on a large scale region according to existing studies, such as at the national scale, but there are still large discrepancies with ground observations at the sub-regional level, especially at the river basin scale. The performance and applicability of CHIRPS at the sub-regional level still need to be validated. What's more, the interpolation performance from the limited and sparse rain gauge stations will be affected by more errors which was evaluated with rain gauge stations shown in Table 5.

As such, due to the poor performance of CHIRPS data at the sub-regional scale and the shortcomings of the modified inverse-distance weighting algorithm, the aim of this article is to offer a novel blending approach to improve the precipitation estimated accuracy at the river basin scale.

**Change:** We changed the sentence from "As such, the aim of this article is to offer a novel approach for blending daily precipitation gauge data, gridded precipitation data and the Climate Hazards Group Infrared Precipitation (CHIRP) satellite-derived precipitation estimates over Jinsha River Basin." to "As such, due to the poor performance of CHIRPS data at the sub-regional scale and the shortcomings of the existing blending algorithms, the aim of this article is to offer a novel approach for blending daily liquid precipitation gauge data, gridded precipitation data and the Climate Hazards Group Infrared Precipitation (CHIRP) satellite-derived precipitation estimates developed by the UC Santa Barbara, over the Jinsha River Basin." for better explanation.

(2)- It would also be desirable to expand the investigated period to get more robust results, e.g. add more summer seasons from other years.

**Answer:** Thanks. Done.

**Change:** we changed the study period from summer of 2016, JJA to a longer study period during June-July-August from 1990 to 2014, to evaluate the model performance more reasonably.

**Specific comments**

(1) - P.1, L.37: There is twice "without adjustment" in the sentence

**Answer:** Thanks. Deleted one "without adjustment".

**Change:** We changed "Without adjustments, inaccurate satellite-based precipitation estimates without adjustment will lead to unreliable assessments of risk and reliability" to "Without adjustments, inaccurate satellite-based precipitation estimates will lead to unreliable assessments of risk and reliability".

(2) - P.2, L.63 and 65: Remove the brackets at Bai et al. and Trejo et al.

**Answer:** Thanks. Done.

**Change:** We removed the brackets at Bai et al. and Trejo et al.

(3) - P.3, L.89: Section 5 is about data availability. Section 6 presents conclusions

**Answer:** Thanks. Done.

**Change:** changed from "The results and discussion are analysed in Section 4, and conclusions and future work are presented in Section 5." to "The results and discussion are analysed in Section 4, the data available is described in Section 5, and conclusions and future work are presented in Section 6."

(4) - P.3, L.102-103: I'm a bit confused here. Does "average annual precipitation", "annual precipitation" and "total annual precipitation" mean the same thing? Or is the total (for me this refers to the sum) of the precipitation north of Shigu almost four times smaller than the mean annual precipitation in the whole Jinsha River Basin?

**Answer:** Thanks. Done. We have used the spatially averaged annual accumulation of precipitation as an indication of precipitation climatology for the study region. The reference (Yuan, Z., Xu, J. J., and Wang, Y. Q.: Projection of Future Extreme Precipitation and Flood Changes of the Jinsha River Basin in China Based on CMIP5 Climate Models, Int. J. Environ. Res. Public Health, 15, 17, 10.3390/ijerph15112491, 2018.) can support the average annual precipitation statistic.

**Change:** We changed "Average annual precipitation in the Jinsha River Basin is approximately 3433.45 mm, the total annual precipitation north of Shigu is 937.25 mm, while south of Shigu annual precipitation is 2496.20 mm." to "The average annual precipitation of the Jinsha River Basin is approximately 710 mm, the average annual precipitation of the lower reaches is approximately 900-1300 mm, while the average annual precipitation of the middle and upper reaches is approximately 600-800 mm (Yuan et al., 2018)."

(5) - P.6, L169: I would remove the numbering here, as it doesn't seem to be another part of the method, but refers to the overview of steps 1-4.

**Answer:** Thanks. Done.

**Change:** We added this sentence into the first phase in section 3.1, as the reference to the overview of steps 1-4.

(6) - P.11, L.309: Nash and Sutcliffe(1970) is missing in the references

**Answer:** Thanks. Done.

**Change:** We have added the Nash and Sutcliffe (1970) in the references.

(Nash, J. E., Sutcliffe, J. V.: River flow forecasting through conceptual models, Part I - A discussion of principles, Journal of Hydrology, 10, 282–290, doi.org/10.1016/0022-1694(70)90255-6, 1970.)

(7) - P.14, Table 4: How is the accuracy assessment of C3 pixels done? What is the reference here? Why is SCC < 0.5?

**Answer:** Thanks. In the previous experiment, the number of C3 pixels accounted for 62.18% of the total pixels inside the river basin and the major of the C3 pixels had the same location with the 30% testing data. So we evaluated the C3 pixels with part of testing set (rain gauge stations and gridded points). While, in the new experiment, due to the leave-one-out cross validation step using all the stations, the performance of WHU-SGCC method would be evaluated on the overall accuracy, not on a certain class of pixels. So we didn't evaluate the C3 pixels separately.

**Change:** Deleted the statistical analysis about the C3 pixels.

(8) - P.17, Fig.10: It might be helpful to present the percentage deviation from the observations for clarification of the model performance. It seems that at some days, all three datasets deviate more than 70% from the observations.

**Answer:** Thanks. Because the daily precipitation of rain stations may be no rain, the percentage deviation from the observations cannot be obtained (the denominator is 0). The statistical analysis of the agreement between daily observations and WHU-SGCC, CHIRP and CHIRPS estimates on leave-one-out cross validation: a) Pearson's correlation coefficient b) root mean square error c) mean absolute error d) relative bias e) Nash-Sutcliffe efficiency coefficient f) probability of detection g) false alarm ratio, and h) critical success index was shown in Fig. 5.

**Change:** We redraw the boxplots of the statistical analysis of the agreement between daily observations and WHU-SGCC, CHIRP and CHIRPS estimates on leave-one-out cross validation.

And now the section 4 was divided into 3 parts: 4.1 Model performance based on overall accuracy evaluations, 4.2 Model performance based on daily accuracy evaluations and 4.3 Model performance on rain events predictions.

[revised manuscript text omitted]

Accordingly, there are many kinds of precipitation estimates combining multiple sources datasets. Since 1997, the Tropical Rainfall Measurement Mission (TRMM) has improved satellite-based rainfall retrievals over tropical regions (Kummerow et al., 1998;Simpson et al., 1988), and then applies a stepwise method for blending daily TRMM Multisatellite Precipitation Analysis (TMPA) output with rain gauges in South America (Vila et al., 2009). The Global Precipitation Measurement (GPM) satellite was launched after the success of the TRMM satellite by the cooperation of National Aeronautics and Space Administration (NASA) and Japan Aerospace Exploration Agency (JAXA) on February 27, 2014 (Mahmoud et al., 2018;Ning et al., 2016). The main core observatory satellite (GPM) cooperates with the ten other satellites (partners) to offer the high spatiotemporal resolution products (0.1° × 0.1°- half- hourly) of the global real-time precipitation estimates (Mahmoud et al., 2019). The Geostationary Operational Environmental Satellite (GOES)-R Series is the geostationary weather satellites, which significantly improves the detection and observation of environmental phenomena. The Advanced Baseline Imager (ABI) onboard the GOES-R platform will provide images in 16 spectral bands, spatial resolution of 0.5 to 2 km (2 km in the infrared and 1–0.5 km in the visible), and full-disk scanning every 5 minutes over the continental United States. The GOES-R Series will offer the enhanced capabilities for satellite-based rainfall estimation and nowcasting (Behrangi et al., 2009;Schmit et al., 2005). The Global Precipitation Climatology Project (GPCP) is one of the successful projects for blending rain gauge analysis and multiple satellite-based precipitation estimates, and constructed a relatively coarse-resolution (monthly, 2.5° × 2.5°) global precipitation dataset (Adler et al., 2003;Huffman et al., 1997). To improve the resolution of this satellite-based dataset, the GPCC network data was incorporated into remote sensing information with Artificial Neural Networks (PERSIANN) rainfall estimates, which provides finer temporal and spatial resolutions (daily, 0.25° × 0.25°) (Ashouri et al., 2015). The CPC Merged Analysis of Precipitation (CMAP) product is a data blending and fusion analysis of gauge data and satellite-based precipitation estimates (Xie and Arkin, 1996). CMAP has a long-term dataset series from 1979, while the resolution is relatively coarse. Although the aforementioned products are widely used and have performed well, the data resolution cannot achieve high accuracy in precipitation monitoring over the Jinsha River Basin, China.

Currently, the Climate Hazards Group Infrared Precipitation with Station data (CHIRPS) developed by the UC Santa Barbara, which has a higher spatial resolution (0.05°), can solve the scale problem. CHIRPS is a long-term precipitation data series, which merges three types of information: global climatology, satellite estimates and in situ observations. Table 1 shows the temporal and spatial resolution of current major satellite-based precipitation datasets. The CHIRPS precipitation dataset with several temporal and spatial scales has been evaluated in Brazil (Nogueira et al., 2018;Paredes-Trejo et al., 2017), Chile (Yang et al., 2016;Zambrano-Bigiarini et al., 2017), China (Bai et al., 2018), Cyprus (Katsanos et al., 2016b;Katsanos et al., 2016a), India (Ali and Mishra, 2017) and Italy (Duan et al., 2016). Nevertheless, the temporal resolutions of the aforementioned applications were mainly at seasonal and monthly scales, lacking the evaluation of daily precipitation. Additionally, despite the great potential of gauge-satellite fusing products for large-scale environmental monitoring, there are still large discrepancies with ground observations at the sub-regional level where these data are applied. Furthermore, the CHIRPS product reliability has not been analysed in detail  over the Jinsha River Basin, China, particularly on a daily scale. The existing research indicates that estimations over mountainous areas with complex topography often have large uncertainties and systematic errors due to the topography, seasonality, climate impact and sparseness of rain gauges (Derin et al., 2016;Maggioni and Massari, 2018;Zambrano-Bigiarini et al., 2017). Moreover, (Bai et al., 2018) evaluates CHIRPS over mainland China and indicates that the performance of CHIRPS is poor over the Sichuan Basin and the Northern China Plain, which have complex terrains with substantial variations in elevation. Additionally, (Trejo et al., 2016) shows that CHIRPS overestimates low monthly rainfall and underestimates high monthly rainfall using several numerical metrics, and rainfall event frequency is overestimated excluding the rainy season.

| | Table 1 Coverage and spatiotemporal resolutions of major satellite precipitation datasets | | | |
|---|---|---|---|---|
| Product | Temporal resolution | Spatial resolution | Period | Coverage |
| TRMM 3B42 | 3hours | 0.25° | 1998-present | 50°S-50°N |
| GPM | 30min/Hourly/ 3hours/Daily/3Day/7 Day/Monthly | 0.1°/0.25°/0.05°/5° | 2014-present | 60°S-60°N 70°N-70°S 90°N-90°S |
| GOES-R | 5min/15min | 0.5-2 km | 2016-present | the continental United States/ western hemisphere |
| GPCP | Monthly/Pentad | 2.5° | 1979-(delayed) present | 90°S-90°N |
| PRERSSIANN-CDR | Daily | 0.25° | 1983-(delayed) present | 60°S-60°N |
| CMAP | Monthly | 2.5° | 1979- present | 90°S-90°N |
| CHIRPS | Annual/Monthly/ Dekad/Pentad/Daily | 0.05°/0.25° | 1981- present | 50°S-50°N |

To overcome these limitations, many studies have focused on proposing effective methodologies for blending rain gauge observations and satellite-based precipitation estimates, and sometimes radar data to take advantage of each dataset. Many numerical models are established among these datasets for high-accuracy precipitation estimations, such as bias adjustment by a quantile mapping (QM) approach (Yang et al., 2016), Bayesian kriging (BK) (Verdin et al., 2015) and a conditional merging technique (Berndt et al., 2014). Among aforementioned methods, the QM approach is a distribution-based approach, which works with historical data for bias adjustment and is effective in reducing the systematic bias of regional climate model precipitation estimates at monthly or seasonal scales (Chen et al., 2013). However, the QM approach offers very limited improvement in removing day-by-day random errors. The BK approach shows very good model fit with precipitation observations. Unfortunately, the Gaussian assumption of the BK model is invalid for daily scales. Overall, there is a lack of effective methods for high-accuracy precipitation estimates over complex terrain on a daily scale.

As such, due to the poor performance of CHIRPS data at the sub-regional scale and the shortcomings of the existing blending algorithms, the aim of this article is to offer a novel approach for blending daily liquid precipitation gauge data, gridded precipitation data and the Climate Hazards Group Infrared Precipitation (CHIRP) satellite-derived precipitation estimates developed by the UC Santa Barbara, over the Jinsha River Basin. Here, we will use precipitation to name liquid precipitation throughout the text. The CHIRP is the raw data of CHIRPS before blending in rain gauge data. The objective is to build corresponding precipitation models that consider terrain factors and precipitation characteristics to produce high-quality precipitation estimates. This novel method is named the Wuhan University Satellite and Gauge precipitation Collaborated Correction (WHU-SGCC) method. We demonstrate this method by applying it to daily precipitation over the Jinsha River Basin during the summer seasons from 1990 to 2014in summer 2016. The results support the validity of the proposed approach for producing refined satellite-gauge precipitation estimates over mountainous areas.

The remainder of this paper is organized as follows: Section 2 describes the study region and precipitation rain gauges, gridded observations and CHIRPS dataset used in this study. Section 3 presents the principle of the WHU-SGCC approach for high-accuracy precipitation estimates. The results and discussion are analysed in Section 4, the data available is described in Section 5, and conclusions and future work are presented in Section 56.

**2 Study Region and Data**

**2.1 Study Region**

The Yangtze River, one of the largest and most important rivers in Southeast Asia, originates on the Tibetan Plateau and extends approximately 6300 km eastward to the East China Sea. The river's catchment proximately covers an area of approximately ~180 × 10⁴ km² and the average annual precipitation is approximately 1100 mm (Zhang et al., 2019). In 2016, the average precipitation in the Yangtze River Basin was 12053 mm and the total precipitation was 21478.71 billion m³, which is 10.9% higher than the annual average total precipitation. Yangtze River is divided into nine sub-regionsbasins, the upper drainage basin is the Jinsha River Basin, which flows through the provinces of Qinghai, Sichuan, and Yunnan in western China. The total river length is 3486 km, accounting for 77% of the length of the upper Yangtze River, and covering a watershed area of $460 \times 10^3$ km$^2$. The location of the Jinsha River Basin is shown in Fig. 1, and covers the eastern part of the Tibetan Plateau and the part of the Hengduan Mountains. The southern portion of the river basin is the Northern Yunnan Plateau and the eastern portion includes a wide area of the southwestern margin of the Sichuan basin. Crossing complex and varied terrains, the elevation of the Jinsha River ranges from 263 to 6575 m above sea level, which results in significant temporal and spatial  weather variation within the basin. The average annual precipitation of the Jinsha River Basin is approximately 710 mm, the average annual precipitation of the lower reaches is approximately 900-1300 mm, while the average annual precipitation of the middle and upper reaches is approximately 600-800 mm (Yuan et al., 2018).  The climate of the Jinsha River Basin has more precipitation during the  summer season (June-July-August, JJA), which is affected by oceanic southwest and southeast monsoons . Therefore, the blending of satellite estimations with gauged observations during the  (JJA) is the main focus of this research.

[Figure]

**Figure 1** Location of the study area with key topographic features.

**2.2 Study Data**

**2.2.1 Precipitation gauge observations**

Daily rain gauge observations at 30 national standard rain stations in the Jinsha River Basin  during the JJA from 1990 to 2016 were provided by the National Climate Centre (NCC) of the China Meteorological Administration (CMA) (http://data.cma.cn/data/cdcdetail/dataCode/SURF_CLI_CHN_MUL_DAY_V3.0.html, last access: 10 December, 2018), which imposes a strict quality control at station-provincial-state levels. Station identification numbers and relevant geographical characteristics are shown in Table 2, and their uneven spatial distribution is shown in Fig. 2. The selected rain gauges are located in Qinghai, Tibet, Sichuan and Yunnan Provinces but are mainly scattered in Sichuan Province, and the number of rain gauges in the northern river basin is less than in the southern river basin. In this study, the gauge observations were used as the reference data in bias adjustment of satellite precipitation estimations.

**Table 2** Geographical characteristics of rain stations.

| Station number | Province | Lat (°N) | Lon (°E) | Elevation (m) |
|---|---|---|---|---|
| 52908 | Qinghai | 35.13 | 93.05 | 4823 |
| 56004 | Qinghai | 34.13 | 92.26 | 4744 |
| 56021 | Qinghai | 34.07 | 95.48 | 5049 |
| 56029 | Qinghai | 33.00 | 96.58 | 4510 |
| 56034 | Qinghai | 33.48 | 97.08 | 4503 |
| 56144 | Tibet | 31.48 | 98.35 | 4743 |
| 56038 | Sichuan | 32.59 | 98.06 | 4285 |
| 56146 | Sichuan | 31.37 | 100.00 | 4703 |
| 56152 | Sichuan | 32.17 | 100.20 | 4401 |
| 56167 | Sichuan | 30.59 | 101.07 | 3374 |
| 56247 | Sichuan | 30.00 | 99.06 | 2948 |
| 56251 | Sichuan | 30.56 | 100.19 | 4284 |
| 56257 | Sichuan | 30.00 | 100.16 | 3971 |
| 56357 | Sichuan | 29.03 | 100.18 | 4280 |
| 56374 | Sichuan | 30.03 | 101.58 | 3902 |
| 56459 | Sichuan | 27.56 | 101.16 | 3002 |
| 56462 | Sichuan | 29.00 | 101.30 | 4019 |
| 56475 | Sichuan | 28.39 | 102.31 | 1850 |
| 56479 | Sichuan | 28.00 | 102.51 | 2470 |
| 56485 | Sichuan | 28.16 | 103.35 | 2060 |
| 56565 | Sichuan | 27.26 | 101.31 | 2578 |
| 56571 | Sichuan | 27.54 | 102.16 | 1503 |
| 56666 | Sichuan | 26.35 | 101.43 | 1567 |
| 56671 | Sichuan | 26.39 | 102.15 | 1125 |
| 56543 | Yunnan | 27.50 | 99.42 | 3216 |
| 56586 | Yunnan | 27.21 | 103.43 | 2349 |
| 56651 | Yunnan | 26.51 | 100.13 | 2449 |
| 56664 | Yunnan | 26.38 | 101.16 | 1540 |
| 56684 | Yunnan | 26.24 | 103.15 | 2184 |
| 56778 | Yunnan | 25.00 | 102.39 | 1975 |

[Figure]

The multi-year (1990-2014) average annual precipitation during the JJA over the Jinsha River Basin increases from north to south (Fig. 2). The spatial distribution of precipitation is uneven, with an average annual precipitation ranging from less than 250 mm to more than 600 mm during the summer seasons. Figure. 2 also shows the multi-year average daily precipitation during the JJA is no more than 10mm.

[Figure]

**Figure 2**

**Figure 2** The multi-year (1990-2014) average annual precipitation during JJA over the Jinsha River Basin.  30 rain stations were provided by the China Meteorological Administration stations, the other 18 CHIRPS fusion stations were provided by the Climate Hazards Group UC Santa Barbara online at ftp://ftp.chg.ucsb.edu/pub/org/chg/products/CHIRPS-2.0/diagnostics/global_monthly_station_density/tifs/p05/ (last access: 10 December, 2018).

~~The gridded precipitation data developed by CMA with 0.5°× 0.5° resolution on a daily scale, was interpolated from 2472 gauge observations with a thin plate spline algorithm from 1961 to the present. Over the Jinsha River Basin, a total of 170 gridded points were selected as the supplementary data for observations in JJA 2016, due to the 2472 gauged station data that were not shared on CMA (http://data.cma.cn/data/cdcdetail/dataCode/SURF_CLI_CHN_PRE_DAY_GRID_0.5.html, last access: 10 December, 2018). The even distribution of daily gridded precipitation observations is shown in Fig. 2.~~

**2.2. 2 CHIRPS satellite-gauge fusion precipitation estimates**

The CHIRPS v.2 dataset, a satellite-based daily rainfall product, is available online at ftp://ftp.chg.ucsb.edu/pub/org/chg/products/CHIRPS-2.0/global_daily/tifs/p05/ (last access: 10 December, 2018). It covers a quasi-global area (land only, 50° S-50° N) with several temporal scales (daily, 3-day, 6-day or monthly time steps) and high spatial resolution (0.05°) (Rivera et al., 2018). This dataset contains a wide variety of satellite-based rainfall products derived from multiple data sources and incorporates four data types: monthly precipitation from CHPClim v.1.0 (Climate Hazards Group's Precipitation Climatology version 1) derived from the combination of the satellite fields, gridded physiographic indicators, and in situ climate normal with the geospatial modelling approach based on moving window regressions and inverse distance weighting interpolation (Funk et al., 2015 b), quasi-global geostationary thermal infrared satellite observations (TRMM 3B42 version 7), atmospheric model rainfall fields CFS (Climate Forecast System) from NOAA, and surface based precipitation observations from various sources including national or regional meteorological services. The differences from other frequently used precipitation products are the higher resolution of 0.05° and the longer-term data series from 1981 to the present (Funk et al., 2015 a).

CHIRPS is the blended product of a two-part process. First, IR precipitation (IRP) pentad rainfall estimates are fused with corresponding CHPClim pentad data to produce an unbiased gridded estimate, called the Climate Hazards Group IR Precipitation (CHIRP), which is available online at ftp://ftp.chg.ucsb.edu/pub/org/chg/products/CHIRP/daily/ (last access: 10 December, 2018). In the second part of the process, CHIRP data is blended with in situ precipitation observations obtained from a variety of sources including national and regional meteorological services by means of a modified inverse-distance weighting algorithm to create the final blended product, CHIRPS (Funk et al., 2014). The daily CHIRP satellite-based data over the Jinsha River Basin during the summer seasons from 1990 to 2014 was selected as the input for WHU-SGCC blending with rain observations, and the corresponding daily CHIRPS blended data was used for comparisons of precipitation accuracy.

The blended in situ daily precipitation observations come from a variety of sources such as: the daily GHCN archive (Durre et al., 2010), the Global Summary of the Day dataset (GSOD) provided by NOAA's National Climatic Data Center, the World Meteorological Organization's Global Telecommunication System (GTS) daily archive provided by NOAA CPC, and over a dozen national and regional meteorological services. The number of daily  observation stations used for CHIRPS data over the Jinsha River Basin was only 18, compared to the 30 rain gauge stations  provided by CMA  (Fig. 2).

**3 Methods**

**3.1 The WHU-SGCC approach**

In this study, the approach of the WHU-SGCC is to estimate the precipitation for every pixel by blending satellite estimates and rain gauge observations considering the terrain factors and precipitation characteristics. There were  four steps to establish the numerical relationship between gauge stations and the corresponding satellite pixels, and interpolation for the remaining pixels. On this basis, the WHU-SGCC method identifies the geographical locations and topographical features of each pixel and applies the four classification and blending rules. A flowchart of the WHU-SGCC method is shown in Fig. 3. The proposed approach was evaluated over the Jinsha River Basin based on 30 gauge stations and CHIRP satellite-based precipitation estimations during the JJA from 1990 to 2014. The leave-one-out cross validation step was applied to computing the out-of-sample adjusted error with gauge stations.

 The basic description of the WHU-SGCC method is given below, with details illustrated separately in later sections.

1)  Classify all regional pixels into  four types: C1 (pixel including one gauge station in its area),  C2 (pixel statistically  similar to C1),  C3 (pixel statistically  similar to C2) and  C4 (remaining pixels).

2) Analyse the relationships between precipitation observations and the C1, C2, and C3 pixel types, and interpolated for  the C4  pixels. These relationships are described by  four rules, detailed below as Rules 1 through 4.

3)  Establish regression models and screen target pixels based on the  four aforementioned rules.

4) Correct all precipitation pixels in daily regional precipitation images.

[Figure]

[Figure]

**Figure 3** Flowchart of the WHU-SGCC approach with the  four rules applied in this study.

**3.1.1 Assumptions**

1) Gauge  observations are the most accurate, or "true", values for reference purposes.

2) No major terrain change occurred during the twenty years.

3)  Pearson's Correlation Coefficient (PCC) can indicate the statistically similarity of rainfall characteristics among pixels over a seasonal scale.

**3.1.2 Rule 1 of the WHU-SGCC method**

$$ \tag{1}$$

$$ \tag{2}$$

$$ \tag{3}$$

$$ \tag{4}$$

$$ \tag{5}$$

In general, satellite precipitation estimations deviated from ground-based measurements, which were assumed to be the true values. Rule 1 aims to establish a regression model between each gauge historical observations and the corresponding CHIRP grid cell values. The regression relationship was derived by random forest regression (RFR) at each gauge station. RFR is a machine-learning algorithm for a predictive model with a large set of regression trees in which each tree in the ensemble is grown from a bootstrap (Johnson, 1998) sample drawn with replacement from the training set. The final prediction is obtained by combining the results of the prediction methods applied to each bootstrap sample (Genuer et al.,

2017). The predicted value is calculated by the mean of all trees.

$$Y_o = f_{RFR}(Y_s) \tag{1}$$

where $Y_o$ denotes each gauge historical observations and $Y_s$ denotes the corresponding CHIRP grid cell values at C1

pixels, $f_{RFR}$ is constructed from the time series $Y_o$ (dependent variable) and $Y_s$ (independent variable) by means of RFR.

The number of decision trees was set to 500, which was determined by out-of-bag (OOB) error (Appendix A). The OOB error reached the minimum value when the number of decision trees was less than 500.

The Rule 1 builds the statistical relationships between gauge observations and the corresponding CHIRP grid cell values, which is the key idea in correcting the satellite-based precipitation estimations in the whole study area. As there are 30 gauge stations in the study area, 30 regression relationships at C1 pixels were derived from Rule1. The values of C1 pixels are not corrected in Rule 1, but interpolated in Rule 4.

**3.1.3 Rule 2 of the WHU-SGCC method**

Commonly, a few of the national standard stations have free access, and these stations are unevenly distributed and do not satisfied the accuracy needed for regional precipitation estimation. Under these circumstances, the gridded precipitation data developed by CMA are applied as the supplementary data for observations with uniform spatial distribution. Therefore, Rule

2 is same as Rule 1 with different input data. $\widehat{C2}_{as}$ is the adjusted target precipitation of one C2 pixel.

**3.1.4 3 Rule 3 2 of the WHU-SGCC method**

It is reasonable to assume that there are some pixels that are statistically physically similar to the precipitation characteristics of C1 pixels in a certain spatial scope. Therefore, it is feasible to adjust the satellite estimation bias of C3 C2 pixels by referring to the appropriate regression relationships at corresponding C1 pixels based on Rule 1.

First, the spatial scope in which pixels may have highly similar characteristics is established. Some studies indicate that geographical location, elevation and other terrain information influences the spatial distribution of rainfall, especially in mountainous areas with complex topography (Anders et al., 2006;Long and Singh, 2013). The size of the spatial range is an important parameter to distinguish spatial similarity and heterogeneity. In the WHU-SGCC method, the approach of fuzzy c- means (FCM) clustering was explored to determine the spatial range considered as each pixel's terrain factors including longitude, latitude, elevation, slope, aspect and curvature. FCM method was developed by J.C. Dunn in 1973 (Dunn, 1973), and improved in 1983 (Wang, 1983). It is an unsupervised fuzzy clustering method and the steps are as follows (Pessoa et al.,

2018):

1) Choose the number of clusters $c$. The optimum number of clusters was determined by $L(c)$ which was derived from the inter-distance and inner-distance of elements samples in Eq. (2). It is ensured that the distance between the same samples is smaller, while the distance between the different samples is larger.

$$L(c) = \frac{\sum_{i=1}^{c}\sum_{j=1}^{n} w_{ij}^m \| c_i - \overline{x} \|^2 /(c-1)}{\sum_{i=1}^{c}\sum_{j=1}^{n} w_{ij}^m \| x_j - c_i \|^2 /(n-c)} \tag{2}$$

In Eq. (2), the denominator is inner-distance and the molecular is inter-distance. The initial value of $c$ is 1 and the maximum value of $c$ is the number of gauge stations in this study area. The optimum number of clusters was optimized to maximize the

$L(c)$. For this reason, $c$ value is conducted in the range of 1 to the number of gauge stations with an incremental interval value of 1 in this study.

2) Assign coefficients randomly to each data point $x_i$ for the degree to which it belongs in the $ji$ th cluster $w_{ij}(x_i)$:

$$c_i^{(t)} = \frac{\sum_{j=1}^{n} w_{ij}^m x_i}{\sum_{j=1}^{n} w_{ij}^m} \quad (6\underline{3}), \qquad w_{ij} = \frac{1}{\sum_{k=1}^{c} (\frac{\| x_i - c_i \|}{\| x_i - c_k \|})^{\frac{2}{m-1}}} \quad (\underline{7}4), \qquad \bar{x} = \frac{\sum_{i=1}^{c} \sum_{j=1}^{n} w_{ij}^m x_j}{n} \quad (5)$$

where $x$ is a finite collection of $n$ elements that will be partitioned into a collection of $c$ fuzzy clusters, $c_i$ is the centre of each cluster, $m$ is the hyper-parameter that controls the level of cluster fuzziness and $w_{ij}$ is the degree to which element $x_i$

belongs to $c_i$ and $\bar{x}$ is the centre vector of collection. In Eq. (6$\underline{3}$), $c_j^{(t)}$ represents the cluster centre in iteration $t$. If the minimum improvement in objective function between two consecutive iterations satisfies the following equation, the algorithm terminates in iteration $t$ (Eq. (6)):

$$\| c_i^{(t)} - c_i^{(t+1)} \| < \varepsilon \quad (6)$$

1)3) Minimize the objective function $F_c$ to achieve data partitioning.

$$F_c = \sum_{j=1}^{n} \sum_{i=1}^{c} w_{ij}^m \| x_j - c_i \|^2 \quad (8\underline{7})$$

The results of FCM are the degree of membership of each pixel to the cluster centre as represented by numerical value.

Pixels in each cluster have similar terrain features and precipitation characteristics.

Second, the adjusted C1 and C2 are employed. SCC was used as the evaluation index for each C1 and C2 with their values after adjustment and gauge observations in JJA:

$$SCC = \frac{\sum_{i=1}^{n} (rgx_i - rg\bar{x})(rgy_i - rg\bar{y})}{\sqrt{\sum_{i=1}^{n} (rgx_i - rg\bar{x})^2} \sqrt{\sum_{i=1}^{n} (rgy_i - rg\bar{y})^2}} \quad (9)$$

Spearman's correlation coefficient is defined as Pearson's correlation coefficient between the ranked variables, and it assesses monotonic relationships (whether linear or not) where $n$ is the number of data points in each set, which was the number of each C1 or C2 in the historical JJA dataset. $x_i$ is the $i$th data value in the first data set (satellite estimations after Rule 1

and Rule 2 adjustment, $\hat{C1}_{as}$ and $\hat{C2}_{as}$), $x_i$ is converted to its rank $rgx_i$, and $rg\bar{x}$ is its average value. Similar definitions exist for $rgy_i$ and $rg\bar{y}$ (gauge and gridded observations at C1 and C2 pixels, $Y_s$). The value range of the $SCC$

is between $-1$ and $+1$. If there are no repeated data values, a perfect $SCC$ of $+1$ or $-1$ occurs when each of the variables is a perfect monotone function of the other. However, if the value is close to zero, there is zero correlation. In addition, correlation is not only determined by the value of the correlation coefficient but also from the correlation test's $p$-value. The critical value is 0.05, thus a $p$ lower than 0.05 indicates the data are significantly correlated. Therefore, the C1 and C2 pixels selected for

Rule 3 must meet the following criteria:

$$|SCC| \geq 0.5 \quad and \quad p < 0.05 \quad (10)$$

Third, the filtered C1 and C2 pixels after adjustment is used to establish a regression model between the historical $\hat{C1}_{as}$,

$\hat{C2}_{as}$ and $Y_s$. To ensure high accuracy, it is necessary to calculate the $SCC$ and $p$ values between $\hat{C1}_{as}$, $\hat{C2}_{as}$ and $Y_s$, and complete the filtering criteria described above in Eq. (7) before building the regression model. The regression relationship was derived by random forest regression (RFR). RFR is a machine-learning algorithm for a predictive model with a large set of regression trees in which each tree in the ensemble is grown from a bootstrap (Johnson, 1998) sample drawn with replacement from the training set. The final prediction is obtained by combining the results of the prediction methods applied to each bootstrap sample (Genuer et al., 2017). The predicted value is calculated by the mean of all trees.

$$\hat{C1}_{as} \text{ or } \hat{C2}_{as} = f_{RFR}(Y_s) \tag{11}$$

where $f_{RFR}$ is constructed from the time series $\hat{C1}_{as}$ or $\hat{C2}_{as}$ (dependent variable) and the corresponding $Y_s$ data (independent variable) at filtered C1 and C2 pixels in JJA by means of RFR. The number of decision trees was set at the default value of 500.

Fourth, as mentioned above, the aim of Rule 3 is to derive an adjustment method for C3 pixels based on learning from Rule 1 and Rule 2. With the establishment of a regression relationship between values before and after adjustment of the C1 and C2 pixels by RFR method, the determination of C3 pixels follows a considerable procedure. Pixels in each cluster represent potential C3 pixels, with exception of the C1 and C2 pixels and are called R pixels. Spearman's $r$ and $p$ values between the satellite estimations (CHIRP grid cell values) at R pixels and the C1 and C2 pixels are the criteria for final determination of C3 pixels. Each R pixel has $m$ SCC and $p$ values (the number of C1 and C2 pixels in the cluster), and the subset of C3 pixels is identified by excluding the data that failed the correlation test and retaining both the data with a maximum SCC of at least 0.5 and the corresponding index of C1 and C2 pixels. The selected C3 pixels are statistically similar to the precipitation characteristics of corresponding C1 and C2 pixels in their defined spatial scope.

After identifying the C3 pixels and their corresponding C1 and C2 pixels, the adjustment method for C3 pixels is derived from the regression model for the C1 and C2 pixels.

$$\hat{C3}_{as} = f_{RFRc}(Y_s) \tag{12}$$

where $\hat{C3}_{as}$ is the adjusted satellite precipitation estimate and $Y_s$ is the CHIRP grid cell value for the C3 pixels, and $f_{RFRc}$ is the $f_{RFR}$ of corresponding C1 and C2 pixels.

Second, as mentioned above, the aim of Rule 2 is to derive an adjustment method for C2 pixels based on learning from Rule 1. With the establishment of a regression relationship between gauge observations and the corresponding CHIRP grid cell values of the C1 pixels by RFR method, the determination of C2 pixels follows a considerable procedure. With exception of the C1 pixels, the remaining pixels in each cluster represent potential C2 pixels called R pixels. Pearson's correlation coefficient (PCC) and $p$-values between the satellite estimations (CHIRP grid cell values) at R pixels and the C1 pixels are the criteria for final determination of C2 pixels. The PCC is defined as follows:

$$PCC_{x,y} = \frac{\sum_{i=1}^{n}(x_i - \bar{x})(y_i - \bar{y})}{\sqrt{\sum_{i=1}^{n}(x_i - \bar{x})^2}\sqrt{\sum_{i=1}^{n}(y_i - \bar{y})^2}} \tag{8}$$

where $n$ is the number of samples, $x_i$ and $y_i$ are individual samples (CHIRP grid cell values at C1 and C2 pixels), $\bar{x}$ is the arithmetic mean of $x$ calculated by $\bar{x} = \frac{1}{n}\sum_{i=1}^{n}x_i$, $\bar{y}$ is the arithmetic mean of $y$ calculated by $\bar{y} = \frac{1}{n}\sum_{i=1}^{n}y_i$.

The value range of the PCC is between -1 and +1. If there are no repeated data values, a perfect PCC of +1 or −1 occurs when each of the variables is a perfect monotone function of the other. However, if the value is close to zero, there is zero correlation. In addition, correlation is not only determined by the value of the correlation coefficient but also from the correlation test's $p$-value. The critical values for PCC and $p$-value are 0.5 and 0.05, thus a PCC value higher than 0.5 and a $p$-value lower than 0.05 indicate the data are significantly correlated (Zhang and Chen, 2016). Therefore, the final determination of C2 pixels must meet the following criteria:

$$|\mathrm{PCC}| \geq 0.5 \quad and \quad p < 0.05 \tag{9}$$

Each R pixel has $m$ PCC and $p$-values (the number of C1 pixels in the cluster), and the subset of C2 pixels is identified by excluding the data that failed the correlation test and retaining both the data with a maximum PCC of at least 0.5 and a $p$-value lower than 0.05, and the corresponding index of C1 pixels. The selected C2 pixels are statistically similar to the precipitation characteristics of corresponding C1 pixels in their defined spatial scope.

After identifying the C2 pixels and their corresponding C1 pixels, the adjustment method for C2 pixels is derived from the regression model for the C1 pixels.

$$C2_{as} = f_{RFRc}(Y_s) \tag{10}$$

where $C2_{as}$ is the adjusted satellite precipitation estimate and $Y_s$ is the CHIRP grid cell value at the C2 pixels, and $f_{RFRc}$ is the $f_{RFR}$ of corresponding C1 pixel.

**3.1.4 Rule 3 of the WHU-SGCC method**

Recognizing that precipitation has a spatial distribution, the assumption that C3 pixels are statistically similar to the precipitation characteristics of C2 pixels is adopted to establish the adjustment method for C3 pixels.

First, the determination of C3 pixels in each spatial cluster is based on the selection of C2 pixels. The satellite-based estimation values at the remaining pixels with exception of the C1 and C2 pixels are used to calculate the PCC and $p$-values with the satellite-based estimation values$Y_s$ at the C2 pixels in the same cluster. The results of each pixel's $k$ PCC and $p$-value (the number of C2 pixels in the cluster) are evaluated based on the correlation test (Eq. (9)), and that the pixels with a maximum PCC of is at least 0.5 and the $p$-value is no more than 0.05, and then the corresponding index of C2 pixels are retained. The selected pixels called C3 pixels, which are statistically similar to the precipitation characteristics of the corresponding C2 pixels in the defined spatial scope.

After identifying the C3 pixels, a method for merging the CHIRP grid cell values at C3 pixels ($Y_s$) and the target reference values of $C2_{as}$ at the corresponding C2 pixels is applied to estimate the adjusted precipitation values at C3 pixels. This method combines $Y_s$ and $C2_{as}$ values in one variable, as shown in Eq. (11):


$$w_i = \frac{C2_{as_i} + \lambda}{Y_{s_i} + \lambda} \quad i=1,\ldots,\ n \tag{11}$$

where $\lambda$ is a positive constant set to 10 mm (Sokol, 2003), $C2_{as}$ is the adjusted precipitation values at the C2 pixels, $Y_{s_i}$ is extracted from the CHIRP grid cell values at the corresponding location of the C2 pixel, and $n$ is the number of C2 pixels in each spatial cluster.

Each $w$ of the C3 pixels is assigned the same value as the corresponding C2 pixel. Therefore, the values of C3 pixels are derived from Eq. (12):

$$C3_{as} = \max(w \times (Y_s + \lambda) - \lambda, 0) \tag{12}$$

where $C3_{as}$ is the adjusted target precipitation value at one C3 pixel and $Y_s$ is the corresponding CHIRP grid cell value. To avoid precipitation estimates below 0, Eq. (12) sets these negative values to 0.

If there is no C3 pixels in a spatial cluster, the C4 pixels are assumed to be physically similar to the precipitation characteristics of the C1 and C2 pixels and adjusted by the above method in Rule 4.

**3.1.6 5 Rule 5 4 of the WHU-SGCC method**

Excluding the C1, C2, C3 and C4 C3 pixels, the number of remaining pixels, called C5 C4 pixels which are adjusted by Inverse Distance Weighted (IDW). IDW is based on the concept of the first law of geography from 1970. It was defined as *everything is related to everything else, but near things are more related than distant things*. Therefore, the attribute value of an unsampled point is the weighted average of known values within the neighbourhood, and the distance weighting can be determined by IDW (Lu and Wong, 2008). In Rule 4, IDW is used to interpolate the unknown spatial precipitation data from the adjusted precipitation values at the C2 and C3 pixels. The IDW formulas are given as Eq. (13) and Eq. (14).

$$R_{as} = \sum_{i=1}^{n} w_i R_i \quad (13)$$

$$w_i = \frac{d_i^{-\alpha}}{\sum_{i=1}^{n} d_i^{-\alpha}} \quad \text{with} \quad \sum_{i=1}^{n} w_i = 1 \quad (14)$$

where $R_{as}$ is the unknown spatial precipitation data, $R_i$ is the adjusted precipitation values at C2 and C3 pixels, $n$ is the number of C2 and C3 pixels, $d_i$ is the distance from each C2 or C3 pixel to be unknown grid cell, $\alpha$ is the power which is generally specified as a geometric form for the weight. Several researches (Simanton and Osborn 1980; Tung 1983) have experimented with variations in a power, the small $\alpha$ tends to estimate values with the averages of sampled grids in the neighbourhood, while large $\alpha$ tends to give larger weights to the nearest points and increasingly down-weights points farther away (Chen and Liu, 2012;Lu and Wong, 2008). The value of $\alpha$ has an influence on the spatial distribution of information from precipitation observations. For this reason, $\alpha$ value is conducted in the range of 0.1 to three (0.1, 0.3, 0.5, 1.0, 1.5, 2.0, 2.5 and 3.0) in this study.

It is noted that the unknown spatial precipitation data including C1 and C4 pixels, because C1 pixels values were not adjusted in Rule 1.

In the end, after applying these  four rules, we obtained complete daily adjusted regional precipitation maps for the summer (JJA) over the Jinsha River basin.

**3.2 Accuracy assessment**

The performance of the WHU-SGCC adjusted precipitation estimates was evaluated by eight statistical metrics:  Pearson's correlation coefficient (PCC), root mean square error (RMSE), mean absolute error (MAE), relative bias (BIAS), the Nash-Sutcliffe efficiency coefficient (NSE), probability of detection (POD) and false alarm ratio (FAR) and critical success index (CSI). PCC, RMSE, MAE and BIAS were used to evaluate how well the WHU-SGCC method adjusted satellite estimation bias, while POD, FAR and CSI were used to evaluate the precipitation event predictions (Su et al., 2011). PCC measures strength of the  correlation relationship between the satellite estimations and observations. RMSE is an absolute measurement used to compare the difference between the satellite estimations and observations. MAE represents the average magnitude of error estimations, considering both systematic and random errors. The NSE (Nash and Sutcliffe, 1970) determines the relative magnitude of the variance of the residuals compared to the variance of the observations, bounded by minus infinity to 1. A negative value indicates a poor precipitation estimate and the value of an optimal estimate is equal to 1. BIAS measures the mean tendency of the estimated precipitation to be larger (positive values) or smaller (negative values) than the observed precipitation, with an optimal value of 0. POD, also known as the hit rate, represents the probability of rainfall detection. FAR is defined as the ratio of the false alarm  of rainfall to the total number of rainfall events. All of the accuracy assessment metrics  are shown  in Table 3.

**Table 3** Accuracy assessment metrics.

| Accuracy assessment Index | Unit | Formula | Range | Optimal value |
|---|---|---|---|---|
|  Pearson's Correlation Coefficient (*SCC*) | NA | $\text{P}SCC = \dfrac{\sum_{i=1}^{n}(Y_{oi} - \bar{Y}_o)(C_i - \bar{C})}{\sqrt{\sum_{i=1}^{n}(Y_{oi} - \bar{Y}_o)^2} \cdot \sqrt{\sum_{i=1}^{n}(C_i - \bar{C})^2}}$ | [-1,1] | 1 |
| Root Mean Square Error (RMSE) | mm | $RMSE = \sqrt{\dfrac{1}{n-1}\sum_{i=1}^{n}(C_i - Y_{oi})^2}$ | [0,+∞) | 0 |
| Mean Absolute Error (MAE) | mm | $MAE = \dfrac{1}{n}\sum_{i=1}^{n}|C_i - Y_{oi}|$ | [0, +∞) | 0 |
| Relative Bias (BIAS) | NA | $BIAS = \dfrac{\sum_{i=1}^{n}(C_i - Y_{oi})}{\sum_{i=1}^{n}Y_{oi}}$ | (-∞, +∞) | 0 |
| Nash-Sutcliffe Efficiency Coefficient (*NSE*) | NA | $NSE = 1 - \dfrac{\sum_{i=1}^{N}(C_i - Y_{oi})^2}{\sum_{i=1}^{N}(C_i - \bar{Y}_o)^2}$ | (-∞,1] | 1 |
| Probability of Detection (POD) | NA | $POD=H/(H+M)$ | [0,1] | 1 |
| False Alarm Ratio (FAR) | NA | $FAR=F/(H+F)$ | [0,1] | 0 |
| Critical Success Index (CSI) | NA | $CSI=H/(H+M+F)$ | [0,1] | 1 |

Note: $Y_{oi}$ is the observation data and $C_i$ is the adjusted value using the WHU-SGCC method for test sample pixel; $\bar{Y}_o$ is the arithmetic mean of $Y_o$ and is given by $\bar{Y}_o = \dfrac{1}{n}\sum_{i=1}^{n}Y_{oi}$ ; $\bar{C}$ is the arithmetic mean of $C$ and is given by $\bar{C} = \dfrac{1}{n}\sum_{i=1}^{n}C_i$ ;

H represents the number of both observed and estimated precipitation events (successfully forecasted), and F is the number of false alarms when observed precipitation was below the threshold and estimated precipitation was above threshold (false alarms). M is the number of events in which the estimated precipitation was below the threshold and observed precipitation was above the threshold (missed forecasts). POD and FAR values are dimensionless numbers ranging from 0 to 1. The precipitation threshold (event/no event) was set to 0.1 mm/day.

**4 Results and Discussion**

There were 18482 daily pixels  adjusted by blending satellite estimations (CHIRP) and observations (rain gauge stations

) using the WHU-SGCC approach over the Jinsha River Basin during the JJA from 1990

to 2014. The number of pixels adjusted by each rule in the WHU-SGCC method is shown in Table 4. The number of C1 pixels was the number of training gauge stations accounting 0.16% of the total pixels (18482) inside the basin.

Due to the leave-one-out cross validation step, the different training samples will have the different number of C2, C3 and C4

pixels respectively inside the Jinsha River Basin. The number of C4 pixels was approximately 10822 with the percentage around 60%, the number of C3 pixels was approximately 4331 with the percentage ranging from 21.72% to 24.40%, and the number of C2 pixels was approximately 3300 with the percentage ranging from 15.59% to 18.36%.

The number of C1 and C2 was nearly 140, as well as 11493 C3 pixels, approximately 6344 C4 pixels, and the number of remaining C5 pixels was no more than 5%.

**Table 4** The number of each class pixels adjusted by each rule using the WHU-SGCC method inside the Jinsha River Basin.

| Validation gauge station | C1 Pixels (%) | C2 Pixels (%) | C3 Pixels (%) | C4 Pixels (%) |
|---|---|---|---|---|
| 52908 | 29 (0.16%) | 3066 (16.59%) | 4224 (22.85%) | 11163 (60.40%) |
| 56004 | 29 (0.16%) | 2882 (15.59%) | 4111 (22.24%) | 11460 (62.01%) |
| 56021 | 29 (0.16%) | 3311 (17.91%) | 4510 (24.40%) | 10632 (57.53%) |
| 56029 | 29 (0.16%) | 3338 (18.06%) | 4447 (24.06%) | 10668 (57.72%) |
| 56034 | 29 (0.16%) | 3300 (17.86%) | 4427 (23.95%) | 10726 (58.03%) |
| 56038 | 29 (0.16%) | 3209 (17.36%) | 4014 (21.72%) | 11230 (60.76%) |
| 56144 | 29 (0.16%) | 3347 (18.11%) | 4442 (24.03%) | 10664 (57.70%) |
| 56146 | 29 (0.16%) | 3183 (17.22%) | 4480 (24.24%) | 10790 (58.38%) |
| 56152 | 29 (0.16%) | 3173 (17.17%) | 4176 (22.59%) | 11104 (60.08%) |
| 56167 | 29 (0.16%) | 3362 (18.19%) | 4346 (23.51%) | 10745 (58.14%) |
| 56247 | 29 (0.16%) | 3385 (18.32%) | 4416 (23.89%) | 10652 (57.63%) |
| 56251 | 29 (0.16%) | 3301 (17.86%) | 4348 (23.53%) | 10804 (58.46%) |
| 56257 | 29 (0.16%) | 3313 (17.93%) | 4043 (21.88%) | 11097 (60.04%) |
| 56357 | 29 (0.16%) | 3352 (18.14%) | 4390 (23.75%) | 10711 (57.95%) |
| 56374 | 29 (0.16%) | 3341 (18.08%) | 4294 (23.23%) | 10818 (58.53%) |
| 56459 | 29 (0.16%) | 3345 (18.10%) | 4334 (23.45%) | 10774 (58.29%) |
| 56462 | 29 (0.16%) | 3380 (18.29%) | 4377 (23.68%) | 10696 (57.87%) |
| 56475 | 29 (0.16%) | 3345 (18.10%) | 4344 (23.50%) | 10764 (58.24%) |
| 56479 | 29 (0.16%) | 3305 (17.88%) | 4212 (22.79%) | 10936 (59.17%) |
| 56485 | 29 (0.16%) | 3393 (18.36%) | 4419 (23.91%) | 10641 (57.57%) |
| 56543 | 29 (0.16%) | 3373 (18.25%) | 4384 (23.72%) | 10696 (57.87%) |
| 56565 | 29 (0.16%) | 3241 (17.54%) | 4450 (24.08%) | 10762 (58.23%) |
| 56571 | 29 (0.16%) | 3306 (17.89%) | 4263 (23.07%) | 10884 (58.89%) |
| 56586 | 29 (0.16%) | 3387 (18.33%) | 4434 (23.99%) | 10632 (57.53%) |
| 56651 | 29 (0.16%) | 3340 (18.07%) | 4432 (23.98%) | 10681 (57.79%) |
| 56664 | 29 (0.16%) | 3368 (18.22%) | 4262 (23.06%) | 10823 (58.56%) |
| 56666 | 29 (0.16%) | 3323 (17.98%) | 4431 (23.97%) | 10699 (57.89%) |
| 56671 | 29 (0.16%) | 3356 (18.16%) | 4367 (23.63%) | 10730 (58.06%) |
| 56684 | 29 (0.16%) | 3335 (18.04%) | 4278 (23.15%) | 10840 (58.65%) |
| 56778 | 29 (0.16%) | 3347 (18.11%) | 4277 (23.14%) | 10829 (58.59%) |

[Figure]

**Figure 4** The number of pixels adjusted by each rule using the WHU-SGCC method.

**4.1 CDFs of Rule 1 and Rule 2 results**

Figure 5 shows the daily average precipitation for observations, CHIRP, C1 (Fig. 5 (a)) and C2 (Fig. 5 (b)) in JJA 2016.
Compared to the gauge or grid observations, CHIRP estimations deviated from the observations in Jinsha River Basin.
However, the adjusted values for the C1 and C2 pixels improved the estimates and approximated the observations with
application of Rule 1 and Rule 2 of the WHU-SGCC method. This result demonstrates that Rule 1 and Rule 2 of WHU-SGCC
method are effective in correcting consistent biases and considerably reduce the systematic biases of CHIRP. These
improvements not only adjust the bias of satellite estimations but also preserve the original CHIRP pixel values which are
close to the corresponding observed data. These adjustments provide reliable precipitation estimates for the C1 and C2 pixels,
which supports further study using the WHU-SGCC method, especially for areas in which rain gauges are limited.

[Figure]

                (a)                                    (b)

**Figure 5** CDFs of seasonal mean daily observations, CHIRP, C1 and C2 estimations for the Jinsha River Basin in JJA 2016

**4.2 Spatial Clustering of Rule 3 results**

To adjust the pixels other than for the gauged and gridded points, the pixels physically similar to the C1 and C2 pixels were selected. According to Rule 3, C3 pixels were identified in a spatial scope defined by the FCM method. Figure 6 shows the twenty spatial clusters with consideration of the terrain factors. Overall, the spatial results of FCM have many of the same characteristics as spatial areas defined by terrain changes, especially with respect to slope and runoff directions, which may influence regional rainfall to some extent.

[Figure]

**Figure 6** Spatial clustering as defined by FCM for the Jinsha River Basin.

After Rule 3, each C3 pixel has a good SCC with a C1 or C2 pixel in its cluster; the statistical analysis is shown in Fig. 7. It was found that the average SCC value was 0.6. Therefore, the regression model established in Rule 3 for C1 and C2 before and after adjustment is applicable for each corresponding C3 pixel.

[Figure]

**Figure 7** Frequency distribution histogram for Spearman's correlation coefficient (SCC) for C3 pixels and their corresponding C1 and C2
pixels using Rule 3.

It is important to note that 62.18% of the pixels satellite precipitation estimates were adjusted by Rule 3 of the WHU-SGCC
method. The accuracy assessment of C3 pixels is shown in Table 4. Validation statistics indicate that compared with the CHIRP
and CHIRPS satellite estimations, the WHU-SGCC approach provides best adjustments based on all the statistical indicators
at C3 pixels. With the improvement of precipitation accuracy by WHU-SGCC of C3 pixels, the adjustments of C4 pixels,
which mainly rely on C3 pixel corrections, are reasonable.

**Table 4** Accuracy assessment of C3 pixels for JJA 2016.

| Statistic | WHU-SGCC | CHIRP | CHIRPS |
| --- | --- | --- | --- |
| SCC | 0.3518 | 0.3176 | 0.2476 |
| RMSE | 5.1776 | 5.6686 | 7.0311 |
| MAE | 3.5226 | 3.7353 | 4.6909 |
| BIAS | -0.0831 | -0.2366 | -0.2404 |
| NSE | -0.0590 | -0.2693 | -0.9528 |
| POD | 1.0000 | 0.8900 | 0.3396 |
| FAR | 0.0687 | 0.0749 | 0.0763 |
| CSI | 0.9313 | 0.8302 | 0.3304 |

**4.3 1 Model performance based on overall accuracy evaluations**

To test the performance of the WHU-SGCC method for precipitation estimates, the statistical analyses of SCCPCC, RMSE,
BAE, BIAS, NSE, POD, FAR, and CSI were calculated and are presented in Table 5 (The results were derived from the 22
clusters for FCM in Rule 2 shown in Appendix B, and $\alpha$ =0.1 for IDW in Rule 4 after the comparison of RMSEs). Compared
with the satellite images of CHIRP and CHIRPS, the results of the WHU-SGCC provide the greatest improvements for regional
daily precipitation estimates over the Jinsha River Basin during thein JJA from 1990 to 20164. After bias adjustment of the
WHU-SGCC, SCC PCC was improved by 17.383.34% and 39.6231.81% compared to CHIRP and CHIRPS, respectively.
Meanwhile, the RMSE and, MAE and BIAS of the WHU-SGCC was decreased by 4.206.91% and, 6.236.59% and 11.83%
compared to CHIRP, and by 19.1022.71% and, 24.4722.15% and 41.93% compared to CHIRPS. Although, the absolute value
of BIAS of WHU-SGCC was no significant improvement than CHIRP and slightly higher than CHIRPS, all of the values were

approximately to 0. This results of BIAS indicates that all the three kinds of data were much the same on the performance of
relative bias. Nevertheless, the NSE of the WHU-SGCC reached -0.0864, an increase of 93.33% and 98.32%
compared to CHIRP and CHIRPS, respectively. The NSE of WHU-SGCC was still negative, but it was improved to be zero
that indicates the adjusted results are close to the average level of the rain gauge observations, while the NSEs of CHIRP and
CHIRPS were much worse. It is noted that the POD of WHU-SGCC was approximate to 1, better than CHIRP and CHIRPS,
and the FAR of WHU-SGCC was 0.11, lower than CHIRP and CHIRPS, which represents the better ability on precipitation
event predictions of the WHU-SGCC.

**Table 5** .

|  |  |  |  |
|---|---|---|---|
|  |  |  |  |
|  |  |  |  |
|  |  |  |  |
|  |  |  |  |
|  |  |  |  |
|  |  |  |  |
|  |  |  |  |
|  |  |  |  |

| Statistic | WHU-SGCC | CHIRP | CHIRPS |
|---|---|---|---|
| PCC | 0.2536 | 0.2454 | 0.1924 |
| RMSE | 8.7608 | 9.4108 | 11.3354 |
| MAE | 5.4564 | 5.8415 | 7.0088 |
| BIAS | -0.0167 | -0.0443 | -0.0134 |
| NSE | -0.0139 | -0.2083 | -0.8293 |
| POD | 0.9932 | 0.9578 | 0.4351 |
| FAR | 0.1146 | 0.2323 | 0.1601 |
| CSI | 0.8799 | 0.7405 | 0.4010 |

The spatial distributions of the statistical comparisons between observations and WHU-SGCC precipitation estimations are
shown in Fig. 4. The variation of  PCC as seen in Fig. 4 (a) shows that low correlations are observed in areas with lower
elevation, particularly in the southern Jinsha River Basin where there is higher precipitation and a greater density of rain gauges.
This result is in contrast to the result in (Rivera et al., 2018), because of the few days for heavy rains in this study area.
The higher correlations noted over the north central area of the river basin are in a drier region with complex terrain and sparse
rain gauges. With respect to the spatial distribution of RMSE, Fig. 4 (b) indicates that smaller errors are scattered in the
northwest area of the river basin, with values lower than 5 mm, while the highest errors, which are over 10 mm, are located
over the border between the lower reaches of the Jinsha Jiang River and the river basin. All the values of MAE are below 12
mm and the spatial behaviour is similar to that of the RMSE. Fig. 4 (c) shows that the lower MAE values  were located
over the mountainous region southwest of Qinghai and west of Sichuan, with values below 6 mm. The spatial distribution of
the BIAS (Fig. 4 (d)) indicates that the WHU-SGCC has good agreement with the observations, with the most values ranging
from -0.1% -0.1%. All the spatial distribution statistics indicate that the statistical relationships established during the
process of the WHU-SGCC method is susceptible to the mode values of the rain gauge stations data. Although the average
annual precipitation in the southern Jinsha River Basin was more than 600 mm (Fig.2), the days of light rain were still in the
great percentage that caused the large biases and limited the performance over the south area, while there were sufficient data with similar precipitation features for WHU-SGCC over the north area. Nevertheless, the WHU-SGCC approach is still effective in adjusting the satellite biases by blending with the observations, particularly in the complicated mountainous region where there  were higher  PCC corresponding to lower values of RMSE, MAE and BIAS.

[Figure]

(a)

(b)

(c)

(d)

[Figure]

(a)                 (b)

(c)                 (d)

**Figure 8 4** Spatial distribution of the statistical analyses of the overall agreement between observations and the WHU-SGCC estimations on leave-one-out cross validation 30% validation for during the JJA from 1990 to 2014 2016: a) Spearman's Pearson's correlation coefficient, b) root mean square error c) mean absolute error, and d) relative bias.

**4.4 Model performance based on daily accuracy evaluations**

After overall accuracy evaluations for JJA were conducted, further evaluations of daily accuracy were undertaken and the results are shown in Fig. 9. The evaluation of daily accuracy indicates that the WHU-SGCC reduces errors and biases compared to CHIRP and CHIRPS, with especially greatly decreases compared to CHIRPS. The RMSE and MAE derived from the WHU-SFCC were reduced by approximately 5% and 30% compared to CHIRP and CHIRPS, respectively. However, the greatest reduction was reflected in the BIAS, with at least an 18% and 30% reduction compared to CHIRP and CHIRPS, respectively. Therefore, the WHU-SGCC approach is effective for adjustments of daily precipitation estimates, and improves estimate performance.

[Figure]

**Figure 9**

 In general, the precipitation estimated using the WHU-SGCC method
are superior to other products.

[Figure]

**Figure 10**

**4.2 Model performance based on daily accuracy evaluations**

After overall accuracy evaluations for JJA were conducted, further evaluations of daily accuracy were investigated and the results were shown in Fig. 5. The evaluation of daily accuracy indicates that the PCCs of WHU-SGCC, CHIRP and CHIRPS

were roughly the same, while WHU-SGCC has the reduction of errors and biases compared to CHIRP and CHIRPS, especially the greater decreases when compared to CHIRPS. Figure. 5 indicates that there was no significant increase in PCC, however,

PCC is a relative metric about the magnitude of the association between paired variables, and a relative consistency may not mean absolute proximity. Thus, the absolute measure indicated by RMSE may be more reasonable. In this study, the RMSE

and MAE derived from the WHU-SGCC were reduced by approximately 15% and 30% compared to CHIRP and CHIRPS, respectively. The slight reduction was reflected in the BIAS, with an 8% to 45% reduction compared to CHIRP and CHIRPS, while all the values were concentrated between -0.5 and 0.5. All the precipitation estimations derived from WHU-SGCC,

CHIRP, and CHIRPS represented well agreement with the observations in relative bias. The WHU-SGCC method shown obvious improvement in the NSE relative to CHIRP and CHIRPS, while the values were still less than 0 which may be due to the inherent uncertainty in the CHIRP. Moreover, in terms of POD, FAR and CSI, the WHU-SGCC method seems to be more promising in detecting precipitation than CHIRP and CHIRPS, although it performs poorly on FAR relative to CHIRPS in some days. However, the POD and CSI of WHU-SGCC were closest to 1. Overall, the WHU-SGCC approach is effective for adjustments of daily precipitation estimates, and improves estimate performance.

[Figure]

**4.5 3 Model performance on rain events predictions**

To measure the WHU-SGCC performance  on different rain events predictions, the daily precipitation thresholds of 0.1, 10, 25, and  mm were considered, and the result  was shown in Table 6 and Table . The days of each class of rain events at the validation gauge station during the JJA from 1990 to 2014 were shown in Table 5. The major rain events inside the Jinsha River Basin were light rain (0.1-10 mm), accounting for 54.76% of the total days (the average percentage of rain event days in its total days at each gauge station), while the days with daily precipitation over the 50 mm was least, only accounting for 0.72%. And the percentage of the daily precipitation of <0.1, 10-25, and 25-50 mm were 26.89%, 14.01% and 3.62% respectively. The result indicated that the average daily precipitation was less than 10 mm, though in the summer seasons during the multi-year. As well as, the spatial distribution of precipitation was also uneven, with an increase from north to south.

~~In terms of performance with respect to different daily rain events, the WHU-SGCC approach had the lowest error, as indicated by RMSE, MAE and BIAS for events with total rainfall between 1 and 20 mm, but WHU-SGCC performance for heavy rain (20-40 mm) events did not improve compared to CHIRP, though it was better than that of CHIRPS. Although the WHU-SGCC approach improved accuracy for light rain events, its behaviour for heavy rain ($\geq$ 40 mm) events was not as good as CHIRP and CHIRPS, as shown in Fig. 11. These results indicate that WHU-SGCC is applicable for the detection of rainfall events with less than 20 mm precipitation, while there is insufficient observational data for the validation of WHU-SGCC performance during heavy rain events, which represented less than 4% of all observational data and were not sufficient to fully test performance of the model.~~

**Table 6** The days of each class of rain events at the validation gauge station during the JJA from 1990 to 2014 inside the Jinsha River

| Rain event (mm) / Validation gauge station | <0.1 | [0.1,10) | [10,25) | [25,50) | >=50 | Total days |
|---|---|---|---|---|---|---|
| 52908 | 637 | 1186 | 134 | 9 | 0 | 1966 |
| 56004 | 628 | 1243 | 128 | 3 | 0 | 2002 |
| 56021 | 535 | 1305 | 166 | 9 | 0 | 2015 |
| 56029 | 556 | 1328 | 190 | 5 | 0 | 2079 |
| 56034 | 558 | 1351 | 185 | 17 | 0 | 2111 |
| 56038 | 459 | 1329 | 222 | 16 | 0 | 2026 |
| 56144 | 562 | 1153 | 321 | 25 | 0 | 2061 |
| 56146 | 467 | 1278 | 267 | 19 | 0 | 2031 |
| 56152 | 466 | 1255 | 307 | 35 | 1 | 2064 |
| 56167 | 565 | 1234 | 278 | 20 | 0 | 2097 |
| 56247 | 591 | 1089 | 246 | 34 | 0 | 1960 |
| 56251 | 466 | 1247 | 320 | 30 | 0 | 2063 |
| 56257 | 336 | 1212 | 429 | 59 | 0 | 2036 |
| 56357 | 313 | 1247 | 373 | 63 | 1 | 1997 |
| 56374 | 393 | 1191 | 351 | 47 | 0 | 1982 |
| 56459 | 487 | 1080 | 377 | 102 | 13 | 2059 |
| 56462 | 185 | 1315 | 430 | 86 | 2 | 2018 |
| 56475 | 544 | 983 | 352 | 148 | 20 | 2047 |
| 56479 | 667 | 931 | 298 | 156 | 28 | 2080 |
| 56485 | 588 | 905 | 232 | 100 | 37 | 1862 |
| 56543 | 332 | 1200 | 289 | 41 | 1 | 1863 |

| | | | | | | |
|---|---|---|---|---|---|---|
| 56565 | 526 | 1020 | 349 | 120 | 13 | 2028 |
| 56571 | 674 | 819 | 301 | 159 | 49 | 2002 |
| 56586 | 730 | 950 | 223 | 79 | 9 | 1991 |
| 56651 | 402 | 1056 | 391 | 137 | 31 | 2017 |
| 56664 | 727 | 797 | 306 | 166 | 56 | 2052 |
| 56666 | 858 | 791 | 226 | 128 | 44 | 2047 |
| 56671 | 616 | 886 | 289 | 148 | 70 | 2009 |
| 56684 | 768 | 899 | 246 | 114 | 19 | 2046 |
| 56778 | 682 | 930 | 274 | 119 | 43 | 2048 |

In terms of performance with respect to different daily rain events, the WHU-SGCC approach had the lowest error, as indicated by RMSE, MAE and BIAS for events with total rainfall between 1lower than  and 2025 mm, but the performance of WHU-SGCC performance for total rainfall higher than 25mm heavy rain (20-40 mm) events did not improve compared to CHIRP and CHIRPS (Table 6), though it was better than that of CHIRPS. This negative performance on the total rainfall higher than 25 mm was probably caused by the precipitation conditions inside the Jinsha River Basin (Table 6). The average daily precipitation was less than 10 mm inside the basin, during the multi-year summer seasons, which provided a large amount of rain gauge stations data with the values lower than 10 mm, that caused a significantly impact on the statistical relationships establishment for WHU-SGCC. In hence, the approach of WHU-SGCC is applicable for the detection of rainfall events over the Jinsha River Basin, with the average daily precipitation less than 10 mm, or even than 25mm. Due to the 4.34% of summer days with the daily precipitation over the 25 mm, the performance of WHU-SGCC on these rain events was poorer than the results of CHIRP and CHRPS.

Although the WHU-SGCC approach improved accuracy for light rain events, its behaviour for heavy rain ($\geq$ 40 mm) events was not as good as CHIRP and CHIRPS, as shown in Fig. 9. These results indicate that WHU-SGCC is applicable for the detection of rainfall events with less than 20 mm precipitation, while there is insufficient observational data for the validation of WHU-SGCC performance during heavy rain events, which represented less than 4% of all observational data and were not sufficient to fully test performance of the model.

**Table 6** Accuracy assessment on wet precipitation events for JJA 2016

| | RMSE | | | MAE | | | BIAS | | |
|---|---|---|---|---|---|---|---|---|---|
| Rain Event | WHU-SGCC | CHIRP | CHIRPS | WHU-SGCC | CHIRP | CHIRPS | WHU-SGCC | CHIRP | CHIRPS |
| [0.1,1) | 4.1609 | 4.5077 | 5.2762 | 2.3569 | 2.2940 | 2.2187 | 4.8423 | 4.9153 | 4.7541 |
| [1 , 2) | 4.2658 | 4.7385 | 6.2943 | 2.4820 | 2.5563 | 3.3707 | 1.3491 | 1.8199 | 2.3996 |
| [2 , 5) | 4.8378 | 5.2392 | 7.7315 | 3.2026 | 3.4011 | 5.2681 | 0.2808 | 1.0023 | 1.5525 |
| [5 , 10) | 4.8765 | 5.5616 | 8.4619 | 4.0646 | 4.5505 | 6.8346 | -0.2292 | 0.6315 | 0.9485 |
| [10,20) | 8.8240 | 9.5254 | 11.5381 | 7.5957 | 8.3153 | 10.0287 | -0.4627 | 0.6142 | 0.7408 |
| [20,40) | 17.3305 | 17.0107 | 18.8758 | 15.5649 | 15.2646 | 16.4080 | -0.6035 | 0.6011 | 0.6461 |
| $\geq$40 | 95.8157 | 95.5185 | 95.2107 | 64.6789 | 64.1252 | 64.6337 | -0.8850 | 0.8774 | 0.8844 |

**Table 7** Accuracy assessment on liquid precipitation events during the JJA from 1990 to 2014

| | RMSE | | | MAE | | | BIAS | | |
|---|---|---|---|---|---|---|---|---|---|
| Rain Event | WHU-SGCC | CHIRP | CHIRPS | WHU-SGCC | CHIRP | CHIRPS | WHU-SGCC | CHIRP | CHIRPS |
| <0.1 | 4.7253 | 5.0802 | 7.1643 | 2.5927 | 2.9562 | 2.9145 | / | / | / |
| [0.1,10) | 4.1661 | 6.8684 | 9.6022 | 3.9885 | 4.5534 | 6.2462 | 0.8021 | 1.4435 | 1.9842 |
| [10,25) | 10.4281 | 11.0848 | 13.4427 | 9.2722 | 9.6866 | 11.5909 | -0.5762 | 0.6342 | 0.7559 |
| [25,50) | 25.7494 | 24.5600 | 25.4975 | 24.8386 | 23.0967 | 23.4927 | -0.7784 | 0.7250 | 0.7388 |

| | ≥50 | 56.6072 | 54.5037 | 52.7875 | 54.4168 | 52.1557 | 49.4318 | -0.8861 | 0.8297 | 0.7852 |

批注 [S1]: This figure was same to the table 7, so we deleted it.

[Figure]

**5 Data availability**

All the resulting dataset derived from the WHU-SGCC approach is available on PANGAEA, with the following DOI: https://doi.pangaea.de/10.1594/PANGAEA.896615 (Shen et al., 2018). The high-resolution (0.05°) daily precipitation estimation data over the Jinsha River Basin in the summer from 1990 to 2014 can be downloaded in TIFF format.

**6 Conclusions**

This study provides a novel approach in the WHU-SGCC method for merging daily satellite-based precipitation estimates with observations. A case study of Jinsha River Basin was conducted to verify the effectiveness of the WHU-SGCC approach   JJA from 1990 to 2014, and the adjusted precipitation estimates were compared to CHIRP and CHIRPS.

WHU-SGCC aims to reduce systematic and random errors in CHIRP over  a region that has complicated mountainous terrain and sparse rain gauges. To the best of the authors' knowledge, this study is the first to use daily CHIRP and CHIRPS data in this area.

According to our findings, the following conclusions can be drawn: (1) The WHU-SGCC method is effective for the adjustment of precipitation biases from point to surface. The precipitation estimated by the WHU-SGCC method can achieve greater accuracy, which was evaluated with PCC, RMSE, MAE, BIAS, NSE, POD, FAR and CSI. Particularly, the  NSE statistic was improved by 93.33% and 98.32% compared to CHIRP and CHIRPS, respectively, and all measured errors were reduced except the BIAS with no significant improvement, but approximately to 0. The results show that compared to CHIRPS, the WHU-SGCC approach can achieve substantial improvements in precipitation estimate accuracy. (2) Moreover, the spatial distribution of precipitation estimate accuracy derived from the WHU-SGCC method is related to the complexity of the topography. These random errors over the lower evaluations and the large size of  light precipitation events with short duration rainstorms in the region resulted in a limited improvement in accuracy, with  PCC values less than 0.3,  However, higher  PCC and lower errors were observed over the north central area of the river basin, which is a drier region with complex terrain and sparse rain gauges. All the spatial distribution statistics indicate that the WHU-SGCC method is promising  for adjustment of satellite biases by blending with the observations over the complex terrain region. (3) The leave-one-out cross validation of WHU-SGCC on daily rain events confirmed that the model was effective in the detection of precipitation events less than 25 mm due to the less average annual precipitation inside the Jinsha River Basin. According to the comparison, the WHU-SGCC approach achieves error reductions for the RMSE, MAE and BIAS statistics for rain events within the range of 1-25 mm. Specifically, compared with CHIRP, the RMSE value was reduced by approximately by 9%, the MAE value by 4.28%  12.41%, and the absoulte BIAS value by 9.15%  44.43%; compared with CHIRPS, the RMSE and MAE values were reduced by 11.04%  56.61%, and the absolute BIAS value by 23.77%  59.58%.

In conclusion, the WHU-SGCC approach can help adjust the biases of daily satellite-based precipitation estimates over the Jinsha River Basin, the complicated mountainous terrains with sparse rain gauges, particularly on the daily precipitation events with less than 25 mm in the summer. This approach is a promising tool to monitor monsoon precipitation over the Jinsha River Basin, considering the spatial correlation and historical precipitation characteristics between raster pixels located in regions with similar topographic features. Future development of the WHU-SGCC approach will focus on the following three aspects: (1) improvement of the adjusted precipitation quality by blending in different rain events and applying in all seasons; (2) introduction of more climatic factors and mulit-model ensemble  to achieve a more accurate spatial distribution of precipitation; and (3) investigation of the performance over the other areas and on the particular hydrological case to validate the WHU-SGCC approach.

**Appendix A: The selection of decision trees for random forest regression**

[Figure]

[Figure]

[Figure]

[Figure]

[Figure]

**Figure A1** The change of out-of-bag (OOB) error with the number of decision trees increase by means of random forest regression at each gauge station.

**Appendix B: Spatial Clustering from the FCM method**

[Figure]

**Figure B1** The optimum number of clusters determined by the maximum $L(c)$ with the iterative process.

[Figure]

**Figure B2** Spatial clustering as defined by FCM for the Jinsha River Basin.

This appendix demonstrates how to set the number of clusters in the FCM method.

To adjust the pixels other than for the gauged stations, the pixels statistically similar to the C1 were selected. According to

Rule 2, C2 pixels were identified in a spatial scope defined by the FCM method. In the following experiments of Rule 2, we set the parameters $m=2, \varepsilon=0.00001$ and the maximum number of iterations was set 1000 (an enough large value with the consideration of the algorithm efficiency). In order to determine the optimal numbers of clusters, $c$ value was conducted in the range from 1 to 30 with an incremental interval value of 1 in this study. During the running of FCM approach, the values of

$L(c)$ were shown in Fig B1. The optimum number of clusters was 22, with the number of iterations was 690 less than the maximum number of iterations.

Therefore, the number of clusters was set to 22 and the number of iterations was still set to 1000 for fully operations by means of FCM. The spatial clusters results with consideration of the terrain factors was shown in Fig. B2. Overall, the spatial

results of FCM have many of the same characteristics as spatial areas defined by terrain changes, especially with respect to
slope and runoff directions, which may influence regional rainfall to some extent.

**7 Acknowledgments**

This work was supported by the National Natural Science Foundation of China program (no. 41771422), the Nature Science
Foundation of Hubei Province (no. 2017CFB616), the fundamental research funds for the central universities (no.
2042017kf0211), and the LIESMARS Special Research Funding.
The authors would like to thank data support: the Climate Hazards Group at the University of California, Santa Barbara, for
providing CHIRP and CHIRPS datasets (http://chg.ucsb.edu/data/), and the National Climate Center (NCC) of the China
Meteorological Administration (CMA) for providing the daily rain gauged observations and gridded precipitation observations
(http://data.cma.cn/). The authors also thank the PANGAEA Data Publisher for Earth & Environmental Science platform for
providing the storage to disseminate the data generated in this experiment.
The authors are grateful for the editor and anonymous reviewers for their useful suggestions that clearly improved this paper.

[revised manuscript text omitted]

---

## Referee Report (RR1)

Review of

*WHU-SGCC: A novel approach for blending daily satellite (CHIRP) and precipitation observations over the Jinsha River Basin*

The is my second review of the manuscript. Compared to the first version the authors have substantially improved the manuscript. They extended the data base and performed a more sophisticated analyses to obtain statistically robust results. They have taken all my suggestions into account. Therefore, I recommend publishing the manuscript after addressing a few minor issues:

- L.49-50: Use the plural form of the verbs in this sentence.
- L.199-210: Please provide a bit more detail about the random forest regression and the bootstrapping used in the study!
- L.357-364: This part is difficult to understand. Please perform a language check and rephrase this part.
- L401: shown → showed
- Figure 5: The labelling (a,b,c,…) is missing in the subfigures.
- L.417: It should be Table 6 here, not Table 5.
- L.453: I guess, it is supposed to be "elevation" instead of "evaluation"?
- L.561: …by approximately by…→ ..by approximately…
- L.466-468: What is the typical daily rainfall amount in during the monsoon season? Considering, that the proposed method doesn't perform very well in case of heavy precipitation, it might not be the appropriate tool for monitoring monsoon precipitation.

---

## Referee Report (RR2)

**WHU-SGCC: A novel approach for blending daily satellite (CHIRP) 1 and precipitation observations over the Jinsha River Basin by Shen et al.**

General comments:

The manuscript describes a method of merging daily gauge measurements with satellite precipitation product (CHIRP) in the mountainous region of Jinsha River Basin in China. The methodology seems to be quite rigorous in statistical sense. The algorithm is not perfect as it does not take into account the skewed nature of precipitation distribution and the improvements are mostly shown on light rain events but not the heavy rain events. But overall it is a commendable effort and worth publishing.

One major question is the choice of CHIRP data instead of CHIRPS to start with. Perhaps because CHIRPS with all additional gauge and model data actually performs worse than CHIRP in this particular area as the statistics later in the paper shows. If this is the reason, the authors should mention it in the beginning. It is expected that the new data will perform better than CHIRPS because WHU-SGCC incorporates more surface gauge measurements. The question is then the improvement is due to more gauge stations or better merging and correction methodology? If the WHU-SGCC method only takes the station data used by CHIRPS, will the results be better than CHRIPS?

Besides this, the methodology and validation seem to be reasonable. The authors should emphasize that the validation is against the gauge measurements (from using leave-one-out) and not against CHIRP and CHIRPS. Only the error statistics of CHIRP and CHIRPS data are used for comparison purpose. There are still quite a lot of English usage problems. More careful proofreading is needed.

Minor comments:

Line 46, 49: Use Huffman et al. (2018) reference for IMERG data and Huffman et al (2010) for TMPA data. These are official IMERG and TMPA references.

Huffman, G. J, D. T. Bolvin, D. Braithwaite, K. Hsu, R. Joyce, P. Xie (2018). NASA Global Precipitation Measurement Integrated Multi-satellitE Retrievals for GPM (IMERG), NASA Algorithm theoretical basis document (ATBD) version 5.2., 35 pp. https://pmm.nasa.gov/sites/default/files/document_files/IMERG_ATBD_V5.2.pdf

Huffman, G.J., R.F. Adler, D.T. Bolvin, and E. Nelkin (2010). The TRMM Multi-satellite Precipitation Analysis (TMPA). In F. Hossain and M. Gebremichael (Ed.), *Satellite Rainfall Applications for Surface Hydrology*. (3-22). Springer Verlag. ISBN: 978-90-481-2914-0

Line 41-62: These satellite precipitation data or merged data seem to be introduced in random order.

Line 65: The sentence "Table 1 shows …" should appear in the beginning of previous paragraph before introducing different data set.

Line 69: Temporal resolution?

Table 1: GPCP has a daily product
CMAP has pentad product?
CHIRPS: what is Dekad/Pentad? The temporal resolution does not match 3-day and 6-day resolution described in Line 151.

Line 137 and Figure 2 caption: annual precipitation during JJA is not correct usage. Should remove "annual".

Line 157: TRMM 3B42 is itself a merged product including both geostationary IR and microwave measurements from polar orbiting satellites and some gauge data.

Line 161: Are you sure that the IR rainfall estimates have pentad resolution?

Line 167-168: Don't understand why CHIRPS is used as validation data. The new data is supposed to perform better because more surface station is available?

Line 187: Analyze

Line 196: replace "statistically" with "statistical"
Line 231: "molecular" should be numerator

Rule 2 generates clusters of precipitation areas for the JinSha River region. Can you plot the cluster map?

Line 291: Equation (12): It seems that you choose to adjust C3 precipitation based on one C2 pixel that provides maximum precipitation value.  What is the rational here?

Line 311: Don't understand this sentence. Aren't C4 pixels adjusted in Rule 4? How are C1 pixels adjusted in the end?

Line 430: "In hence", there is no such usage. Use "Therefore" instead.

Line 470: Should be "multi-model"

Table 5: Why CHIRPS has much worse performance than CHIRP?

---

## Author Response (AR2)

**Reply to Report #1 (Referee #3)**

**Manuscript ID:** essd-2018-150
**Title:** WHU-SGCC: A novel approach for blending daily satellite (CHIRP) and precipitation observations over the Jinsha River Basin
**Journal:** Earth System Science Data
**Type:** Article

Dear Reviewer,

Thank you for your insight comments and suggestions. We have modified the manuscript accordingly. We trust that all of your comments have been addressed accordingly in the revised manuscript. If you have further suggestions for changes, please let us know. The detailed corrections are listed below point by point:

All changes in the manuscript are marked with red color.

**Major comments**

This is my second review of the manuscript. Compared to the first version the authors have substantially improved the manuscript. They extended the data base and performed a more sophisticated analyses to obtain statistically robust results. They have taken all my suggestions into account. Therefore, I recommend publishing the manuscript after addressing a few minor issues:

**Minor comments**

1. L.49-50: Use the plural form of the verbs in this sentence.
**Answer:** Thanks. Done. We have changed the introduction and deleted the GOES-R.

2. L.199-210: Please provide a bit more detail about the random forest regression and the bootstrapping used in the study!
**Answer:** Thanks. Done. We have added a bit more detail about forest regression and the bootstrapping.
**Change:** changed

"RFR is a machine-learning algorithm for a predictive model with a large set of regression trees in which each tree in the ensemble is grown from a bootstrap (Johnson, 1998) sample drawn with replacement from the training set. The final prediction is obtained by combining the results of the prediction methods applied to each bootstrap sample (Genuer et al., 2017). The predicted value is calculated by the mean of all trees.

$$Y_o = f_{RFR}(Y_s) \qquad (1)$$

where $Y_o$ denotes each gauge historical observations and $Y_s$ denotes the corresponding CHIRP grid cell values at C1 pixels, $f_{RFR}$ is constructed from the time series $Y_o$ (dependent variable) and $Y_s$ (independent variable) by means of RFR. The number of decision trees was set to 500, which was determined by out-of-bag (OOB) error (Appendix B). The OOB error reached the minimum value when the number of decision trees was less than 500. "

to

"RFR is a machine-learning algorithm for a predictive model with a large set of regression trees in which each tree in the ensemble is grown from a bootstrap sample (Johnson, 1998) drawn with a replacement from the training set. In the process of establishing regression trees, a subset of variables for each node is selected to avoid overfitting. The final prediction is obtained by combining the results of the prediction methods applied to each bootstrap sample (Genuer et al., 2017). The predicted value is calculated by the average of the values from all of the decision trees. Each tree can be expressed as

$$Tree_k = f_{RFR}(Y_o, \Theta_{Y_{s_k}}), \quad k=1\ldots n \tag{1}$$

where $Y_o$ denotes the historical observations at each gauge at the C1 pixels, $\Theta_{Y_{s_k}}$ is a randomly selected vector from $Y_s$, $Y_s$ denotes the corresponding CHIRP grid cell values at the C1 pixels, $n$ is the number of trees, and $f_{RFR}$ is constructed from the time series $Y_o$ (dependent variable) and $Y_s$ (independent variable) by means of RFR. The bootstrap sample will be the training set used for growing the tree. The error rate (out-of-bag, OOB) left out of one-third of the training data is also monitored to determine the number of decision trees. In this study, the minimum OOB error rate was reached when the number of decision trees $n$ was less than 500 (Appendix C)."

3. L.357-364: This part is difficult to understand. Please perform a language check and rephrase this part.
**Answer:** Thanks. Done. We have modified the grammar.
**Change:** changed
"Although, the absolute value of BIAS of WHU-SGCC was no significant improvement than CHIRP and slightly higher than CHIRPS, all of the values were approximately to 0. This results of BIAS indicates that all the three kinds of data were much the same on the performance of relative bias. Nevertheless, the NSE of the WHU-SGCC reached -0.0137, an increase of 93.33% and 98.32% compared to CHIRP and CHIRPS, respectively. The NSE of WHU-SGCC was still negative, but it was improved to be zero that indicates the adjusted results are close to the average level of the rain gauge observations, while the NSEs of CHIRP and CHIRPS were much worse. It is noted that the POD of WHU-SGCC was approximate to 1, better than CHIRP and CHIRPS, and the FAR of WHU-SGCC was 0.11, lower than CHIRP and CHIRPS, which represents the better ability on precipitation event predictions of the WHU-SGCC."
To
"Although the absolute value of the BIAS of the WHU-SGCC in fall are not significantly better than those of CHIRP and CHIRPS, all of the values are approximately 0. The NSEs of the WHU-SGCC reached 0.2836, 0.2944 and 0.1853 in the spring, fall and winter, respectively, which are substantially better than the negative or zero values of CHIRP and CHIRPS. In the summer, the NSE of the WHU-SGCC is still negative, but it is improved to be nearly zero, which indicates that the adjusted results are similar to the average level of the rain gauge observations. It is worth noting that in the spring, summer and fall, the POD values of the WHU-SGCC are in the range of 0.95 to 1, better than CHIRP and CHIRPS, and the FAR values of the WHU-SGCC are approximately 0.3, lower than CHIRP and CHIRPS; these results represent the better ability of the WHU-SGCC method to predict precipitation events. The rainfall detection ability is the worst in the winter compared to the other seasons. This can be explained by the seasonal distribution of precipitation in the Jinsha River Basin, in which the most rainfall occurs in the summer, followed by the fall, spring and winter. The worst errors of forecasting performance in the summer may be attributed to the highest precipitation. The limited precipitation event detection in the winter could also be explained by the lowest precipitation (Xu et al., 2019)."

4. L401: shown –> showed
**Answer:** Thanks. Done.
**Change:** changed "shown" to "provides".

5. Figure 5: The labelling (a,b,c,…) is missing in the subfigures.
**Answer:** Thanks. Done.
**Change:** We have added the labelling (a,b,…,h) into Figure 5.

6. L.417: It should be Table 6 here, not Table 5.
**Answer:** Thanks. Done.
**Change:** changed Table 6 to Table 5 here.

7. L.453: I guess, it is supposed to be "elevation" instead of "evaluation"?
**Answer:** Thanks. Done.
**Change:** changed "evaluations" to "elevation".

8. L.561: …by approximately by… ..by approximately…
**Answer:** Thanks. This mistake may on L.461.
**Change:** "reduced by approximately by" to "reduced by approximately".

9. What is the typical daily rainfall amount in during the monsoon season? Considering, that the proposed method doesn't perform very well in case of heavy precipitation, it might not be the appropriate tool for monitoring monsoon precipitation.
**Answer:** Thanks. We added all four seasons for WHU-SHCC
The multi-year (1990-2014) average seasonal precipitation over the Jinsha River Basin increases from north to south (Fig. 2). The dynamic and uneven distribution of precipitation is influenced distinctly by the seasonal climate. Most of the precipitation falls in the summer, with the average seasonal precipitation ranging from less than 250 mm to more than 600 mm, while the average seasonal precipitation during the winter is no more than 50 mm. The average seasonal precipitation and spatial distribution in the spring are similar with those in the fall, with values concentrated in the range of 50 mm to 200 mm.

**Change:** "This approach is a promising tool to monitor monsoon precipitation over the Jinsha River Basin" to "This approach is a promising tool to monitor daily precipitation over the Jinsha River Basin".

**Reply to Report #2 (Referee #4)**

Dear Reviewer,

Thank you for your insight comments and suggestions. We have modified the manuscript accordingly. We trust that all of your comments have been addressed accordingly in the revised manuscript. If you have further suggestions for changes, please let us know. The detailed corrections are listed below point by point:

All changes in the manuscript are marked with red color.

The manuscript "WHU-SGCC:A novel approach for blending daily satellite (CHIRP) and precipitation observations over Jinsha River Basin" by Shen et al. describes an approach to blend and homogenize daily satellite (CHIRP) and precipitation surface observations over Jinsha River Basin. The approach is named WHU-SGCC.

The manuscript "WHU-SGCC: A novel approach for blending daily satellite (CHIRP) and precipitation observations over Jinsha River Basin" by Shen et al. describes an approach to blend and homogenize daily satellite (CHIRP) and precipitation surface observations over Jinsha River Basin. The approach is named WHU-SGCC.

The manuscript is quite well written though sometimes it is too schematic and should be instead, to my opinion, more discursive. The manuscript is in the scope of ESSD. However, I suggest below a few potential major revisions of the manuscript along with specific line-by-line suggestions or corrections.

**General comments**

1. The introduction of the dataset underlines the central role of the observations used as the "reference dataset" to build up the homogenized dataset with the needed bias adjustment. Nevertheless, surface rain gauge observations at present have a very low level of metrologic and uncertainty characterization, therefore the manuscript can benefit from a more accurate description of the real and potential limitations due to the use of rain gauge observations as the reference for the homogenization of other rain gauge data.

**Answer:** Thanks. The gauge stations were provided by the National Climate Centre (NCC) of the China Meteorological Administration (CMA) (http://data.cma.cn/data/cdcdetail/dataCode/SURF_CLI_CHN_MUL_DAY_V3.0.htm l, last access: 10 December, 2018), which imposes strict quality control at the station, provincial and state levels. Therefore, the gauge stations can be regarded as the reference data due to their lower errors and uncertainties.

2. Table 6 and 7 allows the reader to quantify the better performances WHU-SGCC dataset compared to CHIRP and CHIRPS for rain event included in the class 0-10 mm, the moderate improvement for the in 10-25 mm class, while, as the authors themselves discussed in the manuscript, performances are negative for rain events larger than 25 mm. Though this class of rain events contains a minor number of events, the authors should discuss in the manuscript if this might indicate that extreme rain events in the Jinsha River Basin are attenuated in their intensity by the adopted homogenization approach.

**Answer:** Thanks.

**Change:** We added the discussion about the homogenization.

"the average daily precipitation within the basin less than 10 mm over the study period, which results in numerous rain gauge station data with values lower than 10 mm, which had a significant impact on the establishment of statistical relationships for the WHU-SGCC. The estimates of extreme rain events might also be attenuated during the process of WHU-SGCC adjustment."

3. The authors should comment more on the effect of and the reason why the pixel spatial autocorrelation (and the temporal autocorrelation only within the same season of the same year) is not considered and how its study can help to improve the quality of their results.

**Answer:** Thanks.

**Change:**

Added "Due to the significant seasonal difference of precipitation, the WHU-SGCC method was applied in the different seasons".

In addition, it is reasonable to assume that some pixels are statistically similar to the historical precipitation characteristics (temporal autocorrelation) of the C1 pixels within a certain area. The spatial area in which pixels may have highly similar characteristics is established. Several studies indicate that the geographical location, elevation and other terrain information influence the spatial distribution of rainfall, especially in mountainous areas with complex topography. The size of the spatial range is an important parameter to distinguish the spatial similarity and heterogeneity. So it can be assumed that the pixels are spatial correlation in these certain area.

4. Table 6 shows that most of the data belongs to C4 pixel category (55-60% of the total dataset), given that these data are homogenized using the IDW technique based on the neighboring observations, the authors should clarify if the application of this technique on the majority of the dataset can smooth out extreme events (very low or very amount) due to the smoothing introduced at the C4 pixels. Moreover, it could be helpful to the highlight the position of C1-C3 pixels in Figure 4 to allow the reader to check the position of that part of the dataset where data homogenization is likely more robust.

**Answer:** Thanks.

**Change:**

We added "Due to the significant seasonal difference of precipitation, the WHU-SGCC method was applied in the different seasons".

"In addition, the spatial distribution of C2 and C3 pixels also significantly impact the overall accuracy in different seasons that the most uniform in the fall, while the sparsest in the winter."
We also added the spatial distribution of C1-C3 pixels in Appendix D

5. In Figure 4, the spatial distribution of those statistical indicators used to study the WHU-SGCC performances is shown. It would be interesting to compare the results in Figure 4 with a similar figure showing the statistics for CHIRP. Though it is true that several table already summarizes the results of the statistics for WHU-SGCC, CHIRP and CHIRPS, a comparison of the reported statistics at the spatial level is missing.

**Answer:** Thanks.

**Change:** We added the comparison with CHIRP and CHIRPS for PCC and RMSE.

[Figure]

[Figure]

**Figure 4** Spatial distribution of the Pearson's correlation coefficient of the overall agreement between observations and the WHU-SGCC, CHIRP and CHIRPS estimations in the four seasons from 1990 to 2014.

**Figure 5** Spatial distribution of the root mean square errors of the overall agreement between observations and the WHU-SGCC, CHIRP and CHIRPS estimations in the four seasons from 1990 to 2014.

6. The statistical analysis shows that CHIRPS is worse than CHIRP in terms of variability but similar in terms of bias: the authors should clarify what is the role of the representativeness uncertainty in the merging of gridded and in-situ point observations as happens in CHIRPS.

**Answer:**

The CHIRPS was derived from blending in-suit precipitation observations and the CHIRP data, with a modified inverse-distance weighting algorithm at a quasi-global area (land only, 50° S-50° N). The blended data (CHIRPS) has an effective performance on a large scale region according to existing studies, such as at the national scale, but there are still large discrepancies with ground observations at the sub-regional level, especially at the river basin scale. The performance and applicability of CHIRPS at the sub-regional level still need to be validated. What's more, the interpolation performance from the limited and sparse rain gauge stations will be affected by more errors which was evaluated with rain gauge stations shown in Table 4.

As such, due to the poor performance of CHIRPS data at the sub-regional scale and the shortcomings of the modified inverse-distance weighting algorithm, the aim of this article is to offer a novel blending approach to improve the precipitation estimated accuracy at the river basin scale.

The Jinsha River Basin is a typically study area, with the complex and varied terrains that the range of elevation is from 263 to 6575 m above sea level, which results in significant temporal and spatial weather variation within the basin. What's more, the multi-year (1990-2014) average annual precipitation increases from north to south and the spatial distribution of precipitation is uneven, with an average annual precipitation ranging from less than 250 mm to more than 600 mm during the summer seasons over the Jinsha River Basin. However, the number of rain gauges stations is limited inside the river basin which cause precipitation estimations bias a lot.

In the previous experiment, the rain gauge stations and gridded points were used as the reference precipitation data. From that data, the training samples represented 70% of total gauged stations and gridded points, and the remaining data were used to verify the model performance. And the WHU-SGCC approach was evaluated for the Jinsha River Basin for JJA 2016.

However, the gridded precipitation used here was from China Meteorological Data Service, interpolated from 2472 rain gauge stations, which was less accurate than the rain gauge stations observations, for example, daily precipitation was more than 1000 mm at one interpolated grid point. So only the 30 rain gauge stations were used in the new WHU-SGCC experiments. In the new experiment, selecting 30% of the stations for validation was not an appropriate validation method, while the leave-one-out cross validation step was a better instead for using all the stations in WHU-SGCC correction algorithm. What's more, in order to evaluate the model performance more reasonably, the study period was changed from summer of 2016, JJA to a longer study period in the four seasons from 1990 to 2014.

In the results, The average percentage of each class of rain events at the validation gauge station during the four seasons from 1990 to 2014 are shown in Table 5 in the paper and the following figure. The major rain events within the Jinsha River Basin were light rain (0.1-10 mm), which accounted for more than 80% of the total days (the average percentage of rain event days of the total days at each gauge station), while the number of days with daily precipitation greater than the 50 mm was the smallest (no more than 1% of the total days) and fewer than 10% of the days had daily precipitation in the range of 25 mm to 50 mm. In the spring, fall and summer, significantly more no-rain days occurred than rainy days, and approximately 5% of the days had daily precipitation of 10-50 mm. The seasonal distribution of rainfall was concentrated in the summer, and 54.76%, 14.01% and 3.62% of the days had daily precipitation of 0.1-10, 10-25, and 25-50 mm, respectively. The results indicated that the average daily precipitation was less than 10 mm throughout the years of the study.

The WHU-SGCC approach blended daily precipitation gauge data and the CHIRP satellite-derived precipitation, considering the spatial correlation and the historical precipitation characteristics. Therefore, the applicability of the WHU-SGCC method over the complicated mountainous terrain with sparse rain gauge data could be confirmed by the multi-year validation.

**Specific comments**

1. Line 11: please here and throughout the manuscript you cannot state that random errors are "removed". I also encourage to have a consistent use of the terms "error" and "uncertainty".

**Answer:** Thanks. Done.

**Change**: changed "random errors" to "errors".

2. Line 21: replace "the" with "a".

**Answer:** Thanks. Done.

**Change**: changed "This study indicates that the WHU-SGCC approach is a promising tool to monitor monsoon precipitation over the Jinsha River Basin, a complicated mountainous terrain with sparse rain gauge data, considering the spatial correlation and the historical precipitation characteristics." to "In spite of the correction, the uncertainties in the seasonal precipitation forecasts in the summer and winter are still large, which may due to the homogenization attenuating the simulation of extreme rain event. However, the WHU-SGCC approach may serve as a promising tool to monitor daily precipitation over the Jinsha River Basin, which contains complicated mountainous terrain with sparse rain gauge data, based on the spatial correlation and the historical precipitation characteristics."

3. Line 62: it looks clear that the Jinsha River Basin is a region of interest for monitoring precipitation, not only for the authors; however, in order to justify the relevance of the dataset presented in the manuscript, it could be appropriate for the authors to say something more general about Jinsha River Basin and the interest to focus on this region as a special place to monitor monsoon precipitations.

**Answer:** The Jinsha River Basin is a typically study area, with the complex and varied terrains that the range of elevation is from 263 to 6575 m above sea level, which results in significant temporal and spatial weather variation within the basin. What's more, the multi-year (1990-2014) average annual precipitation increases from north to south and the spatial distribution of precipitation is uneven, with an average annual precipitation ranging from less than 250 mm to more than 600 mm during the summer seasons over the Jinsha River Basin. However, the number of rain gauges stations is limited inside the river basin which cause precipitation estimations bias a lot. So the precipitation over Jinsha River Basin is poor at high accuracy precipitation monitoring. What's more, the climate of the Jinsha River Basin has more precipitation during the summer season (June-July-August, JJA), which is affected by oceanic southwest and southeast monsoons. Therefore, the blending of satellite estimations with gauged observations during the JJA is the main focus of this research.

**Change:** We added the sentence "The Jinsha River Basin is a typical study area with complex and varied terrain, an uneven spatial distribution of precipitation, and a sparse spatial distribution of rain gauges, which limit high accuracy precipitation monitoring."

4. Line 71: gauge-satellite fusing means "combined gauge-satellite products"?

**Answer:** Thanks. Gauge-satellite fusing means "blending gauge observations and satellite estimates to produce a new dataset".

5. Line 76: please check the style to report citation in the text.

**Answer:** Thanks. Done.

**Changed:** changed "Bai et al., 2018" to "Bai et al. (2018)" and changed "Trejo et al., 2016" to "Trejo et al. (2016)".

6. Line 102: please add "weather and climate variations" instead of only climate.

**Answer:** Thanks. Done.

**Change:** We changed "which results in significant temporal and spatial climate variation within the basin" to "which results in significant temporal and spatial climate and weather variations within the basin."

7. Line 117-118: Discuss in more detailed the limitation of your "reference" dataset, here in the text.

**Answer:** Thanks. We discuss in more detailed the limitation.

**Change:** We added the sentence "However, the sparseness of gauges, their uneven spatial distribution, and high proportion of missing data may limit the high accuracy estimation in rainfall monitoring."

8. Line 131: what does it indicates the word "strict" here? Please be more specific because this is critical to learn more about the data quality used as "reference" in your approach.

**Answer:** Daily rain gauge observations at 30 national standard rain stations inside the

Jinsha River Basin during the JJA from 1990 to 2014 were provided by the National Climate Centre (NCC) of the China Meteorological Administration (CMA) (http://data.cma.cn/data/cdcdetail/dataCode/SURF_CLI_CHN_MUL_DAY_V3.0.htm l, last access: 10 December, 2018), which imposes a strict quality control at station-provincial-state levels. The process of quality control is as follows:

(1) Climate threshold or allowable value check;
(2) Extreme values at gauge stations check;
(3) Internal consistency check between fixed value, daily average value and daily extreme value;
(4) Time consistency check;
(5) Manual verification and correction;

This quality control approach is provided by the official document from CMA. So the daily rain gauge observations were used as the "reference dataset".

9. Line 159: "longer length" instead of "longer-term"
**Answer:** Thanks. Done.
**Change:** changed "longer-term" instead of "longer length".

10. Line 195: This sentence should be supported by metadata, a reference, maps, or photos.
**Answer:** Thanks. Done.
**Change:** The multi-year land cover types in the Jinsha River Basin from 2001 to 2013 are shown in Fig. B1. All of the land cover type maps were derived from the MODIS/Terra+Aqua Land Cover Type Yearly L3 Global 500 m SIN Grid V051 data set, which is available online at https://search.earthdata.nasa.gov/search/granules?p=C200106111-LPDAAC_ECS&q=MCD12&ok=MCD12 (last access: 23 July 2019). Fig. B1 shows that the land use had no obvious changes over the study period. In addition, the upstream area of the Jinsha River is an untraversed region that has not been affected significantly by human activities. Thus, the land use in the study area has hardly changed.

[Figure]

[Figure]

**Figure B1 Land cover types over the Jinsha River Basin from 2001 to 2013.**

11. Line 196: Are you assuming that there are no outliers? Please explain more.

**Answer:** Yes. Thanks. We explained more.

**Change:** changed "Spearman's Correlation Coefficient (SCC) can indicate the similarity of rainfall characteristics among pixels over a seasonal scale." to "There are no long-held outliers at one pixel of the CHIRP dataset, so Pearson's Correlation Coefficient (PCC) can represent the statistical similarity of rainfall characteristics among pixels over a seasonal scale."

12. Line 219-220: these seems to be a repetition of previous text, please check.

**Answer:** Thanks. This sentence mainly explains the terrain factors influencing the spatial distribution of rainfall. And the previous text in introduction discusses various factors affecting precipitation monitor inside mountainous areas, including topography, seasonality, climate impact and sparseness of rain gauges: "The existing research indicates that estimations over mountainous areas with complex topography often have large uncertainties and errors due to the topography, seasonality, climate impact and sparseness of rain gauges (Derin et al., 2016;Maggioni and Massari, 2018;Zambrano-Bigiarini et al., 2017)."

13. Line 246: can the authors validate their cluster analysis with a GIS map, for example, showing the terrain features?

**Answer:** The results of spatial clustering derived from the FCM method were shown in Appendix E.

14. Line 295-296: "Excluding the C1, C2 and C3 pixels, the number of remaining pixels, called C4 pixels which are adjusted by Inverse Distance Weighted (IDW)" there is something in this sentence.

**Answer:**

**Change:** changed "Excluding the C1, C2 and C3 pixels, the number of remaining pixels, called C4 pixels which are adjusted by Inverse Distance Weighted (IDW)" to "The pixels other than the C1, C2 and C3 pixels are called C4 pixels and they are adjusted by inverse distance weighting (IDW)."

15. Line 321-322: the word strength here is incorrect. SCC measures if a process is linear or not, you cannot infer more than this.

**Answer:** Thanks. In the new experiment, we changed the statistical metric of SCC to Pearson's correlation coefficient (PCC) because PCC measures the linear correlation between two series better than SCC to evaluate the adjusted precipitation accuracy.

**Change:**

"The performance of the WHU-SGCC adjusted precipitation estimates was evaluated by eight mathematic metrics: the Pearson's correlation coefficient (PCC), root mean square error (RMSE), mean absolute error (MAE), relative bias (BIAS), Nash-Sutcliffe efficiency coefficient (NSE), probability of detection (POD), false alarm ratio (FAR) and critical success index (CSI). The results of accuracy assessment are the average values validated by the leave-one-out cross method. Each validated pixel will probably be a C2, C3 or C4 pixel in the process of the WHU-SGCC algorithm. The PCC, RMSE, MAE and BIAS were used to evaluate how well the WHU-SGCC method adjusted the satellite estimation bias, while POD, FAR and CSI were used to evaluate the performance of precipitation forecasting (Su et al., 2011). The PCC measures the strength of the correlation relationship between the satellite estimations and observations."

16. Line 350: To perform the accuracy evaluation, did you use the 30% of in-situ observation mentioned at the beginning? From this section, this is not clear and how this fraction of observation was selected, is it nor clear as well.

**Answer:** Thanks. Selecting 30% of the stations for validation was not an appropriate validation method, while the leave-one-out cross validation step was a better instead.

**Change:** The validation method was changed from "selecting 30% of the stations for validation" to "leave-one-out cross validation for using all the rain gauge stations";

"The leave-one-out cross validation step was applied to computing the out-of-sample adjusted error with gauge stations."; "The WHU-SGCC algorithm was repeated 30

times, each time leaving one station as the validation station. ”

”

**Reply to Report #3 (Referee #5)**

**Manuscript ID:** essd-2018-150
**Title:** WHU-SGCC: A novel approach for blending daily satellite (CHIRP) and precipitation observations over the Jinsha River Basin
**Journal:** Earth System Science Data
**Type:** Article

Dear Reviewer,

Thank you for your insight comments and suggestions. We have modified the manuscript accordingly. We trust that all of your comments have been addressed accordingly in the revised manuscript. If you have further suggestions for changes, please let us know. The detailed corrections are listed below point by point:

All changes in the manuscript are marked with red color.

**General comments**

**(1) Phase1:** The manuscript describes a method of merging daily gauge measurements with satellite precipitation product (CHIRP) in the mountainous region of Jinsha River Basin in China. The methodology seems to be quite rigorous in statistical sense. The algorithm is not perfect as it does not take into account the skewed nature of precipitation distribution and the improvements are mostly shown on light rain events but not the heavy rain events. But overall it is a commendable effort and worth publishing.

**(2) Phase2:** One major question is the choice of CHIRP data instead of CHIRPS to start with. Perhaps because CHIRPS with all additional gauge and model data actually performs worse than CHIRP in this particular area as the statistics later in the paper shows. If this is the reason, the authors should mention it in the beginning. It is expected that the new data will perform better than CHIRPS because WHU-SGCC incorporates more surface gauge measurements. The question is then the improvement is due to more gauge stations or better merging and correction methodology? If the WHU-SGCC method only takes the station data used by CHIRPS, will the results be better than CHRIPS?
(1)
**Answer:** Thanks. We chose CHIRP data instead of CHIRPS for gauge-satellite fusing due to the poor performance of CHIRPS data at the sub-regional scale and the shortcomings of the existing blending algorithms.
**Change:**
We mention the reason for the choice of CHIRP data in the introduction. Changed "As such, due to the poor performance of CHIRPS data at the sub-regional scale and the shortcomings of the existing blending algorithms, the aim of this article is to offer a novel approach for blending daily liquid precipitation gauge data and the Climate Hazards Group Infrared Precipitation (CHIRP) satellite-derived precipitation estimates developed by the UC Santa Barbara, over the Jinsha River Basin." to "As such, due to the poor performance at the sub-regional scale, the gauge-satellite fusing algorithms can be assumed to limit high accuracy estimations in the process of CHIRPS data production. Therefore, the aim of this article is to present a novel approach for reblending daily liquid precipitation gauge data and the Climate Hazards Group Infrared Precipitation (CHIRP) satellite-derived precipitation estimates developed by UC Santa Barbara, over the Jinsha River Basin."

(2)

**Answer:**

The CHIRPS station processing stream incorporates data from five public data streams and several private archives. The public data streams are the GHCN monthly, GHCN daily, Global Summary of the Day (GSOD), GTS and Southern African Science Service Centre for Climate Change and Adaptive Land Management (SASSCAL, www.sasscalweathernet.org). GTS data are collected daily; GHCN, GSOD and SASSCAL data are updated monthly.

However, the stations for daily CHIRPS data have a different spatial distribution than those downloaded from the CMA, and the precipitation values used for CHIRPS production are the monthly values available online (ftp://ftp.chg.ucsb.edu/pub/org/chg/products/CHIRPS-2.0/diagnostics/monthly_station_data/). For the daily precipitation adjustments over the Jinsha River Basin, the daily gauge observations from the CMA are blended with the daily CHIRP data due to the unknown spatial distribution and precipitation values of gauge stations used in the process of daily CHIRPS merging.

**Change:** We reconfirmed the gauge stations used for CHIRPS merging at the official website and redrawn the map of daily gauge stations (Figure 2).

[Figure]

(a) Spring                           (b) Summer

(c) Fall           (d) Winter

10   50   100   200   300   400   500   600   700   800   900

**Average precipitation (mm)**

**Figure 2** The multi-year (1990-2014) average seasonal precipitation over the Jinsha River Basin interpolated from30 rain gauges downloaded from the China Meteorological Administration stations.

The following figure shows the daily gauge stations from CMA and monthly gauge stations in the process of CHIRPS merging which includes the unknown spatial distribution and precipitation values of daily stations

[Figure]

**Figure** The daily gauge stations from CMA and monthly gauge stations in the process of CHIRPS merging

**(3) Phase3:** Besides this, the methodology and validation seem to be reasonable. The authors should emphasize that the validation is against the gauge measurements (from using leave-one-out) and not against CHIRP and CHIRPS. Only the error statistics of CHIRP and CHIRPS data are used for comparison purpose. There are still quite a lot of English usage problems. More careful proofreading is needed.

**Answer:** Thanks. The rainfall from CHIRP, CHIRPS, and the adjusted rainfall from WHU-SGCC are all evaluated with gauge observations so that we can know how the performance of WHU-SGCC on daily precipitation monitoring over the mountainous area before and after correction, and the limitations of CHIRPS data.

**Change:** We corrected English usage problems.

**Minor comments:**

1. Line 46, 49: Use Huffman et al. (2018) reference for IMERG data and Huffman et al (2010) for TMPA data. These are official IMERG and TMPA references.

Huffman, G. J, D. T. Bolvin, D. Braithwaite, K. Hsu, R. Joyce, P. Xie (2018). NASA Global Precipitation Measurement Integrated Multi-satellitE Retrievals for GPM (IMERG), NASA Algorithm theoretical basis document (ATBD) version 5.2., 35 pp. https://pmm.nasa.gov/sites/default/files/document_files/IMERG_ATBD_V5.2.pdf

Huffman, G.J., R.F. Adler, D.T. Bolvin, and E. Nelkin (2010). The TRMM Multi-satellite Precipitation Analysis (TMPA). In F. Hossain and M. Gebremichael (Ed.), Satellite Rainfall Applications for Surface Hydrology. (3-22). Springer Verlag. ISBN: 978-90-481-2914-0

**Answer:** Thanks. Done;

**Change:** We have added these two references.

2. Line 41-62: These satellite precipitation data or merged data seem to be introduced in random order.

**Answer:** Thanks. We have introduced the multi-satellite precipitation products in order.

**Change:** changed from

"Accordingly, there are many kinds of precipitation estimates combining multiple sources datasets. Since 1997, the Tropical Rainfall Measurement Mission (TRMM) has improved satellite-based rainfall retrievals over tropical regions (Kummerow et al., 1998;Simpson et al., 1988), and then applies a stepwise method for blending daily TRMM Multisatellite Precipitation Analysis (TMPA) output with rain gauges in South America (Vila et al., 2009). The Global Precipitation Measurement (GPM) satellite was launched after the success of the TRMM satellite by the cooperation of National Aeronautics and Space Administration (NASA) and Japan Aerospace Exploration Agency (JAXA) on February 27, 2014 (Mahmoud et al., 2018;Ning et al., 2016). The main core observatory satellite (GPM) cooperates with the ten other satellites (partners) to offer the high spatiotemporal resolution products (0.1° × 0.1°- half- hourly) of the global real-time precipitation estimates (Mahmoud et al., 2019). The Geostationary Operational Environmental Satellite (GOES)-R Series is the geostationary weather satellites, which significantly improves the detection and observation of environmental phenomena. The Advanced Baseline Imager (ABI) onboard the GOES-R platform will provide images in 16 spectral bands, spatial resolution of 0.5 to 2 km (2 km in the infrared and 1-0.5 km in the visible), and full-disk scanning every 5 minutes over the continental United States. The GOES-R Series will offer the enhanced capabilities for satellite-based rainfall estimation and nowcasting (Behrangi et al., 2009;Schmit et al., 2005). The Global Precipitation Climatology Project (GPCP) is one of the successful projects for blending rain gauge analysis and multiple satellite-based precipitation estimates, and constructed a relatively coarse-resolution (monthly, 2.5° × 2.5°) global precipitation dataset (Adler et al., 2003;Huffman et al., 1997). To improve the resolution of this satellite-based dataset, the GPCC network data was incorporated into remote sensing information with Artificial Neural Networks (PERSIANN) rainfall estimates, which provides finer temporal and spatial resolutions (daily, 0.25° × 0.25°) (Ashouri et al., 2015). The CPC Merged Analysis of Precipitation (CMAP) product is a data blending and fusion analysis of gauge data and satellite-based precipitation estimates (Xie and Arkin, 1996). CMAP has a long-term dataset series from 1979, while the resolution is relatively coarse. Although the aforementioned products are widely used and have performed well, the data resolution cannot achieve high accuracy in precipitation monitoring over the Jinsha River Basin, China."

to

[revised manuscript text omitted]

3. Line 65: The sentence "Table 1 shows …" should appear in the beginning of previous paragraph before introducing different data set.
**Answer:** Thanks. Done.
**Change:** The sentence "Table 1 shows …" was changed to appear in the beginning of previous paragraph before introducing different data set.

4. Line 69: Temporal resolution?
**Answer:** Thanks. The CHIRPS data has the different temporal resolutions, such as annual, monthly, daily and so on.

5. Table 1: GPCP has a daily product
CMAP has pentad product?
CHIRPS: what is Dekad/Pentad? The temporal resolution does not match 3-day and 6- day resolution described in Line 151.

**Answer:** Thanks. We changed Table 1.

**Change:** We deleted the CMAP product, which is not very relevant to this study.

(1) 6 pentads = 1 calendar month. Each of first 5 pentads in a month have 5 days. The last pentad contains all the days from the 26th to the end of the month.

(2) A dekad = sum of 2 pentads. There are 3 dekads in a calendar month.

So we changed "It covers a quasi-global area (land only, 50° S-50° N) with several temporal scales (daily, pentad, dekad or monthly time steps)" to "It covers a quasi-global area (land only, 50° S-50° N) with several temporal scales (daily, pentad, dekad or monthly temporal resolutions)"

**Table 1** Coverage and spatiotemporal resolutions of major satellite precipitation datasets.

| Product | Temporal resolution | Spatial resolution | Period | Coverage |
|---|---|---|---|---|
| TRMM 3B42-RT | 3 hourly | 0.25° | 1998-present | 50°S-50°N |
| CMORPH | 30 min | 8 km | 1998- | 60°S-60°N |
| PERSIANN-CDR | Daily | 0.25° | 1983-(delayed) present | 60°S-60°N |
| GsMaP-NRT | hourly | 0.01° | 2007 | 60°S-60°N |
| GsMaP-MVK | hourly | 0.01° | 2000 | 60°S-60°N |
| GPM | 30 min/Hourly/ 3 hourly/Daily/3Day/7 Day/Monthly | 0.1°/0.25°/0.05°/5° | 2014-present | 60°S-60°N 70°N-70°S |
| MSWEP | 3 hourly/Daily/Monthly | 0.1° | 1979-2017 | 90°N-90°S 90°N-90°S |
| CHIRPS | Annual/Monthly/ Dekad/Pentad/Daily | 0.05°/0.25° | 1981- present | 50°S-50°N |

6. Line 137 and Figure 2 caption: annual precipitation during JJA is not correct usage. Should remove "annual".

**Answer:** Thanks. We changed the study period from summer season to the four seasons.

**Change:** changed "**Figure** 2 The multi-year (1990-2014) average annual precipitation during JJA over the Jinsha River Basin. 30 rain stations were provided by the China Meteorological Administration stations, the other 18 CHIRPS fusion stations were provided by the Climate Hazards Group UC Santa Barbara online at ftp://ftp.chg.ucsb.edu/pub/org/chg/products/CHIRPS-2.0/diagnostics/global_monthly_station_density/tifs/p05/ (last access: 10 December, 2018)." to "**Figure 2** The multi-year (1990-2014) average seasonal precipitation over the Jinsha River Basin interpolated from30 rain gauges downloaded from the China Meteorological Administration stations. "

7. Line 157: TRMM 3B42 is itself a merged product including both geostationary IR and microwave measurements from polar orbiting satellites and some gauge data.

**Answer:** Thanks. There are some errors with CHIRPS data introduction.

**Change:** changed

"This dataset contains a wide variety of satellite-based rainfall products derived from multiple data sources and incorporates four data types: monthly precipitation from

CHPClim v.1.0 (Climate Hazards Group's Precipitation Climatology version 1) derived from the combination of the satellite fields, gridded physiographic indicators, and in situ climate normal with the geospatial modelling approach based on moving window regressions and inverse distance weighting interpolation (Funk et al., 2015 b), quasi-global geostationary thermal infrared satellite observations (TRMM 3B42 version 7), atmospheric model rainfall fields CFS (Climate Forecast System) from NOAA, and surface based precipitation observations from various sources including national or regional meteorological services. The differences from other frequently used precipitation products are the higher resolution of 0.05° , wider coverage and the longer-term data series from 1981 to the present (Funk et al., 2015 a)."

to

"This dataset contains a wide variety of satellite-based rainfall products derived from multiple data sources and incorporates five data types: (1) the montxinhly precipitation from CHPClim v.1.0 (Climate Hazards Group's Precipitation Climatology version 1) derived from the combination of the satellite fields, gridded physiographic indicators, and in situ climate normal with the geospatial modelling approach based on moving window regressions and inverse distance weighting interpolation (Funk et al., 2015 b); (2) quasi-global geostationary thermal infrared (IR) satellite observations; (3)TRMM 3B42 product (Huffman et al., 2007); (4) atmospheric model rainfall fields CFS (Climate Forecast System, version 2) from NOAA; (5) and surface based precipitation observations from various sources including national and regional meteorological services. The differences from other frequently used precipitation products are the higher resolution of 0.05° , wider coverage and the longer-term data series from 1981 to the near-real time (Funk et al., 2015 a). "

8. Line 161: Are you sure that the IR rainfall estimates have pentad resolution?
**Answer:** Yes. Thanks.
"CHIRPS is the product of a two part process. First, IR Precipitation (IRP) pentad rainfall estimates are created from satellite data by calculating the percentage of time during the pentad that the IR observations indicate cold cloud tops (<235° K), and converting that value into millimeters of precipitation by means of previously determined local regression with TRMM 3B42 precipitation pentads." (Funk et al., 2014)

Funk, C. C., Peterson, P. J., Landsfeld, M. F., Pedreros, D. H., Verdin, J. P., Rowland, J. D., Romero, B. E., Husak, G. J., Michaelsen, J. C., and Verdin, A. P.: A quasi-global precipitation time series for drought monitoring, US Geological Survey Data Series, 832, 1-12, 2014.

9. Line 167-168: Don't understand why CHIRPS is used as validation data. The new data is supposed to perform better because more surface station is available?
**Answer:** Thanks.
(1) The rainfall from CHIRP, CHIRPS, and the adjusted rainfall from WHU-SGCC are all evaluated with gauge observations so that we can know how the performance of WHU-SGCC on daily precipitation monitoring over the mountainous area before and after correction, and the limitations of CHIRPS data.

(2) The WHU-SGCC approach is a promising tool to monitor the summer precipitation over the Jinsha River Basin, considering the spatial correlation and historical precipitation characteristics which is different from the process of CHIRPS merging.

And then, the CHIRPS station processing stream incorporates data from five public data streams and several private archives. The public data streams are the GHCN monthly, GHCN daily, Global Summary of the Day (GSOD), GTS and Southern African Science Service Centre for Climate Change and Adaptive Land Management (SASSCAL, www.sasscalweathernet.org). GTS data are collected daily; GHCN, GSOD and SASSCAL data are updated monthly. However, the stations for daily CHIRPS are different from those download from CMA in the spatial distribution, and the precipitation values used for CHIRPS production are only monthly values available online: ftp://ftp.chg.ucsb.edu/pub/org/chg/products/CHIRPS-2.0/diagnostics/monthly_station_data/). With the aim of daily precipitation adjustment over the Jinsha River Basin, the daily gauge observations from CMA are considered into the study to reblend with daily CHIRP data, due to the unknown spatial distribution and precipitation values of gauge stations in the process of daily CHIRPS merging.

**Change:** We reconfirmed the gauge stations used for CHIRPS merging at the official website and redrawn the map of daily gauge stations (Figure 2).

[Figure]

(a) Spring                                    (b) Summer

[Figure]

(c) Fall                                (d) Winter

**Figure 2** The multi-year (1990-2014) average seasonal precipitation over the Jinsha River Basin interpolated from30 rain gauges downloaded from the China Meteorological Administration stations.

The following figure shows the daily gauge stations from CMA and monthly gauge stations in the process of CHIRPS merging which includes the unknown spatial distribution and precipitation values of daily stations

[Figure]

**Figure** The daily gauge stations from CMA and monthly gauge stations in the process of CHIRPS merging

10. Line 187: Analyze
**Answer:** Thanks. Done.
**Change:** changed "Analyse the relationships" to "Analyze the relationships"

11. Line 196: replace "statistically" with "statistical"
**Answer:** Thanks. Done.
**Change:** changed "Pearson's Correlation Coefficient (PCC) can indicate the statistically similarity of rainfall characteristics among pixels over a seasonal scale." to "Pearson's Correlation Coefficient (PCC) can represent the statistical similarity of the rainfall characteristics among the pixels in a certain spatial area at a seasonal scale.."

12. Line 231: "molecular" should be numerator Rule 2 generates clusters of precipitation areas for the JinSha River region. Can you plot the cluster map?
**Answer:** Thanks. Done.
**Change:** changed "molecular" to "numerator" and the cluster map is in Appendix E.

13. Line 291: Equation (12): It seems that you choose to adjust C3 precipitation based on one C2 pixel that provides maximum precipitation value. What is the rational here?
**Answer:** Thanks. Equation (12) is to avoid precipitation estimates adjusted by Rule 3 below 0 and the negative values is set to 0.

14. Don't understand this sentence. Aren't C4 pixels adjusted in Rule 4? How are C1 pixels adjusted in the end?
**Answer:** Thanks.
Rule 1 is only to establish a regression model between each gauge historical observations and the corresponding CHIRP grid cell values, so the C1 pixels are not adjusted in this process. Then after Rule 1, Rule 2 and Rule 3, the unknown spatial precipitation data include both C1 and C4 pixels. Finally, all the remaining unknown pixels are adjusted by Rule 4.

15. Line 430: "In hence", there is no such usage. Use "Therefore" instead.
**Answer:** Thanks. Done.
**Change:** changed "In hence" to "Therefore".

16. Line 470: Should be "multi-model"
**Answer:** Thanks. Done.
**Change:** changed "mulit-model" to "multi-model".

17. Table 5: Why CHIRPS has much worse performance than CHIRP?
**Answer:**
   The CHIRPS was derived from blending in-suit precipitation observations and the CHIRP data, with a modified inverse-distance weighting algorithm at a quasi-global area (land only, 50° S-50° N). The blended data (CHIRPS) has an effective performance on a large scale region, such as at the national scale, but there are still large discrepancies with ground observations at the sub-regional level, especially at the river basin scale. The performance and applicability of CHIRPS at the sub-regional level still need to be validated. What's more, the interpolation performance from the limited and sparse rain gauge stations will be affected by more errors which was evaluated with rain gauge stations shown in Table 5.

As such, due to the poor performance at the sub-regional scale, the gauge-satellite fusing algorithms can be assumed to limit high accuracy estimations in the process of CHIRPS data production. Therefore, the aim of this article is to present a novel approach for reblending daily liquid precipitation gauge data and the Climate Hazards Group Infrared Precipitation (CHIRP) satellite-derived precipitation estimates developed by UC Santa Barbara to improve the precipitation estimated accuracy at the river basin scale.

**Reply to Report #4 (Referee #6)**

**Manuscript ID:** essd-2018-150
**Title:** WHU-SGCC: A novel approach for blending daily satellite (CHIRP) and precipitation observations over the Jinsha River Basin
**Journal:** Earth System Science Data
**Type:** Article

Dear Reviewer,

Thank you for your insight comments and suggestions. We have modified the manuscript accordingly. We trust that all of your comments have been addressed accordingly in the revised manuscript. If you have further suggestions for changes, please let us know. The detailed corrections are listed below point by point:

All changes in the manuscript are marked with red color.

In this manuscript, authors have proposed a new method to combine satellite and rain gauge precipitation estimates at fine spatial resolution over a river basin. The merged precipitation product is also compared with CHIRPS dataset.

**Specific comments**

My specific comments are given below.

1. L43-44: TMPA uses monthly rain gauge analysis over the tropical land, not only over the South America. For the production of TMPA product, reference of Huffman et al. (2008, JHM/2010, Book chapter) must be there.

**Answer:** Thanks. Done.

**Change:** changed "and then applies a stepwise method for blending daily TRMM Multisatellite Precipitation Analysis (TMPA) output with rain gauges in South America (Vila et al., 2009)." to "the TRMM Multisatellite Precipitation Analysis (TMPA) products, which are derived from gauge-satellite fusing (Huffman et al., 2010;Vila et al., 2009);"

Huffman, G. J., Adler, R. F., Bolvin, D. T., and Nelkin, E. J.: The TRMM Multi-Satellite Precipitation Analysis (TMPA), in: Satellite Rainfall Applications for Surface Hydrology, edited by: Gebremichael, M., and Hossain, F., Springer Netherlands, Dordrecht, 3-22, 2010.

2. L46-49: These references for GPM are not relevant here. It must be replaced by Hou et al. (2014, BAMS) and Skofronick-Jackson et al. (2017, BAMS).

**Answer:** Thanks.

**Change:** changed. We replaced the references.

Changed from "The Global Precipitation Measurement (GPM) satellite was launched after the success of the TRMM satellite by the cooperation of National Aeronautics and

Space Administration (NASA) and Japan Aerospace Exploration Agency (JAXA) on February 27, 2014 (Mahmoud et al., 2018;Ning et al., 2016). The main core observatory satellite (GPM) cooperates with the ten other satellites (partners) to offer the high spatiotemporal resolution products (0.1° × 0.1°- half- hourly) of the global real-time precipitation estimates (Mahmoud et al., 2019;Huffman et al., 2018)."

To

"In 2014, the Global Precipitation Measurement (GPM) satellite was launched after the success of the TRMM satellite by a cooperation between the National Aeronautics and Space Administration (NASA) and Japan Aerospace Exploration Agency (JAXA) (Mahmoud et al., 2018;Ning et al., 2016). The main core observatory satellite (GPM) integrates advanced radar and radiometer systems to obtain the precipitation physics and takes advantages of TMPA, the Climate Prediction Center morphing technique (CMORPH), and PERSIANN algorithms to offer high spatiotemporal resolution products (0.1° × 0.1°, half-hourly) of global real-time precipitation estimates (Huffman et al., 2018; Skofronick-Jackson et al., 2017; Hou et al., 2014)."

Hou, A. Y., Kakar, R. K., Neeck, S., Azarbarzin, A. A., Kummerow, C. D., Kojima, M., Oki, R., Nakamura, K., and Iguchi, T.: The Global Precipitation Measurement Mission, Bulletin of the American Meteorological Society, 95, 701-722, 10.1175/bams-d-13-00164.1, 2014.

Skofronick-Jackson, G., Petersen, W. A., Berg, W., Kidd, C., Stocker, E. F., Kirschbaum, D. B., Kakar, R., Braun, S. A., Huffman, G. J., Iguchi, T., Kirstetter, P. E., Kummerow, C., Meneghini, R., Oki, R., Olson, W. S., Takayabu, Y. N., Furukawa, K., and Wilheit, T.: The Global Precipitation Measurement (GPM) Mission for Science and Society, Bulletin of the American Meteorological Society, 98, 1679-1695, 10.1175/bams-d-15-00306.1, 2017.

3. L49-53: Why authors introduced GOES-R here? It is suggested to maintain a hierarchy such as infrared-only, MW-only followed by multi-satellite precipitation products. Actually, this whole paragraph does not read good to me.

**Answer:** Thanks. We deleted the introduction of GOES-R and modified the explanation

**Change:** changed from

"Accordingly, there are many kinds of precipitation estimates combining multiple sources datasets. Since 1997, the Tropical Rainfall Measurement Mission (TRMM) has improved satellite-based rainfall retrievals over tropical regions (Kummerow et al., 1998;Simpson et al., 1988), and then applies a stepwise method for blending daily TRMM Multisatellite Precipitation Analysis (TMPA) output with rain gauges in South America (Vila et al., 2009). The Global Precipitation Measurement (GPM) satellite was launched after the success of the TRMM satellite by the cooperation of National Aeronautics and Space Administration (NASA) and Japan Aerospace Exploration Agency (JAXA) on February 27, 2014 (Mahmoud et al., 2018;Ning et al., 2016). The main core observatory satellite (GPM) cooperates with the ten other satellites (partners) to offer the high spatiotemporal resolution products (0.1° × 0.1°- half- hourly) of the global real-time precipitation estimates (Mahmoud et al., 2019). The Geostationary Operational Environmental Satellite (GOES)-R Series is the geostationary weather satellites, which significantly improves the detection and observation of environmental phenomena. The Advanced Baseline Imager (ABI) onboard the GOES-R platform will provide images in 16 spectral bands, spatial resolution of 0.5 to 2 km (2 km in the infrared and 1-0.5 km in the visible), and full-disk scanning every 5 minutes over the continental United States. The GOES-R Series will offer the enhanced capabilities for satellite-based rainfall estimation and nowcasting (Behrangi et al., 2009;Schmit et al., 2005). The Global Precipitation Climatology Project (GPCP) is one of the successful projects for blending rain gauge analysis and multiple satellite-based precipitation estimates, and constructed a relatively coarse-resolution (monthly, 2.5° × 2.5°) global precipitation dataset (Adler et al., 2003;Huffman et al., 1997). To improve the resolution of this satellite-based dataset, the GPCC network data was incorporated into remote sensing information with Artificial Neural Networks (PERSIANN) rainfall estimates, which provides finer temporal and spatial resolutions (daily, 0.25° × 0.25°) (Ashouri et al., 2015). The CPC Merged Analysis of Precipitation (CMAP) product is a data blending and fusion analysis of gauge data and satellite-based precipitation estimates (Xie and Arkin, 1996). CMAP has a long-term dataset series from 1979, while the resolution is relatively coarse. Although the aforementioned products are widely used and have performed well, the data resolution cannot achieve high accuracy in precipitation monitoring over the Jinsha River Basin, China."

to

[revised manuscript text omitted]

4. L69: Recent study (https://doi.org/10.1016/j.jhydrol.2019.01.036) needs to be included.
**Answer:** Thanks. Done.
**Change:** Added the reference.
"The CHIRPS precipitation dataset with several temporal and spatial scales has been evaluated in Brazil (Nogueira et al., 2018;Paredes-Trejo et al., 2017), Chile (Yang et al., 2016;Zambrano-Bigiarini et al., 2017), China (Bai et al., 2018), Cyprus (Katsanos et al., 2016a;Katsanos et al., 2016b), India Ali and Mishra, 2017) and Italy (Duan et al., 2016)." to
"The CHIRPS precipitation dataset with several temporal and spatial scales has been evaluated in Brazil (Nogueira et al., 2018;Paredes-Trejo et al., 2017), Chile (Yang et al., 2016;Zambrano-Bigiarini et al., 2017), China (Bai et al., 2018), Cyprus (Katsanos et al., 2016a;Katsanos et al., 2016b), India (Ali and Mishra, 2017; Prakash, 2019) and Italy (Duan et al., 2016)."

Prakash, S.: Performance assessment of CHIRPS, MSWEP, SM2RAIN-CCI, and TMPA precipitation
products across India, Journal of Hydrology, 571, 50-59, 10.1016/j.jhydrol.2019.01.036, 2019.

5. Table 2 should be kept as "Supplementary Material".
**Answer:** Thanks. Done.
**Change:** changed Table 2 to Appendix A.

6. Fig. 2: Symbols are not clearly visible. The mean precipitation is derived from 30
rain gauges or from CHIRPS?
**Answer:** Thanks. Done. The mean precipitation is derived from 30 rain gauges.
**Change:** We redrawn the Fig. 2 in the four seasons.

[Figure]

**Figure 2** The multi-year (1990-2014) average seasonal precipitation over the Jinsha River Basin interpolated from30 rain gauges downloaded from the China Meteorological Administration stations.

7. L172-173: Whether 18 gauges of CHIRPS are subset of 30 gauges of CMA? It needs to be elaborated.

**Answer:**

The CHIRPS station processing stream incorporates data from five public data streams and several private archives. The public data streams are the GHCN monthly, GHCN daily, Global Summary of the Day (GSOD), GTS and Southern African Science Service Centre for Climate Change and Adaptive Land Management (SASSCAL, www.sasscalweathernet.org). GTS data are collected daily; GHCN, GSOD and SASSCAL data are updated monthly.

However, the stations for daily CHIRPS data have a different spatial distribution than those downloaded from the CMA, and the precipitation values used for CHIRPS production are the monthly values available online (ftp://ftp.chg.ucsb.edu/pub/org/chg/products/CHIRPS-2.0/diagnostics/monthly_station_data/). For the daily precipitation adjustments over the Jinsha River Basin, the daily gauge observations from the CMA are blended with the daily CHIRP data due to the unknown spatial distribution and precipitation values of gauge stations used in the process of daily CHIRPS merging.

**Change:** We reconfirmed the gauge stations used for CHIRPS merging at the official website and redrawn the map of daily gauge stations (Figure 2).

The following figure shows the daily gauge stations from CMA and monthly gauge stations in the process of CHIRPS merging which includes the unknown spatial distribution and precipitation values of daily stations

[Figure]

**Figure** The daily gauge stations from CMA and monthly gauge stations in the process of CHIRPS merging

8. This study proposes a new method to merge satellite and rain gauge observations. However, it is still not clear that the proposed method is better than any existing merging technique. I would suggest to compare your technique with few operational techniques and present the improvement in accuracy of precipitation estimation by the proposed method.

**Answer:** Thank you for your valuable comments.

As such, due to the poor performance at the sub-regional scale, the gauge-satellite fusing algorithms can be assumed to limit high accuracy estimations in the process of CHIRPS data production. Therefore, the aim of this article is to present a novel approach (WHU-SGCC) for reblending daily liquid precipitation gauge data and CHIRP satellite-derived precipitation estimates developed by UC Santa Barbara, over the Jinsha River Basin.   The results adjusted by WHU-SGCC were compared with those of CHIRP and CHIRPS at the overall, daily and rain events evaluations. According to the performance of the WHU-SGCC, we can see that the precipitation adjusted by the WHU-SGCC method can achieve greater accuracy compared with CHIRP and CHIRPS, with average improvements of Pearson's correlation coefficient (PCC) of 0.01-0.23 and 0.06-0.32, respectively. The PCCs were improved to more than 0.5 in the spring and fall and to approximately 0.5 in the winter, and they were the worst in the summer, which may be attributed to the greater precipitation in the summer and lower precipitation in the winter. In addition, the NSE of the WHU-SGCC provides substantial improvements over CHIRP and CHIRPS, which reached 0.2836, 0.2944 and

0.1853 in the spring, fall and winter, respectively. In the summer, the NSE of the WHU-SGCC is still negative, but it is improved to be nearly zero, which indicates that the adjusted results are similar to the average level of the rain gauge observations. All of the measured errors were reduced except for the BIAS, which showed no significant improvement in the summer but was approximately 0. The results indicate that the WHU-SGCC has the best ability to reduce precipitation errors for daily accuracy evaluations, with average reductions of 15% and 34% compared to CHIRP and CHIRPS, respectively. Overall, the WHU-SGCC approach achieves good performance in error correction of CHIRP and CHIRPS.

In the future, we will further investigate the performance of the WHU-SGCC method when time is sufficient by comparing with more methods, such as Quantile Mapping and Bayesian methods.

9. I would suggest to use normalized parameters for error/bias computation for better assessment.

**Answer:** Thanks. The BIAS used in this study is the normalized bias.

$$BIAS = \frac{\sum_{i=1}^{n}(C_i - Y_{oi})}{\sum_{i=1}^{n} Y_{oi}}$$

**Reply to Report #5 (Referee #7)**

**Manuscript ID:** essd-2018-150
**Title:** WHU-SGCC: A novel approach for blending daily satellite (CHIRP) and precipitation observations over the Jinsha River Basin
**Journal:** Earth System Science Data
**Type:** Article

Dear Reviewer,

Thank you for your insight comments and suggestions. We have modified the manuscript accordingly. We trust that all of your comments have been addressed accordingly in the revised manuscript. If you have further suggestions for changes, please let us know. The detailed corrections are listed below point by point:

All changes in the manuscript are marked with red color.

1. L11: The authors mention that the existing blending algorithms are very bad at removing random errors at the daily scale. However, in the introduction, the authors only mention three articles related to blending techniques. The authors must present a better literature review. I do not agree that all the algorithms are very bad at removing random errors at the daily scale; see MSWEP as an example. MSWEP is a merged precipitation product (using rain gauges, reanalysis, and satellite-based precipitation products) which has good performance in many areas.
**Answer:** Thanks. We changed the description of the shortcomings of existing methods in abstract.
**Changed:** changed "and the existing data blending algorithms are very bad at removing the day-by-day random errors." to "and most of the existing data blending algorithms are not good at removing the day-by-day errors"

2. L70: The authors mention that the articles presented between L67-69 evaluated the products mainly at monthly and seasonal scales lacking the evaluation of daily precipitation. In the case of Chile, Zambrano-Bigiarini et al., (2017) include a daily evaluation and Yang et al., (2016) is a study of bias correction and its also at daily scale.
**Answer:** Thanks. The evaluations of CHIRPS at daily scale are indeed conducted in Chile, however, the bias correction is still limited.
**Change:** changed from "Nevertheless, the temporal resolutions of the aforementioned applications were mainly at seasonal and monthly scales, lacking the evaluation of daily precipitation." to "However, the temporal resolutions of these applications were mainly at seasonal and monthly scales, lacking the evaluation and correction of daily precipitation."

3. Table 1: I think that MSWEP should be included in the study because is a state-of-the-art merged product, it can be compared to the product obtained applying this method.

**Answer:** Thanks. We changed the introduction.

**Changed:**

"To fill the gap in high resolution and long-term global multi-satellite precipitation monitoring, the Multi-Source Weighted-Ensemble Precipitation (MSWEP) product (Beck et al., 2017; Beck et al., 2019), and the Climate Hazards Group Infrared Precipitation with Station data (CHIRPS) product from UC Santa Barbara (Funk et al., 2015 a) were developed. MSWEP is a precipitation data set with global coverage available at 0.1° spatial resolution and at three-hourly, daily, and monthly temporal resolutions. MSWEP is multi-source data that takes advantage of the complementary strengths of gauge-, satellite-, and reanalysis-based data. However, to provide precipitation estimates at a higher spatial resolution, the CHIRPS data set is used in this study."

**Table 1** Coverage and spatiotemporal resolutions of major satellite precipitation datasets.

| Product | Temporal resolution | Spatial resolution | Period | Coverage |
|---|---|---|---|---|
| TRMM 3B42-RT | 3 hourly | 0.25° | 1998-present | 50°S-50°N |
| CMORPH | 30 min | 8 km | 1998- | 60°S-60°N |
| PERSIANN-CDR | Daily | 0.25° | 1983-(delayed) present | 60°S-60°N |
| GsMaP-NRT | hourly | 0.01° | 2007 | 60°S-60°N |
| GsMaP-MVK | hourly | 0.01° | 2000 | 60°S-60°N |
| GPM | 30 min/Hourly/ 3 hourly/Daily/3Day/7 Day/Monthly | 0.1°/0.25°/0.05°/5° | 2014-present | 60°S-60°N 70°N-70°S 90°N-90°S |
| MSWEP | 3 hourly/Daily/Monthly | 0.1° | 1979-2017 | 90°N-90°S |
| CHIRPS | Annual/Monthly/ Dekad/Pentad/Daily | 0.05°/0.25° | 1981- present | 50°S-50°N |

4. L122: Here the authors mention that: 'because of the Jinsha River Basin has more precipitation during the summer season the blending of satellite estimations with gauged observations during JJA is the main focus of the research'. I completely disagree with this... If the authors are presenting a novel approach for blending ground-based measurements with a satellite product it has to be evaluated throughout the year and not only for one season.

**Answer:** Thanks.

**Change:** We demonstrated the WHU-SGCC method by applying it to daily precipitation over the Jinsha River Basin in the different seasons from 1990 to 2014.

5. L196: Assumption number three states that the correlation coefficient can indicate the statistical similarity of rainfall characteristics among pixels. What would happen if two different areas show similar correlation values but different precipitation amounts?

**Answer:** Thanks. We assumed that there are no long-held outliers at one pixel of the CHIRP dataset, so Pearson's Correlation Coefficient (PCC) can represent the statistical similarity of rainfall characteristics among pixels over a seasonal scale. Here, PCC are calculated between raster pixels from long-term CHIRP data set at a certain spatial scope, which will have the same precipitation over a seasonal scale.

**Change:** changed "Pearson's Correlation Coefficient (PCC) can indicate the statistically similarity of rainfall characteristics among pixels over a seasonal scale." to "There are no long-held outliers at one pixel in the CHIRP dataset, so Pearson's Correlation Coefficient (PCC) can represent the statistical similarity of the rainfall characteristics among the pixels in a certain spatial area at a seasonal scale."

6. The authors present four classes of pixels: C1 (pixels with at least 1 station); C2 (pixels statistically similar to C1); C3 (similar to C2); and C4 (the rest of the pixels): How does the total precipitation amount is considered when the authors apply the RF models to the C2 pixels?

**Answer:** Thanks. The historical precipitation features between CHIRP pixels at C1 and C2 pixels are the main consideration of this study, but not the total precipitation amount. It is reasonable to assume that there are some pixels that are statistically similar to the historical precipitation characteristics of C1 pixels in a certain spatial scope. Therefore, it is feasible to adjust the satellite estimation bias of C2 pixels by referring to the appropriate regression relationships at corresponding C1 pixels based on Rule 1. The determination of C2 pixels is as the following procedure:

With exception of the C1 pixels, the remaining pixels in each cluster represent potential C2 pixels called R pixels. Pearson's correlation coefficient (PCC) and p-values between the satellite estimations (multi-year daily CHIRP grid cell values during the summer seasons from 1990 to 2014) at R pixels and the C1 pixels are the criteria for final determination of C2 pixels. The critical values for PCC and p-value are 0.5 and 0.05, so the selected C2 pixels can then be considered statistically similar to the precipitation characteristics of the corresponding C1 pixels in their defined spatial scope.

7. The authors mention that the development of methods for high-accuracy precipitation estimates over complex terrain is of vital importance, how is this considered in the

**Answer:** In Rule 2, the terrain factors were considered to divide rainfall area.

Some studies indicate that geographical location, elevation and other terrain information influences the spatial distribution of rainfall, especially in mountainous areas with complex topography (Anders et al., 2006;Long and Singh, 2013). Therefore, it is reasonable to assume that there are some pixels statistically similar to historical precipitation characteristics in a certain spatial scope. In Rule 2, the approach of fuzzy c-means (FCM) clustering was explored to determine the spatial range considered as each pixel's terrain factors including longitude, latitude, elevation, slope, aspect and curvature. The pixels with statistically similar precipitation characteristics are determined at a physically similar area (the region with similar terrain features) in WHU-SGCC algorithm, which can contribute to improve the accuracy of precipitation estimates.

8. The amount of C4 pixels is around 60%, and for these pixels the authors apply an IDW interpolation, how do the authors justify that some pixels are blended with machine learning and others are simply interpolated? Which are the expected errors of this methodology for each type of pixels (C1-4)? What happen with the small scale convective events (which are present in summer) neover the C4 pixels?

**Answer:** Thanks.

The WHU-SGCC process shown that the adjustment method for C2 pixels was derived from C1 pixels, the adjustment method for C3 pixels was derived from C2 pixels, and the adjusted values for C1 and C4 pixels were interpolated by IDW with C2 and C3 pixels. There were statistically relationship among C1, C2, C3 and C4 pixels. Thus, the performance of WHU-SGCC method would be evaluated on the overall accuracy, not on a certain class of pixels.

In addition, we added the spatial distribution of C1-C3 pixels in Appendix D. The C4 pixels can be interpolated by the IDW based on the C2 and C3 pixels due to the amount and spatial distribution of the C2 and C3 pixels.

We also added the pixel type in Appendix D. Each validation gauge station could be identified as either C2, C3 or C4 pixels to evaluate the performances of all the rules in the WHU-SGCC method.

| Pixel type / Validation gauge station | Spring | Summer | Fall | Winter |
|---|---|---|---|---|
| 52908 | C4 | C4 | C4 | C4 |
| 56004 | C4 | C4 | C4 | C4 |
| 56021 | C2 | C2 | C2 | C3 |
| 56029 | C2 | C3 | C2 | C3 |
| 56034 | C2 | C3 | C2 | C3 |
| 56038 | C4 | C4 | C4 | C4 |
| 56144 | C4 | C4 | C4 | C4 |
| 56146 | C4 | C4 | C4 | C4 |
| 56152 | C2 | C3 | C3 | C4 |
| 56167 | C4 | C2 | C2 | C4 |
| 56247 | C4 | C4 | C4 | C4 |
| 56251 | C2 | C2 | C3 | C3 |
| 56257 | C4 | C4 | C4 | C4 |
| 56357 | C4 | C4 | C4 | C4 |
| 56374 | C4 | C4 | C3 | C4 |
| 56459 | C4 | C4 | C4 | C4 |
| 56462 | C4 | C4 | C4 | C4 |
| 56475 | C4 | C3 | C3 | C3 |
| 56479 | C4 | C4 | C4 | C4 |
| 56485 | C3 | C2 | C2 | C3 |
| 56543 | C3 | C3 | C4 | C4 |

| | | | | |
|---|---|---|---|---|
| 56565 | C2 | C2 | C3 | C3 |
| 56571 | C2 | C4 | C4 | C4 |
| 56586 | C2 | C3 | C2 | C3 |
| 56651 | C3 | C2 | C2 | C3 |
| 56664 | C4 | C4 | C4 | C4 |
| 56666 | C3 | C3 | C3 | C3 |
| 56671 | C3 | C2 | C2 | C3 |
| 56684 | C2 | C2 | C2 | C4 |
| 56778 | C4 | C3 | C3 | C4 |

9. The authors should present a map with the mean annual precipitation for the study area using CHIRP and the blended product. This would be interesting because it would help the reader to see if the transition of precipitation between the different classes of pixels is smooth.

**Answer:** Thanks. We added the maps with the average seasonal precipitation for the study area in Fig. 4.

**Changed**: The multi-year (1990-2014) average seasonal precipitation over the Jinsha River Basin interpolated from WHU-SGCC, CHIRP and CHIRPS is shown in Fig. 4. There exist some differences in the spatial pattern of precipitation estimates. Overall, the WHU-SGCC method exhibits the similar spatial distribution of precipitation to the CHIRP and CHIRPS, while the WHU-SGCC method attenuated the intense rain in the central area. The statistical accuracy evaluations are needed to further analyze the performance of the WHU-SGCC method.

[Figure]

**Figure 4** The multi-year (1990-2014) average seasonal precipitation over the Jinsha River Basin interpolated from WHU-SGCC, CHIRP and CHIRPS.

10. Table 5: which values are represented here? mean, median?
**Answer:** These are mean values of the results validated by leave-one-out cross methods.

11. L360: The authors mention an increase of NSE of 93% which sounds very good, however, this value is still negative. The authors should present the values and not the percentage of improvement as it is relative to the performance of the product and it can be misleading.

**Answer:** Thanks. Done.

**Change:** changed "Nevertheless, the NSE of the WHU-SGCC reached -0.0137, an increase of 93.33% and 98.32% compared to CHIRP and CHIRPS, respectively." to "In addition, the NSE of the WHU-SGCC provides substantial improvements over CHIRP and CHIRPS, which reached 0.2941, 0.3087 and 0.1853 in the spring, fall and winter, respectively. In the summer, the NSE of the WHU-SGCC is still negative, but it is improved to be nearly zero, which indicates that the adjusted results are similar to the average level of the rain gauge observations."

12. Table 5: There is some improvement in POD, however, the PCC and the BIAS are almost similar. The categorical indices (POD, FAR, and CSI) can be decomposed to different precipitation intensities in order to see if the method improves the detection of medium and high intensities or only the no-rain events which are higher in number.

**Answer:** Thanks. We added the POD and CSI for rain events assessment in Fig. 8. The POD and CSI are improved at the different rain events, especially with the total rainfall less than 25 mm.

However, the FAR for different rain events cannot be conducted due to the values of 0 for the number of false alarms at the gauge stations.

**Change**:

[Figure]

[Figure]

[Figure]

[Figure]

[Figure]

**Figure 8** Accuracy assessment of liquid precipitation events from 1990 to 2014.

13. L381: The authors mention that the presented approach is still effective in adjusting the satellite biases. However, in L357 it is mentioned that the bias showed no significant improvement (which is in agreement with Table 5).
**Answer: Thanks.** We added the spring, fall and winter and changed the discussion.
**Change:** changed from "Although, the absolute value of BIAS of WHU-SGCC was no significant improvement than CHIRP 357 and slightly higher than CHIRPS, all of the values were approximately to 0." to "The absolute values of the BIAS of the WHU-SGCC are substantial improved in the spring, followed by the summer, winter and fall. Although the absolute value of the BIAS of the WHU-SGCC in fall are not significantly better than those of CHIRP and CHIRPS, all of the values are approximately 0."

14. Figure 4a and 4d: The legend should be improved. The correlation (PCC) goes from 0.1-0.2; 0.2-0.3 and then just >0.3. If 1.0 is the maximum and 0.3 is still low correlation why not showing the complete ranges?
**Change:** Thanks. We redraw the maps about the spatial distribution of the Pearson's correlation coefficient with the complete ranges.

[Figure]

**Figure 5** Spatial distribution of the Pearson's correlation coefficient of the overall agreement between observations and the WHU-SGCC, CHIRP and CHIRPS estimations in the four seasons from 1990 to 2014.

15. L393: If PCCs of WHU-SGCC, CHIRP and CHIRPS are roughly the same, then there was no improvement in the daily correction of events. Can the authors discuss this?

**Answer:** Figure 7 shows the evaluation of daily accuracy.

The evaluation of the daily accuracy indicates that the PCCs of the WHU-SGCC were slightly better than those of CHIRP and CHIRPS in the spring, fall and winter but were not as good in the summer and winter. The WHU-SGCC had lower RMSEs and MAEs than CHIRP and CHIRPS, especially compared to CHIRPS. The daily RMSE and MAE in the summer are the highest, although the WHU-SGCC still corrects the errors. Figure 7 indicates that there is a slight increase in the PCC, with average improvements of 0.02-0.04 and 0.04-0.14, respectively; however, the PCC is a relative metric of the magnitude of the association between paired variables, and a relative consistency may not indicate absolute proximity. Thus, the absolute measure indicated by the RMSE may be more reasonable. In this study, the RMSE and MAE derived from the WHU-SGCC are reduced by approximately 15% and 34% on average compared to CHIRP and CHIRPS, respectively.

Overall, the WHU-SGCC approach can be regarded as an effective tool for daily precipitation adjustments.

16. L403: The authors mention that the WHU-SGCC method seems to be more promising in detecting precipitation compared to CHIRP and CHIRPS altough it performs poorly when evaluated with the FAR. However the objective is to provide high quality precipitation estimates.
**Answer:** Thanks.
Although, the FAR of WHU-SGCC is lower than CHIRP and CHIRPS in some days, the overall assessment of FAR is the best.
In terms of the POD, FAR and CSI, except for the results in winter, the WHU-SGCC method appears to be better at detecting precipitation than CHIRP and CHIRPS; the results of POD and CSI are closest to 1, although FAR is worse than CHIRPS on some days. However, the overall result of FAR is the best in the WHU-SGCC. The POD and FAR results are the worst in the winter, and the CSI is slightly higher, which may be attributed to the overestimation of no-rain events and the inherent uncertainty in the CHIRP.

Overall, the WHU-SGCC approach can be regarded as an effective tool for daily precipitation adjustments.

17. Figure 5: There are not labels in each boxplot.
**Answer:** Thanks. Done.
**Change:** We have added the labelling (a,b,…,h) into Fig. 7.

18. Table 7: Please include POD, FAR, and CSI to see the categorical performance over different rainfall intensities.
**Answer:** We added the POD and CSI for rain events assessment in Fig. 8. The FAR for different rain events cannot be conducted due to the values of 0 for the number of false alarms at the gauge stations.
**Change**:

[Figure]

**Figure 8** Accuracy assessment of liquid precipitation events from 1990 to 2014.

**Reply to Interactive comment**

1. <[authors should] explain more clearly how they did the leave-one-out validation as this important information is not emphasized in the paper. By leaving-one-out if I understand correctly, they will have to repeat the algorithm 30 more times with each time leaving one station as validation station. The pixel that has that station therefore could be identified as either C2, C3 or even C4 pixels. If this is the case, this validation will allow them to validate corrections for all types of pixels. I would then highly recommend that they add (C2, C3, C4 pixel type) near the station number in Table 4. If however, all the station shows as C2 pixels (can't be C1), then the validation is indeed not complete.>

**Answer:** Thanks. We explain more clearly about the leave-one-out validation.
**Change:**
(1) Added the following sentence into the first paragraph of section 3.1:

[revised manuscript text omitted]

The Global Precipitation Climatology Project (GPCP) is one of the successful projects for blending rain gauge analysis and multiple satellite-based precipitation estimates, and constructed a relatively coarse-resolution (monthly, 2.5° × 2.5°) global precipitation dataset (Adler et al., 2003;Huffman et al., 1997). To improve the resolution of this satellite-based dataset, the GPCC network data was incorporated into remote sensing information with Artificial Neural Networks (PERSIANN) rainfall estimates, which provides finer temporal and spatial resolutions (daily, 0.25° × 0.25°) (Ashouri et al., 2015). The CPC Merged Analysis of Precipitation (CMAP) product is a data blending and fusion analysis of gauge data and satellite-based precipitation estimates (Xie and Arkin, 1996). CMAP has a long-term dataset series from 1979, while the resolution is relatively coarse. Although the aforementioned products are widely used and have performed well, the data resolution cannot achieve high accuracy in precipitation monitoring over the Jinsha River Basin, China.

[revised manuscript text omitted]

As such, due to the poor performance of CHIRPS data at the sub-regional scale, the gauge-satellite fusing algorithms can be assumed to limit high accuracy estimations in the process of CHIRPS data production. and the shortcomings of the existing blending algorithms,Therefore, the aim of this article is to offer present a novel approach for reblending daily liquid precipitation gauge data and the Climate Hazards Group Infrared Precipitation (CHIRP) satellite-derived precipitation estimates developed by the UC Santa Barbara, over the Jinsha River Basin. Here, we willWe use precipitation to name denote liquid precipitation throughout the text. The CHIRP data are is the raw data of CHIRPS before blending with thein rain gauge data. The objective is to build corresponding precipitation models that consider terrain factors and precipitation characteristics to produce high-quality precipitation estimates. This novel method is named called the Wuhan University Satellite and Gauge precipitation Collaborated Correction (WHU-SGCC) method. We demonstrate this method by applying it to daily precipitation over the Jinsha River Basin during in the differentsummer seasons from 1990 to 2014. The results support the validity of the proposed approach for producing refined satellite-gauge precipitation estimates over mountainous areas.

The remainder of this paper is organized as follows: Section 2 describes the study region, and rain gauges and CHIRPS dataset used in this study. Section 3 presents the principle of the WHU-SGCC approach for high-accuracy daily precipitation estimates. The results and discussion are analysed in Section 4, the data available is are described in Section 5, and the conclusions and future work are presented in Section 6.

**2 Study Region and Data**

**2.1 Study Region**

The Yangtze River, one of the largest and most important rivers in Southeast Asia, originates on the Tibetan Plateau and extends approximately 6300 km eastward to the East China Sea. The river's catchment covers an area of approximately ~180 × 10$^4$ km$^2$ and the average annual precipitation is approximately 1100 mm (Zhang et al., 2019).The Yangtze River is divided into nine sub-basins, the upper drainage basin is the Jinsha River Basin, which flows through the provinces of Qinghai, Sichuan, and Yunnan in western China. Inside Within the Jinsha River Basin, the total river length is 3486 km, accounting for 77% of the length of the upper Yangtze River, and covering a watershed area of 460 × 10$^3$ km$^2$. The location of the Jinsha River Basin is shown in Fig. 1, and it covers the eastern part of the Tibetan Plateau and the part of the Hengduan Mountains. The southern portion of the river basin is the Northern Yunnan Plateau and the eastern portion includes a wide area of the southwestern margin of the Sichuan basinBasin. Crossing complex and varied terrains, the elevation of the Jinsha River ranges from 263 to 6575 m above sea level, which results in significant temporal and spatial climate and weather variations inside the basin. The average annual precipitation of the Jinsha River Basin is approximately 710 mm, the average annual precipitation of the lower reaches is approximately 900-1300 mm, while and the average annual precipitation of the middle and upper reaches is approximately 600-800 mm (Yuan et al., 2018). The Jinsha River Basin has four seasons: spring (March-April-May), summer (June-July-August), fall (September-October-November) and winter (December-January-February). The The climate of the Jinsha River Basin has more precipitation during the summer season (June-July-August, JJA), which is affected by oceanic southwest and southeast monsoons, resulting in more precipitation during the summer. Therefore, the blending of satellite estimations with gauged observations during the different seasons JJA is the main focus of this research.

[Figure]

**Figure 1** Location of the study area with key topographic features.

**2.2 Study Data**

**2.2.1 Precipitation gauge observations**

Daily rain gauge observations at 30 national standard rain stations inside within the Jinsha River Basin during the JJA from 1 March 1990 to February 2015 were provided by the National Climate Centre (NCC) of the China Meteorological Administration (CMA) (http://data.cma.cn/cn/cdcdetail/dataCode/SURF_CLI_CHN_MUL_DAY_V3.0.html, last access: 10 December, 2018), which imposes a strict quality control at the station, -provincial -and state levels. The Sstation identification numbers and relevant geographical characteristics are shown in Table Appendix 2A, and their uneven spatial distribution is shown in Fig. 2. The selected rain gauges are located in Qinghai, Tibet, Sichuan and Yunnan Provinces but are mainly scattered in Sichuan Province, and the number of rain gauges in the northern river basin contains fewer rain gaugesis less than in 
[revised manuscript text omitted]

The number of C4 pixels was approximately 10822 with the percentage around 60%, the number of C3 pixels was approximately 4331 with the percentage ranging from 21.72% to 24.40%, and the number of C2 pixels was approximately 3300 with the percentage ranging from 15.59% to 18.36%.

**Table 3** The percentage of each class pixels adjusted by each rule using the WHU-SGCC method within the Jinsha River Basin.

| Validation | C2 Pixels (%) | C3 Pixels (%) | C4 Pixels (%) |
|---|---|---|---|

| gauge station | Spring | Summer | Fall | Winter | Spring | Summer | Fall | Winter | Spring | Summer | Fall | Winter |
|---|---|---|---|---|---|---|---|---|---|---|---|---|
| 52908 | 20.80% | 16.59% | 29.15% | 15.52% | 17.76% | 22.85% | 20.82% | 18.16% | 61.29% | 60.40% | 49.87% | 66.16% |
| 56004 | 20.89% | 15.59% | 29.40% | 15.65% | 16.29% | 22.24% | 20.64% | 18.83% | 62.66% | 62.01% | 49.81% | 65.36% |
| 56021 | 21.38% | 17.91% | 32.46% | 15.65% | 17.55% | 24.40% | 21.85% | 19.91% | 60.91% | 57.53% | 45.53% | 64.28% |
| 56029 | 21.77% | 18.06% | 32.60% | 16.03% | 17.31% | 24.06% | 21.61% | 19.64% | 60.76% | 57.72% | 45.63% | 64.18% |
| 56034 | 21.09% | 17.86% | 31.22% | 14.86% | 17.78% | 23.95% | 23.07% | 20.19% | 60.97% | 58.03% | 45.55% | 64.79% |
| 56038 | 20.48% | 17.36% | 30.72% | 15.56% | 16.12% | 21.72% | 23.74% | 17.63% | 63.23% | 60.76% | 45.39% | 66.65% |
| 56144 | 21.42% | 18.11% | 31.97% | 16.00% | 16.46% | 24.03% | 21.78% | 19.38% | 61.96% | 57.70% | 46.09% | 64.46% |
| 56146 | 21.33% | 17.22% | 31.77% | 15.70% | 17.12% | 24.24% | 21.42% | 18.34% | 61.39% | 58.38% | 46.65% | 65.81% |
| 56152 | 21.32% | 17.17% | 31.27% | 15.57% | 17.56% | 22.59% | 22.32% | 18.94% | 60.96% | 60.08% | 46.26% | 65.34% |
| 56167 | 21.46% | 18.19% | 32.36% | 15.84% | 16.90% | 23.51% | 21.72% | 19.03% | 61.48% | 58.14% | 45.76% | 64.98% |
| 56247 | 21.66% | 18.32% | 31.44% | 16.10% | 17.16% | 23.89% | 22.19% | 19.55% | 61.03% | 57.63% | 46.21% | 64.20% |
| 56251 | 21.09% | 17.86% | 31.28% | 15.73% | 17.39% | 23.53% | 22.88% | 18.50% | 61.36% | 58.46% | 45.68% | 65.62% |
| 56257 | 21.17% | 17.93% | 30.99% | 15.95% | 16.15% | 21.88% | 23.55% | 19.13% | 62.53% | 60.04% | 45.30% | 64.77% |
| 56357 | 21.62% | 18.14% | 31.59% | 15.64% | 17.12% | 23.75% | 22.54% | 19.52% | 61.10% | 57.95% | 45.71% | 64.68% |
| 56374 | 21.52% | 18.08% | 31.92% | 14.32% | 17.38% | 23.23% | 21.90% | 19.20% | 60.95% | 58.53% | 46.02% | 66.32% |
| 56459 | 21.30% | 18.10% | 32.14% | 15.64% | 16.92% | 23.45% | 21.16% | 19.17% | 61.62% | 58.29% | 46.54% | 65.03% |
| 56462 | 21.67% | 18.29% | 32.68% | 15.92% | 17.28% | 23.68% | 21.55% | 19.14% | 60.90% | 57.87% | 45.61% | 64.78% |
| 56475 | 21.49% | 18.10% | 32.49% | 15.98% | 16.36% | 23.50% | 22.08% | 19.53% | 62.00% | 58.24% | 45.28% | 64.33% |
| 56479 | 20.42% | 17.88% | 31.34% | 15.69% | 16.35% | 22.79% | 19.36% | 18.74% | 63.07% | 59.17% | 49.14% | 65.41% |
| 56485 | 21.44% | 18.36% | 32.78% | 15.64% | 17.43% | 23.91% | 21.82% | 19.85% | 60.97% | 57.57% | 45.24% | 64.35% |
| 56543 | 21.52% | 18.25% | 32.51% | 15.87% | 16.97% | 23.72% | 21.78% | 18.90% | 61.35% | 57.87% | 45.56% | 65.06% |
| 56565 | 21.21% | 17.54% | 30.93% | 15.52% | 17.81% | 24.08% | 23.55% | 19.96% | 60.83% | 58.23% | 45.36% | 64.37% |
| 56571 | 21.62% | 17.89% | 31.31% | 14.94% | 17.03% | 23.07% | 20.83% | 18.94% | 61.19% | 58.89% | 47.70% | 65.97% |
| 56586 | 21.73% | 18.33% | 21.73% | 15.49% | 17.35% | 23.99% | 17.35% | 19.59% | 60.76% | 57.53% | 60.76% | 64.77% |
| 56651 | 20.90% | 18.07% | 32.46% | 15.38% | 17.78% | 23.98% | 22.13% | 19.95% | 61.16% | 57.79% | 45.25% | 64.51% |
| 56664 | 20.94% | 18.22% | 32.43% | 15.50% | 16.64% | 23.06% | 21.00% | 18.70% | 62.26% | 58.56% | 46.42% | 65.64% |
| 56666 | 21.06% | 17.98% | 31.59% | 15.39% | 18.03% | 23.97% | 22.41% | 19.64% | 60.76% | 57.89% | 45.84% | 64.82% |
| 56671 | 20.71% | 18.16% | 32.55% | 15.67% | 16.53% | 23.63% | 21.89% | 20.03% | 62.61% | 58.06% | 45.41% | 64.14% |
| 56684 | 21.36% | 18.04% | 32.65% | 15.46% | 17.72% | 23.15% | 21.95% | 19.28% | 60.76% | 58.65% | 45.24% | 65.10% |
| 56778 | 21.63% | 18.11% | 32.25% | 15.91% | 17.31% | 23.14% | 22.11% | 19.52% | 60.90% | 58.59% | 45.48% | 64.41% |

Table 4 The number of each class pixels adjusted by each rule using the WHU-SGCC method inside the Jinsha River Basin.

| Validation gauge station | C1 Pixels (%) | C2 Pixels (%) | C3 Pixels (%) | C4 Pixels (%) |
|---|---|---|---|---|
| 52908 | 29 (0.16%) | 3066 (16.59%) | 4224 (22.85%) | 11163 (60.40%) |
| 56004 | 29 (0.16%) | 2882 (15.59%) | 4111 (22.24%) | 11460 (62.01%) |
| 56021 | 29 (0.16%) | 3311 (17.91%) | 4510 (24.40%) | 10632 (57.53%) |
| 56029 | 29 (0.16%) | 3338 (18.06%) | 4447 (24.06%) | 10668 (57.72%) |
| 56034 | 29 (0.16%) | 3300 (17.86%) | 4427 (23.95%) | 10726 (58.03%) |
| 56038 | 29 (0.16%) | 3209 (17.36%) | 4014 (21.72%) | 11230 (60.76%) |
| 56144 | 29 (0.16%) | 3347 (18.11%) | 4442 (24.03%) | 10664 (57.70%) |
| 56146 | 29 (0.16%) | 3183 (17.22%) | 4480 (24.24%) | 10790 (58.38%) |
| 56152 | 29 (0.16%) | 3173 (17.17%) | 4176 (22.59%) | 11104 (60.08%) |
| 56167 | 29 (0.16%) | 3362 (18.19%) | 4346 (23.51%) | 10745 (58.14%) |
| 56247 | 29 (0.16%) | 3385 (18.32%) | 4416 (23.89%) | 10652 (57.63%) |
| 56251 | 29 (0.16%) | 3301 (17.86%) | 4348 (23.53%) | 10804 (58.46%) |
| 56257 | 29 (0.16%) | 3313 (17.93%) | 4043 (21.88%) | 11097 (60.04%) |
| 56357 | 29 (0.16%) | 3352 (18.14%) | 4390 (23.75%) | 10711 (57.95%) |
| 56374 | 29 (0.16%) | 3341 (18.08%) | 4294 (23.23%) | 10818 (58.53%) |
| 56459 | 29 (0.16%) | 3345 (18.10%) | 4334 (23.45%) | 10774 (58.29%) |
| 56462 | 29 (0.16%) | 3380 (18.29%) | 4377 (23.68%) | 10696 (57.87%) |
| 56475 | 29 (0.16%) | 3345 (18.10%) | 4344 (23.50%) | 10764 (58.24%) |
| 56479 | 29 (0.16%) | 3305 (17.88%) | 4212 (22.79%) | 10936 (59.17%) |
| 56485 | 29 (0.16%) | 3393 (18.36%) | 4419 (23.91%) | 10641 (57.57%) |
| 56543 | 29 (0.16%) | 3373 (18.25%) | 4384 (23.72%) | 10696 (57.87%) |
| 56565 | 29 (0.16%) | 3241 (17.54%) | 4450 (24.08%) | 10762 (58.23%) |
| 56571 | 29 (0.16%) | 3306 (17.89%) | 4263 (23.07%) | 10884 (58.89%) |
| 56586 | 29 (0.16%) | 3387 (18.33%) | 4434 (23.99%) | 10632 (57.53%) |
| 56651 | 29 (0.16%) | 3340 (18.07%) | 4432 (23.98%) | 10681 (57.79%) |
| 56664 | 29 (0.16%) | 3368 (18.22%) | 4262 (23.06%) | 10823 (58.56%) |
| 56666 | 29 (0.16%) | 3323 (17.98%) | 4431 (23.97%) | 10699 (57.89%) |
| 56671 | 29 (0.16%) | 3356 (18.16%) | 4367 (23.63%) | 10730 (58.06%) |
| 56684 | 29 (0.16%) | 3335 (18.04%) | 4278 (23.15%) | 10840 (58.65%) |
| 56778 | 29 (0.16%) | 3347 (18.11%) | 4277 (23.14%) | 10829 (58.59%) |

**4.1 Model performance based on overall accuracy evaluations**

The multi-year (1990-2014) average seasonal precipitation over the Jinsha River Basin interpolated from WHU-SGCC, CHIRP
and CHIRPS is shown in Fig. 4.There exist some differences in the spatial pattern of precipitation estimates. Overall, the
WHU-SGCC method exhibits the similar spatial distribution of precipitation to the CHIRP and CHIRPS, while the WHU-
SGCC method attenuated the intense rain in the central area. The statistical accuracy evaluations are needed to further analyze
the performance of the WHU-SGCC method.

[Figure]

Spring

Summer

Fall

Winter

WHU-SGCC          CHIRP          CHIRPS

10   50   100   200   300   400   500   600   700   800   900
Average precipitation (mm)

**Figure 4** The multi-year (1990-2014) average seasonal precipitation over the Jinsha River Basin interpolated from WHU-SGCC, CHIRP and CHIRPS.

To test the performance of the WHU-SGCC method for precipitation estimates, the statistical analyses of PCC, RMSE, BAE, BIAS, NSE, POD, FAR, and CSI were calculated and are presented in Table 5 4 (tThe results were derived from the 22 clusters for the FCM in Rule 2, as shown in Appendix BE, and $\alpha$ =0.1 for the IDW in Rule 4 after the comparison of the RMSEs). After the correction, the PCC in the WHU-SGCC method shows an improvement relative to the CHIRP and CHIRPS estimates. The spring and fall have better correlations than the summer and winter. In addition, the NSE of the WHU-SGCC provides substantial improvements over CHIRP and CHIRPS, especially in the spring and fall which were better than the summer and winter. The RMSE and MAE are the largest in the summer, followed by the fall, spring, and winter; however, the performances of the BIAS in the summer and fall are better than those in the spring and winter, which might be influenced by
the greater precipitation in the summer and fall than in the spring and winter. The assessments of the POD and CSI are lowest,
and the FAR is largest in the winter due to the overestimation of no rain events estimated by the satellite-based data set.

Compared with the estimates of CHIRP and CHIRPS, the PCCs of the WHU-SGCC method are improved to more than 0.5
in the spring and fall and to approximately to 0.5 in the winter. In addition, the RMSE and MAE of the WHU-SGCC were all
lower than those of CHIRP and CHIRPS. The absolute values of the BIAS of the WHU-SGCC are substantial improved in the
spring, followed by the summer, winter and fall. Although the absolute value of the BIAS of the WHU-SGCC in fall are not
significantly better than those of CHIRP and CHIRPS, all of the values are approximately 0. The NSEs of the WHU-SGCC
reached 0.2836, 0.2944 and 0.1853 in the spring, fall and winter, respectively, which are substantially better than the negative
or zero values of CHIRP and CHIRPS. In the summer, the NSE of the WHU-SGCC is still negative, but it is improved to be
nearly zero, which indicates that the adjusted results are similar to the average level of the rain gauge observations. It is worth
noting that in the spring, summer and fall, the POD values of the WHU-SGCC are in the range of 0.95 to 1, better than CHIRP
and CHIRPS, and the FAR values of the WHU-SGCC are no more than 0.3, lower than CHIRP and CHIRPS; these results
represent the better ability of the WHU-SGCC method to predict precipitation events. The rainfall detection ability is the worst
in the winter compared to the other seasons. This can be explained by the seasonal distribution of precipitation in the Jinsha
River Basin, in which the most rainfall occurs in the summer, followed by the fall, spring and winter. In addition, the spatial
distribution of C2 and C3 pixels also significantly impact the overall accuracy in different seasons that the most uniform in the
fall, while the sparsest in the winter. The worst errors of forecasting performance in the summer may be attributed to the
highest precipitation. The limited precipitation event detection in the winter could also be explained by the lowest precipitation
(Xu et al., 2019).

Compared with the satellite images of CHIRP and CHIRPS, the results of the WHU-SGCC method provide the greatest
improvements for regional daily precipitation estimates over the Jinsha River Basin during the JJA from 1990 to 2014. After
bias adjustment of the WHU-SGCC, PCC was improved by 3.34% and 31.81% compared to CHIRP and CHIRPS, respectively.
Meanwhile, the RMSE and MAE of the WHU-SGCC was decreased by 6.91% and 6.59% compared to CHIRP, and by 22.71%
and 22.15% compared to CHIRPS. Although, the absolute value of BIAS of WHU-SGCC was no significant improvement
than CHIRP and slightly higher than CHIRPS, all of the values were approximately to 0. This results of BIAS indicates that
all the three kinds of data were much the same on the performance of relative bias. Nevertheless, the NSE of the WHU-SGCC
reached -0.0137, an increase of 93.33% and 98.32% compared to CHIRP and CHIRPS, respectively. The NSE of WHU-SGCC
was still negative, but it was improved to be zero that indicates the adjusted results are close to the average level of the rain
gauge observations, while the NSEs of CHIRP and CHIRPS were much worse. It is noted that the POD of WHU-SGCC was
approximate to 1, better than CHIRP and CHIRPS, and the FAR of WHU-SGCC was 0.11, lower than CHIRP and CHIRPS,
which represents the better ability on precipitation event predictions of the WHU-SGCC.

**Table 5 4** Overall accuracy assessments during for the four seasonsJJA from 1990 to 2014.

[revised manuscript text omitted]

In terms of performance with respect to different daily rain events, the The WHU-SGCC approach had the lowerest errors
than CHIRP and CHIRPS, as indicated by the RMSE, MAE and BIAS for events with total rainfall lower than 25 mm, but the
performance of WHU-SGCC for total rainfall higher than 25 mm is not promising for events with total rainfall greater than 25
mmdid in the summer (Fig. 8). not improve compared to CHIRP and CHIRPS. This negative performance for total rainfall
higher than 25 mm in the summer might be attributed to the overestimation of rainfall by CHIRP and CHIRPS. For the seasonal
distribution of precipitation (Table 5) This negative performance on the total rainfall higher than 25 mm was probably caused
by (Table 6)., Tthe average daily precipitation within the basin was less than 10 mm inside the basin over the study period,
during the multi-year summer seasons, which provided results in numerousa large amount of rain gauge stations data with the
values lower than 10 mm, which hadthat caused a significantly impact on the establishment of statistical relationships
establishment for the WHU-SGCC. The estimates of extreme rain events might also be attenuated during the process of WHU-
SGCC adjustment. In hence
Besides, the POD and CSI results of CHIRPS are the worst, while the results of the WHU-SGCC are the highest, which
indicate its superiority for the detection of precipitation events. As for the results of the WHU-SGCC, the assessments of POD
and CSI are the best in the summer, followed by the fall, spring, and winter, which are related to the seasonal rainfall pattern
of more rain in the summer and less in the winter.
Therefore, the approach of WHU-SGCC approach is applicable for the detection of rainfall events over in the Jinsha River
Basin, while in the summer is better with the rainfall less than or approximately equal to the average daily precipitation. Due
to the homogenization of the WHU-SGCC method, its performance for short intense and extreme rain events was poorer than
those of CHIRP and CHIRPS, which should be improved in a future study.
less than 10 mm, or even than 25mm. Due to the 4.34% of summer days with the daily precipitation over the 25 mm, the
performance of WHU-SGCC on these rain events was poorer than the results of CHIRP and CHRPS.

[Figure]

**Figure 8** Accuracy assessment of liquid precipitation events from 1990 to 2014.

**Table 7** Accuracy assessment on liquid precipitation events during the JJA from 1990 to 2014.

| Rain Event | RMSE | | | MAE | | | BIAS | | |
| --- | --- | --- | --- | --- | --- | --- | --- | --- | --- |
| | WHU-SGCC | CHIRP | CHIRPS | WHU-SGCC | CHIRP | CHIRPS | WHU-SGCC | CHIRP | CHIRPS |
| <0.1 | 4.7253 | 5.0802 | 7.1643 | 2.5927 | 2.9562 | 2.9145 | / | / | / |
| [0.1,10) | 4.1661 | 6.8684 | 9.6022 | 3.9885 | 4.5534 | 6.2462 | 0.8021 | 1.4435 | 1.9842 |
| [10,25) | 10.4281 | 11.0848 | 13.4427 | 9.2722 | 9.6866 | 11.5909 | -0.5762 | 0.6342 | 0.7559 |
| [25,50) | 25.7494 | 24.5600 | 25.4975 | 24.8386 | 23.0967 | 23.4927 | -0.7784 | 0.7250 | 0.7388 |
| 50 | 56.6072 | 54.5037 | 52.7875 | 54.4168 | 52.1557 | 49.4318 | -0.8861 | 0.8297 | 0.7852 |

**5 Data availability**

[revised manuscript text omitted]

The authors are grateful for the editor and anonymous reviewers for their useful suggestions that clearly improved this paper.

**References**

Adler, R. F., Huffman, G. J., Chang, A., Ferraro, R., Xie, P. P., Janowiak, J., Rudolf, B., Schneider, U., Curtis, S., Bolvin, D., Gruber, A., Susskind, J., Arkin, P., and Nelkin, E.: The version-2 global precipitation climatology project (GPCP) monthly precipitation analysis (1979-present), J. Hydrometeorol., 4, 1147-1167, doi:10.1175/1525-7541(2003)004<1147:tvgpcp>2.0.co;2, 2003.

[revised manuscript text omitted]

---

## Author Response (AR3)

**Reply to Report #1 (Anonymous Referee #7)**

Dear Reviewer,

Thank you for your insight comments and suggestions. We have modified the manuscript accordingly. We trust that all of your comments have been addressed accordingly in the revised manuscript. If you have further suggestions for changes, please let us know. The detailed corrections are listed below point by point:

All changes in the manuscript are marked with red color.

**Major comments**

The authors have substantially improved this version of the manuscript. They have taken all my suggestions into account and therefore, I recommend the publication of this manuscript after addressing the following points.

**Answer:** Thank you for your valuable comments. In addition, we updated the DOI of the resulting dataset.

The newly uploaded files are the results of precipitation bias adjustment by WHU-SGCC method about four seasons.

Shen, G. Y., Chen, N. C., Wang, W., and Chen, Z. Q.: Improving the Climate Hazards Group Infrared Precipitation (CHIRP) using WHU-SGCC method over the Jinsha River Basin from 1990 to 2014. PANGAEA, https://doi.org/10.1594/PANGAEA.905376, 2019.

In addition, these four files without gauge information are the results of the WHU-SGCC method with all gauges as the input.1. Figure 5: Here the authors show Pearson's correlation coefficients. I believe that the legend is a bit misleading. Generally, the green is used to represent well performance but the authors use it in the range between 0.2 and 0.5. It would be better to use a two-color palette (e.g., from red to blue).

**Answer:** Thanks. We changed the color of legend.

**Change:**

[Figure]

**Figure 5** Spatial distribution of the Pearson's correlation coefficient of the overall agreement between observations and the WHU-SGCC, CHIRP and CHIRPS estimations in the four seasons from 1990 to 2014.

2. There is still not clear the influence in the number of gauge stations in the approach. I assume that the performance of this method will decrease with a reduction in the number of C1 pixels. The authors need to include an evaluation of the performance of the method with a varying number of rain gauge stations.

**Answer:** Thank you for your valuable comments. In this study, the WHU-SGCC approach is to estimates the precipitation at every pixel by blending satellite estimates and rain gauge observations considering the terrain factors and precipitation characteristics. The leave-one-out cross validation step was applied to compute the out-of-sample adjusted error with the gauge stations. The WHU-SGCC algorithm was repeated 30 times, each time leaving one station as the validation station. The main objective of this study is to present a novel approach for reblending gauge observations and CHIRP satellite-based estimates.

The statistical comparisons between the different spatial distribution of observations and the WHU-SGCC, CHIRP and CHIRPS precipitation were conducted as shown in Fig. 5 and Fig. 6. The results show that, the variation in the PCC shows low correlations in areas with lower elevation, particularly in the southeast Jinsha River Basin, where there is higher precipitation and a greater density of rain gauges. The PCC is highest in the fall, followed by the spring and winter, and finally by summer. The higher correlations are located in the north-central area along the Tongtian River, Jinsha River and upstream part of the Yalong River, which has complex terrain and few rain gauges. The RMSE is lowest in the winter than in the spring, fall and summer, which can be attributed to the lower precipitation in the winter and the greatest in the summer. The spatial distribution of the RMSE shows that, the smaller errors are scattered in the northwest area of the river basin, with values lower than 5 mm, while the highest errors are located along the border between the lower reaches of the Jinsha Jiang River and the river basin. This is related to the climate regimes of the Jinsha River Basin, which includes more rainfall in the south and southeast areas than in the north, and northwest. Therefore, the total rainfall and spatial distribution of rain gauges (C1 pixels) and will influence the estimates accuracy. But we are not sure that the performance of this method will decrease with a reduction in the number of C1 pixels. In the future, we will further investigate the performance of the different number of C1 pixels on the WHU-SGCC adjustment.

3. One of my main concerns is that the main objective of this approach is to improve the characterization of precipitation over mountainous regions. However, ~50-60% of the pixels are classified as C4 and interpolated using IDW which does not account for the influence of elevation in the interpolation of precipitation. The authors may apply another interpolation method that accounts for the precipitation gradient related to elevation.

**Answer:** Thanks. It is reasonable to assume that some pixels are statistically similar to the historical precipitation characteristics of the C1 pixels within a certain area. Several studies indicate that the geographical location, elevation and other terrain information influence the spatial distribution of rainfall, especially in mountainous areas with complex topography (Anders et al., 2006; Long and Singh, 2013). The size of the spatial range is an important parameter to distinguish the spatial similarity and heterogeneity. In the WHU-SGCC method, the fuzzy c-means (FCM) clustering approach was used to determine the spatial range considered for each pixel's terrain factors, including longitude, latitude, elevation, slope, aspect and curvature. Therefore, the elevation was considered into the process of the WHU-SGCC. In Rule 4, the IDW method was used to interpolate the unknown pixels based on C2 and C3 pixels. IDW is based on the concept of the first law of geography from 1970, which was defined as *everything is related to everything else, but near things are more related than distant things*. Therefore, the attribute value of an unsampled point is the weighted average of the known values within the neighbourhood, and the distance weighting can be determined by means of IDW (Lu and Wong, 2008). Consider the quality of the results and the algorithm efficiency, the IDW method is applicable into the WHU-SGCC.

In the future, we will further investigate another interpolation method that accounts for the precipitation gradient related to elevation when time is sufficient, such as geographically weighted regression.

**Minor comments**

1. Line 10: What do the authors mean with "existing fusion precipitation estimates"?
**Answer:** Thanks. "Existing fusion precipitation estimates" means the existing precipitation products of multi-source data fusion.

2. Line 27: might be due to the ...
**Answer:** Thanks. Done.
**Change:** changed from "which may due to the" to "which might be due to the".

3. Line 28: events.
**Answer:** Thanks. Done.
**Change:** changed from "rain event" to "rain events".

4. Please add a space between the references.
**Answer:** Thanks. Done.
**Change:** We added the space between the references.

5. Line 55: Please replace "spacing" with "spatial resolution".
**Answer:** Thanks. Done.
**Change:** changed from "spacing" to "spatial resolution".

6. Line 89: evaluated and indicated.

**Answer:** Thanks. Done.

**Change:** changed from "Moreover, Bai et al. (2018) evaluates CHIRPS over mainland China and indicates that" to "Moreover, Bai et al. (2018) evaluated CHIRPS over mainland China and indicated that"

7. Table 1: Some of the temporal resolutions of the products start with uppercase and some of them with lowercase. Also, the period of CMORPH is not complete.

**Answer:** Thanks. Done.

**Changed:** We changed the temporal resolutions of the products starting with lowercase and added the temporal resolutions of CMORPH.

**Table 1** Coverage and spatiotemporal resolutions of major satellite precipitation datasets.

| Product | Temporal resolution | Spatial resolution | Period | Coverage |
|---|---|---|---|---|
| TRMM 3B42-RT | 3 Hourly | 0.25° | 1998-present | 50°S-50°N |
| CMORPH | 0.5 Hourly /3 Hourly/Daily | 8 km/0.25° | 1998- | 60°S-60°N |
| PERSIANN-CDR | Daily | 0.25° | 1983-(delayed) present | 60°S-60°N |
| GsMaP-NRT | Hourly | 0.01° | 2007 | 60°S-60°N |
| GsMaP-MVK | Hourly | 0.01° | 2000 | 60°S-60°N |
| GPM | 0.5 Hourly/Hourly/ 3 Hourly/Daily/3 Day/ 7 Day/Monthly | 0.1°/0.25°/0.05°/5° | 2014-present | 60°S-60°N 70°N-70°S 90°N-90°S |
| MSWEP | 3 Hourly/Daily/Monthly | 0.1° | 1979-2017 | 90°N-90°S |
| CHIRPS | Daily/Pentad/Dekad/ Monthly/Annual | 0.05°/0.25° | 1981- present | 50°S-50°N |

8. Line 112: probably present fits better than demonstrate.

**Answer:** Thanks. Done.

**Changed:** changed "demonstrate" to "present".

9. Line 121: replace the comma with "is".

Answer: Thanks. We replaced the comma with "is".

Change: changed from "The Yangtze River, one of the largest and most important rivers in Southeast Asia, originates on the Tibetan Plateau and extends approximately 6300 km eastward to the East China Sea." to "The Yangtze River is one of the largest and most important rivers in Southeast Asia, originating on the Tibetan Plateau and extending approximately 6300 km eastward to the East China Sea."

10. Lines 135-136: I think that now that the research is not focused in summer, it is better to remove this sentence.

**Answer:** Thanks. We removed this sentence about the summer precipitation.

11. Line 178: Recommendation, replace "in situ" with ground-based.
**Answer:** Thanks. Done.
**Change:** changed from "in situ" to "ground-based".

12. Line 194: remove "is to".
**Answer:** Thanks. We removed "is to".
**Changed:** changed from "the WHU-SGCC approach estimates the precipitation at every pixel by blending satellite estimates" to "the WHU-SGCC approach estimates the precipitation at every pixel by blending satellite estimates".

13. Recommendation: change "multi-year" for "multi-annual" throughout the manuscript
**Answer:** Thanks. We changed "multi-year" for "multi-annual" throughout the manuscript

**Reply to Report #3 (Anonymous Referee #4)**

Dear Reviewer,

Thank you for your insight comments and suggestions. We have modified the manuscript accordingly. We trust that all of your comments have been addressed accordingly in the revised manuscript. If you have further suggestions for changes, please let us know. The detailed corrections are listed below point by point:

All changes in the manuscript are marked with red color.

After the last review stage, the authors largely revised the manuscript to improve its quality and to show quantitatively what are the qualities of WHU-SGCC vs CHIRP and CHIRPS datasets. I think the manuscript has reached a fair quality and it can be subject to minor revisions, though a few of the comments reported below, if not seriously address, can definitely affect the consistency of the manuscript and contains serious mistakes.

**Answer:** Thank you for your valuable comments. In addition, we updated the DOI of the resulting dataset.

The newly uploaded files are the results of precipitation bias adjustment by WHU-SGCC method about four seasons.

Shen, G. Y., Chen, N. C., Wang, W., and Chen, Z. Q.: Improving the Climate Hazards Group Infrared Precipitation (CHIRP) using WHU-SGCC method over the Jinsha River Basin from 1990 to 2014. PANGAEA, https://doi.org/10.1594/PANGAEA.905376, 2019.

**Major comments**

1. First of all, concepts of errors and uncertainties are still not correctly used in the manuscript and I have the impression the use of the word "error" is sometimes abused by the authors which also mention in their reply the words "more errors", which definitely makes no sense.

Error is the difference between the true value of the measurand and the measured value. Accuracy is an expression of the lack of error. Uncertainty characterizes the range of values within which the true value is asserted to lie with some level of confidence.

Looking at the abstract I am amazed to see at line 17 the term "error adjustment". Error is error and cannot be adjusted. Values of climate variables can be adjusted if the presence of systematic effects is detected. At line 23-24, the authors state that "precipitation errors" can be reduced. May be the authors are talking about the RMSE

which is not the error. Going through, the text I can provide other examples.

I recommend again to define all the quantities discussed in the manuscript in a proper way. This is mandatory to my opinion and the authors can also check the GUM (https://www.bipm.org/en/publications/guides/gum.html) to stay consistent.

**Answer:** Thank you for your valuable comments. We misunderstood the concepts of "errors" and "bias".

**Change:** changed from "error adjustment" to "bias adjustment";

Changed from "precipitation errors" to "precipitation bias"

2. It is also not clear to me why a few information reported in the reply to the reviewers are not provided also in the manuscript. For example, the authors could describe the way CMA imposes strict quality control on rain gauge observations.

In their comments the authors say:

"The process of quality control is as follows:

(1) Climate threshold or allowable value check; (2) Extreme values at gauge stations check; (3) Internal consistency check between fixed value, daily average value and daily extreme value; (4) Time consistency check; (5) Manual verification and correction; this quality control approach is provided by the official document from CMA. So, the daily rain gauge observations were used as the "Reference dataset".

I think the information above must be provided when the CMA dataset is introduced in the manuscript to increase confidence in this dataset.

**Answer:** Thanks. We have added the description of strict quality on rain gauge observations in the manuscript.

Change: Added "The process of quality control conducted by the CMA is as follows: (1) Climate threshold or allowable value check; (2) Extreme values at gauge stations check; (3) Internal consistency check between fixed value, daily average value and daily extreme value; (4) Time consistency check; and (5) Manual verification and correction." in to section 2.2.1.

3. The authors spent an appropriate effort to improve the quality of the analysis used and to smooth the tone of the writing which was often too enthusiastic in the previous version. Issues which may affect the quality of the dataset are now better highlighted though this is done still in a qualitative way. Generally, this may be sufficient to allow the reader to properly use the WHU-SGCC dataset; nevertheless, wherever possible, I

recommend the authors to be more quantitative.

For example, when the authors say: "In addition, the spatial distribution of C2 and C3 pixels also significantly impact the overall accuracy in different seasons that the most uniform in the fall, while the sparsest in the winter.", it would be desirable to report a quantification of the impact which the authors are referring to. This happens also in other parts of the manuscript.

**Answer:** Thank you for your valuable comments.

**Change:**

(**1**) This sentence is not right, so we changed from "In addition, the spatial distribution of C2 and C3 pixels also significantly impact the overall accuracy in different seasons that the most uniform in the fall, while the sparsest in the winter." to "In addition, the spatial distribution of C2 and C3 pixels might slightly impact the overall accuracy in different seasons that the sparest in the winter, while more uniform in the summer. However, the performances of PCC, RMSE, MAE and NSE in the winter are better than those in the summer. The worst errors of forecasting performance in the summer may be attributed to the highest precipitation. The limited precipitation events detection in the winter could also be explained by the lowest precipitation (Xu et al., 2019)."

(**2**) Changed from "The WHU-SGCC method provides an obvious improvement in the NSE relative to CHIRP and CHIRPS, though the median and average values are still less than 0, which may be due to the inherent uncertainty in the CHIRP" to "The WHU-SGCC method provides an obvious improvement in the NSE, with average improvements of 0.1742-13.8322 and 2.0131-14.7052 relative to CHIRP and CHIRPS, though the median and average values are still less than 0, which may be due to the inherent uncertainty in the CHIRP"

(**3**) Changed from "The estimates of extreme rain events might also be attenuated during the process of WHU-SGCC adjustment." to "Besides the WHU-SGCC dataset has almost always a negative bias, while CHIRP and CHIRPS has a positive bias in the different rain events. After bias correction of the WHU-SGCC, some precipitation estimates are lower than observations. The estimates of extreme rain events might also be attenuated during the process of WHU-SGCC adjustment."

**Other comments:**

1. I am not sure why the authors speak about "attenuating the simulation" of extreme rain events at line 29 and other times thereinafter; I'd simply write "attenuating extreme rain events".

**Answer:** Thanks. We changed the sentence.

**Change:** changed from "which may due to the homogenization attenuating the simulation of extreme rain event." to "which might be due to the homogenization attenuating the extreme rain events estimates."

2.   Check if all the acronyms are reported, for example NSE is not introduced in the abstract

**Answer:** Thanks. We added the full name of NSE in the abstract.

**Change:** changed from "the NSE of the WHU-SGCC" to "the Nash-Sutcliffe efficiency coefficient (NSE) of the WHU-SGCC".

3.   At line 235-236 the sentence could be clearer, adding the term "separately" at the end of the sentence.

Answer: Thanks. Done.

Change: changed from "Because there are 30 gauge stations in the study area, 30 regression relationships at the C1 pixels were derived from Rule 1." to "In the process of Rule 1, the regression relationships at the C1 pixels were established at 30 gauge stations separately."

4.   At line 260, there is the assumption of "no long-held outliers", whose impact is anyhow not assessed in the manuscript. A few words more to demonstrate the robustness of this assumption would be useful.

**Answer:** Thanks. "no long-held outliers" means no abnormal values during the long time series.

Change: changed from "There are no long-held outliers at one pixel in the CHIRP dataset" to "There are no abnormal values at one pixel in the CHIRP dataset during the long time series"

5.   Table 3 shows interesting information which if reported in a plot style would be much clearer and useful for the reader, otherwise is quite confusing

**Answer:** Thanks. We added the spatial distribution of C1-C3 pixels in Figure D1.

**Change:** Added the sentence in the first paragraph in section 4.

  "Besides, the pixel type of the validation gauge stations is shown in Table D1 and the spatial distribution of C1-C3 pixels in Figure D1 with the most uniform in the fall, while the sparsest in the winter."

6.   Line 627-632: this is one of the points where the limitations affecting the WHU-SGCC dataset are discussed. I think that, at least for this case, the paragraph should be reported also in the conclusions to give the right balance to the final section of the manuscript.

**Answer:** Thanks. We added these sentences to the conclusions.

**Change:** "As for the results of the WHU-SGCC, the assessments of POD and CSI are the best in the summer, followed by the fall, spring, and winter, which are related to the seasonal rainfall pattern of more rain in the summer and less in the winter. In spite of the corrections, the performance of the WHU-SGCC for short intense and extreme rain events was poorer than those of CHIRP and CHIRPS, and the bias in the precipitation forecasts in the summer are still large, which may due to the homogenization attenuating the extreme rain events estimates."

7.    Figure 8: this is the core of the presented analysis and it is not clear why the WHU-SGCC dataset has almost always a negative bias, while CHIRP and CHIRPS has a positive bias. The authors should comment more on this important aspect of their homogenization approach.

**Answer:** Thanks. The equation of BIAS is as follows:

$$\mathrm{BIAS} = \frac{\sum_{i=1}^{n}(C_i - Y_{oi})}{\sum_{i=1}^{n} Y_{oi}}$$

In different rain events, the WHU-SGCC dataset has almost always a negative bias, while CHIRP and CHIRPS has a positive bias. After bias correction of the WHU-SGCC, some precipitation estimates are lower than observations. The estimates of extreme rain events might also be attenuated during the process of WHU-SGCC adjustment.

**Change:** Added more comments on this important aspect of their homogenization approach. "Besides the WHU-SGCC dataset has almost always a negative bias, while CHIRP and CHIRPS has a positive bias in the different rain events. After bias correction of the WHU-SGCC, some precipitation estimates are lower than observations. The estimates of extreme rain events might also be attenuated during the process of WHU-SGCC adjustment."

8.    In addition, Figure 8 clearly states that CHIRP and WHU-SGCC have very similar performance, expect for the bias (which for WHU-SGCC is slightly better in absolute value). This should be clearly stated in the analysis and in the conclusions.

**Answer:** Thanks. As shown in Fig. 8, the WHU-SGCC approach had lower errors than CHIRP and CHIRPS, as indicated by the RMSE, MAE and BIAS, but the performance of WHU-SGCC is not promising for events with total rainfall greater than 25 mm in the summer. Besides, the POD and CSI results of CHIRPS are the worst, while the results of the WHU-SGCC are the highest, which indicate its superiority for the detection of precipitation events. As for the results of the WHU-SGCC, the assessments of POD and CSI are the best in the summer, followed by the fall, spring, and winter, which are related to the seasonal rainfall pattern of more rain in the summer and less in the winter. Therefore, the WHU-SGCC approach is applicable for the detection of rainfall events in the Jinsha River Basin, while in the summer it is better with rainfall less than or approximately equal to the average daily precipitation. Due to the homogenization of the WHU-SGCC method, its performance for short intense and extreme rain events was poorer than those of CHIRP and CHIRPS, which should be improved in a future study. **And we have added the relevant analysis in the conclusion** "The leave-one-out cross validation of WHU-SGCC on daily rain events confirmed that the model is effective in the detection of precipitation events that are less than or approximately equal to the average annual precipitation in the Jinsha River Basin. The WHU-SGCC approach achieves reductions of the RMSE, MAE and BIAS metrics, while on rain events less than 25 mm in the summer. Specifically, the WHU-SGCC has the best ability to reduce precipitation errors for daily accuracy evaluations, with average reductions of 15% and 34% for compared to CHIRP and CHIRPS, respectively. As for the results of the WHU-SGCC, the assessments of POD and CSI are the best in the summer, followed by the fall, spring, and winter, which are related to the seasonal rainfall pattern of more rain in the summer and less in the winter. In spite of the corrections, the performance of the WHU-SGCC for short intense and extreme rain events was poorer than those of CHIRP and CHIRPS, and the bias in the precipitation forecasts in the summer are still large, which may due to the homogenization attenuating the extreme rain events estimates."

9. Finally, I recommend an appropriate use of decimal places throughout the presented analysis. The manuscript can also benefit from a final English review (I recommend).
**Answer:** Thanks. We retain four decimal.

[revised manuscript text omitted]

**Figure C1** Changes in the out-of-bag (OOB) error with increasing number of decision trees by means of random forest regression at each gauge station.

**Appendix D: Spatial distribution of C1, C2 and C3 pixels**

**Table D1** Pixel type of the validation gauge station.

| Validation gauge station \ Pixel type | Spring | Summer | Fall | Winter |
|---|---|---|---|---|
| 52908 | C4 | C4 | C4 | C4 |
| 56004 | C4 | C4 | C4 | C4 |
| 56021 | C2 | C2 | C2 | C3 |
| 56029 | C2 | C3 | C2 | C3 |
| 56034 | C2 | C3 | C2 | C3 |
| 56038 | C4 | C4 | C4 | C4 |
| 56144 | C4 | C4 | C4 | C4 |
| 56146 | C4 | C4 | C4 | C4 |
| 56152 | C2 | C3 | C3 | C4 |
| 56167 | C4 | C2 | C2 | C4 |
| 56247 | C4 | C4 | C4 | C4 |
| 56251 | C2 | C2 | C3 | C3 |
| 56257 | C4 | C4 | C4 | C4 |
| 56357 | C4 | C4 | C4 | C4 |
| 56374 | C4 | C4 | C3 | C4 |
| 56459 | C4 | C4 | C4 | C4 |
| 56462 | C4 | C4 | C4 | C4 |
| 56475 | C4 | C3 | C3 | C3 |
| 56479 | C4 | C4 | C4 | C4 |
| 56485 | C3 | C2 | C2 | C3 |
| 56543 | C3 | C3 | C4 | C4 |
| 56565 | C2 | C2 | C3 | C3 |
| 56571 | C2 | C4 | C4 | C4 |
| 56586 | C2 | C3 | C2 | C3 |
| 56651 | C3 | C2 | C2 | C3 |
| 56664 | C4 | C4 | C4 | C4 |
| 56666 | C3 | C3 | C3 | C3 |
| 56671 | C3 | C2 | C2 | C3 |
| 56684 | C2 | C2 | C2 | C4 |
| 56778 | C4 | C3 | C3 | C4 |

52908

[Figure]

[revised manuscript text omitted]